

# Global Methane Budget 2000-2020

Marielle Saunois[1], Adrien Martinez[1], Benjamin Poulter[2], Zhen Zhang[3,4], Peter A. Raymond[5], Pierre Regnier[6], Josep G. Canadell[7], Robert B. Jackson[8], Prabir K. Patra[9,10], Philippe Bousquet[1], Philippe Ciais[1], Edward J. Dlugokencky[11], Xin Lan[11,12], George H. Allen[13], David Bastviken[14], David J. Beerling[15], Dmitry A. Belikov[16], Donald R. Blake[17], Simona Castaldi[18], Monica Crippa[19,20], Bridget R. Deemer[21], Fraser Dennison[22], Giuseppe Etiope[23,24], Nicola Gedney[25], Lena Höglund-Isaksson[26], Meredith A. Holgerson[27], Peter O. Hopcroft[28], Gustaf Hugelius[29], Akihiko Ito[30], Atul K. Jain[31], Rajesh Janardanan[32], Matthew S. Johnson[33], Thomas Kleinen[34], Paul B. Krummel[22], Ronny Lauerwald[35], Tingting Li[36], Xiangyu Liu[37], Kyle C. McDonald[38], Joe R. Melton[39], Jens Mühle[40], Jurek Müller[41], Fabiola Murguia-Flores[42], Yosuke Niwa[32,43], Sergio Noce[44], Shufen Pan[45], Robert J. Parker[46], Changhui Peng[47,48], Michel Ramonet[1], William J. Riley[49], Gerard Rocher-Ros[50], Judith A. Rosentreter[51], Motoki Sasakawa[32], Arjo Segers[52], Steven J. Smith[53,54], Emily H. Stanley[55], Joël Thanwerdas[56,*], Hanqin Tian[57], Aki Tsuruta[58], Francesco N. Tubiello[59], Thomas S. Weber[60], Guido R. van der Werf[61], Douglas E. J. Worthy[62], Yi Xi[1], Yukio Yoshida[32], Wenxin Zhang[63], Bo Zheng[64,65], Qing Zhu[49], Qiuan Zhu[66], and Qianlai Zhuang[37]

[1]Laboratoire des Sciences du Climat et de l'Environnement, LSCE-IPSL (CEA-CNRS-UVSQ), Université Paris-Saclay 91191 Gif-sur-Yvette, France
[2]NASA Goddard Space Flight Center, Biospheric Science Laboratory, Greenbelt, MD 20771, USA
[3]National Tibetan Plateau Data Center (TPDC), State Key Laboratory of Tibetan Plateau Earth System, Environment and Resource (TPESER), Institute of Tibetan Plateau Research, Chinese Academy of Sciences, Beijing, 100101, China
[4]Earth System Science Interdisciplinary Center, University of Maryland, College Park, MD 20740, USA
[5]Yale School of the Environment, Yale University, New Haven, CT 06511, USA
[6]Department Geoscience, Environment & Society (BGEOSYS), Université Libre de Bruxelles, 1050 Bruxelles, Belgium
[7]Global Carbon Project, CSIRO Environment, Canberra, ACT 2601, Australia
[8]Department of Earth System Science, Woods Institute for the Environment, and Precourt Institute for Energy, Stanford University, Stanford, CA 94305-2210, USA
[9]Research Institute for Global Change, JAMSTEC, 3173-25 Showa-machi, Kanazawa, Yokohama, 236-0001, Japan
[10]Research Institute for Humanity and Nature, Kyoto 6038047, Japan
[11]NOAA Global Monitoring Laboratory, 325 Broadway, Boulder, CO 80305, USA
[12]Cooperative Institute for Research in Environmental Sciences, University of Colorado Boulder, CO 80303, USA
[13]Department of Geosciences, Virginia Polytechnic Institute and State University, Blacksburg, VA, USA
[14]Department of Thematic Studies – Environmental Change, Linköping University, 581 83 Linköping, Sweden
[15]School of Biosciences, University of Sheffield, UK
[16]Center for Environmental Remote Sensing, Chiba University, Chiba, 263-8522, Japan
[17]Department of Chemistry, University of California Irvine, 570 Rowland Hall, Irivine, CA 92697, USA
[18]Dipartimento di Scienze Ambientali, Biologiche e Farmaceutiche, Università degli Studi della Campania Luigi





Vanvitelli, via Vivaldi 43, 81100 Caserta, Italy
[19]European Commission, Joint Research Centre (JRC), Ispra, Italy
[20]Unisystems S.A., Milan, Italy
[21]U.S. Geological Survey, Southwest Biological Science Center, Flagstaff, AZ, USA
[22]CSIRO Environment, Aspendale, Victoria 3195, Australia
[23]Istituto Nazionale di Geofisica e Vulcanologia, Sezione Roma 2, via V. Murata 605 00143 Rome, Italy
[24]Faculty of Environmental Science and Engineering, Babes Bolyai University, Cluj-Napoca, Romania
[25] Met Office Hadley Centre, Joint Centre for Hydrometeorological Research, Maclean Building, Wallingford
OX10 8BB, UK
[26]Pollution Management Group (PM), International Institute for Applied Systems Analysis (IIASA), 2361
Laxenburg, Austria
[27] Department of Ecology & Evolutionary Biology, Cornell University, Ithaca, NY, USA
[28] School of Geography, Earth & Environmental Sciences, University of Birmingham, UK
[29] Department of Physical Geography and Bolin Centre for Climate Research, Stockholm University, 106 91
Stockholm, Sweden
[30] Graduate School of Agricultural and Life Sciences, The University of Tokyo, Tokyo, Japan
[31] Department of Atmospheric Sciences, University of Illinois, Urbana, IL 61821, USA
[32] Earth System Division, National Institute for Environmental Studies (NIES), Onogawa 16-2, Tsukuba, Ibaraki
305-8506, Japan
[33]Earth Science Division, NASA Ames Research Center, Moffett Field, CA USA.
[34]Max Planck Institute for Meteorology, Bundesstraße 53, 20146 Hamburg, Germany
[35] Université Paris-Saclay, INRAE, AgroParisTech, UMR EcoSys, Palaiseau, France
[36] LAPC, Institute of Atmospheric Physics, Chinese Academy of Sciences, Beijing, 100029, China
[37]Department of Earth, Atmospheric, and Planetary Sciences, Purdue University, West Lafayette, IN, USA
[38]Department of Earth and Atmospheric Sciences, City College of New York, City University of New York, NY,
USA
[39]Climate Research Division, Environment and Climate Change Canada, Victoria, BC, V8W 2Y2, Canada
[40]Scripps Institution of Oceanography, University of California San Diego, La Jolla, CA, 92037, USA
[41]Climate and Environmental Physics, Physics Institute and Oeschger Centre for Climate Change Research,
University of Bern, Sidlerstr. 5, 3012 Bern, Switzerland
[42]Instituto de Investigaciones en Ecología y Sustentabilidad, Universidad Nacional Autónoma de México,
Morelia, Mexico
[43]Department of Climate and Geochemistry Research, Meteorological Research Institute (MRI), Nagamine 1-1, Tsukuba,
Ibaraki 305-0052, Japan
[44]CMCC Foundation - Euro-Mediterranean Center on Climate Change, Italy
[45]Department of Engineering and Environmental Studies Program, Boston College, Chestnut Hill, MA 02467,
USA
[46]National Centre for Earth Observation, School of Physics and Astronomy, University of Leicester, Leicester,
LE1 7RH, UK
[47]Department of Biology Sciences, Institute of Environment Science, University of Quebec at Montreal,
Montreal, QC H3C 3P8, Canada
[48] School of Geographic Sciences, Hunan Normal University, Changsha 410081, China
[49]Climate and Ecosystem Sciences Division, Lawrence Berkeley National Lab, 1 Cyclotron Road, Berkeley, CA
94720, US
[50]Department of Forest Ecology and Management, Swedish University of Agricultural Sciences, 90183 Umeå,
Sweden
[51]Centre for Coastal Biogeochemistry, Faculty of Science and Engineering, Southern Cross University, Lismore,
NSW 2480, Australia
[52]TNO, dep. of Climate Air & Sustainability, P.O. Box 80015, NL-3508-TA, Utrecht, The Netherlands





[53]Joint Global Change Research Institute, Pacific Northwest National Lab, College Park, MD, USA
[54]Center for Global Sustainability, University of Maryland, College Park, MD, USA
[55]Center for Limnology, University of Wisconsin-Madison, Madison, WI, USA
[56]Empa, Swiss Federal Laboratories for Materials Science and Technology, Dübendorf, Switzerland
[57]Center for Earth System Science and Global Sustainability, Schiller Institute for Integrated Science and
Society, Department of Earth and Environmental Sciences, Boston College, Chestnut Hill, MA 02467, USA
[58]Finnish Meteorological Institute, P.O. Box 503, FI-00101, Helsinki, Finland
[59]Statistics Division, Food and Agriculture Organization of the United Nations (FAO), Viale delle
Terme di Caracalla, Rome 00153, Italy
[60]Department of Earth and Environmental Sciences, University of Rochester, Rochester, NY 14627,
USA
[61]Meteorology and Air Quality Group, Wageningen University and Research, Wageningen, the Netherlands
[62]Environment and Climate Change Canada, 4905, Dufferin Street, Toronto, Canada
[63]Department of Physical Geography and Ecosystem Science, Lund University, Sölvegatan 12, 223 62, Lund, Sweden
[64]Institute of Environment and Ecology, Tsinghua Shenzhen International Graduate School, Tsinghua University,
Shenzhen 518055, China
[65]State Environmental Protection Key Laboratory of Sources and Control of Air Pollution Complex, Beijing 100084,
China
[66]College of Geography and Remote Sensing, Hohai University, Nanjing, 210098, China
*formerly at LSCE [1]
*Correspondence to*: Marielle Saunois (marielle.saunois@lsce.ipsl.fr)
**Abstract.** Understanding and quantifying the global methane ($CH_4$) budget is important for assessing realistic pathways to
mitigate climate change. Emissions and atmospheric concentrations of $CH_4$ continue to increase, maintaining   $CH_4$ as the
second most important human-influenced greenhouse gas in terms of climate forcing after carbon dioxide ($CO_2$). The relative
importance of $CH_4$ compared to $CO_2$ for temperature change is related to its shorter atmospheric lifetime, stronger radiative
effect, and acceleration in atmospheric growth rate over the past decade, the causes of which are still debated. Two major
challenges in reducing uncertainties in the factors explaining the well-observed atmospheric growth rate arise from diverse,
geographically overlapping $CH_4$ sources and from the uncertain magnitude and temporal change in the destruction of $CH_4$
by short-lived and highly variable hydroxyl radicals (OH). To address these challenges, we have established a consortium
of multi-disciplinary scientists under the umbrella of the Global Carbon Project to improve, synthesise and update the global
$CH_4$ budget regularly and to stimulate new research on the methane cycle. Following Saunois et al. (2016, 2020), we present
here the third version of the living review paper dedicated to the decadal $CH_4$ budget, integrating results of top-down $CH_4$
emission estimates (based on in-situ and greenhouse gas observing satellite (GOSAT) atmospheric observations and an
ensemble of atmospheric inverse-model results) and bottom-up estimates (based on process-based models for estimating
land-surface emissions and atmospheric chemistry, inventories of anthropogenic emissions, and data-driven extrapolations).
We present a budget for the most recent 2010-2019 calendar decade (the latest period for which full datasets are available),
for the previous decade of 2000-2009 and for the year 2020.



The revision of the bottom-up budget in this edition benefits from important progress in estimating inland freshwater
emissions, with better accounting of emissions from lakes and ponds, reservoirs, and streams and rivers. This budget also
reduces double accounting across freshwater and wetland emissions and, for the first time, includes an estimate of the
potential double accounting that still exists (average of 23 Tg $CH_4$ $yr^{-1}$). Bottom-up approaches show that the combined
wetland and inland freshwater emissions average 248 [159-369] Tg $CH_4$ $yr^{-1}$ for the 2010-2019 decade. Natural fluxes are
perturbed by human activities through climate, eutrophication, and land use. In this budget, we also estimate, for the first
time, this anthropogenic component contributing to wetland and inland freshwater emissions. Newly available gridded
products also allowed us to derive an almost complete latitudinal and regional budget based on bottom-up approaches.
For the 2010-2019 decade, global $CH_4$ emissions are estimated by atmospheric inversions (top-down) to be 575 Tg $CH_4$ $yr^{-}$
$^1$ (range 553-586, corresponding to the minimum and maximum estimates of the model ensemble). Of this amount, 369 Tg
$CH_4$ $yr^{-1}$ or ~65% are attributed to direct anthropogenic sources in the fossil, agriculture and waste and anthropogenic
biomass burning (range 350-391 Tg $CH_4$ $yr^{-1}$ or 63-68%). For the 2000-2009 period, the atmospheric inversions give a
slightly lower total emission than for 2010-2019, by 32 Tg $CH_4$ $yr^{-1}$ (range 9-40). Since 2012, global direct anthropogenic
$CH_4$ emission trends have been tracking scenarios that assume no or minimal climate mitigation policies proposed by the
Intergovernmental Panel on Climate Change (shared socio-economic pathways SSP5 and SSP3). Bottom-up methods
suggest 16% (94 Tg $CH_4$ $yr^{-1}$) larger global emissions (669 Tg $CH_4$ $yr^{-1}$, range 512-849) than top-down inversion methods
for the 2010-2019 period. The discrepancy between the bottom-up and the top-down budgets has been greatly reduced
compared to the previous differences (167 and 156 Tg $CH_4$ $yr^{-1}$ in Saunois et al. (2016, 2020), respectively), and for the first
time uncertainty in bottom-up and top-down budgets overlap. The latitudinal distribution from atmospheric inversion-based
emissions indicates a predominance of tropical and southern hemisphere emissions (~65% of the global budget, <30°N)
compared to mid (30°N-60°N, ~30% of emissions) and high-northern latitudes (60°N-90°N, ~4% of global emissions). This
latitudinal distribution is similar in the bottom-up budget though the bottom-up budget estimates slightly larger contributions
for the mid and high-northern latitudes, and slightly smaller contributions from the tropics and southern hemisphere than
the inversions. Although differences have been reduced between inversions and bottom-up, the most important source of
uncertainty in the global $CH_4$ budget is still attributable to natural emissions, especially those from wetlands and inland
freshwaters.
We identify five major priorities for improving the $CH_4$ budget: i) producing a global, high-resolution map of water-saturated
soils and inundated areas emitting $CH_4$ based on a robust classification of different types of emitting ecosystems; ii) further
development of process-based models for inland-water emissions; iii) intensification of $CH_4$ observations at local (e.g.,
FLUXNET-$CH_4$ measurements, urban-scale monitoring, satellite imagery with pointing capabilities) to regional scales
(surface networks and global remote sensing measurements from satellites) to constrain both bottom-up models and
atmospheric inversions; iv) improvements of transport models and the representation of photochemical sinks in top-down
inversions, and v) integration of 3D variational inversion systems using isotopic and/or co-emitted species such as ethane



as well as information    in the bottom-up inventories on anthropogenic super-emitters detected by remote sensing (mainly
oil and gas sector but also coal, agriculture and landfills) to improve source partitioning.
The data presented here can be downloaded from https://doi.org/10.18160/GKQ9-2RHT (Martinez et al., 2024).
**1 Introduction**
The average surface dry air mole fraction of atmospheric methane ($CH_4$) reached 1912 ppb in 2022 (Fig. 1, Lan et
al., 2024), 2.6 times greater than its estimated pre-industrial value in 1750. This increase is attributable in large part to
increased anthropogenic emissions arising primarily from agriculture (e.g., livestock production, rice cultivation, biomass
burning), fossil fuel production and use, waste disposal, and alterations to natural $CH_4$ fluxes due to increased atmospheric
$CO_2$ concentrations, land use (Woodward et al., 2010, Fluet-Chouinard et al., 2023) and climate change (Ciais et al., 2013;
Canadell et al., 2021). Atmospheric $CH_4$ is a stronger absorber of Earth's emitted thermal infrared radiation than carbon
dioxide ($CO_2$), as assessed by its global warming potential (GWP) relative to $CO_2$. For a 100-yr time horizon and without
considering climate feedbacks the GWP of $CH_4$-fossil is 29.8 ($CH_4$-non fossil GWP is 27), whereas the values reach 82.5
over a 20-year horizon for $CH_4$-fossil and 79.7 for $CH_4$-non fossil (Forster et al., 2021). Although global anthropogenic
emissions of $CH_4$ are estimated at around 359 Tg $CH_4$ yr$^{-1}$ (Saunois et al., 2020), representing around 2.5% of the global
$CO_2$ anthropogenic emissions when converted to units of carbon mass flux for the recent decade, the emissions-based
effective radiative forcing of  $CH_4$ concentrations has contributed ~31% (1.19 W m$^{-2}$) to the additional radiative forcing
from anthropogenic emissions of greenhouse gases and their precursors (3.84 W m$^{-2}$) over the industrial era (1750-2019)
(Forster et al., 2021). Changes in other chemical compounds such as nitrogen oxides ($NO_x$) or carbon monoxide (CO) also
influence atmospheric $CH_4$ through changes to its atmospheric lifetime. Emissions of $CH_4$ contribute to the production of
ozone, stratospheric water vapour, and $CO_2$, and most importantly affect its own lifetime (Myhre et al., 2013; Shindell et
al., 2012). $CH_4$ has a short lifetime in the atmosphere (about 9 years for the year 2010, Prather et al., 2012). Hence a
stabilisation or reduction of $CH_4$ emissions leads to the stabilisation or reduction of its atmospheric concentration (assuming
no change in the chemical oxidants), and therefore its radiative forcing, in only a few decades. While reducing $CO_2$ emissions
is necessary to stabilise long-term warming, reducing $CH_4$ emissions is recognized as an effective option to limit climate
warming in the near-term (Shindell et al., 2012; Jackson et al., 2020; Ocko et al., 2021; UNEP, 2021), because of its shorter
lifetime compared to $CO_2$.
The momentum around the potential of $CH_4$ to limit near-term warming has led to the launch of the Global Methane
Pledge at the November 2021 Conference of the Parties (COP 26). Signed by 150 countries, this collective effort aims at
reducing global $CH_4$ anthropogenic emissions at least 30 percent from 2020 levels by 2030 (Global Methane Pledge, 2023).
Given that global baseline $CH_4$ emissions are expected to grow through 2030 (by an additional 20-50 Million tons (Mt) of
$CH_4$, UNEP 2022), the $CH_4$ emission reductions currently needed to reach the Global Methane Pledge objective (UNEP,



2022) correspond to 36% of the projected baseline emissions in 2030 (ie. if no further emission reductions were
implemented). This implies that large reductions of $CH_4$ emissions are needed to meet the Global Methane Pledge that is
consistent also with the 1.5-2°C target of the Paris Agreement (UNEP, 2022). Moreover, because $CH_4$ is a precursor of
important air pollutants such as ozone, $CH_4$ emissions reductions are required by two international conventions: the United
Nations Framework Convention on Climate Change (UNFCCC) and the Convention on Long Range Transport of Air
Pollution (CLRTAP), making this global $CH_4$ budget assessment all the more critical.
Changes in the magnitude and temporal variation (annual to inter-annual) of $CH_4$ sources and sinks over the past
decades are characterised by large uncertainties (e.g., Kirschke et al., 2013; Saunois et al., 2017; Turner et al., 2019). Also,
the decadal budget suggests relative uncertainties (hereafter reported as min-max ranges) of 20-35% for inventories of
anthropogenic emissions in specific sectors (e.g., agriculture, waste, fossil fuels (Tibrewal et al., 2024)), 50% for biomass
burning and natural wetland emissions, and up to 100% for other natural sources (e.g., inland waters, geological sources).
The uncertainty in the chemical loss of $CH_4$ by OH, the predominant sink of atmospheric $CH_4$, has been s estimated using
Prather et al. (2012) and Rigby et al. (2017) estimated this uncertainty at ~10% from the uncertainty in the reaction rate
between $CH_4$ and OH, or using methyl-chloroform measurements. Bottom-up approaches (chemistry transport models)
estimate the uncertainty of the chemical loss by OH at around 15-20% (Saunois et al., 2016, 2020). This uncertainty on the
OH induced loss translates, in the top-down methods, into the minimum relative uncertainty associated with global $CH_4$
emissions, as other $CH_4$ sinks (atomic oxygen and chlorine oxidations, soil uptake) are much smaller and the atmospheric
growth rate is well-defined (Dlugokencky et al., 2009). Globally, the contribution of natural $CH_4$ emissions to total emissions
can be quantified by combining lifetime estimates with reconstructed pre-industrial atmospheric $CH_4$ concentrations from
ice cores (assuming natural emissions have not been perturbed during the anthropocene) (e.g., Ehhalt et al., 2001).
Regionally or nationally, uncertainties in emissions may reach 40-60% (e.g., for South America, Africa, China, and India,
see Saunois et al., 2016).
To monitor emission reductions, for example to help conduct the Paris Agreement's stocktake, sustained and long-
term monitoring of anthropogenic emissions per sector is needed in particular for hotspots of emissions that may be missed
in inventories (Bergamaschi et al., 2018a; Pacala, 2010; Lauvaux et al., 2022). At the same time, reducing uncertainties in
all individual $CH_4$ sources, and thus in the overall $CH_4$ budget remains challenging for at least four reasons. First, $CH_4$ is
emitted by multiple processes, including natural and anthropogenic sources, point and diffuse sources, and sources
associated with at least three different production origins (i.e., microbial, thermogenic, and pyrogenic). These multiple
sources and processes require the integration of data from diverse scientific communities and across multiple temporal and
spatial scales. The production of accurate bottom-up estimates is complicated by the fact that anthropogenic emissions result
from leakage from fossil fuel production with large differences between countries depending on technologies and practices,
the fact that many large leak events are sporadic, and the location of many emissions hotspots is not well known, and from
uncertain emission factors used to summarise complex microbial processes in the agriculture and waste sectors. For the





latter, examples include difficulties in upscaling methane emissions from livestock without considering the variety of animal
weight, diet and environment, and difficulties in assessing emissions from landfills depending on waste type and waste
management technology. Second, atmospheric $CH_4$ is removed mainly by chemical reactions in the atmosphere involving
OH and other radicals that have very short lifetimes (typically ~1s). Due to the short lifetime of OH, the spatial and temporal
distributions of OH are highly variable. While OH can be measured locally, calculating global $CH_4$ loss through OH
measurements requires high-resolution global OH measurements (typically half an hour to integrate cloud cover, and 1 km
spatially to consider OH high reactivity and heterogeneity) which is impossible from direct OH observations. As a result,
OH can only be calculated through large scale atmospheric chemistry modelling. Those simulated OH concentrations from
transport-chemistry models prescribed with emissions of precursor species affecting OH still show uncertain spatio-temporal
distribution from regional to global scales (Zhao et al., 2019). Third, only the net $CH_4$ budget (sources minus sinks) is well
constrained by precise observations of atmospheric growth rates (Dlugokencky et al., 2009), leaving the sum of sources and
the sum of sinks uncertain. One distinctive feature of $CH_4$ sources compared to $CO_2$ fluxes is that the oceanic contribution
to the global $CH_4$ budget is small (~1-3%), making CH4 source estimation predominantly a terrestrial endeavour (USEPA,
2010b). Finally, we lack comprehensive observations to constrain 1) the areal extent of different types of wetlands and
inland freshwater (Kleinen et al., 2012, 2020, 2021, 2023; Stocker et al., 2014; Zhang et al., 2021), 2) models of wetland
and inland freshwater emission rates (Melton et al., 2013; Poulter et al., 2017; Wania et al., 2013; Bastviken et al., 2011;
Wik et al., 2016a; Rosentreter et al., 2021; Bansal et al., 2023; Lauerwald et al., 2023a; Stanley et al. 2023), 3) inventories
of anthropogenic emissions (Höglund-Isaksson et al., 2020; Crippa et al., 2023; USEPA, 2019), and 4) atmospheric
inversions, which aim to estimate $CH_4$ emissions from global to regional scales (Houweling et al., 2017; Jacob et al., 2022).
The global $CH_4$ budget inferred from atmospheric observations by atmospheric inversions relies on regional
constraints from atmospheric sampling networks, which are relatively dense for northern mid-latitudes, with various high-
precision and high-accuracy surface stations, but are sparser at tropical latitudes and in the Southern Hemisphere
(Dlugokencky et al., 2011). Recently, the density of atmospheric observations has increased in the tropics due to satellite-
based platforms that provide column-average $CH_4$ mixing ratios. Despite continuous improvements in the precision and
accuracy of space-based measurements (e.g., Buchwitz et al., 2016), systematic errors greater than several ppb on total
column observations can still limit the usage of such data to constrain surface emissions (e.g., Jacob et al., 2022). The
development of robust bias corrections on existing data can help overcome this issue (e.g., Inoue et al., 2016) and satellite
data are now widely used in atmospheric inversions where they provide more global information on the distribution of fluxes
and highly complement the surface networks (e.g., Lu et al., 2021).
In this context, the Global Carbon Project (GCP) seeks to develop a complete picture of the carbon cycle by
establishing common, consistent scientific knowledge to support policy development and actions to mitigate greenhouse gas
emissions to the atmosphere (www.globalcarbonproject.org). The objective of this paper is to analyse and synthesise the
current knowledge of the global $CH_4$ budget, by gathering results of observations and models to better understand and



quantify the main robust features of this budget, its remaining uncertainties, and to make recommendations for improvement.
We combine results from a large ensemble of bottom-up approaches (e.g., process-based models for natural wetlands, data-
driven approaches for other natural sources, inventories of anthropogenic emissions and biomass burning, and atmospheric
chemistry models), and top-down approaches (including $CH_4$ atmospheric observing networks, atmospheric inversions
inferring emissions and sinks from the assimilation of atmospheric observations into models of atmospheric transport and
chemistry). The focus of this work is to update the previous assessment made for the period 2000-2017 (Saunosi et al.,
2020)to the more recent 2000-2020 period. More in-depth analyses of trends and year-to-year changes are left to future
publications. Our current paper is a living review, published at about four-year intervals, to provide an update and new
synthesis of available observational, statistical, and model data for the overall $CH_4$ budget and its individual components.
Kirschke et al. (2013) was the first $CH_4$ budget synthesis followed by Saunois et al. (2016) and Saunois et al.
(2020), with companion papers by Stavert et al. (2021) on regional $CH_4$ budgets and Jackson et al. (2020) focusing on the
last year of the budget (2017). Saunois et al. (2020) covered 2000-2017 and reported $CH_4$ emissions and sinks for three time
periods: 1) the latest calendar decade at that time (2000-2009), 2) data for the latest available decade (2008-2017), and 3)
the latest available year (2017) at the time. Here, the Global Methane Budget (GMB) covers 2000-2020 split into the 2000-
2009 decade, the 2010-2019 decade (where data are available), the year 2020 affected by COVID induced changes in human
activity, and briefly for 2021-2023 as per data availability (Section 6). The $CH_4$ budget is presented at global, latitudinal,
and regional scales and data can be downloaded from   https://doi.org/10.18160/GKQ9-2RHT (Martinez et al., 2024).
Six sections follow this introduction. Section 2 presents the methodology used in the budget: units, definitions of
source categories, regions, data analysis; and discusses the delay between the period of study of the budget and the release
date. Section 3 presents the current knowledge about $CH_4$ sources and sinks based on the ensemble of bottom-up approaches
reported here (models, inventories, data-driven approaches). Section 4 reports atmospheric observations and top-down
atmospheric inversions gathered for this paper. Section 5, based on Sections 3 and 4, provides the updated analysis of the
global $CH_4$ budget by comparing bottom-up and top-down estimates and highlighting differences. Section 6 discusses the
recent changes in atmospheric $CH_4$ in relation with changes in $CH_4$ sources and sinks. Finally, Section 7 discusses future
developments, missing components, and the most critical remaining uncertainties based on our update to the global $CH_4$
budget.
**2 Methodology**
**2.1 Units used**
Unless specified, fluxes are expressed in teragrams of $CH_4$ per year (1 Tg $CH_4$ $yr^{-1}$ = $10^{12}$ g $CH_4$ $yr^{-1}$), while atmospheric
mixing ratios are expressed as dry air mole fractions, in parts per billion (ppb), with atmospheric $CH_4$ annual increases,
$G_{ATM}$, expressed in ppb $yr^{-1}$. In the tables, we present mean values and ranges for the two decades 2000-2009 and 2010-



2019, together with results for the most recent available year (2020). Results obtained from previous syntheses (i.e., Saunois
et al., 2020 and Saunois et al., 2016) are also given for the decade 2000-2009. Following Saunois et al. (2016) and
considering that the number of studies is often relatively small for many individual source and sink estimates, uncertainties
are reported as minimum and maximum values of the available studies, given in brackets. In doing so, we acknowledge that
we do not consider the uncertainty of the individual estimates, and we express uncertainty as the range of available mean
estimates, i.e., differences across measurements/methodologies considered. These minimum and maximum values are those
presented in Section 2.5 and exclude identified outliers.
The $CH_4$ emission estimates are provided with up to three significant digits, for consistency across all budget flux
components and to ensure the accuracy of aggregated fluxes. Nonetheless, given the values of the uncertainties in the $CH_4$
budget, we encourage the reader to consider not more than two digits as significant for the global total budget.

## 2.2 Period of the budget and availability of data

The bottom-up estimates rely on global anthropogenic emission inventories, an ensemble of process-based models for
wetlands emissions, and published estimates in the literature for other natural sources. The global gridded anthropogenic
inventories (see Section (3.1.1) are updated irregularly, generally every 3 to 5 years. The last reported years of available
inventories were 2018 or 2019 when we started the top-down modelling activity. In order to cover the period 2000-2020, it
was necessary to extrapolate the anthropogenic inventory EDGARv6 (Crippa et al., 2021) to 2020 to use it as prior
information for the anthropogenic emissions in the atmospheric inversion systems as explained in the supplementary
material. The land surface (wetland) models were run over the full period 2000-2020 using dynamical wetland areas, derived
by remote sensing data or other models of flooded area variability (Sect. 3.2.1).
The atmospheric inversions run until mid-2021, but the last year of reported inversion results is 2020, which represents a
three-year lag with the present. This is due to the long time period it takes to acquire atmospheric in-situ data and integrate
models. Even though satellite observations are processed operationally and are generally available with a latency of days to
weeks, by contrast surface observations can lag from months to years because of the time for flask analyses and data quality
checks in (mostly) non-operational chains. In addition, the final six months of inversions must be generally ignored because
the estimated fluxes are not constrained by as many observations as the previous periods. Lastly, this budget presents an
extended synthesis of the most recent development regarding inland water emissions (Sect. 3.2.2) and corrections associated
with double counting with wetlands.

## 2.3 Definition of regions

Geographically, emissions are reported globally and for three latitudinal bands (90°S-30°N, 30-60°N, 60-90°N, only for
gridded products). When extrapolating emission estimates forward in time (see Sect. 3.1.1), and for the regional budget
presented by Stavert et al. (2021), a set of 19 regions (oceans and 18 continental regions, see supplementary Fig. S3) were



used. As anthropogenic emissions are often reported by country, we define these regions based on a country list (Table S1).
This approach was compatible with all top-down and bottom-up approaches considered. The number of regions was chosen
to be close to the widely used TransCom inter-comparison map (Gurney et al., 2004) but with subdivisions to separate the
contribution from important countries or regions for the $CH_4$ cycle (China, South Asia, Tropical America, Tropical Africa,
United States of America, and Russia). The resulting region definition is the same as that used for the Global Carbon Project
(GCP) $N_2O$ budget (Tian et al., 2020). Compared to Saunois et al. (2020), the Oceania region has been replaced by
Australasia including only Australia and New Zealand. Other territories formerly in Oceania were included in Southeast
Asia.
**2.4 Definition of source and sink categories**
$CH_4$ is emitted by different processes (i.e., biogenic, thermogenic, or pyrogenic) and can be of anthropogenic or natural
origin. Biogenic $CH_4$ is the final product of the decomposition of organic matter by methanogenic *Archaea* in anaerobic
environments, such as water-saturated soils, swamps, rice paddies, marine and freshwater sediments, landfills, sewage and
wastewater treatment facilities, or inside animal digestive systems. Thermogenic methane is formed on geological time
scales by the breakdown of buried organic matter due to heat and pressure deep in the Earth's crust. Thermogenic $CH_4$
reaches the atmosphere through marine and land geological gas seeps. These $CH_4$ emissions are increased by human
activities, for instance, the exploitation and distribution of fossil fuels. Pyrogenic $CH_4$ is produced by the incomplete
combustion of biomass and other organic materials. Peat fires, biomass burning in deforested or degraded areas, wildfires,
and biofuel burning are the largest sources of pyrogenic $CH_4$. $CH_4$ hydrates, ice-like cages of frozen $CH_4$ found in continental
shelves and slopes and below sub-sea and land permafrost, can be of either biogenic or thermogenic origin. Each of these
three process categories has both anthropogenic and natural components.
In the following, we present the different $CH_4$ sources depending on their anthropogenic or natural origin, which is relevant
to climate policy. Compared to the previous budgets, marginal changes have been made regarding source categories (naming
and grouping), to reflect the improved estimates for inland water sources and their indirect anthropogenic component. In the
previous Global Methane Budgets (Saunois et al., 2016, 2020), natural and anthropogenic emissions were split in a way that
did not correspond exactly to the definition used by the UNFCCC following the IPCC guidelines (IPCC, 2006), where, for
pragmatic reasons, all emissions from managed land are typically reported as anthropogenic. For instance, we considered
all wetlands as natural emissions, despite some wetlands being on managed land and their emissions being partly reported
as anthropogenic in UNFCCC national communications. The human induced perturbation of climate, atmospheric $CO_2$, and
nitrogen and sulfur deposition may cause changes in wetland sources we classified as natural. Following our previous
definition, emissions from wetlands, inland freshwaters, thawing permafrost, or geological leaks are accountable for
"natural" emissions, even though we acknowledge that climate change and other human perturbations (e.g., eutrophication)
may cause changes in those emissions. $CH_4$ emissions from reservoirs were also considered as natural even though reservoirs



are human-made. Indeed, since the 2019 refinement to the IPCC guidelines (IPCC, 2019) emissions from reservoirs and
other flooded lands are considered to be anthropogenic by the UNFCCC and should be reported as such. However these
estimates are not provided by inventories and not systematically reported by all countries (especially non Annex-I countries).
In this budget we rename "natural sources" to "natural and indirect anthropogenic sources" to acknowledge that $CH_4$
emissions from reservoirs, as well as from water bodies that were perturbed by agricultural activities (drainage,
eutrophication, land use change) are indirect anthropogenic emissions. As a result, here, "natural and indirect anthropogenic
sources" refer to "emissions that do not directly originate from fossil, agricultural, waste, and biomass burning sources"
even if they are perturbed by anthropogenic activities and climate change. Natural and indirect anthropogenic emissions are
split between "Wetlands and Inland Freshwaters" and "Other natural" emissions (e.g., wild animals, termites, land
geological sources, oceanic geological and biogenic sources, and terrestrial permafrost). "Anthropogenic direct sources" are
caused by direct human activities since pre-industrial/pre-agricultural time (3000-2000 BC, Nakazawa et al., 1993) including
agriculture, waste management, fossil fuel-related activities and biofuel and biomass burning (yet we acknowledge that a
small fraction of wildfires are naturally ignited). Direct anthropogenic emissions are split between: "Agriculture and waste
emissions", "Fossil fuel emissions", and "Biomass and biofuel burning emissions", assuming that all types of fires are caused
by anthropogenic activities. To conclude, this budget reports "direct anthropogenic", and "natural and indirect
anthropogenic" methane emissions for the five main source categories explained above for both bottom-up and top-down
approaches.
The sinks of methane are split into the soil uptake that can be derived from land-surface models in the bottom-up budget,
and the chemical sinks. The chemical sinks are estimated by either chemistry climate or chemistry transport models in the
bottom-up budget, and are further detailed in terms of vertical distribution (troposphere and stratosphere) and oxidants.
Bottom-up estimates of $CH_4$ emissions for some processes are derived from process-oriented models (e.g., biogeochemical
models for wetlands, models for termites), inventory models (agriculture and waste emissions, fossil fuel emissions, biomass
and biofuel burning emissions), satellite-based models (large scale biomass burning), or observation-based upscaling models
for other sources (e.g., inland water, geological sources). From these bottom-up approaches, it is possible to provide
estimates for more detailed source subcategories inside each main category described above (see budget in Table 3).
However, the total $CH_4$ emission derived from the sum of independent bottom-up estimates remains unconstrained.
For atmospheric inversions (top-down approach), atmospheric methane concentration observations provide a constraint on
the global methane total source if we assume the global sink is known (OH and other oxidant prescribed), or inversions are
optimising also for the chemical sink. OH estimates are constrained by methyl chloroform-inversion (Montzka et al., 2011;
Rigby et al., 2017; Patra et al., 2021). The inversions reported in this work solve for the total net $CH_4$ flux at the surface
(sum of sources minus soil uptake) (e.g., Pison et al., 2013), or a limited number of source categories (e.g., Bergamaschi et
al., 2013). In most of the inverse systems the atmospheric oxidant concentrations were prescribed with pre-optimized or
scaled OH fields, and thus the atmospheric sink is not optimised. The assimilation of $CH_4$ observations alone, as reported in





this synthesis, can help to separate sources with different locations or temporal variations but cannot fully separate individual
sources where they overlap in space and time in some regions. Top-down global and regional $CH_4$ emissions per source
category were nevertheless obtained from gridded optimised fluxes, for the inversions that separated emissions into the five
main GCP categories. Alternatively, for the inversion that only solved for total emissions (or for other categories other than
the five described above), the prior contribution of each source category at the spatial resolution of the inversion was scaled
by the ratio of the total (or embedding category) optimised flux divided by the total (or embedding category) prior flux
(Kirschke et al., 2013). In other words, the prior relative mix of sources at model resolution is kept in each grid cell while
total emissions are given by the atmospheric inversions. The soil uptake was provided separately to report total gross surface
emissions instead of net fluxes (sources minus soil uptake).
In summary, bottom-up models and inventories emissions are presented for all relevant source processes and grouped if
needed into the five main categories defined above. Top-down inversion emissions are reported globally and for the five
main emission categories.

**2.5 Processing of emission maps and box-plot representation of emission budgets**

Common data analysis procedures have been applied to the different bottom-up models, inventories and atmospheric
inversions whenever gridded products exist. Gridded emissions from atmospheric inversions, land-surface models for
wetland or biomass burning were provided at the monthly scale. Emissions from anthropogenic inventories are usually
available as yearly estimates. These monthly or yearly fluxes were provided on a 1°x1° grid or re-gridded to 1°x1°, then
converted into units of Tg $CH_4$ per grid cell. Inversions with a resolution coarser than 1° were downscaled to 1° by each
modelling group. Land fluxes in coastal pixels were reallocated to the neighbouring land pixel according to our 1° land-sea
mask, and vice-versa for ocean fluxes. Annual and decadal means used for this study were computed from the monthly or
yearly gridded 1°x1° maps.
Budgets are presented as boxplots with quartiles (25%, median, 75%), outliers, and minimum and maximum values without
outliers. Outliers were determined as values below the first quartile minus three times the interquartile range, or values above
the third quartile plus three times the interquartile range. Mean values reported in the tables are represented as "+" symbols
in the corresponding figures.

**3 Methane sources and sinks: bottom-up estimates**

For each source category, a short description of the relevant processes, original data sets (measurements, models) and related
methodology are given. More detailed information can be found in original publication references, in Annex A2 where the
sources of data used to estimate the different sources and sinks are summarised and compared with those used in Saunois et
al. (2020) and in the Supplementary Material of this study when specified in the text. The emission estimates for each source



category are compared with Saunois et al. (2020) in Table 3 and with Saunois et al. (2016) in Table S12 for the decade 2000-
2009.

### 3.1 Anthropogenic direct sources

### 3.1.1 Global inventories

The main bottom-up global inventory datasets covering direct anthropogenic emissions from all sectors (Table 1) are from
the United States Environmental Protection Agency (USEPA, 2019), the Greenhouse gas and Air pollutant Interactions and
Synergies (GAINS) model developed by the International Institute for Applied Systems Analysis (IIASA) (Höglund-
Isaksson et al., 2020) and the Emissions Database for Global Atmospheric Research (EDGARv6 and v7, Crippa et al., 2021,
2023) compiled by the European Commission Joint Research Centre (EC-JRC) and Netherlands Environmental Assessment
Agency (PBL). We also used the Community Emissions Data System for historical emissions (CEDS) (Hoesly et al., 2018)
developed for climate modelling and the Food and Agriculture Organization (FAO) FAOSTAT emission database (Tubiello
et al., 2022), which covers emissions from agriculture and land use (including peatland fires and biomass fires). These
inventories are not independent as they may use the same activity data or emission factors, as discussed below.
These inventory datasets report emissions from fossil fuel production, transmission, and distribution; livestock enteric
fermentation; manure management and application; rice cultivation; solid waste and wastewater. Since the level of detail
provided by country and by sector varies among inventories, the data were reconciled into common categories according to
Table S2. For example, agricultural waste-burning emissions treated as a separate category in EDGAR, GAINS and FAO,
are included in the biofuel sector in the USEPA inventory and in the agricultural sector in CEDS. The GAINS, EDGAR and
FAO estimates of agricultural waste burning were excluded from this analysis (these amounted to 1-3 Tg $CH_4$ yr$^{-1}$ in recent
decades) to prevent any potential overlap with separate estimates of biomass burning emissions (e.g., GFEDv4.1s; Giglio et
al. (2013); van der Werf et al (2017)). In the inventories used here, emissions for a given region/country and a given sector
are usually calculated following IPCC methodology (IPCC, 2006), as the product of an activity factor and its associated
emission factor. An abatement coefficient may also be used, to account for any regulations implemented to control emissions
(see e.g., Höglund-Isaksson et al., 2015). These datasets differ in their assumptions and data used for the calculation;
however, they are not completely independent because they often use the same activity data and some of them follow the
same IPCC guidelines (IPCC, 2006). While the USEPA inventory adopts emissions reported by the countries to the
UNFCCC, other inventories (FAOSTAT, EDGAR and the GAINS model) produce their own estimates using a consistent
approach for all countries, typically IPCC Tier 1 methods or deriving IPCC Tier 2 emission factors from country-specific
information using a consistent methodology. These other inventories compile country-specific activity data and emission
factor information or, if not available, adopt IPCC default factors (Tibrewal et al., 2024; Oreggioni et al., 2021; Höglund-
Isaksson et al., 2020; Tubiello, 2019). CEDS takes a different approach (Hoesly et al., 2018) and combines data from
GAINS, EDGAR and FAO depending on the sector. Then their first estimates are scaled to match other individual or region-



specific inventory values when available. This process maintains the spatial information in the default emission inventories
while preserving consistency with country level data. The FAOSTAT dataset (hereafter FAO-CH$_4$) provides estimates at
the country level and is limited to agriculture (CH$_4$ emissions from enteric fermentation, manure management, rice
cultivation, energy usage, burning of crop residues, and prescribed burning of savannahs) and land-use (peatland fires and
biomass burning). FAO-CH$_4$ uses activity data mainly from the FAOSTAT crop and livestock production database, as
reported by countries to FAO (Tubiello et al., 2013), and applies mostly the Tier 1 IPCC methodology for emissions factors
(IPCC, 2006), which depends on geographic location and development status of the country. For manure, the country-scale
temperature was obtained from the FAO global agro-ecological zone database (GAEZv3.0, 2012). Although country
emissions are reported annually to the UNFCCC by annex I countries, and episodically by non-annex I countries, data gaps
of those national inventories do not allow the inclusion of these estimates in this analysis.
In this budget, we use the following versions of these inventories that were available at the start and during the analysis (see
Table 1):

- ● EDGARv6 which provides yearly gridded emissions by sectors from 1970 to 2018 (Crippa et al., 2021; Oreggioni
- et al., 2021; EDGARv6 website https://edgar.jrc.ec.europa.eu/dataset_ghg60; Monforti Ferrario et al., 2021),

- ● EDGARv7, which provides yearly gridded emissions by sectors from 1970 to 2020 (monthly for some sectors),
- but emissions from fossil fuel energy  are not separated (oil and gas, and coal are lumped together - see Table S2)
- (EDGARv7 website https://edgar.jrc.ec.europa.eu/dataset_ghg70; Crippa et al., 2023).

- ● GAINS model scenario version 4.0 (Höglund-Isaksson et al., 2020) which provides an annual sectorial gridded
- product from 1990 to 2020 both by country and gridded. USEPA (USEPA, 2019), which provides 5-year sectorial
- totals by country from 1990 to 2020 (estimates from 2015 onward are a projection), with no gridded distribution
- available. The USEPA dataset was linearly interpolated to provide yearly values from 1990-2020.

- ● CEDS version v_2021_04_21 which provides gridded monthly and annual country-based emissions by sectors
- from 1970 to 2019 (Hoesly et al., 2018; O'Rourke et al., 2021). Fossil fuel emissions for 2020 have been updated
- using the methodology described for CO in Zheng et al. (2023).

- ● FAO-CH$_4$ (database accessed in December 2022, FAO, 2022) containing annual country level data for the period
- 1961-2020, for rice, manure, and enteric fermentation; and 1990-2020 for burning savannah, crop residue and non-
- agricultural biomass burning.

**3.1.2 Total anthropogenic direct emissions**
We calculated separately the total anthropogenic emissions for each inventory by adding its values for "Agriculture and
waste", "Fossil fuels" and "Biofuels" with additional large-scale biomass burning emissions data (Sect. 3.1.5). This method
avoids double counting and ensures consistency within each inventory. This approach was used for the EDGARv6 and v7,
CEDS and GAINS inventories, but we kept the USEPA inventory as originally reported because it includes its own estimates





of biomass burning emissions. FAO-CH4 was only included in the range reported for the "Agriculture and waste" category.
For the latter, we calculated the range and mean value as the sum of the mean and range of the three anthropogenic
subcategory estimates "Enteric fermentation and Manure", "Rice", and "Landfills and Waste". The values reported for the
upper-level anthropogenic categories ("Agriculture and waste", "Fossil fuels" and "Biomass burning & biofuels") are
therefore consistent with the sum of their subcategories, although there might be small percentage differences between the
reported total anthropogenic emissions and the sum of the three upper-level categories. This approach provides a more
accurate representation of the range of emission estimates, avoiding an artificial expansion of the uncertainty attributable to
subtle differences in the definition of sub-sector categorisations between inventories.
Based on the ensemble of databases detailed above, total direct anthropogenic emissions were 358 [329-387] Tg $CH_4$ yr$^{-1}$
for the decade 2010-2019 (Table 3, including biomass and biofuel burning) and 331 [305-365] Tg $CH_4$ yr$^{-1}$ for the decade
2000-2009. Our estimate for the 2000-2009 decade is within the range of Saunois et al. (2020) (334 [321-358]), Saunois et
al. (2016) (338 Tg CH4 yr-1 [329-342]) and Kirschke et al. (2013) (331 Tg $CH_4$ yr$^{-1}$ [304-368]) for the same period. The
slightly larger range reported herein with respect to previous estimates is due to the USEPA lower estimate for agriculture,
waste and fossil emissions associated with the lowest estimate of biomass burning.
Figure 2 (left) summarises or projects global $CH_4$ emissions of anthropogenic sources (including biomass and biofuel
burning) by different datasets between 2000 and 2050. The datasets consistently estimate total anthropogenic emissions of
~300 Tg $CH_4$ yr$^{-1}$ in 2000. For the Sixth Assessment Report of the IPCC, seven main Shared Socioeconomic Pathways
(SSPs) were defined for future climate projections in the Coupled Model Intercomparison Project 6 (CMIP6) (Gidden et al.,
2019; O'Neill et al., 2016) ranging from 1.9 to 8.5 W m$^{-2}$ radiative forcing by the year 2100 (as shown by the number in the
SSP names). For the 1970-2015 period, historical emissions used in CMIP6 (Feng et al., 2019) combine anthropogenic
emissions from CEDS (Hoesly et al., 2018) and a climatological value from the GFEDv4.1s biomass burning inventory (van
Marle et al., 2017). The harmonised scenarios used for CMIP6 activities start in 2015 at 388 Tg $CH_4$ yr$^{-1}$, which corresponds
to the higher range of our estimates. Since $CH_4$ emissions continue to track scenarios that assume no or minimal climate
policies (SSP5 and SSP3), it may indicate that climate policies, when present, have not yet produced sufficient results to
change the emissions trajectory substantially (Nisbet et al., 2019). After 2015, the SSPs span a range of possible outcomes,
but current emissions appear likely to follow the higher-emission trajectories over the next decade in terms of trend, and the
peak year has not yet been reached. This illustrates the challenge of methane mitigation that lies ahead to help reach the
goals of the Paris Agreement. In addition, estimates of methane atmospheric concentrations (Meinshausen et al., 2017, 2020)
from the harmonised scenarios (Riahi et al., 2017) indicate that observations of global $CH_4$ concentrations fall well within
the range of scenarios (Fig. 2 right). The $CH_4$ concentrations are estimated using a simple exponential decay with inferred
natural emissions (Meinshausen et al., 2011), and the emergence of any trend between observations and scenarios needs to
be confirmed in the following years. However, the current observed concentrations and emissions estimates lie in the upper
range of the former RCPs scenarios starting in 2005 (Fig. S1). In the future, it will be important to monitor the trends from



2015 (the Paris Agreement) and from 2020 (Global Methane Pledge) estimated in inventories and from atmospheric
observations, and compare them to various scenarios.

### 3.1.3 Fossil fuel production and use

Most anthropogenic $CH_4$ emissions related to fossil fuels come from the exploitation, transportation, and usage of coal, oil,
and natural gas. Additional emissions reported in this category include small industrial contributions such as the production
of chemicals and metals, fossil fuel fires (e.g., underground coal mine fires and the Kuwait oil and gas fires), and transport
(road and non-road transport). $CH_4$ emissions from the oil processing industry (e.g., refining) and production of charcoal
are estimated to be a few Tg $CH_4$ $yr^{-1}$ only     and are included in the transformation industry sector in the inventory. Fossil
fuel fires are included in the subcategory "Oil & Gas". Emissions from industries, road and, non-road transport are reported
apart from the two main subcategories "Oil & Gas" and "Coal", as in Saunois et al. (2020) and contrary to Saunois et al.
(2016); each of these amounts to about 2 to 5 Tg $CH_4$ $yr^{-1}$ (Table 3). The large range (1-9 Tg $CH_4$ $yr^{-1}$) is attributable to
difficulties in allocating some sectors to these sub-sectors consistently among the different inventories (See Table S2). The
spatial distribution of $CH_4$ emissions from fossil fuels is presented in Fig. 3 based on the mean gridded maps provided by
CEDS, EDGARv6, and GAINS for the 2010-2019 decade; USEPA lacks a gridded product.
Global mean emissions from fossil fuel-related activities, other industries and transport are estimated from the four global
inventories (Table 1) to be of 120 [117-125] Tg $CH_4$ $yr^{-1}$ for the 2010-2019 decade (Table 3), but with large differences in
the rate of change during this period across inventories. The sector accounts on average for 34% (range 31-42%) of total
global anthropogenic emissions.

**Coal mining.**

During mining, $CH_4$ is emitted primarily from ventilation shafts, where large volumes of air are pumped in and out of the
mine to keep the $CH_4$ mixing ratio below 0.5% to avoid accidental ignition, and from dewatering operations. In countries of
the Organization for Economic Co-operation and Development (OECD), coalbed $CH_4$ is often extracted as fuel up to ten
years before the coal mine starts operation, thereby reducing the $CH_4$ channelled through ventilation shafts during mining.
In many countries, large quantities of ventilation air $CH_4$ are still released to the atmosphere or flared, despite efforts to
extend coal mine gas recovery under the UNFCCC Clean Development Mechanisms (http://cdm.unfccc.int). $CH_4$ leaks also
occur during post-mining handling, processing, and transportation. Some $CH_4$ is released from coal waste piles and
abandoned mines; while emissions from these sources were believed to be low (IPCC, 2000), recent work has estimated
these at 22 billion $m^3$ (compared to 103 billion $m^3$ from functioning coal mines) in 2010 with emissions projected to increase
into the future (Kholod et al., 2020).
In 2020, more than 35% (IEA, 2023a) of the world's electricity is still produced from coal. This contribution grew in the
2000s at the rate of several percent per year, driven by Asian economic growth where large reserves exist, but global coal



consumption declined between 2014 and 2020. In 2020, the top ten largest coal producing nations accounted for ~90% of
total world $CH_4$ emissions from coal mining; among them, the top three producers (China, United States of America, and
India) produced almost two-thirds (66%) of the world's coal (IEA, 2021).
Global estimates of $CH_4$ emissions from coal mining show a reduced range of 37-44 Tg $CH_4$ yr$^{-1}$ for 2010-2019, compared
to the previous estimate for 2008-2017 in Saunois et al. (2020) reporting a range of 29-61 Tg $CH_4$ yr$^{-1}$ for 2008-2017. This
reduced range probably results from using similar activity data (mostly from IEA statistics) in the different inventories. The
highest value of the range in Saunois et al. (2020) came from the CEDS inventory while the lowest came from USEPA.
CEDS seems to have revised downward their estimate compared to the previous version used in Saunois et al. (2020). There
were previously large discrepancies in Chinese coal emissions, with a large overestimation from EDGARv4.2 on which
CEDS was based. As highlighted by Liu et al. (2021a), a county-based inventory of Chinese methane emissions also
confirms the overestimation of previous EDGAR inventories and estimated total anthropogenic Chinese emissions at
38.2±5.5 Tg $CH_4$ yr$^{-1}$ for 2000-2008 (Liu et al., 2021a). Coal mining emission factors depend strongly on the type of coal
extraction (underground mining emits up to 10 times more than surface mining), the geological underground structure
(region-specific), history (basin uplift), and the quality of the coal (brown coal (lignite) emits more than hard coal
(anthracite)). Finally, the different emission factors derived for coal mining is the main reason  for the differences between
inventories globally (Fig. 2).
For the 2010-2019 decade, methane emissions from coal mining represent 33% of total fossil fuel-related emissions of $CH_4$
(40 [37-44] Tg $CH_4$ yr$^{-1}$). An additional very small source corresponds to fossil fuel fires (mostly underground coal fires,
~0.15 Tg yr$^{-1}$ any year in EDGARv7).

**Oil and natural gas systems.**

This sub-category includes emissions from both conventional and shale oil and gas exploitation. Natural gas is composed
primarily of $CH_4$, so both fugitive and planned emissions during the drilling of wells in gas fields, extraction, transportation,
storage, gas distribution, end use, and incomplete combustion in gas flares emit $CH_4$ (Lamb et al., 2015; Shorter et al., 1996).
Persistent fugitive emissions (e.g., due to leaky valves and compressors) should be distinguished from intermittent emissions
due to maintenance (e.g., purging and draining of pipes) or incidents. During transportation, fugitive emissions can occur in
oil tankers, fuel trucks and gas transmission pipelines, attributable to corrosion, manufacturing, and welding faults.
According to Lelieveld et al. (2005), $CH_4$ fugitive emissions from gas pipelines should be relatively low, however, old
distribution networks in some cities may have higher rates, especially those with cast-iron and unprotected steel pipelines
(Phillips et al., 2013). Measurement campaigns in cities within the USA (e.g., McKain et al., 2015) and Europe (e.g.,
Defratyka et al., 2021) revealed that significant emissions occur in specific locations (e.g., storage facilities, city natural gas
fueling stations, well and pipeline pressurisation/depressurisation points, sewage systems, and furnaces of buildings) along
the distribution networks (e.g., Jackson et al., 2014a; McKain et al., 2015; Wunch et al., 2016). However, $CH_4$ emissions





vary significantly from one city to another depending, in part, on the age of city infrastructure and the quality of its
maintenance, making urban emissions difficult to scale-up from measurement campaigns, although attempts have been made
(e.g., Defratyka et al., 2021). In many facilities, such as gas and oil fields, refineries, and offshore platforms, most of the
associated and other waste gas generated will be flared for security reasons with almost complete conversion to $CO_2$,
however, due to the large quantities of waste gas generated, small fractions of gas still being vented make up relatively large
quantities of methane. These two processes are usually considered together in inventories of oil and gas industries. In
addition, single-point failure of natural gas infrastructure can leak $CH_4$ at high rate for months, such as at the Aliso Canyon
blowout in the Los Angeles, CA (Conley et al., 2016) or the shale gas well blowout in Ohio (Pandey et al., 2019), thus
hampering emission control strategies. Production of natural gas from the exploitation of hitherto unproductive rock
formations, especially shale, began in the 1970s in the US on an experimental or small-scale basis, and then, from the early
2000s, exploitation started at a large commercial scale. The shale gas contribution to total dry natural gas production in the
United States reached 82% in 2023, growing rapidly from 48% in 2013 (IEA, 2023b). The possibly larger emission factors
from shale gas compared to conventional gas, have been widely debated (e.g., Cathles et al., 2012; Howarth, 2019; Lewan,
2020). However, the latest studies tend to infer similar emission factors in a narrow range of 1-3% (Alvarez et al., 2018;
Peischl et al., 2015; Zavala-Araiza et al., 2015), different from the widely spread rates of 3-17% from previous studies (e.g.,
Caulton et al., 2014; Schneising et al., 2014).
$CH_4$ emissions from oil and natural gas systems vary greatly in different global inventories (67 to 80 Tg yr$^{-1}$ in 2020, Table
3). The inventories generally rely on the same sources and magnitudes for activity data, with the derived differences
therefore resulting primarily from different methodologies and parameters used, including emission factors. Those factors
are country- or even site-specific and the few field measurements available often combine oil and gas activities (Brandt et
al., 2014), resulting in high uncertainty in emission estimates for many major oil and gas producing countries. Depending
on the region, the IPCC 2006 default emission factors may vary by two orders of magnitude for oil production and one order
for gas production. For instance, the GAINSv4.0 estimate of $CH_4$ emissions from US oil and gas systems in 2015   is 16
Tg, which is almost twice as high as EDGARv8.0 at 8.4 Tg and USEPA (UNFCCC, 2023) at 9.5 Tg. The difference can
partly be explained by GAINS using a bottom-up methodology to derive country- and year-specific flows of associated
petroleum gas and attributing these to recovery/reinjection, flaring or venting (Höglund-Isaksson, 2017), and partly to
GAINS using a higher emission factor for unconventional gas production (Höglund-Isaksson et al., 2020). Recent
quantifications using satellite observations and inversion estimate a relatively stable trend for US oil and gas systems
emissions since 2010, with Lu et al. (2023) estimating 14.6 Tg for 2010, 15.9 Tg for 2014 and 15.6 Tg for 2019, Shen et al.
(2022) estimating a mean of 12.6 Tg for 2018-2020, and Maasakkers et al (2021) a mean of 11.1 Tg for 2010 to 2015. The
stable top-down trend for the US appears not well captured in the bottom-up inventories from GAINS and EDGAR, which
tend to show an increasing trend driven by increase in production volumes.



Most studies (Alvarez et al., 2018; Brandt et al., 2014; Jackson et al., 2014b; Karion et al., 2013; Moore et al., 2014; Olivier
and Janssens-Maenhout, 2014; Pétron et al., 2014; Zavala-Araiza et al., 2015), albeit not all (Allen et al., 2013; Cathles et
al., 2012; Peischl et al., 2015), suggest that the methane emissions from oil and gas industry are underestimated by
inventories, industries, and agencies, including the USEPA. Lauvaux et al. (2022) showed that emissions from a few high-
emitting facilities, i.e., super-emitters (> 20 t hr$^{-1}$), which are usually sporadic in nature, and not accounted for in the
inventories, could represent 8-12% of global oil & gas emissions, or around 8 Tg $CH_4$ yr$^{-1}$. These high emitting points,
located on the conventional part of the facilities, could be avoided through better operating conditions and repair of
malfunctions. Over the last decade, absolute $CH_4$ emissions almost certainly increased, since USA crude oil production
doubled and natural gas production rose by about 50% (IEA, 2023a). However, global implications of the rapidly growing
shale gas activity in the US remain to be determined precisely.
For the 2010-2019 decade, $CH_4$ emissions from upstream and downstream oil and natural gas sectors are estimated to
represent about 56% of total fossil $CH_4$ emissions (67 [57-74] Tg $CH_4$ yr$^{-1}$, Table 3) based on global inventories, with a
lower uncertainty range than for coal emissions for most countries. However, it is worth noting that 8 Tg $CH_4$ yr$^{-1}$ should
be added on top of this estimate to acknowledge the ultra-emitters contribution, as done in Tibrewal et al (2024).
**3.1.4 Agriculture and waste sector**
This main category includes $CH_4$ emissions related to livestock production (i.e., enteric fermentation in ruminant animals
and manure management), rice cultivation, landfills, and wastewater handling. Of these activities, globally and in most
countries, livestock is by far the largest source of $CH_4$, followed by waste handling and rice cultivation. Conversely, field
burning of agricultural residues is a minor source of $CH_4$ reported in emission inventories (a few Tg at the global scale).
The spatial distribution of $CH_4$ emissions from agriculture and waste handling is presented in Fig. 3 based on the mean
gridded maps provided by CEDS, EDGARv6 and GAINS over the 2010-2019 decade.
Global emissions from agriculture and waste for the period 2010-2019 are estimated to be 211 [195-231] Tg $CH_4$ yr$^{-1}$ (Table
3), representing 60% of total direct anthropogenic emissions. Agriculture emissions amount to 144 Tg $CH_4$ yr$^{-1}$, 40% of the
direct anthropogenic emissions, with the rest coming from the fossil fuel sector (34%), waste (19%) and biomass (5%) and
biofuel (3%) burning .
**Livestock: Enteric fermentation and manure management.** Domestic ruminants such as cattle, buffalo, sheep, goats, and
camels emit $CH_4$ as a by-product of the anaerobic microbial activity in their digestive systems (Johnson et al., 2002). The
very stable temperatures (about 39°C) and pH (6.5-6.8) within the rumen of domestic ruminants, along with a constant plant
matter flow from grazing (cattle graze many hours per day), allow methanogenic *Archaea* residing within the rumen to
produce $CH_4$. $CH_4$ is released from the rumen mainly through the mouth of multi-stomached ruminants (eructation, ~90%
of emissions) or absorbed in the blood system. The $CH_4$ produced in the intestines and partially transmitted through the
rectum is only ~10%.



The total number of livestock continues to grow steadily. There are currently (2020) about 1.5 billion cattle globally, almost
1.3 billion sheep, and nearly as many goats (http://www.fao.org/faostat/en/#data/GE). Livestock numbers are linearly related
to $CH_4$ emissions in inventories using the Tier 1 IPCC approach such as FAOSTAT. In practice, some non-linearity may
arise due to dependencies of emissions on the total weight of the animals and their diet, which are better captured by Tier 2
and higher approaches. Cattle, due to their large population, large individual size, and particular digestive characteristics,
account for the majority of enteric fermentation $CH_4$ emissions from livestock worldwide (Tubiello, 2019; FAO, 2022),
particularly in intensive agricultural systems in wealthier and emerging economies, including the United States (USEPA,
2016). $CH_4$ emissions from enteric fermentation also vary from one country to another as cattle may experience diverse
living conditions that vary spatially and temporally, especially in the tropics (Chang et al., 2019).
Anaerobic conditions often characterise manure decomposition in a variety of manure management systems globally (e.g.,
liquid/slurry treated in lagoons, ponds, tanks, or pits), with the volatile solids in manure producing $CH_4$. In contrast, when
manure is handled as a solid (e.g., in stacks or dry-lots) or deposited on pasture, range, or paddock lands, it tends to
decompose aerobically and to produce little or no $CH_4$. However aerobic decomposition of manure tends to produce nitrous
oxide ($N_2O$), which has a larger global warming impact than $CH_4$. Ambient temperature, moisture, energy contents of the
feed, manure composition, and manure storage or residency time affect the amount of $CH_4$ produced. Despite these
complexities, most global datasets used herein apply a simplified IPCC Tier 1 approach, where amounts of manure treated
depend on animal numbers and simplified climatic conditions by country.
Global $CH_4$ emissions from enteric fermentation and manure management are estimated in the range of 114-124 Tg $CH_4$ yr
$^{-1}$, for the year 2020, in the GAINS model and CEDS, USEPA, FAO-$CH_4$ and EDGARv7 inventories. Using the Tier 2
method adopted from the 2019 Refinement to 2006 IPCC guidelines, a recent study (Zhang et al., 2022) estimated that
global $CH_4$ emissions from livestock increased from 31.8 [26.5–37.1] (mean [minimum−maximum of 95% confidence
interval) Tg $CH_4$ yr$^{-1}$ in 1890 to 131.7 [109.6–153.7] Tg $CH_4$ yr$^{-1}$ in 2019, a fourfold increase in the past 130 years. Chang
et al. (2021) estimates enteric fermentation and manure management emissions based on mixed Tier 1&2 and Tier1
approaches and calculate livestock emissions being 120±13 and 136±15 Tg $CH_4$ yr$^{-1}$ respectively for 2018. Chang et al.
(2021) and Zhang et al. (2022) estimates for 2018 or 2019 are on average a bit higher than the inventories estimates but in
agreement considering the uncertainties.
For the period 2010-2019, we estimated total emissions of 112 [107-118] Tg $CH_4$ yr$^{-1}$ for enteric fermentation and manure
management, about one third of total global anthropogenic emissions.
**Rice cultivation.** Most of the world's rice is grown in flooded paddy fields (Baicich, 2013). The water management systems,
particularly flooding, used to cultivate rice are one of the most important factors influencing $CH_4$ emissions and one of the
most promising approaches for $CH_4$ emission mitigation: periodic drainage and aeration not only cause existing soil $CH_4$ to
oxidise, but also inhibit further $CH_4$ production in soils (Simpson et al., 1995; USEPA, 2016; Zhang, 2016). Upland rice
fields are not typically flooded, and therefore are not a significant source of $CH_4$. Other factors that influence $CH_4$ emissions





from flooded rice fields include fertilisation practices (i.e., the use of urea and organic fertilisers), soil temperature, soil type
(texture and aggregated size), rice variety and cultivation practices (e.g., tillage, seeding, and weeding practices) (Conrad et
al., 2000; Kai et al., 2011; USEPA, 2011; Yan et al., 2009). For instance, $CH_4$ emissions from rice paddies increase with
organic amendments (Cai et al., 1997) but can be mitigated by applying other types of fertilisers (mineral, composts, biogas
residues) or using wet seeding (Wassmann et al., 2000).
The geographical distribution of rice emissions has been assessed by global (e.g., Janssens-Maenhout et al., 2019; Tubiello,
2019; USEPA, 2012) and regional (e.g., Castelán-Ortega et al., 2014; Chen et al., 2013; Chen and Prinn, 2006; Peng et al.,
2016; Yan et al., 2009; Zhang and Chen, 2014) inventories and land surface models (Li et al., 2005; Pathak et al., 2005; Ren
et al., 2011; Spahni et al., 2011; Tian et al., 2010, 2011; Zhang, 2016). The emissions show a seasonal cycle, peaking in the
summer months in the extra-tropics associated with monsoons and land management. Emissions from rice paddies are
influenced not only by the extent of rice field area, but also by changes in the productivity of plants (Jiang et al., 2017) as
these alter the $CH_4$ emission factor used in inventories. However, the inventories considered herein are largely based on
IPCC Tier 1 methods, which mainly scale with cultivated areas and include regional specific emission factors but do not
account for changes in plant productivity and detailed cultivation practices.
The largest emissions from rice cultivation are found in Asia accounting for 30 to 50% of global emissions (Fig. 3). The
decrease of $CH_4$ emissions from rice cultivation over recent decades is confirmed in most inventories, because of the
decrease in rice cultivation area, changes in agricultural practices, and a northward shift of rice cultivation since the 1970s,
as in China (e.g., Chen et al., 2013).
Based on the global inventories considered in this study, global $CH_4$ emissions from rice paddies are estimated to be 32 [25-
37] Tg $CH_4$ yr$^{-1}$ for the 2010-2019 decade (Table 3), or about 9% of total global anthropogenic emissions of $CH_4$. These
estimates are consistent with the 29 Tg $CH_4$ yr$^{-1}$ estimated for the year 2000 by Carlson et al. (2017).
**Waste management.** This sector includes emissions from managed and non-managed landfills (solid waste disposal on
land), and wastewater handling, where all kinds of waste are deposited. $CH_4$ production from waste depends on the pH,
moisture, and temperature of the material. The optimum pH for $CH_4$ emission is between 6.8 and 7.4 (Thorneloe et al.,
2000). The development of carboxylic acids leads to low pH, which limits methane emissions. Food or organic waste, such
as leaves and grass clippings, ferment quite easily, while wood and wood products generally ferment slowly, and cellulose
and lignin even more slowly (USEPA, 2010a).
Waste management was responsible for about 11% of total global direct anthropogenic $CH_4$ emissions in 2000 (Kirschke et
al., 2013). A recent assessment of $CH_4$ emissions in the USA found landfills to account for almost 26% of total USA
anthropogenic $CH_4$ emissions in 2014, the largest contribution of any single $CH_4$ source in the United States of America
(USEPA, 2016). In Europe, gas control has been mandatory on all landfills since 2009, and more importantly for $CH_4$
emissions, the EU Landfill Directive (1999) with subsequent amendments, has diverted most biodegradable waste away





from landfills towards source separation, recycling, composting and energy recovery, and with a legally binding target not
to landfill more than 10% of municipal solid waste by 2035.
Wastewater from domestic and industrial sources is treated in municipal sewage treatment facilities and private effluent
treatment plants. The principal factor in determining the $CH_4$ generation potential of wastewater is the amount of degradable
organic material in the wastewater. Wastewater with high organic content is treated anaerobically, which leads to increased
emissions (André et al., 2014). Excessive and rapid urban development worldwide, especially in Asia and Africa, could
enhance methane emissions from waste unless adequate mitigation policies are designed and implemented rapidly.
The GAINS model and CEDS and EDGAR inventories give robust emission estimates from solid waste in the range of 37-
42 Tg $CH_4$ yr$^{-1}$ for the year 2019, and more uncertain wastewater emissions in the range 20-45 Tg $CH_4$ yr$^{-1}$1.
In our study, the global emission of $CH_4$ from waste management is estimated in the range of 56-80 Tg $CH_4$ yr$^{-1}$ for the
2010-2019 period with a mean value of 69 Tg $CH_4$ yr$^{-1}$, about 19% of total global anthropogenic emissions.

### 3.1.5 Biomass and biofuel burning

This category includes $CH_4$ emissions from biomass burning in forests, savannahs, grasslands, peats, agricultural residues,
as well as, from the burning of biofuels in the residential sector (stoves, boilers, fireplaces). Biomass and biofuel burning
emit $CH_4$ under incomplete combustion conditions (i.e., when oxygen availability is insufficient for complete combustion),
for example in charcoal manufacturing and smouldering fires. The amount of $CH_4$ emitted during the burning of biomass
depends primarily on the amount of biomass, burning conditions, fuel moisture and the specific material burned.
In this study, we use large-scale biomass burning (forest, savannah, grassland, and peat fires) from five biomass burning
inventories (described below) and the biofuel burning contribution from anthropogenic emission inventories (EDGARv6
and v7, CEDS, GAINS and USEPA). The spatial distribution of emissions from the burning of biomass and biofuel over
the 2010-2019 decade is presented in Fig. 3 based on data listed in Table 1.
At the global scale, during the period of 2010-2019, biomass and biofuel burning generated $CH_4$ emissions of 28 [21-39] Tg
$CH_4$ yr$^{-1}$ (Table 3), of which 30-50% is from biofuel burning.

**Biomass burning.** Fire is an important disturbance event in terrestrial ecosystems globally (van der Werf et al., 2010), and
can be of either natural (typically ~10% of fires, ignited by lightning strikes or started accidentally) or anthropogenic origin
(~90%, human initiated fires) (USEPA, 2010b, chapter 9.1). As previously noted all fires are accounted as anthropogenic in
Table 3. Anthropogenic fires are concentrated in the tropics and subtropics, where forests, savannahs and grasslands may
be burned to clear land for agricultural purposes or to maintain pastures and rangelands. Small fires associated with
agricultural activity, such as field burning and agricultural waste burning, are often not well detected by remote sensing
methods and are instead estimated based on cultivated area.





Emission rates of biomass burning vary with biomass loading (depending on the biomes) at the location of the fire, the
efficiency of the fire (depending on the vegetation type), the fire type (smouldering or flaming) and emission factor (mass
of the considered species / mass of biomass burned). Depending on the approach, these parameters can be derived using
satellite data and/or biogeochemical model, or through simpler IPCC default approaches.
In this study, we use five products to estimate biomass burning emissions. The Global Fire Emission Database (GFED) is
the most widely used global biomass burning emission dataset and provides estimates from 1997 onwards. Here, we use
GFEDv4.1s (van der Werf et al., 2017), based on the Carnegie-Ames-Stanford-Approach (CASA) biogeochemical model
(van der Werf et al., 2010) driven by satellite derived vegetation characteristics and burned area mostly from the MODerate
resolution Imaging Sensor, MODIS (Giglio et al., 2013). GFEDv4.1s (with small fires) is available at a 0.25° resolution and
on a daily basis from 1997 to 2020. One characteristic of the GFEDv4.1s burned area is that small fires are better accounted
for compared to GFEDv4.1 (Randerson et al., 2012), increasing carbon emissions by approximately 35% at the global scale.
The latest version GFEDv5 (Chen et al., 2023) suggest 61% higher burned area than GFEDv4.1s, in closer agreement with
burned area products from higher resolution satellite sensors. The next budget would benefit from GFEDv5 to revisit the
estimates of biomass burning emissions (which would likely go up) based on more specific comparison studies.
The Quick Fire Emissions Dataset (QFED) is calculated using the fire radiative power (FRP) approach, in which the thermal
energy emitted by active fires (detected by MODIS) is converted to an estimate of $CH_4$ flux using biome specific emissions
factors and a unique method of accounting for cloud cover. Further information related to this method and the derivation of
the biome specific emission factors can be found in Darmenov and da Silva (2015). Here we use the historical QFEDv2.5
product available daily on a 0.1x0.1 grid for 2000 to 2020.
The Fire INventory from the National Center for Atmospheric Research (FINNv2.5, Wiedinmyer et al., 2023) provides
daily, 1 km resolution estimates of gas and particle emissions from open burning of biomass (including wildfire, agricultural
fires and prescribed burning) over the globe for the period 2002-2020. FINNv2.5 uses MODIS and VIIRS satellite
observations for active fires, land cover and vegetation density.
We use v1.3 of the Global Fire Assimilation System (GFAS, Kaiser et al., 2012), which calculates emissions of biomass
burning by assimilating Fire Radiative Power (FRP) observations from MODIS at a daily frequency and 0.5° resolution and
is available for 2000-2020.
The FAO-$CH_4$ yearly biomass burning emissions are based on the most recent MODIS 6 burned area products (Prosperi et
al., 2020), coupled with a pixel level (500 m) implementation of the IPCC Tier 1 approach, and are available from 1990 to
2020 (Table 1).
The differences in emission estimates for biomass burning arise from specific geographical and meteorological conditions
and fuel composition, which strongly impact combustion completeness and emission factors. The latter vary greatly
according to fire type, ranging from 2.2 g $CH_4$ $kg^{-1}$ dry matter burned for savannah and grassland fires up to 21 g $CH_4$ $kg^{-1}$
dry matter burned for peat fires (van der Werf et al., 2010). Biomass burning emissions encountered large inter annual





variability related to meteorological conditions, with generally higher emissions during El-Nino periods as in 2019 (20 [14-
28] Tg CH$_4$ yr$^{-1}$), 2015 (22 [15-28] Tg CH$_4$ yr$^{-1}$) and 2010 to a lesser extent (18 [15-29] Tg CH$_4$ yr$^{-1}$).
In this study, based on the five aforementioned products, biomass burning emissions are estimated at 17 Tg CH$_4$ yr$^{-1}$ [12-
24] for 2010-2019, representing about 5% of total global anthropogenic CH$_4$ emissions.

**Biofuel burning.** Burning of biomass to produce energy for domestic, industrial, commercial, or transportation purposes is
hereafter called biofuel burning. A largely dominant fraction of CH$_4$ emissions from biofuel burning comes from domestic
cooking or heating in stoves, boilers, and fireplaces, mostly in open cooking fires where wood, charcoal, agricultural
residues, or animal dung are burned. It is estimated that more than two billion people, mostly in developing countries, use
solid biofuels to cook and heat their homes daily (André et al., 2014), and yet CH$_4$ emissions from biofuel combustion have
received relatively little attention. Biofuel burning estimates are gathered from the CEDS, USEPA, GAINS and EDGAR
inventories. Due to the sectoral breakdown of the EDGAR and CEDS inventories the biofuel component of the budget has
been estimated as equivalent to the "RCO - Energy for buildings" sector as defined in Worden et al. (2017) and Hoesly et
al. (2018) (Table S2). This is equivalent to the sum of the IPCC 1A4a_Commercial-institutional, 1A4b_Residential,
1A4c_Agriculture-forestry-fishing and 1A5_Other-unspecified reporting categories. This definition is consistent with that
used in Saunois et al. (2016) and Kirschke et al. (2013). While this sector incorporates biofuel use, it also includes the use
of other combustible materials (e.g., coal or gas) for small-scale heat and electricity generation within residential and
commercial premises. Data provided by the GAINS inventory suggests that this approach may overestimate biofuels
emissions by between 5 and 50%. Further study into this category would be needed to better disentangle biofuels from fossil
combustibles.
In our study, biofuel burning is estimated to contribute 11 [8-14] Tg CH$_4$ yr$^{-1}$ to the global CH$_4$ budget, about 3% of total
global anthropogenic CH$_4$ emissions for 2010-2019.
**3.1.6 Other anthropogenic sources (not explicitly included in this study)**
Other anthropogenic sources not included in this study are related to agriculture and land-use management. In particular,
increases in agricultural areas (such as global palm oil production) have led to the clearing of natural peat forests, reducing
natural peatland area and associated natural CH$_4$ emissions. Peatlands planted to forests (like in Northern Europe) also lead
to reduced CH$_4$ emissions. While studies have long suggested that CH$_4$ emissions from peatland drainage ditches are likely
to be significant (e.g., Minkkinen and Laine, 2006, Peacock et al., 2021), CH$_4$ emissions related to palm oil plantations have
yet to be properly quantified (e.g., Manning et al, 2019). Taylor et al. (2014) have quantified global palm oil wastewater
treatment fluxes to be $4 \pm 32$ Tg CH$_4$ yr$^{-1}$ for 2010-2013. This currently represents a small and highly uncertain source of
methane but one potentially growing in the future.



### 3.2 Natural and indirect anthropogenic sources

As introduced in section 2.4, natural and indirect anthropogenic sources refer to pre-agricultural $CH_4$ emissions even if they are perturbed by anthropogenic climate change or other global change factors (e.g., eutrophication), and indirect emissions resulting from anthropogenic perturbation of the landscape (reservoirs) and the biogeochemical characteristics of soil. They include vegetated wetland emissions and inland freshwater systems (lakes, small ponds, reservoirs, and rivers), land geological sources (gas-oil seeps, mud volcanoes, microseepage, geothermal manifestations, and volcanoes), wild animals, wildfires, termites, thawing terrestrial and marine permafrost, and coastal and oceanic sources (biogenic, geological and hydrate). In water-saturated or flooded ecosystems, the decomposition of organic matter gradually depletes most of the oxygen in the soil or the sediment zone, resulting in anaerobic conditions and $CH_4$ production. Once produced, $CH_4$ can reach the atmosphere through a combination of three processes: (1) diffusive loss of dissolved $CH_4$ across the air-water boundary; (2) ebullition flux from sediments; and (3) flux mediated by emergent aquatic macrophytes and terrestrial plants (plant transport). On its way to the atmosphere, in the soil or water columns, $CH_4$ can be partly or completely oxidised by microorganisms, which use $CH_4$ as a source of energy and carbon (USEPA, 2010b). Concurrently, methane from the atmosphere can diffuse into the soil column and be oxidised (See Sect. 3.3.4 on soil uptake).

### 3.2.1 Wetlands

Wetlands are generally defined as ecosystems in which mineral or peat soils are water-saturated at some depth or where surface inundation (permanent or not) has a dominating influence on the soil biogeochemistry and determines the ecosystem species composition (USEPA, 2010b). To refine such an overly broad definition for $CH_4$ emissions, we define wetlands as ecosystems with inundated or saturated soils or peats where anaerobic conditions below the water table lead to $CH_4$ production (Matthews and Fung, 1987; USEPA, 2010b). Brackish water emissions are discussed separately in Sect. 3.2.6. Our definition of wetlands includes ombrotrophic and minerotrophic peatlands (i.e., bogs and fens), mineral soil wetlands (swamps and marshes), and seasonal or permanent floodplains. It excludes exposed water surfaces without emergent macrophytes, such as lakes, rivers, estuaries, ponds, and reservoirs (addressed in the next section), as well as rice agriculture (see Sect. 3.1.4, rice cultivation paragraph), and wastewater ponds. It also excludes coastal vegetated ecosystems (mangroves, seagrasses, salt marshes) with salinities usually >0.5 (See Sect. 3.2.6). Even with this definition, some wetlands could be considered as anthropogenic systems, being affected by human land-use changes such as impoundments, drainage, or restoration (Woodward et al., 2012). In the following, we retain the generic denomination "wetlands" for natural and human-influenced wetlands, as discussed in Sect. 2.2.

The three most important factors influencing $CH_4$ production in wetlands are the spatial and temporal extent of anoxia (linked to water saturation), temperature, and substrate availability (Valentine et al., 1994; Wania et al., 2010; Whalen, 2005; Delwiche et al., 2021; Knox et al., 2021).





Land surface models estimate $CH_4$ emissions through a series of processes, including $CH_4$ production, oxidation, and
transport. The models are then forced with inputs accounting for changing environmental factors (Melton et al., 2013;
Poulter et al., 2017; Tian et al., 2010; Wania et al., 2013; Xu et al., 2010). $CH_4$ emissions from wetlands are computed as
the product of an emission flux density and a $CH_4$ producing area or surface extent (see Supplementary Material; Bohn et
al., 2015; Melton et al., 2013). The areal extent of different wetland types (having large differences in areal $CH_4$ emission
rates) appears to be a primary contributor to uncertainties in the absolute flux of $CH_4$ emissions from wetlands, with
meteorological response being the main source of uncertainty for seasonal and interannual variability (Poulter et al., 2017;
Kuhn et al., 2021; Parker et al., 2022; McNicol et al., 2023; Karlson and Bastviken 2023).
In this work, sixteen land surface models computing net $CH_4$ emissions (Table 2) were run under a common protocol with
a  spin-up using repeated climate data from 1901-1920 to pre-industrial conditions followed by a transient simulation through
the end of 2020. Of the 16 models, 13 previously contributed to Saunois et al. (2020), and three models were new to this
release (CH4MOD$_{wetland}$ (Li et al., 2010), ISAM (Shu et al., 2020; Xu et al., 2021), and SDGVM (Beerling and Woodward,
2001; Hopcroft et al., 2011; Hopcroft et al., 2020 )) (Table 2, see also in the Supplementary Material Table S3 for a history
of the contributing models). Climatic forcing uncertainties are considered in the ensemble estimate by using two climate
datasets, CRU/CRU-JRA55 (Harris, 2014) and GSWP3-W5E5 (Dirmeyer et al., 2006; Kim 2017; Lange, 2019; Cucchi et
al., 2020). Atmospheric $CO_2$ was also prescribed in the models. For all models, two wetland area dynamic schemes were
applied: a diagnostic scheme using a remote sensing-based wetland area and dynamics dataset called WAD2M (Wetland
Area Dynamics for Methane Modeling; Zhang et al., 2021a; 2021b) available at 0.25 degree of horizontal resolution, as in
Saunois et al. (2020), and a prognostic scheme using internal model-specific hydrologic models.
The diagnostic wetland extent product WAD2Mv1.0 (Zhang et al., 2021a) has been updated since Saunois et al. (2020) to
WAD2Mv2.0 (Zhang et al., 2021b) and extended to 2020. It uses the same Surface Water Microwave Product Series
(SWAMPSv3.2) for capturing inundation dynamics (Jenson and McDonald, 2019), which was extended to 2020. To reduce
potential double-counting with the freshwater budget, the surface areas of rivers/streams and lakes/ponds are excluded by
using the products Global River Widths from Landsat (GRWL) database v01.01 (Allen and Pavelsky, 2018) and HydroLakes
v1.0 (Messenger et al., 2016), instead of the Global Surface Water (GSW) product (Pekel et al., 2016) used in WAD2Mv1.0.
The GRWL and Hydrolakes are also the datasets used separately in the upscaling of the freshwater budget allowing for a
more consistent approach between the wetland and freshwater $CH_4$ budgets (Sect. 3.2.2). This update in WAD2M leads to
a downward revised annual average wetland extent by 0.5 $Mkm^2$ for the mid-high latitudes (mainly due to larger lake extent
in HydroLakes than in the GSW dataset) with small impacts in other regions. However, since HydroLakes includes only
vectorized lakes larger than 0.1 $km^2$, smaller lakes/ponds under 0.1 $km^2$ are implicitly still included as wetlands in
WAD2Mv2.0. For the high-latitude region, the recent peatland extent product from Hugelius et al. (2020) is applied, which
indicates a slightly higher peatland area by 0.2 $Mkm^2$ primarily in regions above 60°N, compared to the Northern
Circumpolar Soil Carbon Database (NCSCD) product (Hugelius et al., 2013) used in WAD2Mv1.0. Rice agriculture was





removed using the Monthly Irrigated and Rainfed Crop Areas (MIRCA2000, Portmann et al. (2010)) dataset from circa
2000, as a fixed distribution.
The combined remote-sensing and inventory WAD2Mv2.0 product leads to a maximum wetland area of 13.6 $Mkm^2$ during
the peak season (7.9 $Mkm^2$ on annual average, with a range of 7.5 to 8.4 $Mkm^2$ from 2000-2020, about 5.2% of the global
land surface). The largest wetland areas in WAD2Mv2.0 are in Amazonia, the Congo Basin, and the Western Siberian
Lowlands, which in previous studies were underestimated by inventories (Bohn et al., 2015). However, the SWAMPS v3.2
dataset which serves as a proxy of temporal variations of wetland extent, has discontinuity issues over a few tropical hotspots
since 2015 and hence affects the temporal variations of WAD2M. Consequently, this affects $CH_4$ emissions estimates for a
subset of land surface models that are particularly sensitive to inundation in these hotspots. Meanwhile, prognostic estimates
show moderate consistency in capturing the spatial distribution of wetland area with WAD2M, with an annual average
wetland area of 8.0±2.0 $Mkm^2$ during the peak season for 2000-2020. The ensemble mean of annual wetland area anomaly
by the prognostic models show reasonable agreement with satellite-based estimates in capturing the response of wetland
area to climate variations (Zhang et al., in review), with higher agreement over temperate and boreal regions than in the
tropics.
For the wetland methane emissions estimate, we use the decadal mean from the prognostic runs and adjust these flux
estimates for double counting from inland waters (described in next section) given the reliance of the prognostic models on
satellite flooded area data like WAD2Mv2 to parameterize maximum wetland extent (Zhang et al., in review). The average
emission from wetlands for 2010-2019 for the 16 models is plotted in Fig. 3. The zones with the largest emissions are the
Amazon basin, equatorial Africa and Asia, Canada, western Siberia, eastern India, and Bangladesh. Regions where $CH_4$
emissions have high inter-model agreement (defined as regions where mean flux is larger than the standard deviation of the
models, on a decadal mean) represent 72% of the total $CH_4$ flux due to natural wetlands. The different sensitivities of the
models to temperature, vapour pressure, precipitation, and radiation can generate substantially different patterns, such as in
India. Emission estimates over regions with lower emissions (in total) are also consistently inferred between models (e.g.,
Scandinavia, Continental Europe, Eastern Siberia, Central United States of America, and Southern Africa).
The resulting global flux range for vegetated wetland emissions from the prognostic runs is 117-195 Tg $CH_4$ $yr^{-1}$ for the
2000-2020 period, with an average of 157 Tg $CH_4$ $yr^{-1}$ and a one-sigma standard deviation of 24 Tg $CH_4$ $yr^{-1}$. Using the
prognostic set of simulations, the average ensemble emissions were 159 [119-203] Tg $CH_4$ $yr^{-1}$ for the 2010-2019 period
(Table 3). The estimated average ensemble annual total from the two sets of simulations by CRU/CRU-JRA55 and GSWP3-
W5E5 are 158 [126-193] and 158 [118-203] for 2010-2019, respectively. Generally, the magnitude and interannual
variability agree between these two sets of simulations (Zhang et al., in review). Wetland emissions represent about 25% of
the total (natural plus anthropogenic) $CH_4$ sources estimated by bottom-up approaches. The large range in the estimates of
wetland $CH_4$ emissions results from difficulties in defining wetland $CH_4$ producing areas as well as in parameterizing
terrestrial anaerobic conditions that drive sources and the oxidative conditions leading to sinks (Melton et al., 2013; Poulter




et al., 2017; Wania et al., 2013). The ensemble mean emission using the same simulation setup (i.e., diagnostic wetland
extent and CRU/CRU-JRA55) in the models is 163 [117-195] Tg $CH_4$ $yr^{-1}$, higher by ~22 Tg $CH_4$ $yr^{-1}$ than the one previously
reported (see Table 3, for 2000-2009 with comparison to Saunois et al., 2020). This difference is mainly due to the updated
model structure and parameterizations in the wetland $CH_4$ models compared to the versions in the previous budget and the
inclusion of three new land surface models.
For the last decade 2010-2019, we report in this budget an average ensemble estimate of 159 Tg $CH_4$ $yr^{-1}$ with a range of
119-203 (based on prognostic wetland extent runs).

### 3.2.2 Inland freshwater systems (lakes, ponds, reservoirs, streams, rivers)

This category includes $CH_4$ emissions from freshwater systems (lakes, ponds, reservoirs, streams, and rivers). Numerous
advances have been made in the freshwater greenhouse gases knowledge base in the last few years (Lauerwald et al., 2023a).
These advances include improvements in the underlying databases used to estimate inland water surface areas and model
their dynamics, a rapidly growing number of direct measurements of methane fluxes, and improvements in our process-
based understanding of methane biogeochemistry. Despite this, aspects of global freshwater methane estimates remain rather
crude and continue to have large uncertainties. This includes the overall temperature dependency of methane emissions, the
relative role of ebullition (i.e., bubble flux, which may represent the most important, but most difficult-to-capture emission
path in many standing water bodies), fluxes from the smallest standing water bodies (sometimes referred to as ponds) having
large emissions per $m^2$ but uncertain area extent, and the magnitude of anthropogenic influence on emissions, all which are
discussed below.
**Streams and rivers.** The last global $CH_4$ budget used an estimate of 27 Tg $CH_4$ $yr^{-1}$ for global streams and rivers based
largely on a data compilation by Stanley et al. (2016). This estimate was scaled from a simple data compilation without a
spatial component or an estimate of ebullition. More recently, Rosentreter et al. (2021) performed a new data compilation
of 652 flux estimates, including diffusive and ebullitive fluxes, coupled to an ice corrected surface area estimate of ~625,000
$km^2$ that was aggregated to 5 latitudinal bands to come up with a global estimate of 6 and 31± 17 Tg $CH_4$ $yr^{-1}$ (respectively
for the median and mean ± c.i. 95%). We believe, due to better data representation in underlying datasets, that the mean
estimate of Rosentreter et al. (2021) is more representative statistically because the median does not capture hotspots and
hot moments of intense ebullitive fluxes.  Finally, Rocher-Ros et al. (2023) used a new Global River Methane (GRiMeDB)
database (Stanley et al., 2023) with > 24,000 observations of $CH_4$ concentrations to predict  ~28±17 Tg $CH_4$ $yr^{-1}$ (±c.i. 95%)
river emissions globally. This approach used machine learning methods coupled to the latest spatially and temporally explicit
mapping of monthly stream surface area (the smallest streams are still extrapolated) which incorporates drying and freezing
effects (yearly average 672,000 $km^2$, Liu et al., 2022) and includes an ebullitive flux estimated from a correlation between
measured diffusive and ebullitive emissions in the GRiMeDB database (Stanley et al., 2023). Thus, for this study we use an



estimate of 29±17 (±c.i. 95%) Tg $CH_4$ $yr^{-1}$ for streams and rivers, which averages the mean estimate of Rosentreter et al.
(2021) and Rocher-Ros et al. (2023). Currently, ebullitive fluxes remain a major unknown quantity in streams and rivers but
appear to be coarsely linearly correlated in a log-space to diffusive fluxes and of similar magnitude (Rocher-Ros et al.,
2023). Methodologically, the high-water velocity of many streams and rivers make measurement of ebullitive fluxes
challenging (Robison et al., 2021). Effluxes are also linked to hydrology (Aho et al., 2021) although very few studies have
sampled over a representative hydrograph. Plant-mediated effluxes of $CH_4$ in running waters also remain difficult to
constrain, with a recent compilation highlighting very few measurements (Bodmer et al. 2024). Connected adjacent wetlands
is a common source of $CH_4$ to streams and rivers (Borges et al., 2019) which may be important for the regulation of running
water emissions but is currently difficult to assess at the global scale. Overall, the poor representation of sites and deficient
mechanistic understanding make it difficult to model and predict methane evasion from streams and rivers using process-
based models.

**Lakes and ponds.** The previous global $CH_4$ budget used an estimate of 71 Tg $CH_4$ $yr^{-1}$ for lakes and 18 Tg $CH_4$ $yr^{-1}$ for
reservoirs. These estimates were based on an early study by Bastviken et al. (2011) coupled with a newer estimate for lakes
north of 50°N (Wik et al., 2016b). There have been three new lake studies that have published their data with global estimates
of 56 and 151± 73 (Rosentreter et al. (2021; (±c.i. 95%) ; respectively for the median and mean ± c.i. 95% ), 22±8 (Zhuang
et al., 2023; ±lake-area-weighted normalised RMSE for all parameterized lake types ), process-based model), and 41±36
Tg $CH_4$ $yr^{-1}$ (Johnson et al., 2022, mean ±c.i. 95%). This large range in estimated emissions can be attributed to the
differences in the datasets and methods used to calculate the surface area of small waterbodies, as well as the differences
between how the flux data were analyzed and extrapolated between studies. For instance, total surface areas of all lakes and
ponds of 3712-5688 × $10^3$ $km^2$ (Rosentreter et al., 2021) and 2806 × $10^3$ $km^2$ (Johnson et al., 2022) were used along with
measurement data from 198 and 575 individual lake systems, respectively. In contrast, Zhuang et al. (2023) generated
estimates using higher temporal resolution data from just 54 lakes to build a process-based model, which generated much
lower flux estimates from tropical lakes than previously implemented statistical approaches, but in line with the most recent
assessments by Borges et al. (2022). For this study, we explicitly excluded lakes <0.1 $km^2$ which are treated separately (see
below). If we re-assess these three studies for only lakes greater than 0.1 $km^2$, we obtain global effluxes of 17 and 42.9±20.8
Tg $CH_4$ $yr^{-1}$ (Rosentreter et al. (2021); median and mean (±95% C.I.) of global flux), 21.9±8.0. (Zhuang et al., 2023, ±lake-
area-weighted normalised RMSE for all parameterized lake types), and 35.3±31.0 Tg $CH_4$ $yr^{-1}$ (Johnson et al. 2022, ±95%
C.I.) (with areas of 2556-3468 x$10^3$, 2640x$10^3$, and 2676x$10^3$ $km^2$ respectively). Thus, for lakes >0.1 $km^2$, we propose an
efflux of ~33±26 Tg $CH_4$ $yr^{-1}$ (an average of the mean from Rosentreter et al., 2021 Zhuang et al., 2023, and Johnson et al.,
2022, with the average 95% C.I. from Rosentreter et al., 2021 and Johnson et al. 2022).
Small waterbody emissions, hereafter small lakes and ponds<0.1 $km^2$, remain difficult to assess. Evidence is emerging that
there is a lower limit to the power scaling laws that early studies used to extrapolate the surface area of these small systems



(Bastviken et al., 2023; Kyzivat et al., 2022). Thus, for small lakes and ponds < 0.1 km$^2$ (and >0.001 km$^2$), we disregard the
higher end surface area used in Rosentreter et al., 2021 which relied on these earlier estimates and scale their numbers to
the evasion estimates to the lower end surface area of 1,002x10$^3$ to obtain a mean flux of 33 Tg CH$_4$ yr$^{-1}$ (Rosentreter et al.,
2021). Johnson et al. (2022) estimated a surface area of only 166,000 km$^2$ for this size class to obtain an efflux of 6.3
Tg CH$_4$ yr$^{-1}$, which we acknowledge as a lower limit. Averaging these two values provide a conservative estimate of ~20
[6-33] Tg CH$_4$ yr$^{-1}$, which is close to the number proposed by Holgerson and Raymond (2016) for diffusion effluxes only
for this size class. The experts involved in this assessment have low confidence in this estimate. This also does not include
artificial ponds, which we discuss below. As a result, CH$_4$ emissions from both large lakes (>0.1 km$^2$) and small lakes and
ponds (<0.1km$^2$) are estimated at 53 [19-86] Tg CH$_4$ yr$^{-1}$, on average lower than the 71 Tg estimated in the previous budget.

**Reservoirs.** New mean estimates of diffusive + ebullitive CH$_4$ emissions from reservoirs include 15 and 24±8 (the median
and mean±95% C.I. from Rosentreter et al., 2021), 10±4 (Johnson et al., 2021, mean±95% C.I.), 10 (Harrison et al., 2021,
low and high 95% CI 7 and 22, respectively), and 2.1 Tg CH$_4$ yr$^{-1}$ (Zhuang et al., 2023). We compile the first three estimates
to a direct efflux of ~14 Tg CH$_4$ yr$^{-1}$ (with ±95% C.I. of 9 and 23). We note the fourth estimate as a lower bound, but exclude
it from this budget given that it was generated via a model that only included data from six reservoir systems (Zhuang et al.,
2023). We also add in an additional 12 Tg CH$_4$ yr$^{-1}$ (95% C.I, 7 and 37) that is estimated to degas in dam turbines (Harrison
et al., 2021), which was not addressed in the studies by Rosentreter et al. (2021), Zhuang et al. (2023), or Johnson et al.
(2021). Rocher-Ros et al. (2023) also excluded river observations below dams when executing their statistical model, and
so did not capture downstream dam emissions. Thus, we use a direct reservoir emission here of ~13 [6-28] Tg CH$_4$ yr$^{-1}$ and
estimate an additional ~12 [7-37] Tg CH$_4$ yr$^{-1}$ from dam turbine degassing fluxes, giving a total of 25 [13-65] Tg CH$_4$ yr$^{-1}$
from reservoirs.

**Uncertainties and confidence levels.** The emission estimates of lakes, reservoirs and ponds described above are limited by
several uncertainties. First, a major unknown for lakes remains the size cut off and the representation of small lakes and
ponds (Deemer and Holgerson, 2021), which are also more variable than larger water bodies in their CH$_4$ concentrations
and fluxes (Rosentreter et al. 2021, Ray et al., 2023). Interestingly, there is also a lack of methane data representation from
large lakes that are a large component of global lake surface area (Deemer and Holgerson, 2021; Messager et al., 2016).
There is also a growing knowledge base on the importance of high CH$_4$ fluxes from lake littoral zones that is not yet well
incorporated into global scaling efforts (e.g., Grinham et al., 2011; Natchimuthu et al., 2016), and emergent vegetation
(Bastviken et al., 2023; Kyzivat et al., 2022). Ebullition is more constrained in lakes/reservoirs compared to streams/rivers
but is still difficult to measure and model accurately. Finally, for all aquatic systems a greater scrutiny of the regulation
(including the impact of ice-cover and seasonality) of different CH$_4$ emission pathways is needed.



The majority of the inland water $CH_4$ estimates are from a limited number of studies, some without spatial representation or
reported statistical uncertainties. Furthermore, as mentioned above the knowledge base of the surface area of these
ecosystems is new and rapidly expanding, but not standardised between studies leading to uncertainty (but see Lauerwald
et al. 2023b), particularly for ponds. For this study, we are able to provide confidence intervals from the original studies for
all fluxes except the smallest lake/pond size class.

**The Surface Area of Inland Freshwaters.** For all of these ecosystems, determining their surface area remains a central
challenge. Since the last GMB, several methodological advances have reduced the uncertainty associated with the surface
area estimates of rivers, streams, lakes, and reservoirs. Using a single geospatial dataset that includes both lakes and
reservoirs (Messager et al., 2016) has decreased double counting of lakes and reservoirs (Johnson et al., 2022; Rosentreter
et al., 2021). For rivers and streams, high-resolution global streamflow simulations, informed by satellite observations,
enabled a much finer scale estimate of surface areas for rivers with a new temporal component (Allen and Pavelsky, 2018;
Lin et al., 2019; Liu et al., 2022), although the surface for the smaller streams are still estimated indirectly, and mapping of
human-created drainage ditches and canals is lacking. Seasonal ice cover and melt turnover corrections also have been newly
incorporated into rivers, streams, lakes, and reservoirs (Harrison et al., 2021; Johnson et al., 2022; Lauerwald et al., 2023b;
Rocher-Ros et al., 2023; Rosentreter et al., 2021; Zhuang et al., 2023). Finally, removing open water body surface areas
from wetland surface areas based on geographic location has reduced double counting between these two land cover types,
as described in the wetlands section of the GMB. Yet, the surface area of small lakes and ponds (<0.1 km2) is still highly
uncertain, and new techniques for counting these systems and determining the overlap with wetland data bases is paramount.

**Anthropogenic Contributions to Inland Freshwater Emissions.** We argue that all reservoirs should be categorised as an
direct anthropogenic source of emissions. Most of the surface area of reservoirs are human-made and reservoir construction
leads to anoxic sediments and/or bottom waters with labile organic matter sourced from the watershed and to in-situ nutrient
augmented phytoplankton production (Deemer et al., 2016; Maavara et al., 2017; Prairie et al., 2018). It is also clear that the
cultural eutrophication of natural lakes is augmenting $CH_4$ emissions (DelSontro et al., 2018; Li et al., 2021), with shallow
lakes particularly likely to experience eutrophication (Qin et al., 2020). For instance, Beaulieu et al. (2019) modelled a 15%
reduction in lake $CH_4$ with a 25% reduction in lake phosphorus concentrations. Several recent studies have estimated that
anywhere between 30 and 50% of lakes are eutrophic (Cael et al., 2022; Qin et al., 2020; Sayers et al., 2015; Wu et al.,
2022). These studies estimate numerical percentages (one by depth class: Qin et al., 2020), but none have estimated the
percent of lake surface area that is eutrophic nor have any determined the extent of anthropogenic vs. natural eutrophication.
Still, numerous studies have noted widespread increases in eutrophication indicators across lakes due to nutrient loading
and warming (Griffiths et al., 2022; Ho et al., 2019; Taranu et al., 2015), thus we estimate that ⅓, or 11 Tg $CH_4$ $yr^{-1}$ of
$CH_4$ emissions from lakes >0.1 $km^2$ could be anthropogenic. Similarly, $CH_4$ emissions from small lakes and ponds are

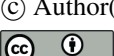



influenced by human factors, with emissions increasing with eutrophication (Deemer and Holgerson, 2021), erosion and runoff in agricultural landscapes (Heathcote et al., 2013), and warming, the latter likely to have a disproportionately greater effect in small, shallow systems (Woolway et al., 2016). Thus, we adopt the same ⅓ number as for lakes for the proportion of anthropogenic emissions in small lakes and ponds (<0.1 km2), which amounts to 6 Tg CH$_4$ yr$^{-1}$.

There are also human-made small lakes and ponds, notably for agriculture, aquaculture, and recreation, that generally have conditions favourable for high CH$_4$ emissions (Downing, 2010; Holgerson and Raymond, 2016; Malerba et al., 2022; Ollivier et al., 2019; Zhao et al., 2021; Dong et al., 2023). Downing (2010) estimated that farm ponds comprise a global surface area of ~77,000 km$^2$; using a conservative emission rate of 265 mg CH$_4$ m$^{-2}$ d$^{-1}$ and an ice correction factor of 0.6 leads to an emission of 4.5 Tg yr$^{-1}$ that is anthropogenically sourced from farm ponds. Here the value is rounded to 5 Tg yr$^{-1}$. Clearly, more work is required to assess the anthropogenic component of CH$_4$ emissions from lakes and ponds.

It remains difficult to parse out an anthropogenic component to stream and river CH$_4$ fluxes. Although some studies have noticed a temperature dependence with stream sediments (Comer-Warner et al., 2018; Zhu et al., 2020), Rocher-Ros et al. (2023) noted a small temperature dependence of CH$_4$ emissions in streams and rivers compared to other freshwater ecosystems, potentially due to the many other external processes affecting fluxes in these dynamic flowing ecosystems. Urbanisation can lead to elevated river CH$_4$ emissions, particularly in regions with elevated organic matter and nutrient loading due to limited wastewater treatment (Begum et al., 2021; Nirmal Rajkumar et al., 2008; Wang et al., 2021a). Some studies have found agricultural streams and ditches can have higher effluxes due to inputs of fine sediments (Comer-Warner et al., 2018; Crawford and Stanley, 2016), organic carbon, and nutrients (Borges et al., 2018) that lead to in-situ methane production. Furthermore, the creation of drainage ditches in organic soils tap CH$_4$ rich waters from water-logged horizons and heighten emissions from ex-situ sources (Peacock et al., 2021), although limitations in both the geographic scope of existing ditch emission estimates our ability to estimate global surface area of ditches precludes their inclusion in this budget. Finally, extremely high rates of CH$_4$ emission have been linked to ongoing permafrost thaw in Asia's Qinghai–Tibet Plateau (Zhang et al., 2020). However, the loss and disconnection of wetlands to rivers may have resulted in a decrease in the input of dissolved CH$_4$ from this source. A recent expert elicitation (Rosentreter, et al. submitted) reported that 35% of all inland freshwater sources were anthropogenic and given that some of the river flux is from upstream reservoirs, we assign a 30% anthropogenic contribution to the stream and river flux (9 Tg CH$_4$ yr$^{-1}$), which approximates the expert elicitation via the impact of eutrophication and urban influences.

**Combination (lakes, ponds, reservoirs, streams and rivers, farm ponds).** Combining the aforementioned emissions from lakes >0.1 km$^2$ (~33 [13-53] Tg CH$_4$ yr$^{-1}$ ), small lakes and ponds < 0.1 km$^2$ (20 [6-33] Tg CH$_4$ yr$^{-1}$), reservoirs (25 [13-65] Tg CH$_4$ yr$^{-1}$), streams and rivers (29 [12-46] Tg CH$_4$ yr$^{-1}$) and farm ponds (5 Tg CH$_4$ yr$^{-1}$), leads to a total of ~112 Tg CH$_4$ yr$^{-1}$ from freshwater systems, with a range of [49-202] Tg CH$_4$ yr$^{-1}$. This estimate is about 50 Tg lower than in Saunois et al. (2020) and is broadly consistent with the recent regionalized estimate by Lauerwald et al. (2023b) compiled for the



Regional Carbon Cycle Assessment and Processes (RECCAP2, https://www.globalcarbonproject.org/reccap/; 103 Tg CH$_4$
yr$^{-1}$, IQR= 82.1–134.8). The updated budget from these ecosystems and their anthropogenic components are represented on
Fig 4. The gridded products for emissions from lakes and ponds by Johnson et al. (2022), from reservoirs by Johnson et al.
(2021) and from streams and rivers by Rocher-Ros et al. (2023) have been combined into a single map presented in Fig. 5.

**Double-counting aquatic systems in the bottom-up estimates.** To address the differences found between bottom-up and
top-down CH$_4$ budgets, and to acknowledge advances in addressing the central issue of double counting CH$_4$ emissions for
inland ecosystems, we introduce here a new correction term. Historically, the bottom-up estimate of global CH$_4$ emissions
has been higher than the top-down estimate, first recognized in Kirschke et al. (2013) and confirmed in Saunois et al. (2016,
2020). The larger bottom-up emissions estimate has been partly attributed to double-counting vegetated wetland emissions
with inland freshwater emissions (including lakes, ponds, rivers, streams, and reservoirs) and also the emissions of CH$_4$
produced in vegetated wetlands and then transported via aquatic processes and emitted from inland freshwaters (Pangala et
al., 2017; Kirk and Cohen, 2023). The Saunois et al. (2020) CH$_4$ budget addressed the issue of double counting through the
use of a revised vegetated wetland area dataset, WAD2M v1.0 (Zhang et al., 2021), that removed inland waters from the
SWAMPS (Jenson and McDonald, 2019) surface-inundation dataset, allowing for independent vegetated wetlands and
inland freshwater CH$_4$ emissions to be compiled. Yet, the Saunois et al. (2020) CH$_4$ budget still had a ~150 Tg CH$_4$ yr$^{-1}$
difference between bottom-up and top-down estimates. In this budget, we refined the vegetated wetland area dataset with
WAD2M v2.0 (see section 3.2.1, where HydroLakes is used to remove lakes and ponds >0.1 km$^2$). Additionally, we applied
numbers from peer-reviewed publications and expert elicitation to account for lateral CH$_4$ flux emissions. This most recent
BU budget estimates 159 [119-203] Tg CH$_4$ yr$^{-1}$ from vegetated wetlands for 2010-2019 and 112 Tg CH$_4$ yr$^{-1}$ from inland
freshwaters that includes 83 Tg CH$_4$ yr$^{-1}$ from lakes, ponds, and reservoirs and 29 Tg CH$_4$ yr$^{-1}$ from rivers and streams,
leading to a combined wetland and inland freshwater flux of 271 Tg CH$_4$ yr$^{-1}$. Here, we propose a correction of 20 Tg CH$_4$
yr$^{-1}$ to account for double counting of small lakes and ponds (< 0.1 km$^2$) that are likely included in our vegetated wetlands
estimate, and removing 1-3 Tg CH$_4$ yr$^{-1}$ from river emissions due to lateral transport of CH$_4$ originating in adjacent vegetated
wetlands. The river flux correction arises from assuming that for catchments with >10% wetlands, rivers provide 5-10% of
vegetated CH$_4$ emissions. The total double-counting correction term of 23 Tg CH$_4$ reduces the BU budget for combined
wetlands and inland waters from 271 Tg CH$_4$ yr$^{-1}$ to 248 Tg CH$_4$ yr$^{-1}$ (see Fig. 4 and Table 3). Comparing the 2000-2009
decadal emissions from wetlands and inland freshwater ecosystems across the last three previous assessments of the budget
shows a significant downward revision with 305 (183+122) Tg CH$_4$ yr$^{-1}$, 356 (147+209) Tg CH$_4$ yr$^{-1}$ and 248 (159+112-23)
Tg CH$_4$ yr$^{-1}$ (respectively from Saunois et al. (2016; 2020) and this work).
Finally, it is worth noting that inland freshwater ecosystems can overlap with geological seepage systems in some areas,
i.e., they may occur in correspondence with geological structures that emit fossil (microbial, thermogenic, or abiotic)
CH$_4$ generated in the Earth's crust. Examples have been documented in the Fisherman Lake in Canada (Smith et al., 2005),



in the Baikal lake (Schmid et al, 2007), and in rice paddies in Japan (Etiope et al., 2011). Thus, some gas emissions in
freshwater environments, particularly as bubble plumes, can be incorrectly attributed to modern biological (ecosystem)
activities if appropriate isotopic and molecular analyses are not performed.

### 3.2.3 Onshore and offshore geological sources

Significant amounts of $CH_4$, produced within the Earth's crust, naturally migrate to the atmosphere through tectonic faults
and fractured rocks. Major emissions are related to hydrocarbon formation in sedimentary basins (microbial and thermogenic
methane), through continuous or episodic exhalations from onshore and shallow marine hydrocarbon seeps and through
diffuse soil microseepage (Etiope, 2015). Specifically, five source categories have been considered. Four are onshore
sources: gas-oil seeps, mud volcanoes, diffuse microseepage, and geothermal manifestations including volcanoes. One
source is offshore: submarine seepage, which may include the same types of gas manifestations occurring on land. Etiope
et al. (2019) have produced the first gridded maps of geological $CH_4$ emissions and their isotopic signature for these five
categories, with a global total of 37.4 Tg $CH_4$ yr$^{-1}$ (reproduced in Fig. 5). However, these maps are based on incomplete
data on geological sites due to missing information and difficulties in defining all current geological emitting sites.
Combining the best estimates for the five categories of geological sources (from grid maps or from previous statistical and
process-based models), the breakdown by category reveals that onshore microseepage dominate (24 Tg $CH_4$ yr$^{-1}$), the other
categories having similar smaller contributions: as mean values, 4.7 Tg $CH_4$ yr$^{-1}$ for geothermal manifestations, about 7 Tg
$CH_4$ yr$^{-1}$ for submarine seepage and 9.6 Tg $CH_4$ yr$^{-1}$ for onshore seeps and mud volcanoes. These values lead to a global
bottom-up geological emission mean of 45 [27-63] Tg $CH_4$ yr$^{-1}$ (Etiope and Schwietzke, 2019).
While all bottom-up and some top-down estimates, following different and independent techniques from different authors,
consistently suggest a global geo-$CH_4$ emission in the order of 40-50 Tg yr-1, the radiocarbon ($^{14}$C-$CH_4$) data in ice cores
reported by Hmiel et al. (2020) appear to give a much lower estimate, with a minimum of about 1.6 Tg $CH_4$ yr$^{-1}$ and a
maximum value of 5.4 Tg $CH_4$ yr$^{-1}$ (95 percent confidence) for the pre-industrial period. The discrepancy between Hmiel et
al. (2020) and all other estimates has been discussed in Thornton et al. (2021), which demonstrated that the global near-zero
geologic $CH_4$ emission estimate in Hmiel et al. (2020) is incompatible with the sum of multiple independent bottom-up
estimates, based on a wide variety of methodologies, from individual natural geological seepage areas: for example, from
the Black Sea (up to 1 Tg $CH_4$ yr$^{-1}$ ), the Eastern Siberian Arctic Shelf (ESAS, up to 4.6 Tg $CH_4$ yr$^{-1}$, referring mostly to
thermogenic gas), onshore Alaska (up to 1.4 Tg $CH_4$ yr$^{-1}$) and a single seepage site in Indonesia (releasing 0.1 Tg $CH_4$ yr$^{-1}$
as estimated by satellite measurement) (see Thornton et al. (2021) and references therein). Jackson et al. (2020) expressed
doubt about the low Hmiel et al. (2020) estimates, noting that they are difficult to reconcile with the results of many other
researchers and with bottom-up approaches in general. This discrepancy highlights another main unresolved uncertainty in
the methane budget.  Waiting for further investigation to better understand discrepancies between radiocarbon approaches
and other studies, we decided to keep the estimates from Etiope and Shwietzke (2019) for the mean values, and associate it





to the lowest estimates reported in Etiope et al. (2019), as in Saunois et al. (2020). Thus, we report a total global geological
emission of 45 [18-63] Tg $CH_4$ yr$^{-1}$, with a breakdown between offshore emissions of 7 [5-10] Tg $CH_4$ yr$^{-1}$ and onshore
emissions of 38 [13-53] Tg $CH_4$ yr$^{-1}$, similar to Saunois et al. (2020). This bottom-up estimate is slightly lower than in the
Saunois et al. (2016) budget mostly due to a reduction of estimated emissions of onshore and offshore seeps (see Sect. 3.2.6
for more offshore contribution explanations).

### 3.2.4 Termites

Termites are decomposers playing a central role in ecosystem nutrient fluxes at tropical and subtropical latitudes, in
particular (Abe et al., 2000). Termites represent a natural CH4 source due to methanogenesis occurring in their hindgut
during the symbiotic metabolic breakdown of lignocellulose (Sanderson, 1996; Brune, 2014). The upscaling of $CH_4$
emissions from termites from site to global level is characterised by high uncertainty (Sanderson, 1996; Kirschke et al.,
2013; Saunois et al., 2016) due to the combination of factors that need to be considered and the scarcity of information for
each of these factors for global upscaling. Needed data include termite biomass density (Sanderson, 1996), species
distribution within and among ecosystems (Sugimoto et al., 1998), variation of termite $CH_4$ emission rates per species and
dietary group (Sanderson, 1996), the role played by the termite mound structure in affecting the fraction of produced $CH_4$
that is effectively released into the atmosphere (Sugimoto et al., 1998; Nauer et al., 2018). In Kirschke et al. (2013) and
Saunois et al. (2016) a global upscaling of termite $CH_4$ emissions was proposed, where $CH_4$ emissions, $E_{CH4}$ (kg $CH_4$ ha$^{-1}$yr$^{-1}$
), were estimated as the product of three terms: termite biomass ($Bio_{TERM}$ g fresh weight m$^{-2}$), a scalar correction factor
(LU) expressing the effect of land use/cover change on termite biomass density, a termite $CH_4$ emission factor ($EF_{TERM}$, μg
$CH_4$ g$^{-1}$ $Bio_{TERM}$ h$^{-1}$) . The approach between the two re-analyses of $CH_4$ emissions varied only for the data sources of gross
primary productivity (GPP) and land use which were used to attribute biomass values of termite per ecosystem surface unit,
in order to cover different time spans, 1980s, 1990s and 2000s in Kirschke et al. (2013) and 2000-2007 and 2010-2016 in
Saunois et al. (2016). For the present update, additional changes have been introduced compared with the previous versions.
Here we summarise the key data used for the new upscaling. $CH_4$ fluxes were modelled between 45°S and 45°N and within
35°S and 35°N. The termite biomass density, $Bio_{TERM}$ for tropical ecosystems was estimated as function (Kirschke et al.,
2013; $Bio_{TERM}=1.21 \cdot e^{0.0008 \cdot GPP}$) of the gross primary production (GPP, g C m$^{-2}$ yr$^{-1}$) using the 0.25° native resolution
VODCA2GPP dataset covering the period 2001-2020 (Wild et al., 2022). Wetlands, barren areas, water bodies and artificial
surfaces were excluded from this estimation and set as no data (no emissions). The scalar correction factor LU of 0.4 (60%)
for agricultural areas (i.e., croplands) (Kirschke et al., 2013) was applied to the GPP value of the nearest natural areas to
account for anthropic disturbance. The annual (2001-2020) land cover information was obtained from the MODIS
Terra+Aqua Combined Land Cover product (MCD12C1v006; https://lpdaac.usgs.gov/products/mcd12c1v006/), using the
International Geosphere-Biosphere Programme (IGP) classification with a 0.05° spatial resolution. For desert and arid lands,
within 35°S and 35°N, a fixed $Bio_{TERM}$ value of 1.56 g m$^{-2}$ was instead used (Sanderson, 1996; Heděnec et al., 2022).



Similarly, fix values from the few available studies reported in literature were used to estimate $Bio_{TERM}$ between 35°- 45° N
and 35°- 45° S as follows: 1.83 g m$^{-2}$ for temperate forests and grasslands (Wood and Sands, 1978; Petersen and Luxton,
1982; Sanderson, 1996; Bignell and Eggleton, 2000; King et al., 2013; conversion factor from dry to fresh biomass is 0.27
from Petersen and Luxton, 1982), 5.3 g m$^{-2}$ for scrublands and Mediterranean areas of Australia (Sanderson, 1996), 1.09 g
m$^{-2}$ for the other Mediterranean shrubland ecosystems (Heděnec et al., 2022). Other climates and land covers were set as no
data. Climate zoning was defined using the Climate Zones Köppen-Geiger dataset (Beck et al., 2018), this product is
representative for the 1980-2016 time period and has a 0.0083° native resolution. The $EF_{TERM}$ was revised compared with
previous estimates (Kirschke et al., 2013; Saunois et al., 2016), in order to consider the different distribution of termite
families and subfamilies in the different continents and ecosystems, characterised by different feeding habits and nest
typologies, as reported by Sugimoto et al. (1998), which might influence the EF. The species of each family and subfamily
of the two major groups of lower and higher termites, listed by Sugimoto et al. (1998) were associated with EF values based
on emissions from in-vitro experiments as reported by Sanderson (1996) and Eggleton et al. (1999), to which a correction
factor ($cf_{MOUND}$) of 0.5 (Nauer et al., 2018) was applied in order to take into account the mound effect on the $CH_4$ produced
by termites, once inside the nest. The average $EF_{TERM}$ for tropical and temperate areas was hence estimated as the weighted
$EF_{TERM}$ derived from the product of the percentage weight of each family or subfamily of termites in the "community
composition" in each geographical area and ecosystem (Sugimoto et al. (1998, Table 6), the respective calculated EF of
each family or subfamily, a scalar or correction factor which considers the nest type (as in Table 5 from Sugimoto et al.
1998). For desert and arid lands and temperate areas, which were not reported in Sugimoto et al. (1998), EF rates were
calculated directly from data reported in literature for the most representative species which were the genus *Amitermes* for
the former (EF from data by Sanderson 1996, Eggleton et al. 1999, Jamali et al. 2011) and the genus *Reticulitermes* (family
Rhinotermitidea) for the latter (EF from data by Odelson and Breznak, 1983; Sanderson, 1996; Eggleton et al., 1999; Myer
et al., 2021). The following $EF_{TERMS}$ were hence obtained to scale up emissions: 3.26 ± 1.79 µg $CH_4$ g$^{-1}$ termite h$^{-1}$ (28.56
mg $CH_4$ g$^{-1}$ termite year$^{-1}$) for tropical ecosystems, 1.82 ± 1.54 µg $CH_4$ g$^{-1}$ termite h$^{-1}$ for temperate forests, grasslands, and
Mediterranean areas, 1.24 ± 1.22 µg $CH_4$ g$^{-1}$ termite h$^{-1}$ for deserts and arid lands (warm climate). Annual $CH_4$ fluxes were
computed for all the years from 2001 to 2020 producing 20 global maps at 0.05° resolution of yearly total emissions. A
further map of the estimated error representative of the entire time period was elaborated at the same resolution as the
emissions dataset.
Termite $CH_4$ emissions over the period 2001-2020 varied between 9.7-10.8 Tg $CH_4$ yr$^{-1}$, with an average of 10.2 ± 6.2 Tg
$CH_4$ yr$^{-1}$. Considering a 20-year average, tropical and subtropical moist broadleaf forests contributed to 46% of the total
average flux, while tropical and subtropical grasslands, savannas, and shrublands to another 36%. In terms of regional
contribution, 37.2% of fluxes were attributed to South America, 31.5% to Africa, 18.1% to Asia, 5.5% to Australia, 7.4%
to North America and less than 1% to Europe. The present estimate value is within the range of previous up-scaling studies,



spanning from 2 to 22 Tg CH$_4$ yr$^{-1}$ (Ciais et al., 2013). In this study, we report a decadal value of 10 Tg CH$_4$ yr$^{-1}$ with a
range of [4-16].

### 3.2.5 Wild animals

Wild ruminants emit CH$_4$ through microbial fermentation that occurs in their rumen, similarly to domesticated livestock
species (USEPA, 2010b). Using a total animal population of 100-500 million, Crutzen et al. (1986) estimated the global
emissions of CH$_4$ from wild ruminants to be in the range of 2-6 Tg CH$_4$ yr$^{-1}$. More recently, Pérez-Barbería (2017) lowered
this estimate to 1.1-2.7 Tg CH$_4$ yr$^{-1}$ using a total animal population estimate of 214 million (range of 210-219), arguing that
the maximum number of animals (500 million) used in Crutzen et al. (1986) was poorly justified. Moreover Pérez-Barbería
(2017) also stated that the value of 15 Tg CH$_4$ yr$^{-1}$ found in the last IPCC reports is much higher than their estimate because
this value comes from an extrapolation of Crutzen's work for the last glacial maximum when the population of wild animals
was much larger, as originally proposed by Chappellaz et al. (1993). Recently, based on the modelling of grassland extent,
Kleinen et al. (2023) also suggest that the population of wild animal during the last glacial maximum proposed by Crutzen
et al. (1986) and further used by Chappellaz et al. (1993) were overestimated.
Based on these findings, the range adopted in this updated CH$_4$ budget is 2 [1-3] Tg CH$_4$ yr$^{-1}$ (Table 3).

### 3.2.6 Coastal and ocean sources

Coastal and oceanic sources comprise CH$_4$ release from estuaries, coastal vegetated habitats, as well as marine waters
including seas and oceans. Possible sources of coastal and oceanic CH$_4$ include (1) in-situ biogenic production through
various pathways in oxygenated sea-surface waters (Oremland, 1979; Karl et al., 2008; Lenhart et al., 2016; Repeta et al.,
2016), a flux that can be enhanced in the coastal ocean because of submarine groundwater discharge (USEPA, 2010b); (2)
production from shallow and marine (bare and vegetated) sediments including free gas or destabilised hydrates and thawing
subsea permafrost containing modern ([14]C-bearing) microbial gas; (3) geological marine seepage (see also Sect. 3.2.3),
including hydrates, containing fossil ([14]C-free) microbial or thermogenic CH$_4$. CH$_4$ produced in marine sediments and
seabed CH$_4$ seepage can be transported across the water column to the sea-surface by upwelling waters (once at the surface
methane can cross the sea-air interface via diffusion) and gas bubble plumes (for instance from geological marine seeps;
e.g., Judd, 2004; Etiope et al., 2019). Gas bubble plumes can generally (but not exclusively, as described below) reach the
atmosphere in relatively shallow waters (<400 m) of continental shelves and coastal zones. In coastal vegetated habitats
CH$_4$ can also be transported to the atmosphere through the aerenchyma of emergent aquatic plants (Purvaja et al., 2004).
We distinguish between coastal and oceanic "geological" and "modern biogenic" CH$_4$ sources. Coastal and oceanic
"geological" emissions refer to CH$_4$ seepage from the Earth's crust (mostly in hydrocarbon-rich sedimentary basins), which
is typically evaluated by combining geochemical analyses (isotopic and molecular, including radiocarbon, [14]C, analyses)
and geological observations (degassing along faults, seeps, mud volcanoes). Geological emissions do not contain modern





biogenic gas that is fossil ([14]C-free). Coastal and oceanic "biogenic" $CH_4$ refers to $CH_4$ formed *in situ* in coastal and marine
sediments and in the water column by recent or modern microbial activity (therefore with measurable amounts of
radiocarbon ([14]C)). To avoid double-counting, we assume that all diffusive $CH_4$ emissions outside of geological seepage
regions (identified in global grid maps; Etiope et al., 2019) are fuelled by biogenic $CH_4$. Finally, we briefly discuss the case
of $CH_4$ hydrates, which can be considered either a "geological" source when they host fossil $CH_4$ or a "biogenic" source
when they host modern $CH_4$.
**Coastal and oceanic modern biogenic methane emissions.** Area-integrated diffusive modern biogenic $CH_4$ emissions
from coastal ecosystems are 1-2 magnitudes lower than from inland freshwaters but significantly higher than biogenic
emissions from the open ocean (Rosentreter et al., 2021; Rosentreter et al., 2023; Weber et al., 2019). Particularly the shallow
vegetated coastline fringed by mangroves, salt marshes, and seagrasses is a $CH_4$ hotspot in the coastal ocean, characterised
by significantly higher flux densities than other coastal settings such as estuaries or the continental shelves (Rosentreter et
al., 2021; Rosentreter et al., 2023). Coastal ecosystems are thus being increasingly recognized as weak global sources to the
atmosphere (Weber et al., 2019; Saunois et al., 2020; Rosentreter et al., 2021). Hydrogenotrophic and acetoclastic
methanogenesis are largely outcompeted by sulphate reduction in coastal/marine sediments, which is often shown by a
decreasing trend of $CH_4$ concentrations with increasing salinity from upper tidal (low salinity) to marine (high salinity)
regions. Much of the $CH_4$ produced below the sulfate-reduction zone is indeed re-oxidized by sedimentary anaerobic
methane oxidation or re-oxidized in the water column, leading to small emissions despite much larger production (Knittel
and Boetius 2009; Regnier et al., 2011). Methylated compounds such as methylamines and methyl sulphides are non-
competitive substrates that are exclusively used by methanogens, therefore methylated methanogenesis can occur in coastal
regions with high sulphate concentrations, for example, in organic-rich (Maltby et al., 2018), vegetated (Schorn et al., 2022),
and hypersaline coastal sediments (Xiao et al., 2018). Coastal $CH_4$ can be driven by the exchange of pore water or
groundwater (high in $CH_4$) with coastal surface waters in tidal systems, referred to as tidal pumping (Ovalle et al., 1990;
Call et al., 2015). Anthropogenic impacts such as wastewater pollution and land-use change can increase $CH_4$ fluxes in
estuaries (Wells et al., 2020). A large increase of $CH_4$ emissions follows the conversion of natural coastal habitats to
aquaculture farms (Yuan et al., 2019; Yang et al., 2022).
Currently available global modern biogenic $CH_4$ flux data show high spatiotemporal variability within and between coastal
systems, but also because of the overall global paucity of data. Therefore, global estimates have high uncertainties and show
large ranges in both empirical (Rosentreter et al., 2021) and machine-learning based approaches (Weber et al., 2019).
According to a recent data-driven meta-data analysis, global estuaries, including tidal systems and deltas, lagoons, and
fjords, are estimated to emit median (Q1-Q3) 0.25 (0.07-0.46) Tg $CH_4$ yr[-1] (Rosentreter et al., 2023). Coastal vegetation,
including mangrove forests, salt marshes, and seagrasses are estimated to emit 0.77 (0.47-1.41) Tg $CH_4$ yr[-1], which is 3
times more than global estuaries (Rosentreter et al., 2023). The combined median (Q1-Q3) emission of 1.01 (0.54-1.87) Tg
$CH_4$ yr[-1] for coastal vegetation and estuaries by Rosentreter et al. (2023) is lower than the recent observation-based global





synthesis including tidal flats and aquaculture ponds (median 1.49 (0.22-6.48) Tg $CH_4$ $yr^{-1}$) by Rosentreter et al. (2021).
Total shallow coastal modern biogenic $CH_4$ emissions based on existing data including emissions from estuaries, coastal
vegetation (Rosentreter et al., 2023), tidal flats, and man-made coastal aquaculture ponds (Rosentreter et al., 2021) amount
to median (Q1-Q3) 1.8 (0.59-5.57) Tg $CH_4$ $yr^{-1}$. This range is about 3-4 times lower than the earlier global assessment by
Borges and Abril (2011) and also lower than the value of 4-5 Tg $CH_4$ $yr^{-1}$reported in the previous $CH_4$ budget for inner and
outer estuaries including marshes and mangroves (Saunois et al., 2020), which was based on a significantly smaller dataset
(n=80) and larger estuarine surface areas (Laruelle et al., 2013) than used here (Laruelle et al., 2023).
The near-shore (0-50 m), inner shelf diffusive modern biogenic $CH_4$ flux of median (Q1-Q3) 1.33 (0.93-2.10) Tg $CH_4$ $yr^{-1}$
by Weber et al. (2019) based on machine-learning is similar to the combined shallow coastal (estuaries and coastal
vegetation) median by Rosentreter et al. (2021, 2023). Adding the diffusive modern biogenic $CH_4$ flux for the outer shelf
(50-200 m) (median (Q1-Q3) of 0.54 (0.40-0.73) Tg $CH_4$ $yr^{-1}$) and for the slope (200-2000m) (median (Q1-Q3) of 0.28
(0.22-0.37) Tg $CH_4$ $yr^{-1}$) (Weber et al., 2019), and excluding geological seepage regions (Etiope et al., 2019; see below),
gives a total median (Q1-Q3) of 3.95 (2.14-8.77) Tg $CH_4$ $yr^{-1}$ for combined coastal shallow, near-shore, outer shelf and
slope diffusive modern biogenic $CH_4$ emissions. The previous budget by Saunois (2020) also included poorly constrained
emissions (upper bound value: 1-2 Tg $CH_4$ $yr^{-1}$) from large river plumes protruding onto the shelves. However, here we
assume that emissions from large river plumes are accounted for in the near-shore and outer shelf estimates by Weber et al.
(2019). Area-integrated diffusive $CH_4$ emissions from the open ocean and deep seas (>2000 m) are much lower than from
other coastal systems but amount to median (Q1-Q3) 0.91 (0.75-1.12) Tg $CH_4$ $yr^{-1}$ because of the large surface area of the
open ocean (>300 x $10^6$ $km^2$) (Weber et al., 2019). Overall, these marine biogenic emissions are sustained by a mixture of
sedimentary production and in-situ production in the sea-surface layers, as shown by, e.g., Karl et al. (2008) and Repeta et
al. (2016). The total coastal and ocean diffusive modern biogenic emissions retained here amount to 5 (3-10) Tg $CH_4$ $yr^{-1}$.

**Coastal and oceanic geological methane emissions** Submarine geological $CH_4$ emission is the offshore component of the
general geological emissions of natural gas from the Earth's crust (Judd, 2004; Etiope, 2009; Etiope et al., 2019). The
onshore components include terrestrial seeps, mud volcanoes, microseepage, and geothermal manifestations, addressed in
Sect. 3.2.3**.** Natural gas seeping at the seabed as bubble plumes can reach the surface, generally occurs in relatively shallow
waters (<400 m), but $CH_4$-rich bubble plumes reaching the atmosphere from depths >500 m have been observed in some
cases (e.g., Solomon et al., 2009), and upwelling of bottom marine waters can, in theory, transport geological
$CH_4$ (dissolved) to the surface from any depth. This represents, however, a small and poorly known fraction of geological
$CH_4$ emission. Geological $CH_4$ can be either microbial or thermogenic, produced throughout diverse geological periods in
hydrocarbon source rocks in sedimentary basins (therefore it is always fossil, $^{14}C$-free). The seepage at the seafloor is
typically related to tectonic faults, sometimes forming mud diapirs and mud volcanoes (Mazzini and Etiope, 2017).





Published estimates of geological $CH_4$ submarine emissions range from 3 to 20 Tg yr$^{-1}$, with a best guess of 7 Tg yr$^{-1}$ (Etiope
and Schwietzke, 2019; Etiope et al., 2019 and references therein).
Here, the diffusive geological $CH_4$ emissions are estimated at 0.16 (0.11-0.24) Tg $CH_4$ yr$^{-1}$ for near-shore (0-50 m), 0.03
(0.02-0.05) Tg $CH_4$ yr$^{-1}$ for outer shelf (50-200 m), and 0.02 (0.01-0.03) Tg $CH_4$ yr$^{-1}$ for slope (200-2000 m) by calculating
the fraction of the Weber et al. (2019) diffusive fluxes that occur within the identified geological seepage regions from
Etiope et al. (2019). No geological seepage regions were identified in the open ocean and deep seas (> 2000 m).
In this study, we consider the ebullitive flux as geologically sourced $CH_4$. While modern biogenic $CH_4$ gas production
appears ubiquitous in shallow sediments (Fleischer et al., 2001; Best et al., 2006), no global dataset is currently available to
estimate the biogenic ebullitive $CH_4$ flux to the atmosphere. Omission of this flux thus constitutes a significant knowledge
gap in the coastal and oceanic $CH_4$ budget. Global geological $CH_4$ ebullition from continental shelf and slope, referring only
to depths <200 m, were estimated at 5.06 (1.99-8.16) Tg $CH_4$ yr$^{-1}$ (Weber et al., 2019). This estimate is based on prior
estimates of the geological flux from the seafloor (Hovland et al., 1993) and bubble transfer efficiency to the ocean surface
(McGinnis et al., 2006). Etiope et al. (2019) estimated a partial fraction of geological emissions in the form of gas bubbles
of 3.9 (1.8-6) Tg $CH_4$ yr$^{-1}$, only referring to the sum of published estimates from 15 geological seepage regions, which are
also deeper than 200 m. Global extrapolation including other 16 identified seepage zones (where flux data are not available)
was suggested to be at least 7 (3-10) Tg $CH_4$ yr$^{-1}$ (Etiope et al., 2019), and this value coincides with the mean emission value
(best guess) derived by combining literature data, see Etiope and Schwietzke (2019) for further details. It is worth noting
that the Weber et al. (2019) estimate of 5.06 (1.99-8.16) Tg $CH_4$ yr$^{-1}$, which considers only the continental shelf at depths
<200 m, is compatible with the overall submarine emission of 7 (3-10) Tg $CH_4$ yr$^{-1}$ (including seeps > 200 m deep) indicated
in Etiope and Schwietzke (2019) and Etiope et al. (2019). Although 300-400 m is considered a general depth limit for
efficient transport (with limited oxidation and dissolution) of $CH_4$ bubbles to the atmosphere (e.g., Judd, 2004; Schmale et
al., 2005; Etiope et al., 2019), in some cases oil coatings on bubbles inhibit gas dissolution so that $CH_4$-rich bubbles can
reach the atmosphere from depths >500 m (e.g., Solomon et al., 2009). As mentioned above, a fraction of geological $CH_4$
released in deep seas (such as in the areas with gas-charged sediments inventoried in Fleischer et al., 2001) can also be
transported to the surface by upwelling bottom waters. Further research is needed to better evaluate the atmospheric impact
of such deep seeps.
Geological submarine emissions, thus, would amount to 0.21 (0.14-0.32) Tg $CH_4$ yr$^{-1}$ in the form of a diffusive flux while
the ebullitive flux would be 5.06 (3.01-7.88) Tg $CH_4$ yr$^{-1}$, considering only < 200 m deep seepage, and 7 (3-10) Tg $CH_4$ yr$^{-}$
$^1$ considering all data available (Etiope and Schwietzke, 2019). Here, we select the Etiope and Schwietzke (2019) assessment
in order to account for all potential seepage areas, including those located at water depths > 200m.

As a result, here we report a (rounded) median of 12 Tg $CH_4$ yr$^{-1}$ with a range of 6-20 Tg $CH_4$ yr$^{-1}$ for all coastal and ocean
sources.





**Methane emissions from gas hydrates.** Among the different origins of coastal and oceanic $CH_4$, hydrates have attracted a lot of attention. $CH_4$ hydrates (or clathrates) are ice-like crystals formed under specific temperature and pressure conditions (Milkov, 2005). Hydrates may host either modern microbial $CH_4$, containing $^{14}C$ and formed *in situ* in shallow sediments (this type of hydrates is also called "autochthonous") or fossil, microbial or thermogenic $CH_4$, migrated from deeper sediments, generally from reservoirs in hydrocarbon-rich sedimentary basins (this type of hydrates is also called "allochthonous"; Milkov, 2005; Foschi et al., 2023). The total stock of marine $CH_4$ hydrates is large but uncertain, with global estimates ranging from hundreds to thousands of Pg $CH_4$ (Klauda and Sandler, 2005; Wallmann et al., 2012). Note that the highly climate-sensitive subsea permafrost reservoir beneath Arctic Ocean shelves also contributes to the hydrate inventory (Ruppel and Kassler, 2017).

Concerning more specifically atmospheric emissions from marine hydrates, Etiope (2015) points out that current estimates of $CH_4$ air–sea flux from hydrates (2–10 Tg $CH_4$ $yr^{-1}$ in Ciais et al., 2013, or Kirschke et al., 2013) originate from the hypothetical values of Cicerone and Oremland (1988). No experimental data or estimation procedures have been explicitly described along the chain of references since then (Denman et al., 2007; IPCC, 2001; Kirschke et al., 2013; Lelieveld et al., 1998). It was estimated that ∼473 Tg $CH_4$ has been released into the water column over 100 years (Kretschmer et al., 2015). Those few teragrams per year become negligible once consumption within the water column has been accounted for. While events such as submarine slumps may trigger local releases of considerable amounts of $CH_4$ from hydrates that may reach the atmosphere (Etiope, 2015; Paull et al., 2002), on a global scale, present-day atmospheric $CH_4$ emissions from hydrates do not appear to be a significant source to the atmosphere, and at least formally, we should consider 0 (< 0.1) Tg $CH_4$ $yr^{-1}$ emissions.

### 3.2.7 Terrestrial permafrost

Permafrost is defined as frozen soil, sediment, or rock having temperatures at or below 0°C for at least two consecutive years (Harris et al., 1988). The total extent of permafrost in the Northern Hemisphere is about 14 million $km^2$ or 15% of the exposed land surface (Obu et al., 2019). As the climate warms, a rise in soil temperatures has been observed across the permafrost region, and permafrost thaw occurs when temperatures pass 0°C, often associated with melting of ice in the ground (Biskaborn et al., 2019). Permafrost thaw is most pronounced in southern and spatially isolated permafrost zones, but also occurs in northern continuous permafrost (Obu et al., 2019). Thaw occurs either as a gradual, often widespread, deepening of the active layer (surface soils that thaw every summer) or as more rapid localised thaw associated with loss of massive ground ice (thermokarst) (Turetsky et al., 2020). A total of 1000 ± 200 Pg of carbon can be found in the upper 3 meters of permafrost region soils, or 1400-2000 Pg C for all permafrost (Hugelius et al., 2014; Strauss et al., 2021 ).





The thawing permafrost can generate direct and indirect $CH_4$ emissions. Direct $CH_4$ emissions are from the release of $CH_4$ contained within the thawing permafrost. This flux to the atmosphere is small and estimated to be a maximum of 1 Tg $CH_4$ yr$^{-1}$ at present (USEPA, 2010b). Increased seepage of geogenic $CH_4$ gas seeps along permafrost boundaries and lake beds may also be considered a direct flux, and this is estimated to be 2±0.4 Tg $CH_4$ yr$^{-1}$ (Walter Anthony et al., 2012). Indirect $CH_4$ emissions are probably more important. They are caused by 1) methanogenesis induced when the organic matter contained in thawing permafrost becomes available for microbial decomposition; 2) thaw induced soil wetting and changes in land surface hydrology possibly enhancing $CH_4$ production (McCalley et al., 2014; Schuur et al., 2022); and 3) the landscape topography changes driven by abrupt thaw processes and loss of ground ice, including the formation of thermokarst lakes, hill-slope thermokarst, and wetland thermokarst (Turetsky et al., 2020). Such $CH_4$ production is probably already significant today and is likely to become more important in the future associated with climate change and strong positive feedback from thawing permafrost (Schuur et al., 2022). However, indirect $CH_4$ emissions from permafrost thawing are difficult to estimate at present, with very few data to refer to, and in any case largely overlap with wetland and freshwater emissions occurring above or around thawing areas. In a recent synthesis of full permafrost region $CH_4$ budgets for the period 2000-2017, Hugelius et al. (2023) compared $CH_4$ budgets from bottom-up and top-down (atmospheric inversion models) approaches. They estimate an integrated bottom-up budget of 50 (23, 53; mean upper and lower 95% CI) Tg $CH_4$ yr$^{-1}$ while the top-down estimate is 19 (15, 24) Tg $CH_4$ yr$^{-1}$. The bottom-up estimate is based on a combination of data-driven upscaling reported by Ramage et al. (2023) and process-based model estimates for wetland $CH_4$ flux calculated from model ensembles used in Saunois et al. (2020). The top-down estimate is calculated from ensembles of atmospheric inversion models used in Saunois et al. (2020). Although it is difficult with direct process-attribution, fluxes of ca. 20-30 Tg $CH_4$ yr$^{-1}$ in the bottom-up budget are caused by land cover types affected by previous permafrost thaw (thermokarst lakes, wetlands, hillslope). Because pre-thaw land cover types often have near neutral $CH_4$ balances (Ramage et al. 2023), these fluxes can largely be seen as driven by permafrost thaw, however the thaw may have occurred decades, or even centuries, before today.

Here, we choose to report only the direct emission range of 0-1 Tg $CH_4$ yr$^{-1}$, keeping in mind that current wetland, thermokarst lakes and other freshwater methane emissions already likely include a significant indirect contribution originating from thawing permafrost.

### 3.2.8 Vegetation

Three distinct pathways for the production and emission of $CH_4$ by living vegetation are considered here (see Covey and Megonigal (2019) and Bastviken et al. (2023) for extensive reviews). Firstly, plants produce $CH_4$ through an abiotic photochemical process induced by stress (Keppler et al., 2006). This pathway was initially questioned (e.g., Dueck et al., 2007; Nisbet et al., 2009), and although numerous studies have since confirmed aerobic emissions from plants and better resolved its physical drivers (Fraser et al., 2015), global estimates still vary by two orders of magnitude (Liu et al., 2015).



This plant source has not been confirmed in-field however, and although the potential implication for the global CH$_4$ budget remains unclear, emissions from this source are certainly much smaller than originally estimated in Keppler et al. (2006) (Bloom et al., 2010; Fraser et al., 2015). Second, and of clearer significance, plants act as "straws", drawing up and releasing microbially produced CH$_4$ from anoxic soils (Cicerone and Shetter, 1981; Rice et al., 2010). For instance, in the forested wetlands of Amazonia, tree stems are the dominant ecosystem flux pathway for soil-produced CH$_4$, therefore, including stem emissions in ecosystem budgets can reconcile regional bottom-up and top-down estimates (Pangala et al., 2017; Gauci et al., 2021). Third, the stems of both living trees (Covey et al., 2012) and dead wood (Covey et al., 2016) provide an environment suitable for microbial methanogenesis. Static chambers demonstrate locally significant through-bark flux from both soil- (Pangala et al., 2013, 2015), and tree stem-based methanogens (Pitz and Megonigal, 2017; Wang et al., 2016). A recent synthesis indicates stem CH$_4$ emissions significantly increase the source strength of forested wetlands, and modestly decrease the sink strength of upland forests (Covey and Megonigal, 2019). The scientific activity covering CH$_4$ emissions in forested ecosystems reveals a far more complex story than previously thought, with an interplay of productive/consumptive, aerobic/anaerobic, and biotic/abiotic processes occurring between upland/wetland soils, trees, and atmosphere. Understanding the complex processes that regulate CH$_4$ source–sink dynamics in forests and estimating their contribution to the global CH$_4$ budget requires cross-disciplinary research, more observations, and new models that can overcome the classical binary classifications of wetland versus upland forest and of emitting versus uptaking soils (Barba et al., 2019; Covey and Megonigal, 2019). Although we recognize these emissions are potentially large (particularly tree transport from inundated soil), global estimates for each of these pathways remain highly uncertain and/or are currently included here within other flux category sources (e/g. inland waters, wetlands, upland soils).

**3.3 Methane sinks and lifetime**

CH$_4$ is the most abundant reactive trace gas in the troposphere and its reactivity is important to both tropospheric and stratospheric chemistry. The main atmospheric sink of CH$_4$ (~90% of the total sink mechanism) is oxidation by the hydroxyl radical (OH), mostly in the troposphere (Ehhalt, 1974). Other losses are by photochemistry in the stratosphere (reactions with chlorine atoms (Cl) and excited atomic oxygen (O($^1$D)), oxidation in soils (Curry, 2007; Dutaur and Verchot, 2007), and by photochemistry in the marine boundary layer (reaction with Cl; Allan et al. (2007), Thornton et al. (2010)). Uncertainties in the total sink of CH$_4$ as estimated by atmospheric chemistry models are in the order of 20-40% (Saunois et al., 2016). It is much less (10-20%) when using atmospheric proxy methods (e.g., methyl chloroform, see below) as in atmospheric inversions (Saunois et al., 2016). In the present release of the global CH$_4$ budget, we estimate bottom-up CH$_4$ chemical sinks and lifetime mainly based on global model results from the Chemistry Climate Model Initiative (CCMI) 2022 activity (Plummer et al., 2021) and CMIP6 simulations (Collins et al., 2017).



### 3.3.1 Tropospheric OH oxidation

OH radicals are produced following the photolysis of ozone ($O_3$) in the presence of water vapour. OH is destroyed by reactions with carbon monoxide (CO), $CH_4$, and non-methane volatile organic compounds.

Following the Atmospheric Chemistry and Climate Model Intercomparison Project (ACCMIP), which studied the long-term changes in atmospheric composition between 1850 and 2100 (Lamarque et al., 2013), a new series of experiments was conducted by several chemistry-climate models and chemistry-transport models participating in the Chemistry-Climate Model Initiative (CCMI) (Plummer et al., 2021). Mass-weighted OH tropospheric concentrations do not directly represent $CH_4$ loss, as the spatial and vertical distributions of OH affect this loss through, in particular, the temperature dependency and the distribution of $CH_4$ (e.g., Zhao et al., 2019). However, estimating OH concentrations and, spatial and vertical distributions is a key step in estimating methane loss through OH. Over the period 2000-2010, the global mass-weighted OH tropospheric concentration is estimated at 13.3 [11.7-18.2] x $10^5$ molecules $cm^{-3}$ by 8 CCMI-2022 models and at 11.5 [7.9-13.5] x $10^5$ molecules $cm^{-3}$ by 10 models contributing CMIP6 (see supplementary Table S4). The ranges calculated here are larger than the ones proposed previously in Saunois et al. (2020), where the multi-model mean (11 models) global mass-weighted OH tropospheric concentration was 11.7±1.0 x $10^5$ molecules $cm^{-3}$ (range 9.9-14.4 x $10^5$ molecules $cm^{-3}$, Zhao et al. (2019)) consistent with the previous estimates from ACCMIP (11.7±1.0 x $10^5$ molecules $cm^{-3}$, with a range of 10.3-13.4 x $10^5$ molecules $cm^{-3}$, Voulgarakis et al. (2013) for year 2000) and the estimates of Prather et al. (2012) of 11.2±1.3 x $10^5$ molecules $cm^{-3}$. Nicely et al. (2017) attribute the differences in OH simulated by different chemistry transport models to, in decreasing order of importance, different chemical mechanisms, various treatments of the photolysis rate of $O_3$, and modelled $O_3$ and CO. Besides the uncertainty on global OH concentrations, there is an uncertainty in the spatial and temporal distribution of OH. Models often simulate higher OH in the northern hemisphere (NH) than in the southern hemisphere (SH), leading to a NH/SH OH ratio greater than 1 (Naik et al., 2013; Zhao et al., 2019). However, there is evidence for parity in inter-hemispheric OH concentrations (Patra et al., 2014), which needs to be confirmed by other observational and model-derived estimates. The analysis of the latest CCMI (Plummer et al., 2021) and CMIP6 (Collins et al., 2021) model outputs show that structural uncertainties in the atmospheric chemistry models remain large, probably due to inherent biases in OH precursors. Based on OH precursor observations and a chemical box model, Zhao et al. (2023) corrected the OH 3D fields simulated by two atmospheric chemistry models, resulting in tropospheric OH mean concentrations lowered by 2. $10^5$ molecules $cm^{-3}$, leading to around 10 x $10^5$ molecules $cm^{-3}$, and a NH/SH OH ratio closer to 1, in better agreement with methyl chloroform (MCF)-based approaches. This study highlights the need for further improvement of the atmospheric chemistry model.

OH concentrations and their changes can be sensitive to climate variability (Dlugokencky et al., 1996; Holmes et al., 2013; Turner et al., 2018), biomass burning (Voulgarakis et al., 2015), and anthropogenic activities. For instance, the increase of the oxidizing capacity of the troposphere in South and East Asia associated with increasing $NO_X$ emissions (Mijling et al., 2013) and decreasing CO emissions (Yin et al., 2015), possibly enhances $CH_4$ oxidation and therefore limits the atmospheric





impact of increasing emissions (Dalsøren et al., 2009). Despite such large regional changes, the global mean OH
concentration was suggested to have changed only slightly over the past 150 years (Naik et al., 2013). This is due to the
compensating effects of the concurrent increases of positive influences on OH (water vapour, tropospheric ozone, nitrogen
oxides ($NO_X$) emissions, and UV radiation due to decreasing stratospheric $O_3$), and of OH sinks ($CH_4$ burden, CO and non-
$CH_4$ volatile organic compound emissions and burden). CCMI models show OH inter-annual variability ranging from 0.4%
to 1.8% (Zhao et al., 2019) over 2000-2010 (similar values are derived in the latest CCMI and CMIP6 activities - see
supplementary Table S4), lower than the value deduced from methyl chloroform measurements (proxy, top-down approach).
However, these simulations consider meteorology variability but not emission interannual variability (e.g., from biomass
burning) and thus are expected to simulate lower OH inter annual variability than in reality. Using an empirical model
constrained by global observations of $O_3$, water vapour, $CH_4$, and temperature as well as the simulated effects of changing
$NO_X$ emissions and tropical expansion, Nicely et al. (2017) found an inter-annual variability in OH of about 1.3-1.6%
between 1980 and 2015, in agreement with methyl chloroform based estimates (Montzka et al., 2011).
Over 2000-2009, the tropospheric loss (tropopause height at 200 hPa) of $CH_4$ by OH oxidation derived from the ten and
CCMI modelling activities (see supplementary Table S5) is estimated at of 546 [446-663] Tg $CH_4$ yr$^{-1}$, which is similar to
the one reported previously in Saunois et al. (2020) from CCMI model (553 [476-677] Tg $CH_4$ yr$^{-1}$) and still slightly higher
than the one from the ACCMIP models (528 [454-617] Tg $CH_4$ yr$^{-1}$ reported in Kirschke et al. (2013) and Saunois et al.
(2016).
For the recent 2010-2019 decade, we report a climatological value based on five models that contributed to CMIP6 runs
(historic followed by SSP3-7.0 projections starting in 2015, Collins et al. (2021)) to acknowledge the impact of the rise in
atmospheric methane on the methane chemical sink. Hence, for 2010-2019, we report the climatological value of 563 [510-
663] Tg $CH_4$ yr$^{-1}$
**3.3.2 Stratospheric loss**
In the stratosphere, $CH_4$ is lost through reactions with excited atomic oxygen $O(^1D)$, atomic chlorine (Cl), atomic fluorine
(F), and OH (Brasseur and Solomon, 2005; le Texier et al., 1988). Uncertainties in the chemical loss of stratospheric $CH_4$ are
large, due to uncertain inter-annual variability in stratospheric transport as well as its chemical interactions and feedbacks
with stratospheric $O_3$ (Portmann et al., 2012). Particularly, the fraction of stratospheric loss due to the different oxidants is
still uncertain, with possibly 20-35% due to halons, about 25% due to $O(^1D)$ mostly in the high stratosphere and the rest due
to stratospheric OH (McCarthy et al., 2003).
In this study, ten chemistry climate models that contributed to CMIP6 and CCMI modelling activities (Table S5) are used
to provide estimates of $CH_4$ chemical loss, including reactions with OH, $O(^1D)$, and Cl; $CH_4$ photolysis is also included but
occurs only above the stratosphere. Considering a 200 hPa tropopause height, the CMIP6 and CCMI results suggest an





estimate of 34 [10-51]Tg CH$_4$ yr$^{-1}$ for the CH$_4$ stratospheric sink for the 2000-2009 decade (Table S5), similar to the value
derived from the previous CCMI activity reported in Saunois et al. (2020) (31 [12-41] Tg CH$_4$ yr$^{-1}$).

For 2010-2019, we report here a climatological range of 11-43 Tg CH$_4$ yr$^{-1}$ associated with a mean value of 33 Tg CH$_4$ yr$^{-1}$
based on five models that contributed to CMIP6 runs (historic followed by SSP3-7.0 projections starting in 2015; Table S5).

### 3.3.3 Tropospheric reaction with Cl

Halogen atoms can also contribute to the oxidation of CH$_4$ in the troposphere. Allan et al. (2005) measured mixing ratios of
methane and $\delta^{13}$C-CH$_4$ at two stations in the southern hemisphere from 1991 to 2003, and found that the apparent kinetic
isotope effect (KIE) of the atmospheric CH$_4$ sink was significantly larger than that explained by OH alone. A seasonally
varying sink due to Cl in the marine boundary layer of between 13 and 37 Tg CH$_4$ yr$^{-1}$ was proposed as the explanatory
mechanism (Allan et al., 2007; Platt et al., 2004). This sink was estimated to occur mainly over coastal and marine regions,
where sodium chloride (NaCl) from evaporated droplets of seawater react with NO$_2$ to eventually form Cl$_2$, which then UV-
dissociates to Cl. However significant production of nitryl chloride (ClNO$_2$) at continental sites has been recently reported
(Riedel et al., 2014) and suggests the broader presence of Cl, which in turn would expand the significance of the Cl sink in
the troposphere. Recently, Hossaini et al. (2016), Sherwen et al. (2016), and Wang et al. (2019b, 2021b) have made
significant improvements in tropospheric chemistry modelling and they conclude to an oxidation contribution of 2.6%, 2%,
1% and 0.8%, respectively. These values correspond to a tropospheric CH$_4$ loss of around 12-13 Tg CH$_4$ yr$^{-1}$, 9 Tg CH$_4$ yr$^{-1}$, 5 Tg yr$^{-1}$, and 3 Tg CH$_4$ yr$^{-1}$ respectively, much lower than the first estimates by Allan et al. (2007). The recent work of
Wang et al. (2021b) is the most comprehensive modelling study and based upon Sherwen et al. (2016) and Wang et al.
(2019b). Both the KIE approach and chemistry transport model simulations carry uncertainties (extrapolations based on
only a few sites and use of indirect measurements, for the former and missing sources, coarse resolution, underestimation
of some anthropogenic sources for the latter). However, Gromov et al. (2018) found that Cl can contribute only 0.23% the
tropospheric sink of CH$_4$ (about 1 Tg CH$_4$ yr$^{-1}$) in order to balance the global $^{13}$C(CO) budget (see their Table S1). While
tropospheric Cl has a marginal impact on the total CH$_4$ sink (few percents), it influences more significantly the atmospheric
isotopic $\delta^{13}$C-CH$_4$ signal and improved estimates of the tropospheric Cl amount should be used for isotopic CH$_4$ modelling
studies (Strode et al., 2020; Thanwerdas et al., 2022b).

Each recent Cl estimate suggests a reduced contribution to the methane loss than previously reported by Allen et al. (2007).
As a result, we suggest here to use the mean, minimum and maximum of the last five estimates published since 2016, leading
to a climatological value of 6 [1-13] Tg CH$_4$ yr$^{-1}$, thus reducing both the magnitude and the uncertainty range compared to
Saunois et al. (2020).





### 3.3.4 Soil uptake

Unsaturated oxic soils are sinks of atmospheric $CH_4$ due to the presence of methanotrophic bacteria, which consume $CH_4$ as a source of energy. Dutaur and Verchot (2007) conducted a comprehensive meta-analysis of field measurements of $CH_4$ uptake spanning a variety of ecosystems. Extrapolating to the global scale, they reported a range of $36 \pm 23$ Tg $CH_4$ yr$^{-1}$, but also showed that stratifying the results by climatic zone, ecosystem, and soil type led to a narrower range (and lower mean estimate) of $22 \pm 12$ Tg $CH_4$ yr$^{-1}$. Modelling studies, employing meteorological data as external forcing, have also produced a considerable range of estimates. Using a soil depth-averaged formulation based on Fick's law with parameterizations for diffusion and biological oxidation of $CH_4$, Ridgwell et al. (1999) estimated the global sink strength at 38 Tg $CH_4$ yr$^{-1}$, with a range 20-51 Tg $CH_4$ yr$^{-1}$ reflecting the model structural uncertainty in the base oxidation parameter. Curry (2007) improved on the latter by employing an exact solution of the one-dimensional diffusion-reaction equation in the near-surface soil layer (i.e., exponential decrease in $CH_4$ concentration below the surface), a land surface hydrology model, and calibration of the oxidation rate to field measurements. This resulted in a global estimate of 28 Tg $CH_4$ yr$^{-1}$ (9-47 Tg $CH_4$ yr$^{-1}$), the result reported by Zhuang et al. (2013), Kirschke et al. (2013) and Saunois et al. (2016). Ito and Inatomi (2012) used an ensemble methodology to explore the variation in estimates produced by these parameterizations and others, which spanned the range 25-35 Tg $CH_4$ yr$^{-1}$. For the period 2000-2020, as part of the wetland emissions modelling activity, JSBACH (Kleinen et al., 2020) and VISIT (Ito and Inatomi, 2012) models compute a global $CH_4$ soil uptake to 18 and 35 Tg $CH_4$ yr$^{-1}$, respectively. Murguia-Flores et al. (2018) further refined the Curry (2007) model's structural and parametric representations of key drivers of soil methanotrophy, demonstrating good agreement with the observed latitudinal distribution of soil uptake (Dutaur and Verchot, 2007). Their model (MeMo) simulates a $CH_4$ soil sink of 37.5 Tg $CH_4$ yr$^{-1}$ for the period 2010-2019 (Fig. S4), compared to 39.5 and 31.3 Tg $CH_4$ yr$^{-1}$ using the Ridgwell et al. (1999) and Curry (2007) parameterizations, respectively, under the same meteorological forcing, run specifically for this study. For the 2000s period, the simulations estimate the soil uptake at 30.4, 36.7 and 38.3 Tg $CH_4$ yr$^{-1}$ based on the parameterization of Curry, MeMo, and Ridgwell, respectively. As part of a more comprehensive model accounting for a range of $CH_4$ sources and sinks, Tian et al. (2010, 2015, 2016) computed vertically-averaged $CH_4$ soil uptake including the additional mechanisms of aqueous diffusion and plant-mediated (*aerenchyma*) transport, arriving at the estimate $30\pm19$ Tg $CH_4$ yr$^{-1}$ (Tian et al., 2016) for the 2000s. The still more comprehensive biogeochemical model of Riley et al. (2011) included vertically resolved representations of the same processes considered by Tian et al. (2016), in addition to grid cell fractional inundation and, importantly, the joint limitation of uptake by both $CH_4$ and $O_2$ availability in the soil column. Riley et al. (2011) estimated a global $CH_4$ soil sink of 31 Tg $CH_4$ yr$^{-1}$ with a structural uncertainty of 15-38 Tg $CH_4$ yr$^{-1}$ (a higher upper limit resulted from an elevated gas diffusivity to mimic convective transport; as this is not usually considered, we adopt the lower upper bound associated with no limitation of uptake at low soil moisture). A model of this degree of complexity is required to explicitly simulate situations where the soil water content increases enough to inhibit the diffusion of oxygen, and the soil becomes a methane source



(Lohila et al., 2016). This transition can be rapid, thus creating areas (for example, seasonal wetlands) that can be either a
source or a sink of methane depending on the season.
The previous Curry (2007) estimate can be revised upward slightly based on subsequent work and the increase in $CH_4$
concentration since that time. Considering the latest estimates (based on VISIT, JSBACH, and Memo models, Table S6 in
the supplementary) we report here a mean estimate of 31 [17-39] Tg $CH_4$ $yr^{-1}$ for 2000-2009 and 32 [18-40] for 2010-2019
Tg $CH_4$ $yr^{-1}$.

### 1545    3.3.5 $CH_4$ lifetime

The atmospheric lifetime of a given gas in steady state may be defined as the global atmospheric burden (Tg) divided by the
total sink (Tg $yr^{-1}$) (IPCC, 2001). Global models provide an estimate of the loss of the gas due to individual sinks, which
can then be used to derive lifetime due to a specific sink. For example, the tropospheric lifetime of $CH_4$ is determined as the
global atmospheric $CH_4$ burden divided by the loss from OH oxidation in the troposphere, sometimes called "chemical
lifetime". The total lifetime of $CH_4$ corresponds to the global burden divided by the total loss including tropospheric loss
from OH oxidation, stratospheric chemistry and soil uptake. The CCMI (Plummer et al., 2021) and CMIP6 (Collins et al.,
2021) runs estimate the tropospheric methane lifetime at about 9.2 years (average over years 2000-2009), with a range of
7.5-11 years (see Table S5). This range agrees with previous values found in ACCMIP and CCMI (9.3 [7.1-10.6] years,
Voulgarakis et al. (2013), 9 [7.2-10.1] years, Saunois et al. (2020)). Adding 31 Tg to account for the soil uptake to the total
chemical loss of the CMIP6 and CCMI models, we derive a total $CH_4$ lifetime of 8.2 years (average over 2000-2009 with a
range of 6.8-9.7 years). The lifetime calculated over 2010-2019 based on CMIP6 simulations is similar (Table S5). These
updated model estimates of total $CH_4$ lifetime agree with the previous estimates from ACCMIP (8.2 [6.4-9.2] years for year
2000, Voulgarakis et al. (2013)) and Saunois et al. (2020) based CCMI models. Reducing the large spread in $CH_4$ lifetime
(between models, and between models and observation-based estimates) would 1) bring an improved constraint on global
total methane emissions, and 2) ensure an accurate forecast of future climate.

### 1561    4 Atmospheric observations and top-down inversions

### 1562    4.1 Atmospheric observations

Systematic atmospheric $CH_4$ observations began in 1978 (Blake et al., 1982) with infrequent measurements from discrete
air samples collected in the Pacific at a range of latitudes from 67°N to 53°S. Because most of these air samples were from
well-mixed oceanic air masses and the measurement technique was precise and accurate, they were sufficient to establish
an increasing trend and the first indication of the latitudinal gradient of methane. Spatial and temporal coverage was greatly
improved soon after (Blake and Rowland, 1986) with the addition of the Earth System Research Laboratory from US
National Oceanic and Atmospheric Administration (NOAA/GML) flask network (Steele et al. (1987); Lan et al. (2024), Fig.



1), and the Advanced Global Atmospheric Gases Experiment (AGAGE) (Cunnold et al., 2002; Prinn et al., 2018), the
Commonwealth Scientific and Industrial Research Organisation (CSIRO, Francey et al. (1999)), the University of California
Irvine (UCI, Simpson et al.. 2012) and in situ and flask measurements from regional networks, such as ICOS (Integrated
Carbon Observation System) in Europe (https://www.icos-ri.eu/). The combined datasets provide the longest time series of
globally averaged $CH_4$ abundances. Since the early-2000s, $CH_4$ column-averaged mole fractions have been retrieved through
passive remote sensing from space (Buchwitz et al., 2005a, 2005b; Butz et al., 2011; Crevoisier et al., 2009; Frankenberg et
al., 2005; Hu et al., 2018). Ground-based Fourier transform infrared (FTIR) measurements at fixed locations also provide
time-resolved $CH_4$ column observations during daylight hours, and a validation dataset against which to evaluate the satellite
measurements such as the Total Carbon Column Observing Network (TCCON) network (e.g., Pollard et al., 2017; Wunch
et al., 2011), or Network for Detection of Atmospheric Composition Change (NDACC) (e.g., Bader et al., 2017).
In this budget, in-situ observations from the different networks were used in the top-down atmospheric inversions to estimate
$CH_4$ sources and sinks over the period 2000-2020. Satellite observations from the TANSO/FTS instrument on board the
satellite GOSAT were used to estimate $CH_4$ sources and sinks over the period 2010-2020. Other atmospheric data (FTIR,
airborne measurements, AirCore, isotopic measurements, etc.) have been used for validation by some groups, but not
specifically in this study. However, further information is provided in Tables S7, S8, S9, S10, and S11 and a more
comprehensive validation of the inversions is planned to use some of these data.

### 4.1.1 In situ $CH_4$ observations and atmospheric growth rate at the surface

We use globally averaged $CH_4$ mole fractions at the Earth's surface from the four observational networks (NOAA/GML,
AGAGE, CSIRO and UCI). The data are archived at the World Data Centre for Greenhouse Gases (WDCGG) of the WMO
Global Atmospheric Watch (WMO-GAW) program (https://gaw.kishou.go.jp/), including measurements from other sites
that are not operated as part of the four networks. The $CH_4$ in-situ monitoring network has grown significantly over the last
decade due to the emergence of laser diode spectrometers which are robust and accurate enough to allow deployments with
low maintenance enabling the development of denser networks in developed countries (Stanley et al., 2018; Yver Kwok et
al., 2015), and new stations in remote environments (Bian et al., 2015; Nisbet et al., 2019).
The networks differ in their sampling strategies, including the frequency of observations, spatial distribution, and methods
of calculating globally averaged $CH_4$ mole fractions. Details are given in the supplementary material of Kirschke et al.
(2013). The global average values of $CH_4$ abundances at Earth's surface presented in Fig. 1 are computed using long-term
measurements from background conditions with minimal influence from immediate emissions. All measurements are
calibrated against gas standards either on the current WMO reference scale or on independent scales with well-estimate
differences from the WMO scale. The current WMO reference scale, maintained by NOAA/ESRL, WMO-X2004A
(Dlugokencky et al., 2005) was updated in July 2015. NOAA and CSIRO global means are on this scale. AGAGE uses an
independent standard scale (based on work by Tohoku University (Aoki et al., 1992) and maintained at Scripps Institution





of Oceanography (SIO)), but direct comparisons of standards and indirect comparisons of atmospheric measurements show that differences are well below 5 ppb (Tans and Zwellberg, 2014; Vardag et al., 2014) and the TU-1987 scale used for AGAGE measurements is only 0.5 ppb difference from WMO-X2004A at 1900 ppb level. UCI uses another independent scale that was established in 1978 and is traceable to NIST (Flores et al., 2015; Simpson et al., 2012), but has not been included in standard exchanges with other networks so differences with the other networks cannot be quantitatively defined. Additional experimental details are presented in the supplementary material from Kirschke et al. (2013) and references therein.

In Fig. 1 (a) globally averaged $CH_4$ and (b) its growth rate (derivative of the deseasonalized trend curve) through to 2022 are plotted for the four measurement programs using a procedure of signal decomposition described in Thoning et al. (1989). We define the annual $G_{ATM}$ as the increase in the atmospheric concentrations from Jan. 1 in one year to Jan. 1 in the next year. Agreement among the four networks is good for the global growth rate, especially since ~1990. The large differences observed mainly before 1990 probably reflect the different spatial coverage of each network. The long-term behaviour of globally averaged atmospheric $CH_4$ shows a positive growth rate (defined as the derivative of the deseasonalized mixing ratio) that is slowing down from the early-1980s through 1998, a near-stabilisation of $CH_4$ concentrations from 1999 to 2006, and a renewed period with positive persistent overall accelerating growth rates since 2007, slightly larger after 2014. When a constant atmospheric lifetime is assumed, the decreasing growth rate from 1983 through 2006 may imply that atmospheric $CH_4$ was approaching steady state, leading to no trend in emissions. The NOAA global mean $CH_4$ concentration was fitted with a function that describes the approach to a first-order steady state ($ss$ index): $[CH_4](t) = [CH_4]_{ss}-([CH_4]_{ss}-[CH_4]_0)e^{-t/\tau}$; solving for the lifetime, $\tau$, gives 9.3 years, which is very close to current literature values (e.g., Prather et al. (2012), $9.1 \pm 0.9$ years). Such an approach includes uncertainties, especially due to the strong assumption of no trend in lifetime. The result of constant emissions does not agree with some study explaining the stabilisation period by decreasing emissions associated with increasing sink (e.g., Bousquet et al., 2006). However, this value seems consistent albeit higher than the chemistry climate estimates (8.2 years, see Sect. 3.3.5)

From 1999 to 2006, the annual increase of atmospheric $CH_4$ was remarkably small at $0.6\pm0.1$ ppb $yr^{-1}$. After 2006, the atmospheric growth rate has increased to a level similar to that of the mid-1990s (~5 ppb $yr^{-1}$), and for 2014 and 2015 even to that of the 1980s (>10 ppb $yr^{-1}$). In the two recent years 2020 and 2021, the highest growth rates of 15 ppb $yr^{-1}$ and 18 ppb $yr^{-1}$ (see Sect. 6 ) were unprecedented since the 1980s. On decadal timescales, the annual increase is on average $2.2\pm0.3$ ppb $yr^{-1}$ for 2000-2009, $7.6\pm0.3$ ppb $yr^{-1}$ for 2010-2019 and $15.2\pm0.4$ ppb $yr^{-1}$ for the year 2020.

### 4.1.2 Satellite data of column average $CH_4$

In this budget, we use satellite data from the JAXA satellite Greenhouse Gases Observing SATellite (GOSAT) launched in January 2009 (Butz et al., 2011; Morino et al., 2011) containing the TANSO-FTS instrument, which observes in the shortwave infrared (SWIR). Different retrievals of $CH_4$ based on TANSO-FTS/GOSAT products are made available to the



community: from NIES (Yoshida et al., 2013), from SRON (Schepers et al., 2012) and from University of Leicester (Parker
et al., 2020; Parker and Boesch, 2020). The three retrievals are used by the top-down systems (Table 4 and S6). Although
GOSAT retrievals still show significant unexplained biases and limited sampling in cloud covered regions and in the high
latitude winter, it represents an important improvement compared to the first satellite measuring $CH_4$ from space,
SCIAMACHY (Scanning Imaging Absorption spectrometer for Atmospheric CartograpHY) both for random and systematic
observation errors (see Table S2 of Buchwitz et al. (2016)).
Here, as in Saunois et al. (2020), only inversions using GOSAT retrievals are used.

## 4.2 Top-down inversions used in the budget

An atmospheric inversion is the optimal combination of atmospheric observations, of a model of atmospheric transport and
chemistry, of a prior estimate of $CH_4$ sources and sinks, and of their uncertainties, to provide improved estimates of the
sources and sinks, and their uncertainty. The theoretical principle of $CH_4$ inversions is detailed in the Supplementary
Material and an overview of the different methods applied to $CH_4$ is presented in Houweling et al. (2017).
We consider an ensemble of inversions gathering various chemistry transport models, differing in vertical and horizontal
resolutions, meteorological forcing, advection and convection schemes, and boundary layer mixing. Including these
different systems is a conservative approach that allows us to cover different potential uncertainties of the inversion, among
them: model transport, set-up issues, and prior dependency. General characteristics of the inversion systems are provided in
Table 4. Further details can be found in the referenced papers and in the Supplementary Material. Each group was asked to
provide gridded flux estimates for the period 2000-2020, using either surface or satellite data, but no additional constraints
were imposed so that each group could use their preferred inversion setup. Two sets of prior emission distributions were
built from the most recent inventories or model-based estimates (see Supplementary Material), but its use was not mandatory
(see Table S8 to S11 for the inversion characteristics). This approach corresponds to a flux assessment, but not to a model
inter-comparison as the protocol was not too stringent. Estimating posterior uncertainty is time and computer resource
consuming, especially for the 4D-var approaches and Monte Carlo methods. Posterior uncertainties have not been requested
for this study, but they were found to be lower than the ensemble spread in Saunois et al. (2020). Indeed, chemistry transport
models differ in inter-hemispheric transport, stratospheric $CH_4$ profiles, and OH distribution, limitations which are not fully
considered in the individual posterior uncertainty. As a result, we report the minimum-maximum range among the different
top-down approaches.
Seven atmospheric inversion systems using global Eulerian transport models were used in this study; they contributed to the
previous budgets that included eight atmospheric inversion systems in Saunois et al. (2016) and nine in Saunois et al. (2020).
Each inversion system provided one or several simulations, including sensitivity tests varying the assimilated observations
(surface or satellite), the OH inter-annual variability, or the prior fluxes ensemble. This represents a total of 24 inversion
runs with different time coverage: generally, 2000-2020 for surface-based observations, and 2010-2020 for GOSAT-based





inversions (Table 4 and Table S7). In poorly observed regions, top-down surface inversions may rely on the prior estimates
and bring little or no additional information to constrain (often) spatially overlapping emissions (e.g., in India, China). Also,
we recall that many top-down systems solve for the total fluxes at the surface only or for some categories that may differ
from the GCP categories. When multiple sensitivity tests were performed the mean of this ensemble was used not to
overweight one particular inverse system. It should also be noticed that some satellite-based inversions are in fact combined
satellite and surface inversions as they use surface-based inversions to correct the latitudinal bias of the satellite retrievals
against the optimised atmosphere measurements to correct for errors in the transport model especially in the stratosphere
(e.g., Segers et al., 2022; Maasakkers et al., 2019). Nevertheless, these inversions are still referred to as satellite-based
inversions. Most of the top-down models use the OH distribution from the TRANSCOM experiment (Patra et al., 2011)
either as fixed over the period or with the inter-annual variability derived by Patra et al. (2021).
Each group provided gridded monthly maps of emissions for both their prior and posterior total and for sources per category
(see the categories Sect. 2.3). Results are reported in Sect. 5. Atmospheric sinks from the top-down approaches have been
provided for this budget, and are compared with the values reported in Saunois et al. (2020). Not all inverse systems report
their chemical sink; as a result, the global mass imbalance for the top-down budget is derived as the difference between total
sources and total sinks for each model when both fluxes were reported.

**5 Methane budget: top-down and bottom-up comparison**

**5.1 Global methane budget**

**5.1.1 Global total methane emissions-**

**Top-down estimates.** At the global scale, the total annual emissions inferred by the ensemble of 24 inversions is 575 Tg $CH_4$ yr$^{-1}$ [553-586] for the 2010-2019 decade (Table 3), with the highest ensemble mean emission of 608 Tg $CH_4$ yr$^{-1}$ [581-627] for 2020. Global emissions for 2000-2009 (543 Tg $CH_4$ yr$^{-1}$) are consistent with Saunois et al. (2016, 2020) and the range for global emissions, 526-558 Tg $CH_4$ yr$^{-1}$ falls within the range in Saunois et al. (2016) (535-569) and Saunois et al. (2020) (524-560), although the ensemble of inverse systems contributing to this budget is different from Saunois et al. (2016, 2020). Changes in ensemble members contributing to the different budgets are a feature of each new GMB release and, therefore, introduce a source of variation (Table S7). The range reported gives the minimum and maximum values among studies and does not reflect the individual full uncertainties. In addition, most of the top-down models use the same OH distribution from the TRANSCOM experiment (Patra et al., 2011), which introduces less variability to the global budget than is likely justified, and so contributes to the rather low range (10%) compared to bottom-up estimates (see below).

**Bottom-up estimates.** The bottom-up estimates considered here differ substantially from the top-down results, with annual global emissions being about 15% larger at 669 Tg $CH_4$ yr$^{-1}$ [512-849] for 2010-2019 (Table 3). Yet, thanks to the double counting corrections in this budget, bottom-up and top-down budgets are in better agreement compared to previous GMB





releases. For the period 2000-2009, the discrepancy between bottom-up and top-down was about 30% of the top-
down estimates in Saunois et al. (2016, 2020) (167 and 156 Tg CH$_4$ yr$^{-1}$, respectively), a value that has been reduced significantly
in this budget (now 95 Tg CH$_4$ yr$^{-1}$ (<17%) for the same 2000-2009 period). This reduction is due to improvements from an
important decrease in the estimate of emissions from natural and indirect anthropogenic emissions from bottom-up
approaches, and more specifically inland freshwater emissions. From the previous budget, the estimate for inland freshwater
emissions (lakes, ponds, reservoirs, rivers, and streams) has decreased from 159 Tg CH$_4$ yr$^{-1}$ to 112 Tg CH$_4$ yr$^{-1}$ (47 Tg
decrease). Then, 23 Tg have been removed in the total freshwater ecosystem emissions due to double counting between
vegetated wetlands and mostly small ponds and lakes (Sect. 3.2.2). As a result the combined wetland and inland freshwater
emissions are estimated to be 242 Tg CH$_4$ yr$^{-1}$ for 2000-2009, compared with  306 Tg CH$_4$ yr$^{-1}$ in Saunois et al. (2020).
This budget is the first that reconciles bottom-up and top-down total emissions within the uncertainty ranges. However, the
uncertainty in the global budget remains high because of the large range reported for emissions from freshwater systems.
Still, the upper bound of global emissions from bottom-up approaches is not consistent with top-down estimates that rely
on OH burden constrained by methyl chloroform atmospheric observations and is still likely overestimated.
**5.1.2 Global methane emissions per source category**
The global CH$_4$ emissions from natural and anthropogenic sources (see Sect. 2.3) for 2010-2019 are presented in Fig. 6, Fig.
7, and Table 3. Top-down estimates attribute about 65% of total emissions to anthropogenic activities (range of 55-70%),
and 35% to natural emissions. Bottom-up estimates attribute 57% of emissions to direct anthropogenic and the rest to natural
plus indirect anthropogenic emissions. A current predominant role of direct anthropogenic sources of CH$_4$ emissions is
consistent with and strongly supported by available ice core and atmospheric CH$_4$ records. These data indicate that
atmospheric CH$_4$ varied around 700 ppb during the last millennium before increasing by a factor of 2.6 to ~1800 ppb since
pre-industrial times. Accounting for the decrease in mean-lifetime over the industrial period, Prather et al. (2012) estimated
from these data a total source of 554±56 Tg CH$_4$ in 2010 of which about 64% (352±45 Tg CH$_4$) was of direct anthropogenic
origin, consistent with the range in our stop-down estimates.

**Natural and indirect anthropogenic emissions.** Although smaller than in previous Global Methane Budget releases, the
main remaining discrepancy between top-down and bottom-up budgets is found for the natural and indirect anthropogenic
emission total (105 Tg), with 311 [183-462] Tg CH$_4$ yr$^{-1}$ for bottom-up and only 206 [188-225] Tg CH$_4$ yr$^{-1}$ for top-down
over the 2010-2019 decade. In the bottom-up estimates, this discrepancy comes first from the estimates in both inland
freshwater sources (64 Tg) and second from other natural sources (20 Tg from geological sources, termites, oceans, and
permafrost). The top-down approaches may be biased due to missing fluxes (mainly inland freshwaters) in their prior
estimates.





For 2010-2019, the top-down and bottom-up derived estimates for wetlands emissions of 165 [145-214] Tg $CH_4$ $yr^{-1}$ and
159 [119-203] Tg $CH_4$ $yr^{-1}$, respectively, are comparable within their range. Based on diagnostic wetland area values (see
notes in Table 3), bottom-up mean wetland emissions for the 2000-2009 period are smaller in this study than those of Saunois
et al. (2016) but larger than in Saunois et al. (2020). The changes in wetland emissions from bottom-up models may be
related to updates on the wetland extent data set (WAD2M), the use of two different meteorological forcings for this study
and a different set of models (see Sect. 3.2.1). Conversely, the current 2000-2009 mean top-down wetland estimates are
lower than those of Saunois et al. (2016) and Saunois et al. (2020) (Table 3). In the bottom-up estimates, the amplitude of
the range of emissions of 116-189 is roughly similar to Saunois et al. (2016) (151-222) and Saunois et al. (2020) (102-179)
for 2000-2009. Here, the larger range in bottom-up estimates of wetland emissions is due to the use of GSWP3-W5E5 and
greater sensibilities of some models to the climate parameters, as discussed in Sect. 3.2.1. Bottom-up and top-down estimates
for wetland emissions agree better in this study (~5 Tg $yr^{-1}$ for 2000-2009) than in Saunois et al. (2016, 2020) (~17 Tg $yr^{-1}$
and ~30 Tg $yr^{-1}$, respectively). Natural emissions from inland freshwater systems were not included in the prior fluxes used
in the top-down approaches, due to unavailable or uncertain gridded products at the start of the modelling activity. However,
emissions from these inland freshwater systems may be implicitly included in the posterior estimates of the top-down
models, as these two sources are close and probably overlap at the rather coarse resolution of the top-down models. This is
the reason why the 'wetland emissions' in the top-down budget in fact correspond to the sum of combined wetland and
inland freshwaters emissions in the bottom-up budget. The double-counting of 23 Tg $CH_4$ reduces the bottom-up budget for
combined wetland and inland freshwaters from 271 Tg $CH_4$ $yr^{-1}$ to 248 Tg $CH_4$ $yr^{-1}$ (Sect. 3.2.2). Comparing the 2000-2009
decadal emissions from wetlands and inland freshwater ecosystems estimated by the bottom-up approaches across the last
three Global Methane Budgets shows an upward and then a downward revision with 305 (183+122) Tg $CH_4$ $yr^{-1}$, 356
(147+209) Tg $CH_4$ $yr^{-1}$ and 248 (159+112-23) Tg $CH_4$ $yr^{-1}$ (respectively from Saunois et al. (2016, 2020) and this work; the
sum in bracket corresponds to the sum of vegetated wetland emissions and inland water emissions estimated through the
different budgets). The combined wetland and inland freshwater emissions discrepancy between bottom-up and top-down
approaches amount to 105 Tg $CH_4$ $yr^{-1}$ for the 2010-2019 decade. From a top-down point of view, the sum of all the natural
sources is more robust than the partitioning between wetlands, inland waters, and other natural sources. Including all known
spatio-temporal distributions of natural emissions in top-down prior fluxes would be a step forward to consistently compare
natural versus anthropogenic total emissions between top-down and bottom-up approaches.
In the top-down budget, wetlands represent 28% on average of the total methane emissions but only 24% in the bottom-up
budget (because of higher total emissions inferred). Given the large uncertainties, neither bottom-up nor top-down
approaches included in this study point to significant changes in wetland emissions between the two decades 2000-2009 and
2010-2019 at the global scale.
For the 2010-2019 decade, top-down inversions infer "Other natural emissions" (Table 3) at 43 Tg $CH_4$ $yr^{-1}$ [40-46], whereas
the sum of the individual bottom-up emissions is 63 Tg $CH_4$ $yr^{-1}$ [24-93], contributing to a 20 Tg discrepancy between




bottom-up and top-down approaches. Atmospheric inversions infer the same amount over the decade 2000-2009 as over
2010-2019, which is almost half of the value reported in Saunois et al. (2016) (68 [21-130] Tg CH$_4$ yr$^{-1}$). This reduction in
magnitude and uncertainty is due to 1) a more consistent way of considering other natural emissions in the various inverse
systems (same prior estimate as in this budget) and 2) a difference in the ensemble of top-down inversions reported here
compared to previous releases. It is worth noting that, most of the top-down models include about the same ocean and
onshore geological emissions and termite emissions in their prior scenarios. However, none include freshwater or permafrost
emissions in their prior fluxes, and thus in their posterior estimates.
Geological emissions are associated with relatively large uncertainties, and marine seepage emissions are still widely
debated (Thornton et al., 2020). However, summing up all bottom-up fossil-CH$_4$ related sources (including anthropogenic
emissions) leads to a total of 165 Tg CH$_4$ yr$^{-1}$ [135-190] in 2010-2019, which is about 29% of the top-down global
CH$_4$ emissions, and 25% of the bottom-up total global estimate. These results agree with the value inferred from $^{14}$C
atmospheric isotopic analyses of 30% contribution of fossil-CH$_4$ to global emissions (Etiope et al., 2008; Lassey et al.,
2007b). This total fossil fuel emissions from bottom-up approaches agrees well with the $^{13}$C-based estimate of Schwietzke
et al. (2016) of 192 ± 32 Tg CH$_4$ yr$^{-1}$. In the bottom-up budget, the larger total emissions (due to uncertainties in bottom-up
estimates of natural emissions) leads to a lower fossil fuel contribution compared to Lassey et al. (2007b).
**Anthropogenic direct emissions.** Total anthropogenic direct emissions for the period 2010-2019 were assessed to be
statistically consistent between top-down (369 Tg CH$_4$ yr$^{-1}$, range 350-391) and bottom-up approaches (358 Tg CH$_4$ yr$^{-1}$,
range 329-387), albeit top-down approaches infer direct anthropogenic emissions larger by 11 Tg CH$_4$ yr$^{-1}$ on average
compared to bottom-up approaches. The partitioning of anthropogenic direct emissions between agriculture and waste, fossil
fuels extraction and use, and biomass and biofuel burning, also shows good consistency between top-down and bottom-up
approaches, though top-down approaches still suggest less fossil fuel and more agriculture and waste emissions than bottom-
up estimates (Table 3 and Fig. 6 and 7). For 2010-2019, agriculture and waste contributed an estimated 228 Tg CH$_4$ yr$^{-1}$
[213-242] in the top-down budget and 211 Tg CH$_4$ yr$^{-1}$ [195-231] in the bottom-up budget. Fossil fuel emissions contributed
115 Tg CH$_4$ yr$^{-1}$ [100-124] in the top-down budget and 120 Tg CH$_4$ yr$^{-1}$ [117-125] in the bottom-up budget. Biomass and
biofuel burning contributed 27 Tg CH$_4$ yr$^{-1}$ [26-27] in the top-down budget and 28 Tg CH$_4$ yr$^{-1}$ [21-39] in the bottom-up
budget. Biofuel CH$_4$ emissions rely on very few estimates currently (Wuebbles and Hayhoe, 2002). Although biofuel is a
small source globally (~12 Tg CH$_4$ yr$^{-1}$), more estimates are needed to allow a proper uncertainty assessment. Overall for
top-down inversions the global fraction of total emissions for the different source categories is 40% for agriculture and
waste, 20% for fossil fuels, and 5% for biomass and biofuel burning. With the exception of biofuel emissions, the uncertainty
associated with global anthropogenic emissions appears to be smaller than that of natural sources but with an asymmetric
uncertainty distribution (mean significantly different than median). The relative agreement between top-down and bottom-
up approaches may indicate a limited capability of the inversion to separate emissions and a dependency to their prior fluxes;
this agreement should therefore be treated with caution. Indeed, in poorly observed regions, top-down inversions rely on the



prior estimates and bring little or no additional information to constrain (often) spatially overlapping emissions (e.g., in India, China). Also, as many top-down systems solve for the total fluxes at the surface or for some categories that may differ from the GCP categories, their posterior partitioning relies on the prior ratio between categories that are prescribed using bottom-up inventories.

### 5.1.3 Global budget of total methane sinks

**Top-down estimates.** The annual $CH_4$ chemical removal from the atmosphere is estimated to be 521 Tg $CH_4$ $yr^{-1}$ averaged over the period 2010-2019, with an uncertainty of about ±2% (range 485-532 Tg $CH_4$ $yr^{-1}$). All the inverse models account for $CH_4$ oxidation by OH and O($^1$D), and some include stratospheric Cl oxidation (Table S8 to S11). Most of the top-down models use the OH distribution from the TRANSCOM experiment (Patra et al., 2011) either as fixed over the period or including inter annual variability from Patra et al. (2021). This study shows no trend in OH and IAV below ±4%, in agreement with Thompson et al. (2024) (no significant OH trend and IAV < 2%). As a result, the range of the top-down sink estimates is rather low compared to bottom-up estimates (see below). Differences between transport models affect the chemical removal of $CH_4$, leading to different chemical loss rates, even with the same OH distribution. However, uncertainties in the OH distribution and magnitude (around ±10% at the global scale, Zhao et al., 2019) are not considered in our study, while they could contribute to a significant change in the chemical sink, and then in the derived posterior emissions through the inverse process ((Zhao et al., 2020), around ±17% at the global scale, much larger than the model spread derived here. The chemical sink represents more than 90% of the total sink, the rest being attributable to soil uptake (35 [35-36] Tg $CH_4$ $yr^{-1}$). The rather narrow range is due to the use of the same climatological soil sink provided within the modelling protocol which is based on Murgia-Flores et al. (2018). This sink estimate used as prior in the inversions is a bit higher than the mean estimate of the soil sink calculated by bottom-up models (30 Tg $CH_4$ $yr^{-1}$, Sec. 3.3.4).

**Bottom-up estimates.** The total chemical loss for the 2010s reported here is 602 Tg $CH_4$ $yr^{-1}$ with an uncertainty of 21% (~125 Tg $CH_4$ $yr^{-1}$). Differences in chemistry schemes in the models (especially in the stratosphere) and in the volatile organic compound treatment probably explain most of the discrepancies among models (Zhao et al., 2019).

### 5.2 Latitudinal and regional methane budgets

The latitudinal and regional breakdown of the bottom-up budget is based on crude assumptions that we acknowledge here. Natural and indirect anthropogenic emissions are based on wetland gridded products from land surface models and the combination of the maps from lakes and ponds from Johnson et al. (2022), reservoirs from Johnson et al. (2022) and streams and rivers from Rocher-Ros et al. (2023), the sum of those three scaled to 89 Tg $CH_4$ $yr^{-1}$ (shown in Fig. 5) to artificially include the double counting (estimated only at the global scale) and match the global estimate. However, we acknowledge that this procedure distributes the double counting relatively to the final emission distribution and not according to the freshwater ecosystems where the double counting probably occurs. Wild animals and permafrost maps do not exist and are





missing from the calculation, leading to around 3 Tg CH$_4$ yr$^{-1}$ of discrepancy. Geological and ocean sources are based on
Etiope et al. (2019) and Weber et al. (2019) gridded products scaled to 50 Tg CH$_4$ yr$^{-1}$ to be consistent to the reported global
values. Finally, we use the termite emission map produced for this budget and used in the global budget. The latitudinal
budget does not include the estimates from FAO and USEPA for the direct anthropogenic emissions as they are only
provided at country scale.

### 5.2.1 Latitudinal budget of total methane emissions

The latitudinal breakdown of emissions inferred from atmospheric inversions reveals a dominance of tropical emissions of
364 Tg CH$_4$ yr$^{-1}$ [337-390], representing 64% of the global total (Table 5 and 6). 32% of the emissions are from the mid-
latitudes (187 Tg CH$_4$ yr$^{-1}$ [160-204]) and 4% from high latitudes (above 60°N). The ranges around the mean latitudinal
emissions are larger than for the global CH$_4$ sources. While the top-down uncertainty is less than ±5% at the global scale, it
increases to ±7% for the tropics, to ±12% the northern mid-latitudes and to more than ±20% in the northern high-latitudes
(for 2010-2019, Table 5). Both top-down and bottom-up approaches consistently show that CH$_4$ decadal emissions have
increased by +21-27 Tg CH$_4$ yr$^{-1}$ in the tropics, and by +5-16 Tg CH$_4$ yr$^{-1}$ in the northern mid-latitudes between 2000-2009
and 2010-2019 using the mean ensemble estimate.
Over 2010-2019, at the global scale, satellite-based inversions infer almost identical emissions to ground-based inversions
(difference of +1 [-3-9] Tg CH$_4$ yr$^{-1}$, with GOSAT based inversion a bit higher than surface measurements-based inversions),
when comparing consistently surface versus satellite-based inversions for each system, similar to Saunois et al. (2020). This
difference is much lower than the range derived between the different systems (range of 20 Tg CH$_4$ yr$^{-1}$ using surface- or
satellite-based inversions). This result reflects that differences in atmospheric transport among the systems probably have
more impact on the estimated global emissions than the types of observations assimilated.
As expected, considering the different coverage of observation datasets, regional distributions of inferred emissions differ
depending on the nature of the observations used (satellite or surface). The largest differences (satellite-based minus surface-
based inversions) are observed over the tropical region, between -10 and +43 Tg CH$_4$ yr$^{-1}$ (90°S to 30°N), and the northern
mid-latitudes (between -36 and -2 Tg CH$_4$ yr$^{-1}$). Satellite data provide stronger constraints on fluxes in tropical regions than
surface data, due to a much larger spatial coverage. It is therefore not surprising that differences between these two types of
observations are found in the tropical band, and consequently in the northern mid-latitudes to balance total emissions, thus
affecting the north-south gradient of emissions. However, the regional patterns of these differences are not consistent
through the different inverse systems. Indeed, some systems found higher emissions in the tropics when using GOSAT
instead of surface observations, while others found the opposite. This difference between inversion systems may depend on
whether or not a bias correction is applied to the satellite data based on surface observations, and also on the modelled
horizontal and vertical transports, in the troposphere and in the stratosphere.





**5.2.2 Latitudinal methane emissions per source category**

The analysis of the latitudinal $CH_4$ budget per source category (Fig. 8 and Table 6) can be performed both for bottom-up and top-down approaches but with limitations. Bottom-up estimates of natural and indirect anthropogenic emissions are based on assumptions as specified at the beginning of this section 5.2. For top-down estimates, as already noted, the partitioning of emissions per source category has to be considered with caution. Indeed, using only atmospheric $CH_4$ observations to constrain $CH_4$ emissions makes this partitioning largely dependent on prior emissions. However, differences in spatial patterns and seasonality of emissions can be utilised to constrain emissions from different categories by atmospheric methane observations (for those inversions solving for different sources categories, see Sect. 2.3).

Agriculture and waste are the largest sources of $CH_4$ emissions in the tropics and southern hemisphere (140 [121-150] Tg $CH_4$ yr$^{-1}$ in the bottom-up budget and 150 [135-168] Tg $CH_4$ yr$^{-1}$ in the top-down budget, about 40% of total $CH_4$ emissions in this region). However, combined wetland and inland freshwater emissions are nearly as large with 151 [85-234] Tg $CH_4$ yr$^{-1}$ in the bottom-up budget and 128 [112-155] Tg $CH_4$ yr$^{-1}$ in the top-down budget. Anthropogenic emissions dominate in the northern mid-latitudes, with the highest contribution from agriculture and waste emissions (40% of total emissions in the top-down budget), closely followed by fossil fuel emissions (32% of total emissions, top-down budget). Boreal regions are largely dominated by inland freshwater emissions (41% and 54% of total emissions, top-down and bottom-up budget, respectively).

The largest discrepancies between the top-down and the bottom-up budgets are found in the mid-latitudes and boreal regions from the natural and indirect sources with bottom-up estimates twice as large as the top-down ones, especially in the inland freshwater category.

The uncertainty for wetlands and inland freshwater emissions is larger in the bottom-up models than in the top-down models (mostly wetlands), while uncertainty in anthropogenic emissions is larger in the top-down models than in the bottom-up inventories. The large uncertainty in tropical inland freshwater emissions (mostly wetlands) of ±44% results from large regional differences between the bottom-up land-surface models. Although they are using the same forcings, their responses in terms of flux density show different sensitivities to temperature, water vapour pressure, precipitation, and radiation.

**5.2.3 Regional budget for total emissions**

The regional breakdown of emissions is provided for 18 continental regions (see map in Fig. S3 and Table S1 with the country aggregation in the supplementary materials).

At the regional scale and, for the 2010-2019 decade, total methane emissions are dominated by South East Asia with 63 [52-71] Tg $CH_4$ yr$^{-1}$, China with 57 [37-72] Tg $CH_4$ yr$^{-1}$, and South Asia with 52 [43-60] Tg $CH_4$ yr$^{-1}$ (top-down budget). These top three emitters contribute 30% of total global $CH_4$ emissions. The following high emitting regions are Brazil 47 [41-58] Tg $CH_4$ yr$^{-1}$, Equatorial Africa 47 [39-59] Tg $CH_4$ yr$^{-1}$, USA 38 [32-46] Tg $CH_4$ yr$^{-1}$, Southwest South America 38 [30-48]





Tg $CH_4$ yr$^{-1}$, Russia 36 [27-45] Tg $CH_4$ yr$^{-1}$, Europe 31 [24-36] Tg $CH_4$ yr$^{-1}$, Middle East 31 [24-39] Tg $CH_4$ yr$^{-1}$, Northern
Africa 25 [23-29] Tg $CH_4$ yr$^{-1}$, and Canada 20 [17-24 ] Tg $CH_4$ yr$^{-1}$. Other regions contribute less than 20 Tg $CH_4$ yr$^{-1}$.
**5.2.4 Regional budget per source category**
**Natural and indirect anthropogenic emissions versus direct anthropogenic emissions.** In agreement with Stavert et al.
(2021), natural and indirect anthropogenic emissions are dominated by Brazil, Canada, Russia, Equatorial Africa and
Southeast Asia, contributing 126 Tg $CH_4$ yr$^{-1}$ in the bottom-up and 105 Tg $CH_4$ yr$^{-1}$ in the top-down budget (Table 7), i.e.,
47% and 50% of the global natural and indirect anthropogenic emissions in these budgets, respectively. At regional scale
also, the range of uncertainty in natural and indirect anthropogenic emissions are much larger in the bottom-up budget than
in the top-down budget (Fig. S5). Except for 4 regions (Canada, Brazil, Northern South America, Southwest South America),
direct anthropogenic emissions contribute more than half of the total regional emissions. Due to the large uncertainty and
discrepancies in natural and indirect emissions estimates, the regional direct anthropogenic fractions may differ between the
bottom-up and top-down budgets. However, in absolute values, the highest direct anthropogenic emitters are the same in
the two budgets with China and South Asia being the top two by far, contributing 56 [51-66] Tg $CH_4$ yr$^{-1}$ and 45 [44-47] Tg
$CH_4$ yr$^{-1}$, respectively (bottom-up values, Fig. 9 and Table 7). These two regions contribute 28% (26%) of the global direct
anthropogenic emissions in the bottom-up (top-down) budget. The ranks of direct anthropogenic emitters are similar to those
presented in the last budget (Stavert et al., 2021). Southeast Asia, United States of America, Middle East, Europe, Equatorial
Africa, and Russia emit between 32 Tg $CH_4$ yr$^{-1}$ and 23 Tg $CH_4$ yr$^{-1}$ as direct anthropogenic emissions (bottom-up values,
Fig 8). Brazil, Northern Africa, and Southwest South America emit between 10 $CH_4$ yr$^{-1}$ and 20 $CH_4$ yr$^{-1}$, while the rest of
the regions emit less than 10 $CH_4$ yr$^{-1}$ direct anthropogenic emissions.

**Sectoral emissions.** The sectoral partitioning at the regional scale has been derived from both bottom-up and top-down
approaches. However, the top-down budget has more limitations, as the sectoral partitioning is usually based on the prior
fluxes fractions at the pixel scale, and assimilating only total methane observations does not allow to disentangle the different
source sectors overlapping in a pixel grid. However, differences in spatial patterns and seasonality of emissions can still be
constrained by atmospheric $CH_4$ observations for those inversions solving for different sources categories (see Sect. 2.3).
Bottom-up approaches allow deeper sectorial splitting, especially in terms of direct anthropogenic emissions (Fig. 9). Table
7, Fig. 9 and Fig. 10 present the estimations of $CH_4$ emissions on average over 2010-2019. Fig. 10 presents the budgets for
three main categories (Combined wetland and inland freshwaters, Fossil fuels and Agriculture & Waste), a more detailed
figure and table including the five categories is available in the supplementary material (Fig. S6 and Table S13 to S18).
Values for each individual data-set for the decades 2000-2009, 2010-2019, and the last year 2020 are made available in a
spreadsheet (see Data Availability).



For most regions, "Combined wetland and inland freshwater emissions" are the most uncertain in the bottom-up budget,
and generally their range is larger than in the top-down budget. In the top-down budget, this category contributes the most
to the regional emissions in Brazil 24 [20-33] Tg $CH_4$ yr$^{-1}$, Southeast Asia 24 [14-29] Tg $CH_4$ yr$^{-1}$ (though similar to their
Agriculture and Waste emissions 24 [21-31] Tg $CH_4$ yr$^{-1}$), Equatorial Africa 22 [19-28] Tg $CH_4$ yr$^{-1}$, Southwest South
America 22 [14-33] Tg $CH_4$ yr$^{-1}$, Canada 12 [9-18] Tg $CH_4$ yr$^{-1}$, Northern South America 8 [6-10] Tg $CH_4$ yr$^{-1}$, Southern
Africa 7 [4-9] Tg $CH_4$ yr$^{-1}$. Agriculture and Waste emissions dominates in South Asia 39 [33-43] Tg $CH_4$ yr$^{-1}$, China 30 [13-
37] Tg $CH_4$ yr$^{-1}$, Europe 19 [16-23] Tg $CH_4$ yr$^{-1}$, United States of America 13 [9-16] Tg $CH_4$ yr$^{-1}$, Northern Africa 13 [12-
14] Tg $CH_4$ yr$^{-1}$, Central America 9 [8-10] Tg $CH_4$ yr$^{-1}$, and Korea and Japan 3 [3-4] Tg $CH_4$ yr$^{-1}$. Fossil fuel emissions
dominate in the Middle East 18 [11-24] Tg $CH_4$ yr$^{-1}$ and Russia 14 [8-23] Tg $CH_4$ yr$^{-1}$ (close to their combined wetland and
inland freshwater emissions of 11 [8-13] Tg $CH_4$ yr$^{-1}$).
The four largest contributors to the Fossil Fuel sector remain China, the Middle East, Russia, and the United States of
America. Altogether they contribute 67 (64) Tg $CH_4$ yr$^{-1}$ in the bottom-up (top-down) budget, around 55% of the global
fossil fuel emissions. The bottom-up and top-down approaches generally agree in terms of ensemble mean, except for China
for which the top-down estimates suggest lower emissions than the inventories. While Chinese fossil fuel emissions occur
mainly through coal mining activity (88%), the Middle East, Russia and the USA extract mainly oil and gas (100%,
80%,72%).

The three largest contributors to the Agriculture and Waste sector remain South Asia, China, and Southeast Asia. Together
they contribute 88 (92) Tg $CH_4$ yr$^{-1}$ in the bottom-up (top-down) budget, around 40% of the global agriculture and Waste
sector. While the ensemble means tend to agree between bottom-up and top-down budgets, the uncertainty derived from the
top-down approaches is larger, especially for these three regions. $CH_4$ emissions due to rice cultivation originate mostly
from these same three regions (South East Asia, China and South Asia). Livestock management emissions occurs mainly in
South Asia 20 [18-22] Tg $CH_4$ yr$^{-1}$, Brazil 12 [11-13] Tg $CH_4$ yr$^{-1}$, China 11 [8-16] Tg $CH_4$ yr$^{-1}$, and Europe 11 [10-12] Tg
$CH_4$ yr$^{-1}$ (bottom-up estimates). The United States of America, Equatorial Africa, Northern Africa and Southwest South
America emit between 7 Tg $CH_4$ yr$^{-1}$ and 10 Tg $CH_4$ yr$^{-1}$ in this sub-sector. Other regions emit less than 4 Tg $CH_4$ yr$^{-1}$ in the
livestock management sector. The Waste sector emissions are dominated by three regions: China 11 [6-14] Tg $CH_4$ yr$^{-1}$,
South Asia 9 [4-11] Tg $CH_4$ yr$^{-1}$, and Europe 8 [6-12] Tg $CH_4$ yr$^{-1}$ (bottom-up estimates). These three regions contribute
around 40% of the global emissions of the Waste sector. It is worth noting that the uncertainty in the inventory estimates at
the regional scale is around 40% (from the min-max range of the estimate, not including the uncertainty from each
inventory).





**6 Insights on the methane cycle from 2020-2022 during which there has been unprecedented high growth rates of methane emissions**

The mean emissions estimate for the last year of the budget (2020) was 608 [581-627] Tg CH$_4$ yr$^{-1}$ (Top-down),) with 65% of the emissions from direct anthropogenic sources. This is 65 Tg CH$_4$ yr$^{-1}$ higher (11%) than the mean emissions of the 2000-2009 decade and 6% higher than 2010-2019. 2020 was a second highest year in terms of atmospheric CH$_4$ growth rate (+15.2 ppb/yr) since systematic measurements began in the late 1980s, coming in just behind the highest in 2021 at 17.97 ppb/yr. A few studies analysed the large growth rate increase between 2019 (+9.7 ppb/yr) and 2020 (+15.2 ppb/yr) of +5.4 ppb/yr (corresponding to +14.4 ± 2.0 Tg CH$_4$ yr$^{-1}$) (Peng et al., 2022; Stevenson et al., 2022). Peng et al. (2022) estimated that the 2019-2020 growth rate change was almost equally due to an increase in wetland emissions (6.9 ± 2.1 Tg CH$_4$ yr$^{-1}$) and a decrease of the OH chemical loss (7.5 ± 0.8 Tg CH$_4$ yr$^{-1}$) due to reduced OH precursor emissions during the COVID lockdown (Laughner et al., 2021). The COVID19 lockdown resulted in decreased NO$_x$ emissions and reduced fossil fuel related CH$_4$ emissions (Thorpe et al., 2023), leading to less OH production. At the global scale, Feng et al. (2023) calculated an emission increase of 27 Tg CH$_4$ yr$^{-1}$ between 2019 and 2020 considering constant OH, and a smaller increase of 21 Tg CH$_4$ yr$^{-1}$ when including a 1.4% decrease of OH. Increased emissions were mainly found in the northern tropics. Qu et al. (2022) also inferred a 31 Tg CH$_4$ yr$^{-1}$ increase of emissions, mostly in the tropics, half of it in Africa. Such a result is compatible with wetland driven abnormal emissions during a consecutive 3-year La Nina event spanning from 2020 to 2022 (Zhang et al., 2023; Nisbet et al., 2023). The difference in terms of methodology and approaches between these three studies make it difficult to compare them quantitatively but provide a robust understanding on the possible causes. Importantly, all the studies indicate, in various proportions, increasing CH$_4$ emissions in the tropics and in the boreal region, potentially driven by microbial emission from wetlands due to wetter and warmer climate , and a significant contribution of reduced OH concentrations due to COVID lockdown.

Based on our ensemble of data, we find that top-down approaches infer a much larger change in CH$_4$ emissions (median [Q1-Q3] at +23 [10-31] Tg CH$_4$ yr$^{-1}$) than bottom-up approaches (-1 [-5-3] Tg CH$_4$ yr$^{-1}$) between 2019 and 2020 (Fig. S7). Bottom-up approaches suggest a very small increase in wetland emissions (around (+1 [0-3] Tg CH$_4$ yr$^{-1}$), while top-down approaches suggest on average a larger increase for wetlands of +8 [5-11] Tg CH$_4$ yr$^{-1}$, mainly in the tropics and mid-latitudes. It is worth noting that large uncertainties exist for a given year and that the inter annual variability is much lower than the ensemble spread. While bottom-up approaches suggest almost constant fossil fuel emissions and slight increase in agriculture and waste (+3 Tg CH$_4$ yr$^{-1}$), top-down approaches tend to derive higher emissions changes (+6 Tg CH$_4$ yr$^{-1}$ from the fossil fuel sector and +11 Tg CH$_4$ yr$^{-1}$ from agriculture and waste as the median over the ensemble). Biomass burning emissions decreased using both approaches by about 5 Tg CH$_4$ yr$^{-1}$ in agreement with Peng et al. (2022). Some inversions were run with IAV of OH from Patra et al. (2021) and others with constant OH. However the inferred OH IAV in 2019 and 2020 are rather low (0.3% and 0.15% on yearly average) in Patra et al. (2021), leading to a small impact in




terms of emissions changes between 2019-2020, with +22 [9-31] (median [Q1-Q3]) based on the inversions with constant
OH and 19 [7-28] based on the inversions with varying OH (Fig S8).
This first analysis based on our ensemble shows how challenging it is to attribute CH$_4$ emissions changes to a specific sector
or region between two years, because related uncertainties remain much larger than the targeted signal to explain. This calls
again for further improvement of both approaches.
NOAA estimates of 2021 and 2022 methane atmospheric growth rates 17.8.0±0.5 ppb/yr and 14.0±0.8 ppb/yr, respectively
(Lan et al., 2024). They show a continuation of very high growth rates, challenging again our understanding of the methane
budget. As of the time of submission of this manuscript, bottom-up estimates for anthropogenic emissions for 2021 and
2022 are only available from the EDGARv8 data set (https://edgar.jrc.ec.europa.eu/dataset_ghg80; EDGAR, 2023). This
research inventory suggests that anthropogenic emissions continued to increase from 2020 (374 Tg CH$_4$ yr$^{-1}$) to 2021 (379
Tg CH$_4$ yr$^{-1}$) and 2022 (386 Tg CH$_4$ yr$^{-1}$) with around 62% of the increase due to the fossil fuel sources, 23 % from the
Waste sector, and 14% from the agriculture sector (Table S19). The bottom-up estimate of wetland emissions for 2021-
2023, derived from a single wetland model, indicates positive anomalies of 26 Tg CH$_4$ yr$^{-1}$ in 2020, 23 Tg CH$_4$ yr$^{-1}$ in
2021, and 21 Tg CH$_4$ yr$^{-1}$ 2022 relative to the 2000-2006 baseline (https://earth.gov/ghgcenter/data-catalog/lpjwsl-
wetlandch4-grid-v1; Zhang et al., 2023).

**7 Future developments, missing elements, and remaining uncertainties**

In this budget, robust features and uncertainties on sources and sinks estimated by bottom-up or top-down approaches have
been highlighted as well as discrepancies between the two budgets. Limitations of the different approaches have also been
highlighted. Four shortcomings of the CH$_4$ budget were already identified in Kirschke et al. (2013) and Saunois et al. (2016,
2020) and are revisited below pointing to key research areas. Although much progress has been made, they are still relevant,
and actions are needed. However, these actions fall into different timescales and actors. Here, we revisit the four
shortcomings of the contemporary methane budget and discuss how each weakness has been addressed since Saunois et al.
(2020). Each section ends by discussing remaining research needs with a list of suggestions, from higher to lower priority.
1. Shortcoming 1: *Towards a decrease of the high uncertainty in the amount of methane emitted by wetland and inland*
*water systems, and a weakened double counting issue.*
This first shortcoming has probably received the largest interest in the last few years with significant improvements. First a
community effort has been made based on more studies, documenting, or modelling more inland freshwater systems and
synthesising emissions from the complex and heterogeneous ensemble of emitting areas: wetlands, ponds, lakes, reservoirs,
streams, rivers, estuaries, and marine systems. The range of wetland and inland water emissions has been narrowed down
with improved wetland extent and refined estimates for inland freshwater systems. Double counting between inland



freshwater systems has been estimated for the first time and accounted for in this budget. All these improvements decreased
the discrepancy between top-down and bottom-up estimate of combined      wetland and inland freshwater emissions from
156 Tg CH$_4$ yr$^{-1}$ in Saunois et al. (2020) down to 85 Tg CH$_4$ yr$^{-1}$ in this update for the 2000-2009 decade. Gridded maps
for lakes, ponds, reservoirs, and streams and rivers freshwater emissions have been produced over the past years (Johnson
et al., 2021, 2022; Rocher-Ros et al., 2023) making the spatial distribution of CH$_4$ sources almost complete for the first time
and allowing better description of prior emissions in future top-down inversions.
Next steps include on the short term from highest to lowest priority include:
(i) integration of spatial distribution of inland waters in atmospheric inversion models to reach a full description of prior
methane sources and sinks.
(ii) refinement of double counting estimation and its possible reduction with more precise spatial and temporal distributions
of the different systems contributing to inland freshwater emissions by using very high-resolution satellite data (down to
metre resolutions) to properly separate them. The development of a dynamical global high-resolution (typically few metres)
classification of saturated soils and inundated surfaces based on satellite data (visible and microwave), surface inventories,
and expert knowledge.
(iii) continuation of ongoing efforts to calibrate and evaluate land surface models for wetland emissions against in-situ
observations such as FLUXNET-CH$_4$ (Knox et al., 2019; Delwiche et al., 2021) or BAWLD-CH4 (Kuhn et al., 2021) for
boreal regions and avoid dependence on top-down estimates. It is still critical to increase the limited number of tropical
observations and to assimilate them in the inverse systems to help address the issue (e.g., Kallingal et al., 2023).
(iv) continuation of ongoing efforts to develop a diversity of modelling approaches (among them process-based model or
machine learning approaches) to estimate wetland and inland freshwater CH4 emissions, including lateral fluxes, and
reducing upscaling issues, as done by e.g. Zhuang et al. (2023) for lakes.
(v) continuous integration of collected flux measurements such as in the FLUXNET-CH$_4$ activity (Knox et al., 2019;
Delwiche et al., 2021) or in BAWLD-CH4 data set (Kuhn et al., 2021) to provide global flux maps based on machine
learning approaches or other approaches (Peltola et al., 2019, McNicol et al., 2023).
Over the long run, developing measurement systems will help to improve estimates of the diversity of wetland and inland
freshwater sources, and further reduce uncertainties:
-    More systematic measurements of CH$_4$ fluxes and their isotopic signatures from sites reflecting the diversity of

environment of wetlands and inland waters, complemented with environmental meta-data (e.g., soil temperature

and moisture, vegetation types, water temperature, acidity, nutrient concentrations, NPP, soil carbon density for

wetlands, lake morphologies) will allow us to better understand and estimate the processes of production and

transport to the atmosphere (diffusive, ebullitive, plants mediated.. ) and to better constrain methane fluxes and

their isotopic signatures in the different  modelling approaches (Glagolev et al., 2011; Turetsky et al., 2014).



2.  Shortcoming 2: *Towards a better assessment of uncertainties for global methane sinks in top-down and bottom-up budgets.*

The inverse systems used here have similar caveats than those described in Saunois et al. (2016, 2020) (same OH field, same kind of proxy method to optimise it) leading to quite constrained atmospheric sink and therefore total global $CH_4$ sources. Although we have used the latest release of CCMI-2022 (Plummer et al., 2021) and CMIP6 simulations (Collins et al., 2017), the uncertainty of derived CH4 chemical loss from the chemistry climate models remains at the same (large) level compared to the previous intercomparison project ACCMIP (Lamarque et al., 2013). The causes of uncertainties on the $CH_4$ loss and the differences between the different OH fields derived from Chemistry Transport Models (CTM) and Climate Chemistry Models (CCM) have been widely discussed (Nicely et al., 2017 ; Zhao et al., 2019). These results emphasise the need to first assess, and then improve, atmospheric transport and chemistry models, especially vertically, and to integrate robust representation of OH fields in atmospheric models. For the latter, Zhao et al. (2023) have proposed a new approach based on OH precursor observations and a chemical box model to improve the 3D distributions of tropospheric OH radicals obtained from atmospheric chemistry models. Finally, soil uptake estimates rely on very few studies, and interannual variations remain underconstrained.

Next steps, in the short term, could include developments by the modelling community in:

- Estimating the soil uptake with different land surface models (creating an ensemble) and discussing its variations over the past decade.

- Assessing the impact of using updated and varying soil uptake estimates, especially considering a warmer climate in the top-down approach. Indeed, for top-down models resolving for the net flux of $CH_4$ at the surface integrating a larger estimate of soil uptake would allow larger emissions, and then reduce the uncertainty with the bottom-up estimates of total $CH_4$ sources.

- Further studying the reactivity of the air parcels in the chemistry climate models and defining new diagnostics to assess modelled $CH_4$ lifetimes.

- Applying Zhao et al. (2023) recipe to several CTM used for top-down inversions in order to increase consistency between source and sink estimates in individual approaches.

- Developing 3D inverse methods to optimise OH using $CH_4$ satellite data (Zhang et al., 2018) or halogenated compounds beyond methyl chloroform (MCF), such as done in box models (Thompson et al., 2024) to derive a 3D dynamical OH field or machine learning methods using satellite data to constrain OH (Anderson et al., 2023).

- Integrating the aforementioned different potential OH chemical fields, including also inter-annual variability, to assess the impact on the methane budget following Zhao et al. (2020).

Over the long run, other parameters should be (better) integrated into top-down approaches, among them:

- The magnitude of the $CH_4$ loss through oxidation by tropospheric Cl, a process debated in the recent literature. More modelling (e.g., Thanwerdas et al., 2022b) and instrumental studies should be devoted to reducing the





uncertainty of this potential additional sink before integrating it in top-down models. This would be especially
critical if inversions using $^{13}$C-CH4 observations are included in GMB in the future.

*3.    Shortcoming 3: Towards a better partitioning of methane sources and sinks by region and process using top-down*
*models*
In this work, we report inversions assimilating satellite data from GOSAT, which bring more constraints than provided by
surface stations alone, especially over tropical continents. However, we still found that satellite- and surface-based
inversions, and the different inversion systems do not consistently infer the same regional flux distribution.
The estimates contributing to the Global Methane budget are further used in more specific studies focusing on the
comparison of the estimates from bottom-up and top-down approaches at national (Deng et al., 2022) and regional scales,
including efforts from the GCP-REgional Carbon Cycle Assessment and Processes (RECCAP2) (Petrescu et al., 2021; 2023;
Tibrewal et al., 2024; Lauerwald et al., 2023b; and other RECCAP-2 publications to come, see
https://www.globalcarbonproject.org/reccap/publications.htm).
Next steps, in the short term, could integrate developments to be made by the top-down community:
-    Including GOSAT 2 retrievals (Noël et al., 2022; Imasu et al., 2023) for the GOSAT-based inversions and
considering TROPOMI-based inversions (as done in Tsuruta et al. (2023), Shen et al. (2023), Chen et al. (2022)
and Qu et al. (2021)) in the next releases once at least 8 years of data are available to provide a decadal estimate
and biases are reduced for global scale use (Lorente et al., 2023; Balasu et al., 2023). Indeed, recent satellite
developments have provided higher temporal and spatial resolutions of CH4 observations in regions with poor in-
situ measurements (Figure S9, such as TROPOMI observations in North Africa).

-    Integrating the newly available updated gridded products for the different natural sources of CH4 in their prior
fluxes (e.g. inland freshwaters) to reach a full spatial description of sources and sinks, and to be able to better
compare the top-down budget with the bottom-up budget.

-    Integration of the newly developed 4D variational inversion systems using isotopic species in the top-down budget
(Basu et al., 2022; Thanwerdas et al., 2024; Drinkwater et al. 2023; Mannisenaho et al., 2023).

-    Improving the availability of in-situ data at high temporal resolution for the scientific community, especially ones
covering poorly documented regions such as China (Liu et al., 2021b; Guo et al., 2020), India (Nomura et al., 2021;
Lin et al., 2015; Tiwari and Kumar, 2012) and Siberia (Sasakawa et al., 2010, 2017; Fujita et al., 2020; Winderlich
et al., 2010), which are not delivered so far to international databases, or only at poor temporal resolution.

-    Integrating the information from imagery satellites (e.g., TROPOMI, Carbon Mapper, Methane Sat, GHG Sat.) of
high to super-emitters to improve prior fluxes of anthropogenic emissions in terms of quantity and locations for
each covered sector.



Over the long run, integrating more measurements and regional studies will help to improve the top-down systems, and
further reduce the uncertainties:
-   Extending the $CH_4$ surface networks to poorly observed regions (e.g., Tropics, China, India, high latitudes) and to
the vertical dimension: aircraft regular measurements (e.g., Filges et al., 2015; Brenninkmeijer et al., 2007; Paris
et al., 2010; Sweeney et al., 2015); Aircore campaigns (e.g., Andersen et al., 2018; Membrive et al., 2017) ; TCCON
observations (e.g., Wunch et al., 2011, 2019) remains critical to complement satellite data that do not observe well
in cloudy regions and at high latitudes, and also to evaluate and eventually correct satellite biases (Buchwitz et al.,
2016).

-   Extending and developing continuous isotopic measurements of $CH_4$ to help partitioning methane sources and to
be integrated in 4D variational isotopic inversions (e.g., Yacovitch et al., 2021).

-   Integrating global data from future satellite instruments with intrinsic low-bias, such as active LIDAR techniques
with MERLIN (Ehret et al., 2017), that are promising to overcome issues of systematic errors (Bousquet et al.,
2018) and should provide measurements over the Arctic, contrary to the existing and planned passive missions.

-   Other co-emitted species such as radiocarbon for fossil/non-fossil emissions (Lassey et al., 2007a, 2007b; Petrenko
et al., 2017), CO (e.g., Zheng et al., 2019) for biomass burning emissions, and ethane for fugitive emissions (e.g.,
Ramsden et al., 2022) could bring additional information for partitioning emissions.


*4.*   Shortcoming 4: *Towards reducing uncertainties in the modelling of atmospheric transport in the models used in the*
*top-down budget*

The TRANSCOM experiment synthesised in Patra et al. (2011) showed a large sensitivity of the representation of
atmospheric transport on $CH_4$ abundances in the atmosphere. In particular, the modelled $CH_4$ budget appeared to depend
strongly on the troposphere-stratosphere exchange rate and thus on the model vertical grid structure and circulation in the
lower stratosphere. Also, regional changes in the $CH_4$ budget depend on the characteristics of the atmospheric transport
models used in the inversion (Bruhwiler et al., 2017; Locatelli et al., 2015). This axis of research is demanding important
development from the atmospheric modelling community. Waiting for future improvements (finer horizontal and vertical
resolutions, more accurate physical parameterization, increase in computing resources…), assessing atmospheric transport
error and the impact on the top-down budget remain crucial and mostly rely on the use of an ensemble of models.
Methodology changes that could be integrated into the next methane budget releases include:
-   Evaluating more deeply the inversions provided against independent measurements such as aircraft regular
campaigns available through for example the CH4 GLOBALVIEWplus v6.0 ObsPack (Schuldt et al., 2023), the
IAGOS     data     portal     (https://iagos.aeris-data.fr/download/),     the     NIES     portal
(https://db.cger.nies.go.jp/ged/en/datasetlist/index.html) for CONTRAIL (e.g., Machida et al., 2008) and Siberian
measurements (e.g., Sasakawa et al., 2017), the WDCGG data portal (https://gaw.kishou.go.jp/) for additional



flights over three other Japanese airports and Orléans, France ; Aircore campaigns data set can be downloaded
through the NOAA Global Monitoring Laboratory website (https://gml.noaa.gov/ccgg/arc/?id=144, Baier et al.,
2021) and the French AIrCore Program for atmospheric sampling (https://aircore.aeris-data.fr, Membrive et al.,
2017); TCCON observations (https://tccondata.org; e.g., Wunch et al., 2011, 2019), and use this evaluation to
weight the different models used in the $CH_4$ budget.

Next steps, in the short term, could include some development to be addressed by the top-down community to reduce
atmospheric transport errors:

-    Developing further methodologies to extract stratospheric partial column abundances from observations such as
TCCON data (Saad et al., 2014; Wang et al., 2014), Aircore (e.g. Andersen et al., 2018; Membrive et al., 2017) or,
ACE-FTS (De Mazière et al., 2018) or MIPAS (Glatthor et al., 2023) satellite data.
-    Combining SWIR and TIR measurements from space to better constrain the tropospheric column, from TROPOMI
and IASI for example in the MethanePlus ESA project (https://methaneplus.eu/#docs, Buchwitz etal., 2023) or
GOSAT (Kuze et al., 2020).
-    Porting transport models codes to run on Graphics processing Units (GPU) to achieve sub-degrees resolution global
inversions (Chevallier et al., 2023).

In the long run, developments within atmospheric transport models such as the implementation of hybrid vertical coordinates
(Patra et al., 2018) or of hexagonal-icosaedric grid with finer resolution (Dubos et al., 2015; Niwa et al., 2017, 2022; Lloret
et al., 2023), and improvements in the simulated boundary layer dynamics are promising to reduce atmospheric transport
errors.

## 8 Conclusions

We have built an updated global methane budget by using and synthesising a large ensemble of published methods and new
results using a consistent, transparent, and traceable approach, including atmospheric observations and inversions (top-down
models), process-based models for land surface emissions and atmospheric chemistry, and inventories of anthropogenic
emissions (bottom-up models and inventories). For the 2010-2019 decade, global $CH_4$ emissions are 575 Tg $CH_4$ yr$^{-1}$ (range
of 553-586 Tg $CH_4$ yr$^{-1}$), as estimated by top-down inversions. About 65% of global emissions are anthropogenic (range of
63-68%). Bottom-up models and inventories suggest larger global emissions (669 Tg $CH_4$ yr$^{-1}$ [512-849]) mostly because
of larger and more uncertain natural emissions from inland freshwater systems, natural wetlands, and geological leaks, and
likely some unresolved double counting of these sources. It is also likely that some of the individual bottom-up emission
estimates are too high, leading to larger global emissions from the bottom-up approach than the atmospheric constraints
suggest. However, the important progress in this update is that for the first time, the bottom-up and top-down budgets agree
within their uncertainty ranges. This is substantial progress toward defining more accurate global methane emissions.



The latitudinal breakdown inferred from the top-down approach reveals a dominant role of tropical emissions (~64%) compared to mid (~32%) and high (~4%) northern latitudes (above 60°N) emissions.

Our results, including an extended set of atmospheric inversions, are compared with the previous budget syntheses of Kirschke et al. (2013) and Saunois et al. (2016; 2020). They show overall good consistency when comparing the same decade (2000-2009) at the global and latitudinal scales. The magnitude and uncertainty of most natural or indirect anthropogenic sources have been revised and updated. In particular, this new budget benefits from large efforts and collaborations from the research community to provide improved estimates of the magnitude and uncertainty of the different freshwater sources and helps reduce the potential double counting at the global scale. Of note, newly available gridded datasets for lakes, ponds, reservoirs, streams, and rivers allow building latitudinal and regional estimates for all these sources for the first time in these estimates. In the next review, we hope to be able to reduce uncertainties in emissions from inland freshwater systems by better quantifying the emission factors of each contributing sub-systems (streams, rivers, lakes, ponds) and estimating double counting at regional scale or avoiding double counting by better defining the surface areas of each ecosystem. Another important priority for improvements is the uncertainty on the chemical loss of $CH_4$ which still needs to be better assessed in both the top-down and the bottom-up budgets. Building on the improvement of the points detailed in Sect. 7, our aim is to update this budget synthesis as a living review paper regularly (~every three or four years). Each update will produce a more recent decadal $CH_4$ budget, highlight changes in emissions and trends, and incorporate newly available data and model improvements.

It is still under debate why exactly there are sustained increase of atmospheric $CH_4$ (more than +5 ppb yr$^{-1}$) since 2007 (Nisbet et al., 2019; Turner et al., 2019). Some likely explanations, already introduced by Saunois et al. (2017) and further investigated by Jackson et al. (2020) and other studies, include, by decreasing order of certainty: 1) a positive contribution from microbial and fossil sources (e.g., Nisbet et al., 2019; Schwietzke et al., 2016; Jackson et al., 2020), a negative contribution from biomass burning emissions before 2014 (Giglio et al., 2013; Worden et al., 2017); 2) a negligible role of Arctic emission changes (e.g., Nisbet et al., 2019; Saunois et al., 2017); and 3) a tropical dominance of the increasing emissions (e.g., Saunois et al., 2017; Jackson et al., 2020; Wilson et al., 2021; Drinkwater et al., 2023). Although the accelerated atmospheric methane growth rate in 2020 (15.2 ppb/yr) has found some explanation with the impact of the world Pandemia in 2020, the sustained observed growth rates in 2021 (17.8 ppb/yr) and 2022 (14 ppb/yr) still challenge our understanding of the global methane cycle. While in Jackson et al. (2020), the increase in $CH_4$ emissions over the last two decades is attributed entirely to direct anthropogenic emissions, the uncertainty range from the GMB ensemble is large, and the contribution from natural emissions (wetlands) is still largely uncertain. Besides the decadal change in $CH_4$ emissions, large inter-annual variability can occur from these natural emissions. The recent high record of $CH_4$ growth rate highlights the potential of large variations from natural emissions from one year to another, in particular wetland emissions (e.g., Peng et al., 2022; Feng et al., 2023). These remain the challenges to be overcome in better quantifying global methane emissions.





The GCP will continue to support and coordinate the development of improved flux estimates for all budget components
and new underlying science to support improved modelling, acquisition of observations, and data integration. At regular
intervals (3-4 years), we will continue to bring all flux components together to produce an improved and updated global
CH₄ budget, and provide a global benchmark for other CH₄ products and assessments.
**9 Data availability**
The data presented here are made available in the belief that their dissemination will lead to greater understanding and new
scientific insights on the methane budget and changes to it, and help to reduce its uncertainties. The free availability of the
data does not constitute permission for publication of the data. For research projects, if the data used are essential to the
work to be published, or if the conclusion or results largely depend on the data, co-authorship should be considered. Full
contact details and information on how to cite the data are given in the accompanying database.
The accompanying database includes a netcdf file defining the regions used, an archive with the maps of prior fluxes used
in the top-down activity, an archive with data corresponding to Fig. 3 and 5, and one Excel file organised in the following
spreadsheets.
The file Global_Methane_Budget_2000-2020_v1.0.xlsx includes (1) a summary, (2) the methane observed mixing ratio and
growth rate from the four global networks (NOAA, AGAGE, CSIRO and UCI), (3) the evolution of global anthropogenic
methane emissions (including biomass burning emissions) used to produce Fig. 2, (4) the global and latitudinal budgets over
2000–2009 based on bottom-up approaches, (5) the global and latitudinal budgets over 2000–2009 based on top-down
approaches, (6) the global and latitudinal budgets over 2010–2019 based on bottom-up approaches, (7) the global and
latitudinal budgets over 2010– 2019 based on top-down approaches, (8) the global and latitudinal budgets for year 2020
based on bottom-up approaches, (9) the global and latitudinal budgets for year 2020  based on top-down approaches, and
(10) the list of contributors to contact for further information on specific data.
This database is available from ICOS Carbon Portal (https://doi.org/10.18160/GKQ9-2RHT, Martinez et al., 2024).
**Author contributions**.
MS, AM, and JT gathered the bottom-up and top-down data sets and performed the post processing and analysis.
MS, BP, PB, PeC, and RJ coordinated the global budget. MS, BP, PB, PeC, RJ, PP and PCi contributed to the update of the
full text and all coauthors appended comments. AM, ED, and XL produced the figures. DJB, NG, PH, AI, AJ, TK, TL, XL,
KMcD, JMe, JMu, SP, CP, WR, HT, YY, WZ, ZZ, Qing Z, Qiuan Z and Qianlai Z performed surface land model simulations
to compute wetland emissions. GA, DB, SC, BRD, GE, MAH, GH, MSJ, RL, SN, GRR, JAR, EHS, PRa, PRe, and TSW
provided data sets useful for natural emission estimates and/or contributed to text on bottom-up natural emissions. LHI, SJS,
TNF, GRvW, and MC provided anthropogenic data sets and contributed to the text for this section.  AM, JT, PP, DBe, RJ,





YN, AS, AT, and BZ performed atmospheric inversions to compute top-down methane emission estimates for sources and sinks. EJD, XL, DRB, PBK, JM, RJP, MR, MS, DWo, and YYo are PI of atmospheric observations used in top-down inversions and/or contributed the text describing atmospheric methane observations. FD, MS, and JT contributed to the bottom-up chemical sink section by providing data sets, processing data and/or contributing to the text. FMF provided data for the soil sink.

**Competing interests.** At least one of the (co-)authors is a member of the editorial board of Earth System Science Data.

**Acknowledgements**
This paper is the result of a collaborative international effort under the umbrella of the Global Carbon Project, a project of Future Earth and a research partner of the World Climate Research Programme (WCRP). We acknowledge all the people and institutions who provided the data used in the global methane budget as well as the institutions funding parts of this effort (see Table A3). We are very grateful for the help provided by Alex Vermeulen in publishing the Global Methane Budget dataset on the Integrated Carbon Observation System (ICOS) website. We acknowledge the modelling groups for making their simulations available for this analysis, the joint WCRP Stratosphere-troposphere Processes And their Role in Climate/International Global Atmospheric Chemistry (SPARC/IGAC) Chemistry-Climate Model Initiative (CCMI) for organising and coordinating the model data analysis activity, and the British Atmospheric Data Centre (BADC) for collecting and archiving the CCMI model output. We acknowledge the long-term support provided by the Commonwealth Scientific and Industrial Research Organisation (CSIRO) and the National Environmental Science Program - Climate Systems Hub to coordinate and support activities of the Global Carbon Project. We are grateful to the Emissions Databse for Global Atmospheric Research (EDGAR) team (M. Crippa, D. Guizzardi, F. Pagani, M. Banja, E. Schaaf, M. Muntean, W. Becker, F. Monforti-Ferrario) for the work needed to publish the EDGAR greenhouse gas emission datasets used in this work (https://edgar.jrc.ec.europa.eu/). We are particularly indebted to the dedicated station/instrumental operators/scientists that have gathered the data and ensured their high quality.
We acknowledge more specifically Katherine Jensen for her contribution to the Surface Water Microwave Product Series SWAMPS, Fortunat Joos for his contribution to simulations with the Land surface Processes and eXchanges model (LPX Bern), Ray Langenfeld for his contribution to CSIRO network, Paul Miller for his contribution to simulations with the Lund-Potsdam-Jena General Ecosystem Simulator (LPJ-GUESS), Peng Shushi for his contribution to simulations with the Organising Carbon and Hydrology In Dynamic Ecosystems (ORCHIDEE) model, Shamil Maksyutov for his contribution for simulations with the inverse model at the National Institute for Environmental Studies (NIES), Isobel Simpson for her contribution to the University of California Irvine (UCI) network, Paul Steele for his former contribution to CSIRO network, Ray Weiss for his contribution to the Advanced Global Atmospheric Gases Experiment (AGAGE) network, Christine Widenmeyer for her contribution with the Fire INventory from the National Center for Atmospheric Research



(FINN) database, Xiaoming Xu for his contribution to simulations with with the The Integrated Science Assessment Model
(ISAM), Yuanzhi Yao for his contribution to simulations with the Dynamicl Land Ecosystem Model (DLEM), Diego
Guizzardi for his contribution to EDGAR, Maria Tenkanen for her contribution with the Crabon Tracker – Europe (CTE)
outputs, Giulia Conchedda for her contribution to the Food and Agriculture Organization (FAO) database. FAOSTAT data
collection, analysis, and dissemination is funded through FAO regular budget funds. The contribution of relevant experts in
member countries is gratefully acknowledged. We acknowledge Juha Hatakka from the Finnish Meteorological Institute
(FMI) for making methane measurements at the Pallas station and sharing the data with the community. We thank Ariana
Sutton-Grier and Lisamarie Windham-Myers for reviewing an earlier version of this manuscript. Any use of trade, firm, or
product names is for descriptive purposes only and does not imply endorsement by the US Government.

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

Glossary of permafrost and related ground-ice terms, National Research Council of Canada. Associate Committee on
Geotechnical Research. Permafrost Subcommittee., 1988.
Harris, I., Jones, P.D., Osborn, T.J. and Lister, D.H.: Updated high-resolution grids of monthly climatic observations – the
CRU TS3.10 Dataset, Int. J. Climatol., 34: 623-642. https://doi.org/10.1002/joc.3711, 2014
Harrison, J.A., Y.T. Prairie, S. Mercier-Blais, and C. Soued: Year-2020 Global Distribution and Pathways of Reservoir
Methane and Carbon Dioxide Emissions According to the Greenhouse Gas From Reservoirs (G-res) Model. Global
Biogeochemical Cycles 35(6): p. e2020GB006888, 2021.
Heathcote, A. J., C. T. Filstrup, and J. A. Downing (2013): Watershed sediment losses to lakes accelerating despite
agricultural soil conservation efforts, *PLoS One*, 8(1), e53554, doi:10.1371/journal.pone.0053554, 2013
Heděnec, P., Jiménez, J.J., Moradi, J. *et al.*: Global distribution of soil fauna functional groups and their estimated litter
consumption across biomes. Sci Rep 12, 17362, https://doi.org/10.1038/s41598-022-21563-z, 2022
Hmiel, B., Petrenko, V. V., Dyonisius, M. N., Buizert, C., Smith, A. M., Place, P. F., Harth, C., Beaudette, R., Hua, Q.,
Yang, B., Vimont, I., Michel, S. E., Severinghaus, J. P., Etheridge, D., Bromley, T., Schmitt, J., Faïn, X., Weiss, R. F.
and Dlugokencky, E.: Preindustrial 14 CH 4 indicates greater anthropogenic fossil CH 4 emissions, Nature, 578(7795),
409–412, doi:10.1038/s41586-020-1991-8, 2020.
Ho, J.C., A.M. Michalak, and N. Pahlevan: Widespread global increase in intense lake phytoplankton blooms since the
1980s. Nature, 574(7780): p. 667-670, 2019
Hoesly, R. M., Smith, S. J., Feng, L., Klimont, Z., Janssens-Maenhout, G., Pitkanen, T., Seibert, J. J., Vu, L., Andres, R.
J., Bolt, R. M., Bond, T. C., Dawidowski, L., Kholod, N., Kurokawa, J. I., Li, M., Liu, L., Lu, Z., Moura, M. C. P.,
O'Rourke, P. R. and Zhang, Q.: Historical (1750–2014) anthropogenic emissions of reactive gases and aerosols from
the Community Emissions Data System (CEDS), Geosci Model Dev, 11(1), 369–408, doi:10.5194/gmd-11-369-2018,

2018.

Höglund-Isaksson, L.: Bottom-up simulations of methane and ethane emissions from global oil and gas systems 1980 to
2012, Environ. Res. Lett., 12(2), 024007, doi:10.1088/1748-9326/aa583e, 2017.
Höglund-Isaksson, L., Thomson, A., Kupiainen, K., Rao, S. and Janssens-Maenhout, G.: Anthropogenic methane sources,
emissions and future projections, Chapter 5 in AMAP Assessment 2015: Methane as an Arctic Climate Forcer, p. 39-
59, available at http://www.amap.no/documents/doc/AMAP-Assessment-2015-Methane-as-an-Arctic-climate-
forcer/1285., 2015.
Höglund-Isaksson, L., Gómez-Sanabria, A., Klimont, Z., Rafaj, P., Schöpp, W.,: Technical potentials and costs for
reducing global anthropogenic methane emissions in the 2050 timeframe -results from the GAINS model, Environ.
Res. Comm. 2(2), https://iopscience.iop.org/article/10.1088/2515-7620/ab7457 , 2020



Holgerson, M. A. and Raymond, P. A.: Large contribution to inland water CO 2 and CH 4 emissions from very small ponds, Nat. Geosci., 9(3), 222–226, doi:10.1038/ngeo2654, 2016.

Holmes, C. D., Prather, M. J., Søvde, O. A. and Myhre, G.: Future methane, hydroxyl, and their uncertainties: key climate and emission parameters for future predictions, Atmospheric Chem. Phys., 13(1), 285–302, doi:10.5194/acp-13-285-2013, 2013.

Hopcroft,P.O. , P.J. Valdes & D.J. Beerling, (2011). Simulating idealised Dansgaard-Oeschger events and their potential influence on the global methane cycle, Quaternary Science Reviews, 30, 3258-3268,

doi: 10.1016/j.quascirev.2011.08.01., 2011

Hossaini, R., Chipperfield, M. P., Saiz-Lopez, A., Fernandez, R., Monks, S., Feng, W., Brauer, P. and Glasow, R. von: A global model of tropospheric chlorine chemistry: Organic versus inorganic sources and impact on methane oxidation, J. Geophys. Res. Atmospheres, 121(23), 14,271-14,297, doi:10.1002/2016JD025756, 2016.

Houweling, S., Bergamaschi, P., Chevallier, F., Heimann, M., Kaminski, T., Krol, M., Michalak, A. M. and Patra, P.: Global inverse modeling of CH4 sources and sinks: an overview of methods, Atmospheric Chem. Phys., 17(1), 235–256, doi:10.5194/acp-17-235-2017, 2017.

M. Hovland, A.G. Judd, R.A. Burke: The global flux of methane from shallow submarine sediments, Chemosphere, Volume 26, Issues 1–4, Pages 559-578, doi:10.1016/0045-6535(93)90442-8., 1993

Howarth, R. W.: Ideas and perspectives: is shale gas a major driver of recent increase in global atmospheric methane?, Biogeosciences, 16(15), 3033–3046, doi:10.5194/bg-16-3033-2019, 2019.

Hu, H., Landgraf, J., Detmers, R., Borsdorff, T., Brugh, J. A. de, Aben, I., Butz, A. and Hasekamp, O.: Toward Global Mapping of Methane With TROPOMI: First Results and Intersatellite Comparison to GOSAT, Geophys. Res. Lett., 45(8), 3682–3689, doi:10.1002/2018GL077259, 2018.

Hugelius, G., Tarnocai, C., Broll, G., Canadell, J. G., Kuhry, P., & Swanson, D. K.. The Northern Circumpolar Soil Carbon Database: spatially distributed datasets of soil coverage and soil carbon storage in the northern permafrost regions. *Earth System Science Data*, 5(1), 3–13. https://doi.org/10.5194/essd-5-3-2013, 2013

Hugelius, G., Strauss, J., Zubrzycki, S., Harden, J. W., Schuur, E. A. G., Ping, C. L., Schirrmeister, L., Grosse, G., Michaelson, G. J., Koven, C. D., O'Donnell, J. A., Elberling, B., Mishra, U., Camill, P., Yu, Z., Palmtag, J. and Kuhry, P.: Estimated stocks of circumpolar permafrost carbon with quantified uncertainty ranges and identified data gaps, Biogeosciences, 11(23), 6573–6593, doi:10.5194/bg-11-6573-2014, 2014.

Hugelius, Gustaf, Loisel, J., Chadburn, S., Jackson, R. B., Jones, M., MacDonald, G., et al. : Large stocks of peatland carbon and nitrogen are vulnerable to permafrost thaw. Proceedings of the National Academy of Sciences, 117(34), 20438–20446. https://doi.org/10.1073/pnas.1916387117, 2020

Hugelius, G., Ramage, J.L., Burke, E.J., Chatterjee, A., Smallman, T.L., Aalto, T., Bastos, A., Biasi, C., Canadell, J.G., Chandra, N. and Chevallier, F., et al. Two decades of permafrost region CO2, CH4, and N2O budgets suggest a small





net greenhouse gas source to the atmosphere. Preprint in ESS Open Archive. September 11, 2023. DOI: 10.22541/essoar.169444320.01914726/v1, 2023

IEA, *Coal Information: Overview*, IEA, Paris https://www.iea.org/reports/coal-information-overview, License: CC BY 4.0, Accessed 17 January 2024, 2021

IEA (2023), Energy Statistics Data Browser, IEA, Paris https://www.iea.org/data-and-statistics/data-tools/energy-statistics-data-browser, Accessed 17 January 2024, 2023a

IEA, US natural gas production by source, 2013-2023, IEA, Paris https://www.iea.org/data-and-statistics/charts/us-natural-gas-production-by-source-2013-2023, IEA. Licence: CC BY 4.0,Accessed 17 January 2024, 2023b

Imasu, R.; Matsunaga, T.; Nakajima, M.; Yoshida, Y.; Shiomi, K.; Morino, I.; Saitoh, N.; Niwa, Y.; Someya, Y.; Oishi, Y.; et al. Greenhouse Gases Observing SATellite 2 (GOSAT-2): Mission Overview. Prog. Earth Planet. Sci., *10*, 33., 2023,

Inoue, M., Morino, I., Uchino, O., Nakatsuru, T., Yoshida, Y., Yokota, T., Wunch, D., Wennberg, P. O., Roehl, C. M., Griffith, D. W. T., Velazco, V. A., Deutscher, N. M., Warneke, T., Notholt, J., Robinson, J., Sherlock, V., Hase, F., Blumenstock, T., Rettinger, M., Sussmann, R., Kyrö, E., Kivi, R., Shiomi, K., Kawakami, S., Mazière, M. D., Arnold, S. G., Feist, D. G., Barrow, E. A., Barney, J., Dubey, M., Schneider, M., Iraci, L. T., Podolske, J. R., Hillyard, P. W., Machida, T., Sawa, Y., Tsuboi, K., Matsueda, H., Sweeney, C., Tans, P. P., Andrews, A. E., Biraud, S. C., Fukuyama, Y., Pittman, J. V., Kort, E. A. and Tanaka, T.: Bias corrections of GOSAT SWIR XCO2 and XCH4 with TCCON data and their evaluation using aircraft measurement data, Atmospheric Meas. Tech., 9(8), 3491–3512, doi:10.5194/amt-9-3491-2016, 2016.

IPCC: Good Practice Guidance and Uncertainty Management in National Greenhouse Gas Inventories. Intergovernmental Panel on Climate Change, National Greenhouse Gas Inventories Programme. Montreal, IPCC-XVI/Doc.10(1.IV.2000), May 2000., 2000.

IPCC: Climate change 2001: The scientific basis. Contribution of working group I to the third assessment report of the Intergovernmental Panel on Climate Change, Cambridge University Press, Cambridge, United Kingdom and New York, NY, USA., 2001.

IPCC: IPCC Guidelines for National Greenhouse Gas Inventories. The National Greenhouse Gas Inventories Programme, Eggleston H.S., Buendia L., Miwa K., Ngara T. and Tanabe K. (eds). The Intergovernmental Panel on Climate Change, IPCC TSU NGGIP, IGES. Institute for Global Environmental Strategy, Hayama, Kanagawa, Japan. Available online at: http://www.ipcc-nggip.iges.or.jp/support/Primer_2006GLs.pdf., 2006.

IPCC: 2019 Refinement to the 2006 IPCC Guidelines for National Greenhouse Gas Inventories — IPCC. [online] Available from: https://www.ipcc.ch/report/2019-refinement-to-the-2006-ipcc-guidelines-for-national-greenhouse-gas-inventories/ (Accessed 17 March 2020), 2019.

Ito, A. and Inatomi, M.: Use of a process-based model for assessing the methane budgets of global terrestrial ecosystems



and evaluation of uncertainty, Biogeosciences, 9(2), 759–773, doi:10.5194/bg-9-759-2012, 2012.
Jacob, D. J., Varon, D. J., Cusworth, D. H., Dennison, P. E., Frankenberg, C., Gautam, R., Guanter, L., Kelley, J.,
McKeever, J., Ott, L. E., Poulter, B., Qu, Z., Thorpe, A. K., Worden, J. R., and Duren, R. M.: Quantifying methane
emissions from the global scale down to point sources using satellite observations of atmospheric methane, Atmos.
Chem. Phys., 22, 9617–9646, https://doi.org/10.5194/acp-22-9617-2022, 2022.
Jackson, R. B., Down, A., Phillips, N. G., Ackley, R. C., Cook, C. W., Plata, D. L. and Zhao, K.: Natural gas pipeline
leaks across Washington, D.C, Environ. Sci. Technol., 48(3), 2051–2058, doi:10.1021/es404474x, 2014a.
Jackson, R. B., Vengosh, A., Carey, J. W., Davies, R. J., Darrah, T. H., O'Sullivan, F. and Pétron, G.: The Environmental
Costs and Benefits of Fracking, Annu. Rev. Environ. Resour., 39, 327–362, doi:10.1146/annurev-environ-031113-
144051, 2014b.
Jackson, R. B., Saunois, M., Bousquet, P., Canadell, J. G., Poulter, B., Stavert, A. R., Poulter, B., Bergamaschi, P., Niwa,
Y., Segers, A., Tsuruta, A.: Increasing anthropogenic methane emissions arise equally from agricultural and fossil fuel
sources, Environmental Research Letters, 15, 7, https://doi.org/10.1088/1748-9326/ab9ed2, 2020
Jamali, H., Livesley, S.J., Dawes, T.Z. *et al.* Termite mound emissions of $CH_4$ and $CO_2$ are primarily determined by
seasonal changes in termite biomass and behaviour. *Oecologia* 167, 525–534, doi.org/10.1007/s00442-011-1991-3,

2011

Janssens-Maenhout, G., Crippa, M., Guizzardi, D., Muntean, M., Schaaf, E., Dentener, F., Bergamaschi, P., Pagliari, V.,
Olivier, J., Peters, J., van Aardenne, J., Monni, S., Doering, U., Petrescu, R., Solazzo, E. and Oreggioni, G.: EDGAR
v4.3.2 Global Atlas of the three major Greenhouse Gas Emissions for the period 1970-2012, Earth Syst Sci Data
Discuss, 2019, 1–52, doi:10.5194/essd-2018-164, 2019.
JAXA: GOSAT-2: Greenhouse gases Observing SATellite-2@ibuki2_JAXA"IBUKI-2", [online] Available from:
https://global.jaxa.jp/projects/sat/gosat2/index.html (Accessed 25 March 2020), 2019.
Jensen, K. and Mcdonald, K.: Surface Water Microwave Product Series Version 3: A Near-Real Time and 25-Year
Historical Global Inundated Area Fraction Time Series From Active and Passive Microwave Remote Sensing, IEEE
Geosci. Remote Sens. Lett., 16(9), 1402–1406, doi:10.1109/LGRS.2019.2898779, 2019.
Jiang, Y., Groenigen, K. J. van, Huang, S., Hungate, B. A., Kessel, C. van, Hu, S., Zhang, J., Wu, L., Yan, X., Wang, L.,
Chen, J., Hang, X., Zhang, Y., Horwath, W. R., Ye, R., Linquist, B. A., Song, Z., Zheng, C., Deng, A. and Zhang, W.:
Higher yields and lower methane emissions with new rice cultivars, Glob. Change Biol., 23(11), 4728–4738,
doi:10.1111/gcb.13737, 2017.
Johnson, D. E., Phetteplace, H. W. and Seidl, A. F.: Methane, nitrous oxide and carbon dioxide emissions from ruminant
livestock production systems, edited by J. Takahashi and B. A. Young, pp. 77–85, Elsevier, Amsterdam, The
Netherlands., 2002.
Johnson, M.S., E. Matthews, J. Du, V. Genovese, and D. Bastviken, Methane Emission From Global Lakes: New

 

Spatiotemporal Data and Observation-Driven Modeling of Methane Dynamics Indicates Lower Emissions, Journal of
Geophysical Research: Biogeosciences, 127(7): p. e2022JG006793, 2022
Johnson, M. S., E. Matthews, D. Bastviken, B. Deemer, J. Du, and V. Genovese, Spatiotemporal methane emission from
global reservoirs, Journal of Geophysical Research: Biogeosciences, 126, e2021JG006305,
https://doi.org/10.1029/2021JG006305, 2021
Judd, A.G. (2004). Natural seabed seeps as sources of atmospheric methane, Environ. Geol., 46, 988–996, 2004.
Jung, M., Reichstein, M., Margolis, H. A., Cescatti, A., Richardson, A. D., Arain, M. A., Arneth, A., Bernhofer, C., Bonal,
D., Chen, J., Gianelle, D., Gobron, N., Kiely, G., Kutsch, W., Lasslop, G., Law, B. E., Lindroth, A., Merbold, L.,
Montagnani, L., Moors, E. J., Papale, D., Sottocornola, M., Vaccari, F., and Williams, C.: Global patterns of land-
atmosphere fluxes of carbon dioxide, latent heat, and sensible heat derived from eddy covariance, satellite, and
meteorological observations, J. Geophys. Res., 116, G00J07,https://doi.org/10.1029/2010jg001566, 2011.
Kai, F. M., Tyler, S. C., Randerson, J. T. and Blake, D. R.: Reduced methane growth rate explained by decreased Northern
Hemisphere microbial sources, Nature, 476(7359), 194–197, 2011.
Kaiser, J. W., Heil, A., Andreae, M. O., Benedetti, A., Chubarova, N., Jones, L., Morcrette, J. J., Razinger, M., Schultz,
M. G., Suttie, M. and van der Werf, G. R.: Biomass burning emissions estimated with a global fire assimilation system
based on observed fire radiative power, Biogeosciences, 9(1), 527–554, doi:10.5194/bg-9-527-2012, 2012.
Kallingal, J. T., Lindström, J., Miller, P. A., Rinne, J., Raivonen, M., and Scholze, M.: Optimising $CH_4$ simulations from
the LPJ-GUESS model v4.1 using an adaptive MCMC algorithm, Geosci. Model Dev. Discuss. [preprint],
https://doi.org/10.5194/gmd-2022-302, in review, 2023.
Karion, A., Sweeney, C., Pétron, G., Frost, G., Michael Hardesty, R., Kofler, J., Miller, B. R., Newberger, T., Wolter, S.,
Banta, R., Brewer, A., Dlugokencky, E., Lang, P., Montzka, S. A., Schnell, R., Tans, P., Trainer, M., Zamora, R. and
Conley, S.: Methane emissions estimate from airborne measurements over a western United States natural gas field,
Geophys. Res. Lett., 40(16), 4393–4397, doi:10.1002/grl.50811, 2013.
Karl, D., Beversdorf, L., Björkman, K. *et al.* Aerobic production of methane in the sea. *Nature Geosci* **1**, 473–478 ,
https://doi.org/10.1038/ngeo234), 2008
Karlson, M., and Bastviken, D.: Multi-Source Mapping of Peatland Types Using Sentinel-1, Sentinel-2, and Terrain
Derivatives—A Comparison Between Five High-Latitude Landscapes. Journal of Geophysical Research:
Biogeosciences 128, e2022JG007195. https://doi.org/10.1029/2022JG007195, 2023
Keppler, F., Hamilton, J. T. G., Brass, M. and Rockmann, T.: Methane emissions from terrestrial plants under aerobic
conditions, Nature, 439, 187–191, doi:10.1038/nature04420, 2006.
Kholod, N., Evans, M., Pilcher, R. C., Roshchanka, V., Ruiz, F., Coté, M. and Collings, R.: Global methane emissions
from coal mining to continue growing even with declining coal production, J. Clean. Prod., 256, 120489,
doi:10.1016/j.jclepro.2020.120489, 2020.



Kim H., Global Soil Wetness Project Phase 3 Atmospheric Boundary Conditions (Experiment 1) [Data set]. Data
Integration and Analysis System (DIAS)., https://doi.org/10.20783/DIAS.501, 2017

King, J.R., Warren, R.J., Bradford, M.A. Correction: Social Insects Dominate Eastern US Temperate Hardwood Forest
Macroinvertebrate Communities in Warmer Regions. PLOS ONE 8(10): 10.1371/annotation/87285c86-f1df-4f8b-
bc08-d64643d351f4, 2013.

Kirk, L, and MJ Cohen: River Corridor Sources Dominate $CO_2$ Emissions From a Lowland River Network. Journal of
Geophysical Research, Biogeosciences, 128(1), e2022JG006954, 2023.

Kirschke, S., Bousquet, P., Ciais, P., Saunois, M., Canadell, J. G., Dlugokencky, E. J., Bergamaschi, P., Bergmann, D.,
Blake, D. R., Bruhwiler, L., Cameron-Smith, P., Castaldi, S., Chevallier, F., Feng, L., Fraser, A., Heimann, M.,
Hodson, E. L., Houweling, S., Josse, B., Fraser, P. J., Krummel, P. B., Lamarque, J. F., Langenfelds, R. L., Le Quere,
C., Naik, V., O'Doherty, S., Palmer, P. I., Pison, I., Plummer, D., Poulter, B., Prinn, R. G., Rigby, M., Ringeval, B.,
Santini, M., Schmidt, M., Shindell, D. T., Simpson, I. J., Spahni, R., Steele, L. P., Strode, S. A., Sudo, K., Szopa, S.,
van der Werf, G. R., Voulgarakis, A., van Weele, M., Weiss, R. F., Williams, J. E. and Zeng, G.: Three decades of
global methane sources and sinks, Nat. Geosci., 6(10), 813–823, doi:10.1038/ngeo1955, 2013.

Klauda, J. B. and Sandler, S. I.: Global distribution of methane hydrate in ocean sediment, Energy Fuels, 19(2), 459–470,
2005.

Kleinen, T., Brovkin, V. and Schuldt, R. J.: A dynamic model of wetland extent and peat accumulation: results for the
Holocene, Biogeosciences, 9(1), 235–248, doi:10.5194/bg-9-235-2012, 2012.

Kleinen, T., Mikolajewicz, U., and Brovkin, V.: Terrestrial methane emissions from the Last Glacial Maximum to the
preindustrial period, Clim. Past, 16, 575–595, doi:10.5194/cp-16-575-2020, 2020.

Kleinen, T., Gromov, S., Steil, B., and Brovkin, V.: Atmospheric methane underestimated in future climate projections,
Environ. Res. Lett., 16, 094006, doi:10.1088/1748-9326/ac1814, 2021.

Kleinen, T., Gromov, S., Steil, B., and Brovkin, V.: Atmospheric methane since the last glacial maximum was driven by
wetland sources, Clim. Past, 19, 1081–1099, doi:10.5194/cp-19-1081-2023, 2023

Knittel K and Boetius A : Anaerobic oxidation of methane: progress with an unknown process methane. Annu Rev
Microbiol 63:311–334, 2009

Knox, S. H., Jackson, R. B., Poulter, B., McNicol, G., Fluet-Chouinard, E., Zhang, Z., Hugelius, G., Bousquet, P.,
Canadell, J. G., Saunois, M., Papale, D., Chu, H., Keenan, T. F., Baldocchi, D., Torn, M. S., Mammarella, I., Trotta,
C., Aurela, M., Bohrer, G., Campbell, D. I., Cescatti, A., Chamberlain, S., Chen, J., Chen, W., Dengel, S., Desai, A.
R., Euskirchen, E., Friborg, T., Gasbarra, D., Goded, I., Goeckede, M., Heimann, M., Helbig, M., Hirano, T., Hollinger,
D. Y., Iwata, H., Kang, M., Klatt, J., Krauss, K. W., Kutzbach, L., Lohila, A., Mitra, B., Morin, T. H., Nilsson, M. B.,
Niu, S., Noormets, A., Oechel, W. C., Peichl, M., Peltola, O., Reba, M. L., Richardson, A. D., Runkle, B. R. K., Ryu,
Y., Sachs, T., Schäfer, K. V. R., Schmid, H. P., Shurpali, N., Sonnentag, O., Tang, A. C. I., Ueyama, M., Vargas, R.,





Vesala, T., Ward, E. J., Windham-Myers, L., Wohlfahrt, G. and Zona, D.: FLUXNET-CH4 Synthesis Activity:
Objectives, Observations, and Future Directions, Bull. Am. Meteorol. Soc., 100(12), 2607–2632, doi:10.1175/BAMS-
D-18-0268.1, 2019.

Knox, S. H., Bansal, S., McNicol, G., Schafer, K., Sturtevant, C., Ueyama, M., et al.:. Identifying dominant environmental
predictors of freshwater wetland methane fluxes across diurnal to seasonal time scales. *Global Change Biology*,
27(15), 3582–3604. https://doi.org/10.1111/gcb.15661, 2021

Kretschmer, K., Biastoch, A., Rüpke, L. and Burwicz, E.: Modeling the fate of methane hydrates under global warming,
Glob. Biogeochem Cycles, 29(5), 610–625, doi:1002/2014GB005011, 2015.

Kuhn, M.A., Varner, R.K., Bastviken, D., Crill, P., MacIntyre, S., Turetsky, M., Walter Anthony, K., McGuire, A.D., and
Olefeldt, D. (2021). BAWLD-CH4: a comprehensive dataset of methane fluxes from boreal and arctic ecosystems.
Earth Syst. Sci. Data 13, 5151-5189. 10.5194/essd-13-5151-2021.

Kuze A, Kikuchi N, Kataoka F, Suto H, Shiomi K, Kondo Y. Detection of Methane Emission from a Local Source Using
GOSAT Target Observations. *Remote Sensing*, 12(2):267. https://doi.org/10.3390/rs12020267, 2020

Kyzivat, E.D., L.C. Smith, F. Garcia-Tigreros, C. Huang, C. Wang, T. Langhorst, J.V. Fayne, M.E. Harlan, Y. Ishitsuka,
D. Feng, W. Dolan, L.H. Pitcher, K.P. Wickland, M.M. Dornblaser, R.G. Striegl, T.M. Pavelsky, D.E. Butman, and
C.J. Gleason, The Importance of Lake Emergent Aquatic Vegetation for Estimating Arctic-Boreal Methane Emissions.
Journal of Geophysical Research: Biogeosciences 127(6): p. e2021JG006635, 2022.

Lamarque, J. F., Shindell, D. T., Josse, B., Young, P. J., Cionni, I., Eyring, V., Bergmann, D., Cameron-Smith, P., Collins,
2946        W. J., Doherty, R., Dalsoren, S., Faluvegi, G., Folberth, G., Ghan, S. J., Horowitz, L. W., Lee, Y. H., MacKenzie, I.
2947        A., Nagashima, T., Naik, V., Plummer, D., Righi, M., Rumbold, S. T., Schulz, M., Skeie, R. B., Stevenson, D. S.,
Strode, S., Sudo, K., Szopa, S., Voulgarakis, A. and Zeng, G.: The Atmospheric Chemistry and Climate Model
Intercomparison Project (ACCMIP): overview and description of models, simulations and climate diagnostics, Geosci.
Model Dev., 6(1), 179–206, doi:10.5194/gmd-6-179-2013, 2013.

Lamb, B. K., Edburg, S. L., Ferrara, T. W., Howard, T., Harrison, M. R., Kolb, C. E., Townsend-Small, A., Dyck, W.,
Possolo, A. and Whetstone, J. R.: Direct Measurements Show Decreasing Methane Emissions from Natural Gas Local
Distribution Systems in the United States, Environ. Sci. Technol., 49(8), 5161–5169, doi:10.1021/es505116p, 2015.

Lan, X., K.W. Thoning, and E.J. Dlugokencky: Trends in globally-averaged CH4, N2O, and SF6 determined from NOAA
Global Monitoring Laboratory measurements. Version 2024-02, https://doi.org/10.15138/P8XG-AA10, 2024

Lange S., WFDE5 over land merged with ERA5 over the ocean (W5E5). V. 1.0. 2019. doi:10.5880/pik.2019.023, 2019
Laruelle, G. G., Dürr, H. H., Lauerwald, R., Hartmann, J., Slomp, C. P., Goossens, N. and Regnier, P. A. G.: Global multi-
scale segmentation of continental and coastal waters from the watersheds to the continental margins, Hydrol. Earth
Syst. Sci., 17(5), 2029–2051, doi:10.5194/hess-17-2029-2013, 2013.



Laruelle, G. G., Rosentreter, J. A. & Regnier P.: Extrapolation based regionalized re-evaluation of the global estuarine
surface area. Preprint at Earth ArXiv https://doi.org/10.31223/X5X664, 2023.
Lassey, K. R., Etheridge, D. M., Lowe, D. C., Smith, A. M. and Ferretti, D. F.: Centennial evolution of the atmospheric
methane budget: what do the carbon isotopes tell us?, Atmospheric Chem. Phys., 7(8), 2119–2139, 2007a.

Lassey, K. R., Lowe, D. C. and Smith, A. M.: The atmospheric cycling of radiomethane and the "fossil fraction" of the
methane source, Atmospheric Chem. Phys., 7(8), 2141–2149, 2007b.

Lauerwald, R., Allen, G.H., Deemer, B.R., Liu, S., Maavara, T., Raymond, P., Alcott, L., Bastviken, D., Hastie, A.,
Holgerson, M.A., Johnson, M.S., Lehner, B., Lin, P., Marzadri, A., Ran, L., Tian, H., Yang, X., Yao, Y. and Regnier,
P. Inland water greenhouse gas budgets for RECCAP2: 1. State-of- the-art of global scale assessments. *Global
Biogeochemical Cycles*, *37*, e2022GB007657. https://doi. org/10.1029/2022GB007657, 2023a.

Lauerwald, R., Allen, G.H., Deemer, B.R., Liu, S., Maavara, T., Raymond, P., Alcott, L., Bastviken, D., Hastie, A.,
Holgerson, M.A., Johnson, M.S., Lehner, B., Lin, P., Marzadri, A., Ran, L., Tian, H., Yang, X., Yao, Y. and Regnier,
P. Inland water greenhouse gas budgets for RECCAP2: 2 Regionalization and homogenization of estimates following
the RECCAP2 framework, *Global Biogeochemical Cycles*, *37*, e2022GB007658. https://doi.
org/10.1029/2022GB007658, 2023b.

Laughner, J. L., Neu, J. L., Schimel, D., Wennberg, P. O., Barsanti, K., Bowman, K. W., Chatterjee, A., Croes, B. E.,
Fitzmaurice, H. L., Henze, D. K., Kim, J., Kort, E. A., Liu, Z., Miyazaki, K., Turner, A. J., Anenberg, S., Avise, J.,
Cao, H., Crisp, D., de Gouw, J., Eldering, A., Fyfe, J. C., Goldberg, D. L., Gurney, K. R., Hasheminassab, S., Hopkins,
F., Ivey, C. E., Jones, D. B. A., Liu, J., Lovenduski, N. S., Martin, R. V., McKinley, G. A., Ott, L., Poulter, B., Ru, M.,
Sander, S. P., Swart, N., Yung, Y. L., and Zeng, Z.-C.: Societal shifts due to COVID-19 reveal large-scale complexities
and feedbacks between atmospheric chemistry and climate change, P. Natl. Acad. Sci. USA, 118, e2109481118,
https://doi.org/10.1073/pnas.2109481118, 2021.

Lauvaux, T., Giron, C., Mazzolini, M., d'Aspremont, A., Duren, R., Cusworth, D., Shindell, D., and Ciais, P.: Global
assessment of oil and gas methane ultra-emitters, Science, 375, 557–561, https://doi.org/10.1126/science.abj4351,
2022.

Lelieveld, J., Crutzen, P. J. and Dentener, F. J.: Changing concentration, lifetime and climate forcing of atmospheric
methane, Tellus Ser. B-Chem. Phys. Meteorol., 50(2), 128–150, doi:10.1034/j.1600-0889. 1998. t01-1-00002.x, 1998.

Lelieveld, J., Lechtenbohmer, S., Assonov, S. S., Brenninkmeijer, C. A. M., Dienst, C., Fischedick, M. and Hanke, T.:
Greenhouse gases: Low methane leakage from gas pipelines, Nature, 434(7035), 841–842, doi:10.1038/434841a, 2005.

Lenhart, K., Klintzsch, T., Langer, G., Nehrke, G., Bunge, M., Schnell, S. and Keppler, F.: Evidence for methane
production by the marine algae *Emiliania huxleyi*, Biogeosciences, 13(10), 3163–3174, doi:10.5194/bg-13-3163-2016,
2016.

Lewan, M. D.: Comment on Ideas and perspectives: is shale gas a major driver of recent increase in global atmospheric





2993 methane? by Robert W. Howarth (2019), Biogeosciences Discuss., 1–10, doi:10.5194/bg-2019-419, 2020.

2994 Li, C., Frolking, S., Xiao, X., Moore, B., Boles, S., Qiu, J., Huang, Y., Salas, W. and Sass, R.: Modeling impacts of farming management alternatives on $CO_2$, $CH_4$, and $N_2O$ emissions: A case study for water management of rice agriculture of China, Glob. Biogeochem. Cycles, 19(3), doi:10.1029/2004gb002341, 2005.

2997 Li T, Huang Y, Zhang W, Song C: CH4MODwetland: A biogeophysical model for simulating methane emissions from natural wetlands. Ecological Modelling 221: 666–680, 2010.

2999 Li, Y., J. Shang, C. Zhang, W. Zhang, L. Niu, L. Wang, and H. Zhang, The role of freshwater eutrophication in greenhouse gas emissions: A review. Science of The Total Environment, 768: p. 144582., 2021

3001 Lin, X., Indira, N. K., Ramonet, M., Delmotte, M., Ciais, P., Bhatt, B. C., Reddy, M. V., Angchuk, D., Balakrishnan, S., Jorphail, S., Dorjai, T., Mahey, T. T., Patnaik, S., Begum, M., Brenninkmeijer, C., Durairaj, S., Kirubagaran, R., Schmidt, M., Swathi, P. S., Vinithkumar, N. V., Yver Kwok, C. and Gaur, V. K.: Long-lived atmospheric trace gases measurements in flask samples from three stations in India, Atmospheric Chem. Phys., 15(17), 9819–9849, doi:10.5194/acp-15-9819-2015, 2015.

3006 Lin, P., Pan, M., Beck, H. E., Yang, Y., Yamazaki, D., Frasson, R., et al.: Global reconstruction of naturalized river flows at 2.94 million reaches. *Water Resources Research*, 55, 6499–6516. https://doi.org/10.1029/2019WR025287, 2019

3009 Liu, Z., Guan, D., Wei, W., Davis, S. J., Ciais, P., Bai, J., Peng, S., Zhang, Q., Hubacek, K., Marland, G., Andres, R. J., Crawford-Brown, D., Lin, J., Zhao, H., Hong, C., Boden, T. A., Feng, K., Peters, G. P., Xi, F., Liu, J., Li, Y., Zhao, Y., Zeng, N. and He, K.: Reduced carbon emission estimates from fossil fuel combustion and cement production in China, Nature, 524(7565), 335–338, doi:10.1038/nature14677, 2015.

3013 Liu, G., Peng, S., Lin, X., Ciais, P., Li, X., Xi, Y., Lu, Z., Chang, J., Saunois, M., Wu, Y., Patra, P., Chandra, N., Zeng, H., and Piao, S.: Recent Slowdown of Anthropogenic Methane Emissions in China Driven by Stabilized Coal Production, Environ. Sci. Technol. Lett., 8, 739–746, https://doi.org/10.1021/acs.estlett.1c00463, 2021a.

3016 Liu, S., Fang, S., Liu, P., Liang, M., Guo, M., and Feng, Z.: Measurement report: Changing characteristics of atmospheric CH4 in the Tibetan Plateau: records from 1994 to 2019 at the Mount Waliguan station, Atmos. Chem. Phys., 21, 393–413, https://doi.org/10.5194/acp-21-393-2021, 2021b

3019 Liu, S., C. Kuhn, G. Amatulli, K. Aho, D.E. Butman, G.H. Allen, P. Lin, M. Pan, D. Yamazaki, C. Brinkerhoff, C. Gleason, X. Xia, and P.A. Raymond: The importance of hydrology in routing terrestrial carbon to the atmosphere via global streams and rivers. Proceedings of the National Academy of Sciences, 119(11): p. e2106322119, 2022.

3022 Lloret, Z., Chevallier, F., Cozic, A., Remaud, M., and Meurdesoif, Y.: Simulating the variations of carbon dioxide in the global atmosphere on the hexagonal grid of DYNAMICO coupled with the LMDZ6 model, Geosci. Model Dev. Discuss. [preprint], https://doi.org/10.5194/gmd-2023-140, in review, 2023.

3025 Locatelli, R., Bousquet, P., Saunois, M., Chevallier, F. and Cressot, C.: Sensitivity of the recent methane budget to LMDz



sub-grid-scale physical parameterizations, Atmospheric Chem. Phys., 15(17), 9765–9780, doi:10.5194/acp-15-9765-2015, 2015.

Lohila, A., Aalto, T., Aurela, M., Hatakka, J., Tuovinen, J.-P., Kilkki, J., Penttilä, T., Vuorenmaa, J., Hänninen, P., Sutinen, R., Viisanen, Y. and Laurila, T.: Large contribution of boreal upland forest soils to a catchment-scale CH4 balance in a wet year, Geophys. Res. Lett., 43(6), 2946–2953, doi:10.1002/2016gl067718, 2016.

Lorente, A., Borsdorff, T., Martinez-Velarte, M. C., and Landgraf, J.: Accounting for surface reflectance spectral features in TROPOMI methane retrievals, Atmos. Meas. Tech., 16, 1597–1608, https://doi.org/10.5194/amt-16-1597-2023, 2023.

Lu, X., Jacob, D. J., Zhang, Y., Maasakkers, J. D., Sulprizio, M. P., Shen, L., Qu, Z., Scarpelli, T. R., Nesser, H., Yantosca, R. M., Sheng, J., Andrews, A., Parker, R. J., Boesch, H., Bloom, A. A., and Ma, S.: Global methane budget and trend, 2010–2017: complementarity of inverse analyses using in situ (GLOBALVIEWplus CH4 ObsPack) and satellite (GOSAT) observations, Atmos. Chem. Phys., 21, 4637–4657, https://doi.org/10.5194/acp-21-4637-2021, 2021

Lu, X., Jacob, D. J., Zhang, Y., Maasakkers, J. D., Zhang, Y., Qu, Z., Chen, Z., Sulprizio, M. P., Varon, D., Hmiel, H., Park, R. J., Boesch, H., and Fan, S.: Observation-derived 2010–2019 trends in methane emissions and intensities from US oil and gas fields tied to activity metrics, P. Natl. Acad. Sci. USA, 120, e2217900120, https://doi.org/10.1073/pnas.2217900120, 2023.

Maasakkers, J. D., Jacob, D. J., Sulprizio, M. P., Scarpelli, T. R., Nesser, H., Sheng, J.-X., Zhang, Y., Hersher, M., Bloom, A. A., Bowman, K. W., Worden, J. R., Janssens-Maenhout, G., and Parker, R. J.: Global distribution of methane emissions, emission trends, and OH concentrations and trends inferred from an inversion of GOSAT satellite data for 2010–2015, Atmos. Chem. Phys., 19, 7859–7881, https://doi.org/10.5194/acp-19-7859-2019, 2019.

Maasakkers, J. D., Jacob, D. J., Sulprizio, M. P., Scarpelli, T. R., Nesser, H., Sheng, J., Zhang, Y., Lu, X., Bloom, A. A., Bowman, K. W., Worden, J. R., and Parker, R. J.: 2010–2015 North American methane emissions, sectoral contributions, and trends: a high-resolution inversion of GOSAT observations of atmospheric methane, Atmos. Chem. Phys., 21, 4339–4356, https://doi.org/10.5194/acp-21-4339-2021, 2021.

Maavara, T., Lauerwald, R., Regnier, P. and Van Capellen P.: Global perturbation of organic carbon cycling by river damming, Nat Commun 8, 153, https://doi.org/10.1038/ncomms15347, 2017.

Machida, T., H. Matsueda, Y. Sawa, Y. Nakagawa, K. Hirotani, N. Kondo, K. Goto, N. Nakazawa, K. Ishikawa and T. Ogawa: Worldwide measurements of atmospheric CO2 and other trace gas species using commercial airlines, J. Atmos. Oceanic Technol., 25(10), 1744-1754, doi:10.1175/2008JTECHA1082.1, 2008

Maksyutov, S., Oda, T., Saito, M., Janardanan, R., Belikov, D., Kaiser, J. W., Zhuravlev, R., Ganshin, A., Valsala, V. K., Andrews, A., Chmura, L., Dlugokencky, E., Haszpra, L., Langenfelds, R. L., Machida, T., Nakazawa, T., Ramonet, M., Sweeney, C. and Worthy, D.: Technical note: A high-resolution inverse modelling technique for estimating surface CO2 fluxes based on the NIES-TM – FLEXPART coupled transport model and its adjoint, Atmospheric Chem.



Phys. Discuss., 1–33, doi:10.5194/acp-2020-251, 2020.

Malerba, M.E., T. de Kluyver, N. Wright, L. Schuster, and P.I. Macreadie: Methane emissions from agricultural ponds are
underestimated in national greenhouse gas inventories. Communications Earth & Environment, 3(1): p. 306, 2022

Maltby, J., L. Steinle, C. R. Löscher, H. W. Bange, M. A. Fischer, M. Schmidt, and T. Treude: Microbial methanogenesis
in the sulfate-reducing zone of sediments in the Eckernförde Bay, SW Baltic Sea. Biogeosciences **15**: 137–157.
doi:10.5194/bg-15-137-2018, 2018

Manning, F. C. , Kho, L. K., Hill, T. C., Cornulier, T., and Teh, Y A: Carbon Emissions From Oil Palm Plantations on
Peat Soil, Front. For. Glob. Change, Sec. Tropical Forests, Volume 2 , https://doi.org/10.3389/ffgc.2019.0003, 2019

Mannisenaho V, Tsuruta A, Backman L, Houweling S, Segers A, Krol M, Saunois M, Poulter B, Zhang Z, Lan X, et al.
Global Atmospheric $\delta^{13}CH_4$ and $CH_4$ Trends for 2000–2020 from the Atmospheric Transport Model TM5 Using $CH_4$
from Carbon Tracker Europe–$CH_4$ Inversions. *Atmosphere*, 14(7):1121. https://doi.org/10.3390/atmos14071121, 2023

van Marle, M. J. E., Kloster, S., Magi, B. I., Marlon, J. R., Daniau, A.-L., Field, R. D., Arneth, A., Forrest, M., Hantson,
S., Kehrwald, N. M., Knorr, W., Lasslop, G., Li, F., Mangeon, S., Yue, C., Kaiser, J. W. and Werf, G. R. van der:
Historic global biomass burning emissions for CMIP6 (BB4CMIP) based on merging satellite observations with
proxies and fire models (1750–2015), Geosci. Model Dev., 10(9), 3329–3357, doi:10.5194/gmd-10-3329-2017, 2017.

Martinez, A., Saunois, M., Poulter B., Zhen, Z., Raymond, P., Regnier, P. Canadell, J. G., Jackson, R. B., Patra, P. K.,
Bousquet, P., Ciais, P., Dlugokencky, E.J., Lan, X., Allen, G., Bastviken,D., Beerling, D. J., Belikov, D., Blake, D.,
Castaldi, S., Crippa, M., Deemer, B.R., Dennison, F., Etiope, G., Gedney, N., Höglund-Isaksson, L., Holgerson, M.A.,
Hopcroft, P. O. , Hugelius, G., Ito, A., Jain, A. K., Janardanan, R., Johnson, M. S., Kleinen, T, Krummel, P. B.,
Lauerwald, R., Li, T., Liu, X., McDonald, K. C., Melton, J. R., Mühle, J., Müller, J., Murguia-Flores, F., Niwa, Y.,
Noce, S., Pan, S., Parker, R. J., Peng, C., Ramonet, M., Riley, W. J., Rocher-Ros, G., Rosentreter, J. A., Sasakawa,
3080        M., Segers A. , Smith, S. J., Stanley, E. H.,Thanwerdas, J., Tian, H., Tsuruta, A., Tubiello, F. N., Weber, T. S., van
der Werf, G. R., Worthy, D. E. J., Xi, Y., Yoshida Y. , Zhang, W. , Zheng, B. , Zhu, Qing , Zhu, Qiuan, and Zhuang,
Q.: *Supplemental data of the Global Carbon Project Methane Budget 2024* v1. [Data set],
https://doi.org/10.18160/GKQ9-2RHT, 2024

Matthews, E. and Fung, I.: Methane emission from natural wetlands: Global distribution, area, and environmental
characteristics of sources, Glob. Biogeochem. Cycles, 1(1), 61–86, doi:d10.1029/GB001i001p00061, 1987.

Mazzini A., Etiope G. (2017). Mud volcanism: an updated review. Earth Sci. Rev., 168, 81-112.
http://dx.doi.org/10.1016/j.earscirev.2017.03.001., 2017

McCalley, C. K., Woodcroft, B. J., Hodgkins, S. B., Wehr, R. A., Kim, E.-H., Mondav, R., Crill, P. M., Chanton, J. P.,
Rich, V. I., Tyson, G. W. and Saleska, S. R.: Methane dynamics regulated by microbial community response to
permafrost thaw, Nature, 514(7523), 478–481, doi:10.1038/nature13798, 2014.

McCarthy, M. C., Boering, K. A., Rice, A. L., Tyler, S. C., Connell, P. and Atlas, E.: Carbon and hydrogen isotopic



compositions of stratospheric methane: 2. Two-dimensional model results and implications for kinetic isotope effects,
J. Geophys. Res. Atmospheres, 108(D15), doi:10.1029/2002JD003183, 2003.
McGinnis, D. F., J. Greinert, Y. Artemov, S. E. Beaubien, and A. Wüest: Fate of rising methane bubbles in stratified
waters: How much methane reaches the atmosphere?, *J. Geophys. Res.*, 111, C09007, doi:10.1029/2005JC003183,
2006

McGuire, A. D., Christensen, T. R., Heroult, A., Miller, P. A., Hayes, D., Euskirchen, E., Kimball, J. S., Yi, Y., Koven,
C., Lafleur, P., Oechel, W., Peylin, P. and Williams, M.: An assessment of the carbon balance of Arctic tundra, Comp.
Obs. Process Models Atmospheric Inversions, 9(Article), 3185–3204, doi:10.5194/bg-9-3185-2012, 2012.
McKain, K., Down, A., Raciti, S. M., Budney, J., Hutyra, L. R., Floerchinger, C., Herndon, S. C., Nehrkorn, T., Zahniser,
M. S., Jackson, R. B., Phillips, N. and Wofsy, S. C.: Methane emissions from natural gas infrastructure and use in the
urban region of Boston, Massachusetts, Proc. Natl. Acad. Sci., 112(7), 1941–1946, doi:10.1073/pnas.1416261112,
2015.

McNicol, G., Fluet-Chouinard, E., Ouyang, Z., Knox, S., Zhang, Z., Aalto, T., et al.: Upscaling wetland methane
emissions from the FLUXNET-CH4 eddy covariance network (UpCH4 v1.0): Model development, network
assessment, and budget comparison. *AGU Advances*, 4, e2023AV000956. https://doi.org/10.1029/2023AV000956,
2023

Meinshausen, M., Smith, S., Calvin, K., Daniel, J., Kainuma, M., Lamarque, J. F., Matsumoto, K., Montzka, S., Raper,
S., Riahi, K., Thomson, A., Velders, G. and van Vuuren, D. P.: The RCP greenhouse gas concentrations and their
extensions from 1765 to 2300, Clim. Change, 109(1), 213–241, doi:10.1007/s10584-011-0156-z, 2011.
Meinshausen, M., Vogel, E., Nauels, A., Lorbacher, K., Meinshausen, N., Etheridge, D. M., Fraser, P. J., Montzka, S. A.,
Rayner, P. J., Trudinger, C. M., Krummel, P. B., Beyerle, U., Canadell, J. G., Daniel, J. S., Enting, I. G., Law, R. M.,
Lunder, C. R., O'Doherty, S., Prinn, R. G., Reimann, S., Rubino, M., Velders, G. J. M., Vollmer, M. K., Wang, R. H.
J., and Weiss, R.: Historical greenhouse gas concentrations for climate modelling (CMIP6), Geosci. Model Dev., 10,
2057–2116, https://doi.org/10.5194/gmd-10-2057-2017, 2017.
Meinshausen, M., Nicholls, Z. R. J., Lewis, J., Gidden, M. J., Vogel, E., Freund, M., Beyerle, U., Gessner, C., Nauels, A.,
Bauer, N., Canadell, J. G., Daniel, J. S., John, A., Krummel, P. B., Luderer, G., Meinshausen, N., Montzka, S. A.,
Rayner, P. J., Reimann, S., Smith, S. J., van den Berg, M., Velders, G. J. M., Vollmer, M. K., and Wang, R. H. J.: The
shared socio-economic pathway (SSP) greenhouse gas concentrations and their extensions to 2500, Geosci. Model
Dev., 13, 3571–3605, https://doi.org/10.5194/gmd-13-3571-2020, 2020.
Melton, J. R. and Arora, V. K.: Competition between plant functional types in the Canadian Terrestrial Ecosystem Model
(CTEM) v. 2.0, Geosci. Model Dev., 9(1), 323–361, doi:10.5194/gmd-9-323-2016, 2016.
Melton, J. R., Wania, R., Hodson, E. L., Poulter, B., Ringeval, B., Spahni, R., Bohn, T., Avis, C. A., Beerling, D. J., Chen,
G., Eliseev, A. V., Denisov, S. N., Hopcroft, P. O., Lettenmaier, D. P., Riley, W. J., Singarayer, J. S., Subin, Z. M.,



Tian, H., Zürcher, S., Brovkin, V., van Bodegom, P. M., Kleinen, T., Yu, Z. C. and Kaplan, J. O.: Present state of global wetland extent and wetland methane modelling: conclusions from a model intercomparison project (WETCHIMP), Biogeosciences, 10(2), 753–788, doi:10.5194/bg-10-753-2013, 2013.

Membrive, O., Crevoisier, C., Sweeney, C., Danis, F., Hertzog, A., Engel, A., Bönisch, H. and Picon, L.: AirCore-HR: a high-resolution column sampling to enhance the vertical description of $CH_4$ and $CO_2$, Atmospheric Meas. Tech., 10(6), 2163–2181, doi:10.5194/amt-10-2163-2017, 2017.

Messager, M. L., Lehner, B., Grill, G., Nedeva, I. and Schmitt, O.: Estimating the volume and age of water stored in global lakes using a geo-statistical approach, Nat. Commun., 7(1), 1–11, doi:10.1038/ncomms13603, 2016.

Mijling, B., van der A, R. J. and Zhang, Q.: Regional nitrogen oxides emission trends in East Asia observed from space, Atmospheric Chem. Phys., 13(23), 12003–12012, doi:doi:10.5194/acp-13-12003-2013, 2013.

Milkov, A. V.: Molecular and stable isotope compositions of natural gas hydrates: A revised global dataset and basic interpretations in the context of geological settings, Org. Geochem., 36(5), 681–702, 2005.

Minkkinen, K. and Laine, J.: Vegetation heterogeneity and ditches create spatial variability in methane fluxes from peatlands drained for forestry, Plant Soil, 285(1), 289–304, doi:10.1007/s11104-006-9016-4, 2006.

Monforti Ferrario, Fabio; Crippa, Monica; Guizzardi, Diego; Muntean, Marilena; Schaaf, Edwin; Lo Vullo, Eleonora; Solazzo, Efisio; Olivier, Jos; Vignati, Elisabetta: EDGAR v6.0 Greenhouse Gas Emissions. European Commission, Joint Research Centre (JRC) [Dataset] PID: http://data.europa.eu/89h/97a67d67-c62e-4826-b873-9d972c4f670b, 2021

Montzka, S. A., Krol, M., Dlugokencky, E., Hall, B., Jockel, P. and Lelieveld, J.: Small Interannual Variability of Global Atmospheric Hydroxyl, Science, 331(6013), 67–69, 2011.

Moore, C. W., Zielinska, B., Pétron, G. and Jackson, R. B.: Air impacts of increased natural gas acquisition, processing, and use: a critical review, Environ. Sci. Technol., 48, 8349–8359, doi:10.1021/es4053472, 2014.

Morgenstern, O., Hegglin, M. I., Rozanov, E., O'Connor, F. M., Abraham, N. L., Akiyoshi, H., Archibald, A. T., Bekki, S., Butchart, N., Chipperfield, M. P., Deushi, M., Dhomse, S. S., Garcia, R. R., Hardiman, S. C., Horowitz, L. W., Jöckel, P., Josse, B., Kinnison, D., Lin, M., Mancini, E., Manyin, M. E., Marchand, M., Marécal, V., Michou, M., Oman, L. D., Pitari, G., Plummer, D. A., Revell, L. E., Saint-Martin, D., Schofield, R., Stenke, A., Stone, K., Sudo, K., Tanaka, T. Y., Tilmes, S., Yamashita, Y., Yoshida, K., and Zeng, G.: Review of the global models used within phase 1 of the Chemistry–Climate Model Initiative (CCMI), Geosci. Model Dev., 10, 639–671, https://doi.org/10.5194/gmd-10-639-2017, 2017.

Morino, I., Uchino, O., Inoue, M., Yoshida, Y., Yokota, T., Wennberg, P. O., Toon, G. C., Wunch, D., Roehl, C. M., Notholt, J., Warneke, T., Messerschmidt, J., Griffith, D. W. T., Deutscher, N. M., Sherlock, V., Connor, B., Robinson, J., Sussmann, R. and Rettinger, M.: Preliminary validation of column-averaged volume mixing ratios of carbon dioxide and methane retrieved from GOSAT short-wavelength infrared spectra, Atmospheric Meas. Tech., 4(6), 1061–1076, 2011.



Murguia-Flores, F., Arndt, S., Ganesan, A. L., Murray-Tortarolo, G. and Hornibrook, E. R. C.: Soil Methanotrophy Model
(MeMo v1.0): a process-based model to quantify global uptake of atmospheric methane by soil, Geosci. Model Dev.,
11(6), 2009–2032, doi:10.5194/gmd-11-2009-2018, 2018.

Myer, A., Myer, M.H., Trettin, C.C. and Forschler, B.T.: The fate of carbon utilized by the subterranean termite
*Reticulitermes flavipes. Ecosphere* 12 (12):e03872,doi:10.1002/ecs2.3872, 2021

Myhre, G., Shindell, D., Bréon, F.-M., Collins, W., Fuglestvedt, J., Huang, J., Koch, D., Lamarque, J.-F., Lee, D.,
Mendoza, B., Nakajima, T., Robock, A., Stephens, G., Takemura, T. and Zhang, H.: Anthropogenic and Natural
Radiative Forcing., in In Climate Change 2013: The Physical Science Basis. Contribution of Working Group I to the
Fifth Assessment Report of the Intergovernmental Panel on Climate Change., edited by T. F. Stocker, D. D. Qin, G.-
3167        K. Plattner, M. Tignor, S. K. Allen, J. Boschung, A. Nauels, Y. Xia, V. Bex, and P. M. Midgley, Cambridge University
Press, Cambridge, United Kingdom and New York, NY, USA., 2013.

Naik, V., Voulgarakis, A., Fiore, A. M., Horowitz, L. W., Lamarque, J. F., Lin, M., Prather, M. J., Young, P. J., Bergmann,
D., Cameron-Smith, P. J., Cionni, I., Collins, W. J., Dalsoren, S. B., Doherty, R., Eyring, V., Faluvegi, G., Folberth,
G. A., Josse, B., Lee, Y. H., MacKenzie, I. A., Nagashima, T., van Noije, T. P. C., Plummer, D. A., Righi, M., Rumbold,
S. T., Skeie, R., Shindell, D. T., Stevenson, D. S., Strode, S., Sudo, K., Szopa, S. and Zeng, G.: Preindustrial to present
3173        day changes in tropospheric hydroxyl radical and methane lifetime from the Atmospheric Chemistry and Climate
Model Intercomparison Project (ACCMIP), Atmospheric Chem. Phys., 13(10), 5277–5298, doi:10.5194/acp-13-5277-
2013, 2013.

Nakazawa, T., Machida, T., Tanaka, M., Fujii, Y., Aoki, S. and Watanabe, O.: Differences of the atmospheric $CH_4$
concentration between the Arctic and Antarctic regions in pre-industrial/pre-agricultural era, Geophys. Res. Lett.,
20(10), 943–946, doi:10.1029/93GL00776, 1993.

Natchimuthu, S., I. Sundgren, M. Gålfalk, L. Klemedtsson, P. Crill, Å. Danielsson, and D. Bastviken, Spatio-temporal
variability of lake $CH_4$ fluxes and its influence on annual whole lake emission estimates. Limnology and
Oceanography, 61(S1): p. S13-S26, 2016.

Nauer, P. A., Hutley, L. B., and Arndt, S. K.: Termite mounds mitigate half of termite methane emissions, P. Natl. Acad.
Sci. USA, 115, 13306–13311, 2018.

Nicely, J. M., Salawitch, R. J., Canty, T., Anderson, D. C., Arnold, S. R., Chipperfield, M. P., Emmons, L. K., Flemming,
3185        J., Huijnen, V., Kinnison, D. E., Lamarque, J.-F., Mao, J., Monks, S. A., Steenrod, S. D., Tilmes, S. and Turquety, S.:
Quantifying the causes of differences in tropospheric OH within global models, J. Geophys. Res. Atmospheres, 122(3),
1983–2007, doi:10.1002/2016JD026239, 2017.

Nirmal Rajkumar, A., J. Barnes, R. Ramesh, R. Purvaja, and R.C. Upstill-Goddard, Methane and nitrous oxide fluxes in
the polluted Adyar River and estuary, SE India. Marine Pollution Bulletin, 56(12): p. 2043-2051, 2008

Nisbet, E. G., Manning, M. R., Dlugokencky, E. J., Fisher, R. E., Lowry, D., Michel, S. E., Myhre, C. L., Platt, S. M.,





Allen, G., Bousquet, P., Brownlow, R., Cain, M., France, J. L., Hermansen, O., Hossaini, R., Jones, A. E., Levin, I., Manning, A. C., Myhre, G., Pyle, J. A., Vaughn, B., Warwick, N. J. and White, J. W. C.: Very strong atmospheric methane growth in the four years 2014-2017: Implications for the Paris Agreement, Glob. Biogeochem. Cycles, 0(ja), doi:10.1029/2018GB006009, 2019.

Nisbet, E. G., Manning, M. R., Dlugokencky, E. J., Michel, S. E., Lan, X., Röckmann, T., van der Denier Gon, H. A., Schmitt, J., Palmer, P. I., Dyonisius, M. N., Oh, Y., Fisher, R. E., Lowry, D., France, J. L., White, J. W. C., Brailsford, G., and Bromley, T.: Atmospheric methane: Comparison between methane's record in 2006–2022 and during glacial terminations, Global Biogeochem. Cy., 37, e2023GB007875, https://doi.org/10.1029/2023GB007875, 2023.

Nisbet, R. E. R., Fisher, R., Nimmo, R. H., Bendall, D. S., Crill, P. M., Gallego-Sala, A. V., Hornibrook, E. R. C., Lopez-Juez, E., Lowry, D., Nisbet, P. B. R., Shuckburgh, E. F., Sriskantharajah, S., Howe, C. J. and Nisbet, E. G.: Emission of methane from plants, Proc. R. Soc. B-Biol. Sci., 276(1660), 1347–1354, 2009.

Niwa, Y., Fujii, Y., Sawa, Y., Iida, Y., Ito, A., Satoh, M., Imasu, R., Tsuboi, K., Matsueda, H. and Saigusa, N.: A 4D-Var inversion system based on the icosahedral grid model (NICAM-TM 4D-Var v1.0) – Part 2: Optimization scheme and identical twin experiment of atmospheric $CO_2$ inversion, Geosci. Model Dev., 10(6), 2201–2219, doi:10.5194/gmd-10-2201-2017, 2017.

Niwa, Y., Ishijima, K., Ito, A. and Iida, Y.: Toward a long-term atmospheric $CO_2$ inversion for elucidating natural carbon fluxes: technical notes of NISMON-$CO_2$ v2021.1. Prog. Earth Planet Sci. 9, 42, doi:10.1186/s40645-022-00502-6, 2022

Noël, S., Reuter, M., Buchwitz, M., Borchardt, J., Hilker, M., Schneising, O., Bovensmann, H., Burrows, J. P., Di Noia, A., Parker, R. J., Suto, H., Yoshida, Y., Buschmann, M., Deutscher, N. M., Feist, D. G., Griffith, D. W. T., Hase, F., Kivi, R., Liu, C., Morino, I., Notholt, J., Oh, Y.-S., Ohyama, H., Petri, C., Pollard, D. F., Rettinger, M., Roehl, C., Rousogenous, C., Sha, M. K., Shiomi, K., Strong, K., Sussmann, R., Té, Y., Velazco, V. A., Vrekoussis, M., and Warneke, T.: Retrieval of greenhouse gases from GOSAT and GOSAT-2 using the FOCAL algorithm, Atmos. Meas. Tech., 15, 3401–3437, https://doi.org/10.5194/amt-15-3401-2022, 2022.

Nomura, S., Naja, M., Ahmed, M. K., Mukai, H., Terao, Y., Machida, T., Sasakawa, M., and Patra, P. K.: Measurement report: Regional characteristics of seasonal and long-term variations in greenhouse gases at Nainital, India, and Comilla, Bangladesh, Atmos. Chem. Phys., 21, 16427–16452, https://doi.org/10.5194/acp-21-16427-2021, 2021

Obu, J., Westermann, S., Bartsch, A., Berdnikov, N., Christiansen, H. H., Dashtseren, A., Delaloye, R., Elberling, B., Etzelmüller, B., Kholodov, A., Khomutov, A., Kääb, A., Leibman, M. O., Lewkowicz, A. G., Panda, S. K., Romanovsky, V., Way, R. G., Westergaard-Nielsen, A., Wu, T., Yamkhin, J. and Zou, D.: Northern Hemisphere permafrost map based on TTOP modelling for 2000–2016 at 1 km2 scale, Earth-Sci. Rev., 193, 299–316, doi:10.1016/j.earscirev.2019.04.023, 2019.



Ocko, I. B, Sun, T., Shindell, D., Oppenheimer, M., Hristov, A. N, Pacala, S. W, Mauzerall, D. L, Xu, Y. and Hamburg,
S. P: Acting rapidly to deploy readily available methane mitigation measures by sector can immediately slow global
warming, Environ. Res. Lett., 16, 054042, doi :10.1088/1748-9326/abf9c8, 2021
Odelson, D.A. and Breznak, J. A. Volatile fatty acid production by the hindgut microbiota of xilophagus termites. Applied
and Environmental Microbiology, 45, 1602-1613, 1983. doi: 10.1128/aem.45.5.1602-1613.1983.
Olivier, J. G. J. and Janssens-Maenhout, G.: Part III: Total Greenhouse Gas Emissions, of $CO_2$ Emissions from Fuel
Combustion (2014 ed.), International Energy Agency, Paris, ISBN-978-92-64-21709-6., 2014.
Ollivier, Q. R., Maher, D. T., Pitfield, C. and Macreadie, P. I.: Punching above their weight: Large release of greenhouse
gases from small agricultural dams, Glob. Change Biol., 25(2), 721–732, doi:10.1111/gcb.14477, 2019.
O'Neill, B. C., Tebaldi, C., Vuuren, D. P. van, Eyring, V., Friedlingstein, P., Hurtt, G., Knutti, R., Kriegler, E., Lamarque,
J.-F., Lowe, J., Meehl, G. A., Moss, R., Riahi, K. and Sanderson, B. M.: The Scenario Model Intercomparison Project
(ScenarioMIP) for CMIP6, Geosci. Model Dev., 9(9), 3461–3482, doi:https://doi.org/10.5194/gmd-9-3461-2016,
2016.

Oreggioni, G. D., F. Monforti Ferrario, M. Crippa, M. Muntean, E. Schaaf, D. Guizzardi, E. Solazzo, M. Duerr, M. Perry
and E. Vignati: Climate change in a changing world: Socio-economic and technological transitions, regulatory
frameworks and trends on global greenhouse gas emissions from EDGAR v.5.0, Global Environmental Change,
doi:10.1016/j.gloenvcha.2021.10235, 2021
Oremland, R. S. Methanogenic activity in plankton samples and fish intestines: a mechanism for *in situ* methanogenesis
in oceanic surface waters. *Limnol. Oceanogr.* **24,** 1136–1141, 1979.
O'Rourke, P. R, Smith, S. J., Mott, A., Ahsan, H., McDuffie, E. E., Crippa, M., Klimont, S., McDonald, B., Z., Wang,
Nicholson, M. B, Feng, L., and Hoesly, R. M., CEDS v-2021-02-05 Emission Data 1975-2019 (Version Feb-05-2021).
Zenodo. http://doi.org/10.5281/zenodo.4509372, 2021
Ovalle, A. R. C., C. E. Rezende, L. D. Lacerda, and C. A. R. Silva: Factors affecting the hydrochemistry of a mangrove
tidal creek, Sepetiba Bay, Brazil. Estuar. Coast. Shelf Sci. 31: 639–650. doi:10.1016/0272-7714(90)90017-L, 1990
Pacala, S. W.: Verifying greenhouse gas emissions: Methods to support international climate agreements, National
Academies Press., 2010
Pandey, S., Gautam, R., Houweling, S., Gon, H. D. van der, Sadavarte, P., Borsdorff, T., Hasekamp, O., Landgraf, J., Tol,
P., Kempen, T. van, Hoogeveen, R., Hees, R. van, Hamburg, S. P., Maasakkers, J. D. and Aben, I.: Satellite
observations reveal extreme methane leakage from a natural gas well blowout, Proc. Natl. Acad. Sci., 116(52), 26376–
26381, doi:10.1073/pnas.1908712116, 2019.
Pangala, S. R., Moore, S., Hornibrook, E. R. C. and Gauci, V.: Trees are major conduits for methane egress from tropical
forested wetlands, New Phytol., 197(2), 524–531, doi:10.1111/nph.12031, 2013.
Pangala, S. R., Hornibrook, E. R. C., Gowing, D. J. and Gauci, V.: The contribution of trees to ecosystem methane





emissions in a temperate forested wetland, Glob. Change Biol., 21(7), 2642–2654, doi:10.1111/gcb.12891, 2015.
Pangala, S. R., Enrich-Prast, A., Basso, L. S., Peixoto, R. B., Bastviken, D., Hornibrook, E. R. C., Gatti, L. V., Marotta,
H., Calazans, L. S. B., Sakuragui, C. M., Bastos, W. R., Malm, O., Gloor, E., Miller, J. B. and Gauci, V.: Large
emissions from floodplain trees close the Amazon methane budget, Nature, 552(7684), 230–234,
doi:10.1038/nature24639, 2017.
Paris, J.-D., Ciais, P., Nedelec, P., Stohl, A., Belan, B. D., Arshinov, M. Y., Carouge, C., Golitsyn, G. S. and Granberg, I.
G.: New insights on the chemical composition of the Siberian air shed from the YAK AEROSIB aircraft campaigns,
Bull. Am. Meteorol. Soc., 91(5), 625–641, doi:10.1175/2009BAMS2663.1., 2010.
Parker, R. J., Webb, A., Boesch, H., Somkuti, P., Barrio Guillo, R., Di Noia, A., Kalaitzi, N., Anand, J. S., Bergamaschi,
P., Chevallier, F., Palmer, P. I., Feng, L., Deutscher, N. M., Feist, D. G., Griffith, D. W. T., Hase, F., Kivi, R., Morino,
I., Notholt, J., Oh, Y.-S., Ohyama, H., Petri, C., Pollard, D. F., Roehl, C., Sha, M. K., Shiomi, K., Strong, K., Sussmann,
R., Té, Y., Velazco, V. A., Warneke, T., Wennberg, P. O., and Wunch, D.: A decade of GOSAT Proxy satellite CH4
observations, Earth Syst. Sci. Data, 12, 3383–3412, https://doi.org/10.5194/essd-12-3383-2020, 2020.
Parker, R. and Boesch, H. (2020): University of Leicester GOSAT Proxy XCH4 v9.0. Centre for Environmental Data
Analysis, 07 May 2020. https://dx.doi.org/10.5285/18ef8247f52a4cb6a14013f8235cc1eb, 2020
Parker, R. J., Wilson, C., Comyn-Platt, E., Hayman, G., Marthews, T. R., Bloom, A. A., Lunt, M. F., Gedney, N., Dadson,
S. J., McNorton, J., Humpage, N., Boesch, H., Chipperfield, M. P., Palmer, P. I., and Yamazaki, D.: Evaluation of
wetland CH4 in the Joint UK Land Environment Simulator (JULES) land surface model using satellite observations,
Biogeosciences, 19, 5779–5805, https://doi.org/10.5194/bg-19-5779-2022, 2022.
Pathak, H., Li, C. and Wassmann, R.: Greenhouse gas emissions from Indian rice fields: calibration and upscaling using
the DNDC model, Biogeosciences, 1(1), 1–11, 2005.
Patra, P. K., Houweling, S., Krol, M., Bousquet, P., Belikov, D., Bergmann, D., Bian, H., Cameron-Smith, P., Chipperfield,
M. P., Corbin, K., Fortems-Cheiney, A., Fraser, A., Gloor, E., Hess, P., Ito, A., Kawa, S. R., Law, R. M., Loh, Z.,
Maksyutov, S., Meng, L., Palmer, P. I., Prinn, R. G., Rigby, M., Saito, R. and Wilson, C.: TransCom model simulations
of CH4 and related species: linking transport, surface flux and chemical loss with CH4 variability in the troposphere
and lower stratosphere, Atmospheric Chem. Phys., 11(24), 12,813-12,837, doi:10.5194/acp-11-12813-2011, 2011.
Patra, P. K., Krol, M. C., Montzka, S. A., Arnold, T., Atlas, E. L., Lintner, B. R., Stephens, B. B., Xiang, B., Elkins, J. W.,
Fraser, P. J., Ghosh, A., Hintsa, E. J., Hurst, D. F., Ishijima, K., Krummel, P. B., Miller, B. R., Miyazaki, K., Moore,
F. L., Mühle, J., O'Doherty, S., Prinn, R. G., Steele, L. P., Takigawa, M., Wang, H. J., Weiss, R. F., Wofsy, S. C. and
Young, D.: Observational evidence for interhemispheric hydroxyl-radical parity, Nature, 513(7517), 219–223,
doi:10.1038/nature13721, 2014.
Patra, P. K., Takigawa, M., Watanabe, S., Chandra, N., Ishijima, K. and Yamashita, Y.: Improved Chemical Tracer
Simulation by MIROC4.0-based Atmospheric Chemistry-Transport Model (MIROC4-ACTM), SOLA, 14(0), 91–96,





doi:10.2151/sola.2018-016, 2018.
Patra, P. K., Krol, M. C., Prinn, R. G., Takigawa, M., Mühle, J., Montzka, S. A., Lal, S., Yamashita, Y., Naus, S., Chandra,
3291        N., Weiss, R. F., Krummel, P. B., Fraser, P. J., O'Doherty, S., and Elkins, J. W.: Methyl Chloroform Continues to
Constrain the Hydroxyl (OH) Variability in the Troposphere, J. Geophys. Res.-Atmos., 126, e2020JD033862,
https://doi.org/10.1029/2020JD033862, 2021.

Paull, C. K., Brewer, P. G., Ussler, W., Peltzer, E. T., Rehder, G. and Clague, D.: An experiment demonstrating that marine
slumping is a mechanism to transfer methane from seafloor gas-hydrate deposits into the upper ocean and atmosphere,
Geo-Mar. Lett., 22(4), 198–203, doi:10.1007/s00367-002-0113-y, 2002.

Peacock, M., J. Audet, D. Bastviken, M.N. Futter, V. Gauci, A. Grinham, J.A. Harrison, M.S. Kent, S. Kosten, C.E.
Lovelock, A.J. Veraart, and C.D. Evans, Global importance of methane emissions from drainage ditches and canals.
Environmental Research Letters, 16(4): p. 044010., 2021

Peischl, J., Ryerson, T. B., Aikin, K. C., de Gouw, J. A., Gilman, J. B., Holloway, J. S., Lerner, B. M., Nadkarni, R.,
Neuman, J. A., Nowak, J. B., Trainer, M., Warneke, C. and Parrish, D. D.: Quantifying atmospheric methane emissions
from the Haynesville, Fayetteville, and northeastern Marcellus shale gas production regions, J. Geophys. Res.
Atmospheres, 120(5), 2119–2139, doi:10.1002/2014jd022697, 2015.

Pekel, J.-F., Cottam, A., Gorelick, N. and Belward, A. S.: High-resolution mapping of global surface water and its long-
term changes, Nature, 540(7633), 418–422, doi:10.1038/nature20584, 2016.

Peltola, O., Vesala, T., Gao, Y., Räty, O., Alekseychik, P., Aurela, M., Chojnicki, B., Desai, A. R., Dolman, A. J.,
Euskirchen, E. S., Friborg, T., Göckede, M., Helbig, M., Humphreys, E., Jackson, R. B., Jocher, G., Joos, F., Klatt, J.,
Knox, S. H., Kowalska, N., Kutzbach, L., Lienert, S., Lohila, A., Mammarella, I., Nadeau, D. F., Nilsson, M. B.,
Oechel, W. C., Peichl, M., Pypker, T., Quinton, W., Rinne, J., Sachs, T., Samson, M., Schmid, H. P., Sonnentag, O.,
Wille, C., Zona, D. and Aalto, T.: Monthly gridded data product of northern wetland methane emissions based on
upscaling eddy covariance observations, Earth Syst. Sci. Data, 11(3), 1263–1289, doi:10.5194/essd-11-1263-2019,
2019.

Peng, S. S., Piao, S. L., Bousquet, P., Ciais, P., Li, B. G., Lin, X., Tao, S., Wang, Z. P., Zhang, Y. and Zhou, F.: Inventory
of anthropogenic methane emissions in Mainland China from 1980 to 2010, Atmospheric Chem. Phys. Discuss., 2016,
1–29, doi:10.5194/acp-2016-139, 2016.

Peng, S., Lin, X., Thompson, R. L., Xi, Y., Liu, G., Hauglustaine, D., Lan, X., Poulter, B., Ramonet, M., Saunois, M.,
Yin, Y., Zhang, Z., Zheng, B., and Ciais, P.: Wetland emission and atmospheric sink changes explain methane growth
in 2020, Nature, 612, 477–482, https://doi.org/10.1038/s41586-022-05447-w, 2022.

Pérez-Barbería, F. J.: Scaling methane emissions in ruminants and global estimates in wild populations, Sci. Total
Environ., 579, 1572–1580, doi:10.1016/j.scitotenv.2016.11.175, 2017.

Petersen, H. and Luxton, M. A comparative analysis of soil fauna populations and their role in decomposition processes.



Oikos 39: 287–388,doi.org/10.2307/3544689., 1982
Petrenko, V. V., Smith, A. M., Schaefer, H., Riedel, K., Brook, E., Baggenstos, D., Harth, C., Hua, Q., Buizert, C., Schilt,
3324        A., Fain, X., Mitchell, L., Bauska, T., Orsi, A., Weiss, R. F. and Severinghaus, J. P.: Minimal geological methane
emissions during the Younger Dryas–Preboreal abrupt warming event, Nature, 548, 443, doi:10.1038/nature23316
https://www.nature.com/articles/nature23316#supplementary-information, 2017.

Petrescu, A. M. R., Qiu, C., Ciais, P., Thompson, R. L., Peylin, P., McGrath, M. M., Solazzo, E., Janssens-Maenhout, G.,
Tubiello, F. N., Bergamaschi, P., Brunner, D., Peters, G. P., Höglund- Isaksson, L., Regnier, P., Lauerwald, R.,
Bastviken, D., Tsuruta, A., Winiwarter, W., Patra, P. P., Kuhnert, M., Oreggioni, G. D., Crippa, M., Saunois, M.,
Perugini, L., Markkanen, T., Aalto, T., Groot Zwaaftink, C. C., Yao, Y., Wilson, C. C., Conchedda, G., Günther, D.,
Leip, A., Smith, P., Haussaire, J. M., Leppänen, A., Manning, A. J., McNorton, J., Brockmann, P., & Dolman, A. J. H.
3332        A. The consolidated European synthesis of CH4 and N2O emissions for the European Union and United Kingdom:
1990-2017. Earth System Science Data, 13(5), 2307-2362. doi:10.5194/ essd-13-2307- 2021, 2021.

Petrescu, A. M. R., Qiu, C., McGrath, M. J., Peylin, P., Peters, G. P., Ciais, P., Thompson, R. L., Tsuruta, A., Brunner,
D., Kuhnert, M., Matthews, B., Palmer, P. I., Tarasova, O., Regnier, P., Lauerwald, R., Bastviken, D., Höglund-
Isaksson, L., Winiwarter, W., Etiope, G., Aalto, T., Balsamo, G., Bastrikov, V., Berchet, A., Brockmann, P., Ciotoli,
G., Conchedda, G., Crippa, M., Dentener, F., Groot Zwaaftink, C. D., Guizzardi, D., Günther, D., Haussaire, J.-M.,
Houweling, S., Janssens-Maenhout, G., Kouyate, M., Leip, A., Leppänen, A., Lugato, E., Maisonnier, M., Manning,
3339        A. J., Markkanen, T., McNorton, J., Muntean, M., Oreggioni, G. D., Patra, P. K., Perugini, L., Pison, I., Raivonen, M.
3340        T., Saunois, M., Segers, A. J., Smith, P., Solazzo, E., Tian, H., Tubiello, F. N., Vesala, T., van der Werf, G. R., Wilson,
C., and Zaehle, S.: The consolidated European synthesis of CH$_4$ and N$_2$O emissions for the European Union and United
Kingdom: 1990–2019, Earth Syst. Sci. Data, 15, 1197–1268, https://doi.org/10.5194/essd-15-1197-2023, 2023.

Pétron, G., Karion, A., Sweeney, C., Miller, B. R., Montzka, S. A., Frost, G. J., Trainer, M., Tans, P., Andrews, A., Kofler,
3344        J., Helmig, D., Guenther, D., Dlugokencky, E., Lang, P., Newberger, T., Wolter, S., Hall, B., Novelli, P., Brewer, A.,
Conley, S., Hardesty, M., Banta, R., White, A., Noone, D., Wolfe, D. and Schnell, R.: A new look at methane and
nonmethane hydrocarbon emissions from oil and natural gas operations in the Colorado Denver-Julesburg Basin, J.
Geophys. Res. Atmospheres, 119(11), 6836–6852, doi:10.1002/2013jd021272, 2014.

Phillips, N. G., Ackley, R., Crosson, E. R., Down, A., Hutyra, L. R., Brondfield, M., Karr, J. D., Zhao, K. and Jackson, R.
B.: Mapping urban pipeline leaks: Methane leaks across Boston, Environ. Pollut., 173, 1–4,
doi:10.1016/j.envpol.2012.11.003, 2013.

Pison, I., Ringeval, B., Bousquet, P., Prigent, C. and Papa, F.: Stable atmospheric methane in the 2000s: key-role of
emissions from natural wetlands, Atmospheric Chem. Phys. Discuss., 13(4), 9017–9049, doi:10.5194/acpd-13-9017-
2013, 2013.

Pitz, S. and Megonigal, J. P.: Temperate forest methane sink diminished by tree emissions, New Phytol., 214(4), 1432–



1439, doi:10.1111/nph.14559, 2017.

Platt, U., Allan, W. and Lowe, D.: Hemispheric average Cl atom concentration from $^{13}C/^{12}C$ ratios in atmospheric methane, Atmos Chem Phys, 4, 2393–2399, 2004.

Plummer, D., Nagashima, T., Tilmes, S., Archibald, A., Chiodo, G., Fadnavis, S., Garny, H., Josse, B., Kim, J., Lamarque, J.-F., Morgenstern, O., Murray, L., Orbe, C., Tai, A., Chipperfield, M., Funke, B., Juckes, M., Kinnison, D., Kunze, M., Luo, B., Matthes, K., Newman, P. A., Pascoe, C. and Peter, T.: CCMI- 2022: a new set of Chemistry–Climate Model Initiative (CCMI) community simulations to update the assessment of models and support upcoming ozone assessment activities. SPARC Newsletter 57, 22–30, 2021.

Pollard, D. F., Sherlock, V., Robinson, J., Deutscher, N. M., Connor, B. and Shiona, H.: The Total Carbon Column Observing Network site description for Lauder, New Zealand, Earth Syst. Sci. Data, 9(2), 977–992, doi:10.5194/essd-9-977-2017, 2017.

Portmann, F. T., Siebert, S. & Döll, P. : MIRCA2000 – Global monthly irrigated and rainfed crop areas around the year 2000: A new high-resolution data set for agricultural and hydrological modeling, Global Biogeochemical Cycles, 24, GB 1011, doi:10.1029/2008GB003435, 2010

Portmann, R. W., Daniel, J. S. and Ravishankara, A. R.: Stratospheric ozone depletion due to nitrous oxide: influences of other gases, Philos. Trans. R. Soc. Lond. B Biol. Sci., 367(1593), 1256–1264, doi:10.1098/rstb.2011.0377, 2012.

Poulter, B., Bousquet, P., Canadell, J. G., Ciais, P., Peregon, A., Saunois, M., Arora, V. K., Beerling, D. J., Brovkin, V., Jones, C. D., Joos, F., Gedney, N., Ito, A., Kleinen, T., Koven, C. D., McDonald, K., Melton, J. R., Peng, C. H., Peng, S. S., Prigent, C., Schroeder, R., Riley, W. J., Saito, M., Spahni, R., Tian, H. Q., Taylor, L., Viovy, N., Wilton, D., Wiltshire, A., Xu, X. Y., Zhang, B. W., Zhang, Z. and Zhu, Q. A.: Global wetland contribution to 2000-2012 atmospheric methane growth rate dynamics, Environ. Res. Lett., 12(9), doi:10.1088/1748-9326/aa8391, 2017.

Prairie, Y.T., J. Alm, J. Beaulieu, N. Barros, T. Battin, J. Cole, P. del Giorgio, T. DelSontro, F. Guérin, A. Harby, J. Harrison, S. Mercier-Blais, D. Serça, S. Sobek, and D. Vachon, Greenhouse Gas Emissions from Freshwater Reservoirs: What Does the Atmosphere See? Ecosystems, 21(5): p. 1058-1071, 2018

Prather, M. J., Holmes, C. D. and Hsu, J.: Reactive greenhouse gas scenarios: Systematic exploration of uncertainties and the role of atmospheric chemistry, Geophys. Res. Lett., 39(9), L09803, doi:10.1029/2012gl051440, 2012.

Prinn, R. G., Weiss, R. F., Arduini, J., Arnold, T., DeWitt, H. L., Fraser, P. J., Ganesan, A. L., Gasore, J., Harth, C. M., Hermansen, O., Kim, J., Krummel, P. B., Li, S., Loh, Z. M., Lunder, C. R., Maione, M., Manning, A. J., Miller, B. R., Mitrevski, B., Mühle, J., O'Doherty, S., Park, S., Reimann, S., Rigby, M., Saito, T., Salameh, P. K., Schmidt, R., Simmonds, P. G., Steele, L. P., Vollmer, M. K., Wang, R. H., Yao, B., Yokouchi, Y., Young, D., and Zhou, L.: History of chemically and radiatively important atmospheric gases from the Advanced Global Atmospheric Gases Experiment (AGAGE), *Earth Syst. Sci. Data*, 10, 985–1018, https://doi.org/10.5194/essd-10-985-2018, 2018.

Prosperi, P., Bloise, M., Tubiello, F.N. *et al.* New estimates of greenhouse gas emissions from biomass burning and peat



fires using MODIS Collection 6 burned areas. *Climatic Change* **161**, 415–432, https://doi.org/10.1007/s10584-020-02654-0, 2020

Purvaja, R., Ramesh, R., & Frenzel, P.: Plant-mediated methane emission from an Indian mangrove, *Global Change Biology*, 10, 1825–1834, 2004

Qin, B., Zhou, J., Elser, J.J., Gardner, W.S., Deng, J., and J.D. Brookes: Water depth underpins the relative roles and fates of nitrogen and phosphorus in lakes, Environmental Science & Technology 2020 *54* (6), 3191-3198, DOI: 10.1021/acs.est.9b05858, 2020.

Qu, Z., Jacob, D. J., Shen, L., Lu, X., Zhang, Y., Scarpelli, T. R., Nesser, H., Sulprizio, M. P., Maasakkers, J. D., Bloom, A. A., Worden, J. R., Parker, R. J., and Delgado, A. L.: Global distribution of methane emissions: a comparative inverse analysis of observations from the TROPOMI and GOSAT satellite instruments, Atmos. Chem. Phys., 21, 14159–14175, https://doi.org/10.5194/acp-21-14159-2021, 2021.

Qu, Z., Jacob, D. J., Zhang, Y., Shen, L., Varon, D. J., Lu, X., Scarpelli, T., Bloom, A., Worden, J., and Parker, R. J.: Attribution of the 2020 surge in atmospheric methane by inverse analysis of GOSAT observations, Environ. Res. Lett., 17, 094003, https://doi.org/10.1088/1748-9326/ac8754, 2022.

Randerson, J. T., Chen, Y., van der Werf, G. R., Rogers, B. M. and Morton, D. C.: Global burned area and biomass burning emissions from small fires, J. Geophys. Res. Biogeosciences, 117, G4, doi:10.1029/2012jg002128, 2012.

Ramage, J.L., Kuhn, M., Virkkala, A.M., Voigt, C., Marushchak, M.E., Bastos, A., Biasi, C., Canadell, J.G., Ciais, P., Ĺopez-Blanco, E. Natali, S.M., et al.: The net GHG balance and budget of the permafrost region (2000-2020) from ecosystem flux upscaling. Preprint in ESS Open Archive. September 11, 2023. DOI: 10.22541/essoar.169447408.86275712/v1, 2023

Ramsden, A. E., Ganesan, A. L., Western, L. M., Rigby, M., Manning, A. J., Foulds, A., France, J. L., Barker, P., Levy, P., Say, D., Wisher, A., Arnold, T., Rennick, C., Stanley, K. M., Young, D., and O'Doherty, S.: Quantifying fossil fuel methane emissions using observations of atmospheric ethane and an uncertain emission ratio, Atmos. Chem. Phys., 22, 3911–3929, https://doi.org/10.5194/acp-22-3911-2022, 2022.

Ray, N.E., Holgerson, M.A., Andersen, M.R., Bikše, J., Bortolotti, L.E., Futter, M., Kokorīte, I., Law, A., McDonald, C., Mesman, J.P., Peacock, M., Richardson, D.C., Arsenault, J., Bansal, S., Cawley, K., Kuhn, M., Shahabinia, A.R. and Smufer, F.: Spatial and temporal variability in summertime dissolved carbon dioxide and methane in temperate ponds and shallow lakes. Limnol Oceanogr, 68: 1530-1545. https://doi.org/10.1002/lno.12362, 2023

Regnier P., Arndt, S., Dale, A.W., LaRowe, D.E., Mogollon, J. and Van Cappellen, P. Advances in the biogeochemical modeling of anaerobic oxidation of methane (AOM). Earth Science Reviews. 106, 105-130, 2011;

Ren, W. E. I., Tian, H., Xu, X., Liu, M., Lu, C., Chen, G., Melillo, J., Reilly, J. and Liu, J.: Spatial and temporal patterns of $CO_2$ and $CH_4$ fluxes in China's croplands in response to multifactor environmental changes, Tellus B, 63(2), 222–240, doi:10.1111/j.1600-0889.2010.00522.x, 2011.





Repeta, D. J., Ferrón, S., Sosa, O. A., Johnson, C. G., Repeta, L. D., Acker, M., DeLong, E. F. and Karl, D. M.: Marine
methane paradox explained by bacterial degradation of dissolved organic matter, Nat. Geosci., 9(12), 884–887,
doi:10.1038/ngeo2837, 2016.

Riahi, K., van Vuuren, D. P., Kriegler, E., Edmonds, J., O'Neill, B. C., Fujimori, S., Bauer, N., Calvin, K., Dellink, R.,
Fricko, O., Lutz, W., Popp, A., Cuaresma, J. C., Kc, S., Leimbach, M., Jiang, L., Kram, T., Rao, S., Emmerling, J.,
Ebi, K., Hasegawa, T., Havlik, P., Humpenöder, F., Da Silva, L. A., Smith, S., Stehfest, E., Bosetti, V., Eom, J.,
Gernaat, D., Masui, T., Rogelj, J., Strefler, J., Drouet, L., Krey, V., Luderer, G., Harmsen, M., Takahashi, K.,
Baumstark, L., Doelman, J. C., Kainuma, M., Klimont, Z., Marangoni, G., Lotze-Campen, H., Obersteiner, M., Tabeau,
3429        A. and Tavoni, M.: The Shared Socioeconomic Pathways and their energy, land use, and greenhouse gas emissions
implications: An overview, Glob. Environ. Change, 42, 153–168, doi:10.1016/j.gloenvcha.2016.05.009, 2017.

Rice, A. L., Butenhoff, C. L., Shearer, M. J., Teama, D., Rosenstiel, T. N. and Khalil, M. A. K.: Emissions of anaerobically
produced methane by trees, Geophys. Res. Lett., 37, L03807, doi:10.1029/2009GL041565, 2010.

Ridgwell, A. J., Marshall, S. J. and Gregson, K.: Consumption of atmospheric methane by soils: A process-based model,
Glob. Biogeochem. Cycles, 13(1), 59–70, doi:10.1029/1998gb900004, 1999.

Riedel, T. P., Wolfe, G. M., Danas, K. T., Gilman, J. B., Kuster, W. C., Bon, D. M., Vlasenko, A., Li, S. M., Williams, E.
3436        J., Lerner, B. M., Veres, P. R., Roberts, J. M., Holloway, J. S., Lefer, B., Brown, S. S. and Thornton, J. A.: An MCM
modeling study of nitryl chloride (ClNO2) impacts on oxidation, ozone production and nitrogen oxide partitioning in
polluted continental outflow, Atmospheric Chem. Phys., 14(8), 3789–3800, doi:10.5194/acp-14-3789-2014, 2014.

Rigby, M., Montzka, S. A., Prinn, R. G., White, J. W. C., Young, D., O'Doherty, S., Lunt, M. F., Ganesan, A. L., Manning,
3440        A. J., Simmonds, P. G., Salameh, P. K., Harth, C. M., Mühle, J., Weiss, R. F., Fraser, P. J., Steele, L. P., Krummel, P.
B., McCulloch, A. and Park, S.: Role of atmospheric oxidation in recent methane growth, Proc. Natl. Acad. Sci.,
114(21), 5373, 2017.

Riley, W. J., Subin, Z. M., Lawrence, D. M., Swenson, S. C., Torn, M. S., Meng, L., Mahowald, N. M. and Hess, P.:
Barriers to predicting changes in global terrestrial methane fluxes: analyses using CLM4Me, a methane
biogeochemistry model integrated in CESM, Biogeosciences, 8(7), 1925–1953, doi:10.5194/bg-8-1925-2011, 2011.

Ringeval, B., Friedlingstein, P., Koven, C., Ciais, P., de Noblet-Ducoudre, N., Decharme, B. and Cadule, P.: Climate-CH4
feedback from wetlands and its interaction with the climate-CO2 feedback, Biogeosciences, 8(8), 2137–2157,
doi:10.5194/bg-8-2137-2011, 2011.

Robison, A.L., W.M. Wollheim, B. Turek, C. Bova, C. Snay, and R.K. Varner, Spatial and temporal heterogeneity of
methane ebullition in lowland headwater streams and the impact on sampling design, Limnology and Oceanography,
66(12): p. 4063-4076, 2021

Rocher-Ros, G., Stanley, E.H., Loken, L.C., Casson, N.J., Raymond, P.A., Liu, S., Amatulli, G. and Sponseller, R.A.,
Global methane emissions from rivers and streams. *Nature*, pp.1-6.,621, 530–535, https://doi.org/10.1038/s41586-023-





06344-6, 2023

Rosentreter, J. A., Maher, D. T., Erler, D. V., Murray, R. H. and Eyre, B. D.: Methane emissions partially offset "blue
carbon" burial in mangroves, Sci. Adv., 4(6), eaao4985, doi:10.1126/sciadv.aao4985, 2018.

Rosentreter, J. A., A. V Borges, B. R. Deemer, and others : Half of global methane emissions come from highly variable
aquatic ecosystem sources. Nat. Geosci. **14**: 225–230. doi:10.1038/s41561-021-00715-2, 2021

Rosentreter, J.A., Laruelle, G.G., Bange, H.W., Bianchi, T.S., Busecke, J.J.M., Cai, W-J, Eyre, B.D., Forbrich, I., Kwon,
E.Y., Mavara, T., Moosdorf, N., Van Dam, B. and Regnier, P. Coastal vegetation and estuaries are collectively a
greenhouse gas sink. Nature Climate Change, 13, 579–587, doi: 10.1038/s41558-023-01682-9, 2023.

Rosentreter, J.A., Alcott, L., Maavara, T., Sun, X., Zhou, Y., Planavsky, N., & Raymond, P. Revisiting the Global Methane
Cycle Through Expert Opinion (submitted to Earth Future).

Ruppel, C. D., and J. D. Kessler (2017), The interaction of climate change and methane hydrates, Rev. Geophys., 55, 126-
168, doi:10.1002/2016RG000534, 2017

Saad, K. M., Wunch, D., Toon, G. C., Bernath, P., Boone, C., Connor, B., Deutscher, N. M., Griffith, D. W. T., Kivi, R.,
Notholt, J., Roehl, C., Schneider, M., Sherlock, V. and Wennberg, P. O.: Derivation of tropospheric methane from
TCCON CH$_4$ and HF total column observations, Atmospheric Meas. Technol., 7(9), 2907–2918, doi:10.5194/amt-7-
2907-2014, 2014.

Sanderson, M. G.: Biomass of termites and their emissions of methane and carbon dioxide: A global database, Glob.
Biogeochem. Cycles, 10(4), 543–557, doi:10.1029/96gb01893, 1996.

Sasakawa, M., Shimoyama, K., Machida, T., Tsuda, N., Suto, H., Arshinov, M., Davydov, D., Fofonov, A., Krasnov, O.,
Saeki, T., Koyama, Y. and Maksyutov, S.: Continuous measurements of methane from a tower network over Siberia,
Tellus B, 62(5), 403–416, doi:10.1111/j.1600-0889.2010.00494.x, 2010.

Sasakawa, M., Machida, T., Ishijima, K., Arshinov, M., Patra, P. K., Ito, A., Aoki, S., and Petrov, V.: Temporal
characteristics of CH4 vertical profiles observed in the West Siberian Lowland over Surgut from 1993 to 2015 and
Novosibirsk from 1997 to 2015. Journal of Geophysical Research: Atmospheres, 122, 11,261– 11,273.
https://doi.org/10.1002/2017JD026836, 2017.

Saunois, M., Bousquet, P., Poulter, B., Peregon, A., Ciais, P., Canadell, J. G., Dlugokencky, E. J., Etiope, G., Bastviken,
D., Houweling, S., Janssens-Maenhout, G., Tubiello, F. N., Castaldi, S., Jackson, R. B., Alexe, M., Arora, V. K.,
Beerling, D. J., Bergamaschi, P., Blake, D. R., Brailsford, G., Brovkin, V., Bruhwiler, L., Crevoisier, C., Crill, P.,
Covey, K., Curry, C., Frankenberg, C., Gedney, N., Höglund-Isaksson, L., Ishizawa, M., Ito, A., Joos, F., Kim, H. S.,
Kleinen, T., Krummel, P., Lamarque, J. F., Langenfelds, R., Locatelli, R., Machida, T., Maksyutov, S., McDonald, K.
C., Marshall, J., Melton, J. R., Morino, I., Naik, V., O'Doherty, S., Parmentier, F. J. W., Patra, P. K., Peng, C., Peng,
S., Peters, G. P., Pison, I., Prigent, C., Prinn, R., Ramonet, M., Riley, W. J., Saito, M., Santini, M., Schroeder, R.,
Simpson, I. J., Spahni, R., Steele, P., Takizawa, A., Thornton, B. F., Tian, H., Tohjima, Y., Viovy, N., Voulgarakis,





A., van Weele, M., van der Werf, G. R., Weiss, R., Wiedinmyer, C., Wilton, D. J., Wiltshire, A., Worthy, D., Wunch,
D., Xu, X., Yoshida, Y., Zhang, B., Zhang, Z. and Zhu, Q.: The global methane budget 2000–2012, Earth Syst Sci
Data, 8(2), 697–751, doi:10.5194/essd-8-697-2016, 2016.
Saunois, M., Bousquet, P., Poulter, B., Peregon, A., Ciais, P., Canadell, J. G., Dlugokencky, E. J., Etiope, G., Bastviken,
D., Houweling, S., Janssens-Maenhout, G., Tubiello, F. N., Castaldi, S., Jackson, R. B., Alexe, M., Arora, V. K.,
Beerling, D. J., Bergamaschi, P., Blake, D. R., Brailsford, G., Bruhwiler, L., Crevoisier, C., Crill, P., Covey, K.,
Frankenberg, C., Gedney, N., Höglund-Isaksson, L., Ishizawa, M., Ito, A., Joos, F., Kim, H. S., Kleinen, T., Krummel,
P., Lamarque, J. F., Langenfelds, R., Locatelli, R., Machida, T., Maksyutov, S., Melton, J. R., Morino, I., Naik, V.,
O'Doherty, S., Parmentier, F. J. W., Patra, P. K., Peng, C., Peng, S., Peters, G. P., Pison, I., Prinn, R., Ramonet, M.,
Riley, W. J., Saito, M., Santini, M., Schroeder, R., Simpson, I. J., Spahni, R., Takizawa, A., Thornton, B. F., Tian, H.,
Tohjima, Y., Viovy, N., Voulgarakis, A., Weiss, R., Wilton, D. J., Wiltshire, A., Worthy, D., Wunch, D., Xu, X.,
Yoshida, Y., Zhang, B., Zhang, Z. and Zhu, Q.: Variability and quasi-decadal changes in the methane budget over the
period 2000-2012, Atmospheric Chem. Phys., 17(18), 11135–11161, doi:10.5194/acp-17-11135-2017, 2017.
Saunois, M., Stavert, A. R., Poulter, B., Bousquet, P., Canadell, J. G., Jackson, R. B., Raymond, P. A., Dlugokencky, E.
J., Houweling, S., Patra, P. K., Ciais, P., Arora, V. K., Bastviken, D., Bergamaschi, P., Blake, D. R., Brailsford, G.,
Bruhwiler, L., Carlson, K. M., Carrol, M., Castaldi, S., Chandra, N., Crevoisier, C., Crill, P. M., Covey, K., Curry, C.
L., Etiope, G., Frankenberg, C., Gedney, N., Hegglin, M. I., Höglund-Isaksson, L., Hugelius, G., Ishizawa, M., Ito, A.,
Janssens-Maenhout, G., Jensen, K. M., Joos, F., Kleinen, T., Krummel, P. B., Langenfelds, R. L., Laruelle, G. G., Liu,
L., Machida, T., Maksyutov, S., McDonald, K. C., McNorton, J., Miller, P. A., Melton, J. R., Morino, I., Müller, J.,
Murguia-Flores, F., Naik, V., Niwa, Y., Noce, S., O'Doherty, S., Parker, R. J., Peng, C., Peng, S., Peters, G. P., Prigent,
C., Prinn, R., Ramonet, M., Regnier, P., Riley, W. J., Rosentreter, J. A., Segers, A., Simpson, I. J., Shi, H., Smith, S.
J., Steele, L. P., Thornton, B. F., Tian, H., Tohjima, Y., Tubiello, F. N., Tsuruta, A., Viovy, N., Voulgarakis, A., Weber,
T. S., van Weele, M., van der Werf, G. R., Weiss, R. F., Worthy, D., Wunch, D., Yin, Y., Yoshida, Y., Zhang, W.,
Zhang, Z., Zhao, Y., Zheng, B., Zhu, Q., Zhu, Q., and Zhuang, Q.: The Global Methane Budget 2000–2017, Earth
Syst. Sci. Data, 12, 1561–1623, https://doi.org/10.5194/essd-12-1561-2020, 2020.
Sayers, M.J., Grimm, A.G., Shuchman, R.A., Deines, A.M., Bunnel, D.B., Raymer, Z.B., Rogers, M.W., Woelmer, W.,
Bennion, D.H., Brooks, C.N., Whitley, MA.A., Warner, D.M, and J. Mychek-Londer: A new method to generate a
high-resolution global distribution map of lake chlorophyll, International Journal of Remote Sensing, 36:7, 1942-1964,
DOI: 10.1080/01431161.2015.1029099, 2015
Schepers, D., Guerlet, S., Butz, A., Landgraf, J., Frankenberg, C., Hasekamp, O., Blavier, J. F., Deutscher, N. M., Griffith,
D. W. T., Hase, F., Kyro, E., Morino, I., Sherlock, V., Sussmann, R. and Aben, I.: Methane retrievals from Greenhouse
Gases Observing Satellite (GOSAT) shortwave infrared measurements: Performance comparison of proxy and physics
retrieval algorithms, J. Geophys. Res. Atmospheres, 117, D10, doi:10.1029/2012jd017549, 2012.



Schmale O, Greinert J, Rehder G (2005) Methane emission from high-intensity marine gas seeps in the Black Sea into the
atmosphere. Geophys Res Lett 32:L07609. doi:10.1029/2004GL021138, 2005

Schmid, M., Batist, M.D., Granin, N.G., Kapitanov, V.A., McGinnis, D.F., Mizandrontsev, I.B., Obzhirov, A.I., and
Wüest, A.. Sources and sinks of methane in Lake Baikal: A synthesis of measurements and modelling. Limnol.
Oceanogr., 52(5), 1824–1837. doi: 10.4319/lo.2007.52.5.1824, 2007

Schneising, O., Burrows, J. P., Dickerson, R. R., Buchwitz, M., Reuter, M. and Bovensmann, H.: Remote sensing of
fugitive methane emissions from oil and gas production in North American tight geologic formations, Earths Future,
2, 548–558, doi:10.1002/2014EF000265, 2014.

Schorn, S., S. Ahmerkamp, E. Bullock, and others. : Diverse methylotrophic methanogenic archaea cause high methane
emissions from seagrass meadows. Proc. Natl. Acad. Sci. **119**: 1–12. doi:10.1073/pnas.2106628119, 2022

Schuldt, K. N., Mund, J., Aalto, T., Arlyn Andrews, Apadula, F., Jgor Arduini, Arnold, S., Baier, B., Bäni, L., Bartyzel,
3531        J., Bergamaschi, P., Biermann, T., Biraud, S. C., Pierre-Eric Blanc, Boenisch, H., Brailsford, G., Brand, W. A.,
Brunner, D., Bui, T. P. V., … Miroslaw Zimnoch. : *Multi-laboratory compilation of atmospheric carbon dioxide data*
*for the period 1983-2022; obspack_ch4_1_GLOBALVIEWplus_v6.0_2023-12-01* [Data set]. NOAA Global
Monitoring Laboratory. https://doi.org/10.25925/20231001, 2023

Schuur, E.A., Abbott, B.W., Commane, R., Ernakovich, J., Euskirchen, E., Hugelius, G., Grosse, G., Jones, M., Koven,
C., Leshyk, V. and Lawrence, D. (2022) Permafrost and climate change: carbon cycle feedbacks from the warming
Arctic. Annual Review of Environment and Resources, 47, pp.343-371. https://doi.org/10.1146/annurev-environ-
012220-011847, 2022

Schwietzke, S., Sherwood, O. A., Bruhwiler, L. M. P., Miller, J. B., Etiope, G., Dlugokencky, E. J., Michel, S. E., Arling,
3540        V. A., Vaughn, B. H., White, J. W. C. and Tans, P. P.: Upward revision of global fossil fuel methane emissions based
on isotope database, Nature, 538(7623), 88–91, doi:10.1038/nature19797, 2016.

Segers, A., Steinke, T., and Houweling, S.: Description of the CH4 Inversion Production Chain, CAMS (Copernicus
Atmospheric Monitoring Service) Report.. [online] Available from:
https://atmosphere.copernicus.eu/sites/default/files/2022-10/CAMS255_2021SC1_D55.5.2.1-
2021CH4_202206_production_chain_CH4_v1.pdf (Accessed 1 février 2024), 2022.

Shen, L., Gautam, R., Omara, M., Zavala-Araiza, D., Maasakkers, J. D., Scarpelli, T. R., Lorente, A., Lyon, D., Sheng, J.,
Varon, D. J., Nesser, H., Qu, Z., Lu, X., Sulprizio, M. P., Hamburg, S. P., and Jacob, D. J.: Satellite quantification of
oil and natural gas methane emissions in the US and Canada including contributions from individual basins, Atmos.
Chem. Phys., 22, 11203–11215, https://doi.org/10.5194/acp-22-11203-2022, 2022.

Shen, L., Jacob, D.J., Gautam, R. et al. National quantifications of methane emissions from fuel exploitation using high
resolution inversions of satellite observations. Nat Commun 14, 4948 , https://doi.org/10.1038/s41467-023-40671-6,
2023



Sherwen, T., Schmidt, J. A., Evans, M. J., Carpenter, L. J., Großmann, K., Eastham, S. D., Jacob, D. J., Dix, B., Koenig,
3554        T. K., Sinreich, R., Ortega, I., Volkamer, R., Saiz-Lopez, A., Prados-Roman, C., Mahajan, A. S., and Ordóñez, C.:
Global impacts of tropospheric halogens (Cl, Br, I) on oxidants and composition in GEOS-Chem, Atmos. Chem. Phys.,
16, 12239–12271, https://doi.org/10.5194/acp-16-12239-2016, 2016.

Shindell, D., Kuylenstierna, J. C. I., Vignati, E., van Dingenen, R., Amann, M., Klimont, Z., Anenberg, S. C., Muller, N.,
Janssens-Maenhout, G., Raes, F., Schwartz, J., Faluvegi, G., Pozzoli, L., Kupiainen, K., Höglund-Isaksson, L.,
Emberson, L., Streets, D., Ramanathan, V., Hicks, K., Oanh, N. T. K., Milly, G., Williams, M., Demkine, V. and
Fowler, D.: Simultaneously Mitigating Near-Term Climate Change and Improving Human Health and Food Security,
Science, 335(6065), 183–189, doi:10.1126/science.1210026, 2012.

Shorter, J. H., Mcmanus, J. B., Kolb, C. E., Allwine, E. J., Lamb, B. K., Mosher, B. W., Harriss, R. C., Partchatka, U.,
Fischer, H., Harris, G. W., Crutzen, P. J. and Karbach, H.-J.: Methane emission measurements in urban areas in Eastern
Germany, J. Atmospheric Chem., 124(2), 121–140, 1996.

Shu, S., Jain, A.K. and Kheshgi, H.S.: Investigating Wetland and Nonwetland Soil Methane Emissions and Sinks Across
the Contiguous United States Using a Land Surface Model. Global Biogeochem. Cycles, 34: e2019GB006251.
https://doi-org.insu.bib.cnrs.fr/10.1029/2019GB006251, 2020

Simpson, I. J., Thurtell, G. W., Kidd, G. E., Lin, M., Demetriades-Shah, T. H., Flitcroft, I. D., Kanemasu, E. T., Nie, D.,
Bronson, K. F. and Neue, H. U.: Tunable diode laser measurements of methane fluxes from an irrigated rice paddy
field in the Philippines, J. Geophys. Res. Atmospheres, 100(D4), 7283–7290, doi:10.1029/94jd03326, 1995.

Simpson, I. J., Sulbaek Andersen, M. P., Meinardi, S., Bruhwiler, L., Blake, N. J., Helmig, D., Rowland, F. S. and Blake,
D. R.: Long-term decline of global atmospheric ethane concentrations and implications for methane, Nature,
488(7412), 490–494, doi:10.1038/nature11342, 2012.

Smith I.R., Grasby S.E., Lane L.S.: An investigation of gas seeps and aquatic chemistry in Fisherman Lake, southwest
Northwest Territories. Geological Survey of Canada, Current Research 2005-A3, 8 p., 2005

Solomon EA, Kastner M, MacDonald IR, Leifer I: Considerable methane fluxes to the atmosphere from hydrocarbon
seeps in the Gulf of Mexico. Nat Geosci 2:561–565, 2009

Spahni, R., Wania, R., Neef, L., van Weele, M., Pison, I., Bousquet, P., Frankenberg, C., Foster, P. N., Joos, F., Prentice,
I. C. and van Velthoven, P.: Constraining global methane emissions and uptake by ecosystems, Biogeosciences, 8(6),
1643–1665, doi:10.5194/bg-8-1643-2011, 2011.

Stanley, E. H., Casson, N. J., Christel, S. T., Crawford, J. T., Loken, L. C. and Oliver, S. K.: The ecology of methane in
streams and rivers: patterns, controls, and global significance, Ecol. Monogr., doi:10.1890/15-1027, 2016.

Stanley, K. M., Grant, A., O'Doherty, S., Young, D., Manning, A. J., Stavert, A. R., Spain, T. G., Salameh, P. K., Harth,
C. M., Simmonds, P. G., Sturges, W. T., Oram, D. E. and Derwent, R. G.: Greenhouse gas measurements from a UK
network of tall towers: technical description and first results, Atmospheric Meas. Tech., 11(3), 1437–1458,





doi:10.5194/amt-11-1437-2018, 2018.

Stanley, E. H., Loken, L. C., Casson, N. J., Oliver, S. K., Sponseller, R. A., Wallin, M. B., Zhang, L., and Rocher-Ros,

G.: GRiMeDB: the Global River Methane Database of concentrations and fluxes, Earth Syst. Sci. Data, 15, 2879–

2926, https://doi.org/10.5194/essd-15-2879-2023, 2023.

Stavert, A. R., Saunois, M., Canadell, J. G., Poulter, B., Jackson, R. B., Regnier, P., Lauerwald, R., Raymond, P. A.,

Allen, G. H., Patra, P. K., Bergamaschi, P., Bousquet, P., Chandra, N., Ciais, P., Gustafson, A., Ishizawa, M., Ito,

3592          A., Kleinen, T., Maksyutov, S., Joe McNorton, Joe R. Melton, Jurek Müller, Yosuke Niwa, Shushi Peng, William

3593          J. Riley, Arjo Segers, Hanqin Tian, Aki Tsuruta, Yi Yin, Zhen Zhang, Bo Zheng, Zhuang, Q. Regional trends

and drivers of the global methane budget. *Global Change Biology*, 28, 182–200. https://doi.org/10.1111/gcb.15901,

2021

Steele, L. P., Fraser, P. J., Rasmussen, R. A., Khalil, M. A. K., Conway, T. J., Crawford, A. J., Gammon, R. H., Masarie,

3597          K. A. and Thoning, K. W.: The global distribution of methane in the troposphere, J. Atmospheric Chem., 5, 125–171,

1987.

Stevenson, D. S., Derwent, R. G., Wild, O., and Collins, W. J.: COVID-19 lockdown emission reductions have the

potential to explain over half of the coincident increase in global atmospheric methane, Atmos. Chem. Phys., 22,

14243–14252, https://doi.org/10.5194/acp-22-14243-2022, 2022.

Stocker, B. D., Spahni, R. and Joos, F.: DYPTOP: a cost-efficient TOPMODEL implementation to simulate sub-grid

spatio-temporal dynamics of global wetlands and peatlands, Geosci. Model Dev., 7(6), 3089–3110, doi:10.5194/gmd-

7-3089-2014, 2014.

Strauss, J., Abbott, B.W., Hugelius, G., Schuur, E., Treat, C., Fuchs, M., Schädel, C., Ulrich, M., Turetsky, M., Keuschnig,

3606          M. and Biasi, C. (2021) Chapter 9. Permafrost. In FAO Recarbonizing global soils–A technical manual of

recommended management practices: Volume 2–Hot spots and bright spots of soil organic carbon, p.130, 2021

Strode, S. A., Wang, J. S., Manyin, M., Duncan, B., Hossaini, R., Keller, C. A., Michel, S. E., and White, J. W. C.: Strong

sensitivity of the isotopic composition of methane to the plausible range of tropospheric chlorine, Atmos. Chem. Phys.,

20, 8405–8419, https://doi.org/10.5194/acp-20-8405-2020, 2020.

Sugimoto, A., Inoue, T., Kitibutr, N., Abe, T: Methane oxidation by termite mounds estimate by the carbon isotope

composition of methane. Glob. Biogeochem. Cy. 12, 595-605. 1998.

Sweeney, C., Karion, A., Wolter, S., Newberger, T., Guenther, D., Higgs, J. A., Andrews, A. E., Lang, P. M., Neff, D.,

Dlugokencky, E., Miller, J. B., Montzka, S. A., Miller, B. R., Masarie, K. A., Biraud, S. C., Novelli, P. C., Crotwell,

3615          M., Crotwell, A. M., Thoning, K. and Tans, P. P.: Seasonal climatology of $CO_2$ across North America from aircraft

measurements in the NOAA/ESRL Global Greenhouse Gas Reference Network, J. Geophys. Res. Atmospheres,

120(10), 5155–5190, doi:10.1002/2014jd022591, 2015.

Tan, Z. and Zhuang, Q.: Methane emissions from pan-Arctic lakes during the 21st century: An analysis with process-



based models of lake evolution and biogeochemistry, J. Geophys. Res. Biogeosciences, 120(12), 2641–2653,
doi:10.1002/2015JG003184, 2015.

Tans, P. and Zwellberg, C.: 17th WMO/IAEA Meeting on Carbon Dioxide, Other Greenhouse Gases and Related Tracers
Measurement Techniques (GGMT-2013), GAW Report, WMO, Geneva. [online] Available from:
https://library.wmo.int/index.php?lvl=notice_display&id=16373#.XnpBPW7jIq8, 2014.

Taranu, Z.E., I. Gregory-Eaves, P.R. Leavitt, L. Bunting, T. Buchaca, J. Catalan, I. Domaizon, P. Guilizzoni, A. Lami, S.
McGowan, H. Moorhouse, G. Morabito, F.R. Pick, M.A. Stevenson, P.L. Thompson, and R.D. Vinebrooke:
Acceleration of cyanobacterial dominance in north temperate-subarctic lakes during the Anthropocene. Ecology
Letters, 18(4): p. 375-384., 2015

Taylor, P. G., Bilinski, T. M., Fancher, H. R. F., Cleveland, C. C., Nemergut, D. R., Weintraub, S. R., Wieder, W. R. and
Townsend, A. R.: Palm oil wastewater methane emissions and bioenergy potential, Nat. Clim. Change, 4(3), 151–152,
doi:10.1038/nclimate2154, 2014.

le Texier, H., Solomon, S. and Garcia, R. R.: The role of molecular hydrogen and methane oxidation in the water vapour
budget of the stratosphere, Q. J. R. Meteorol. Soc., 114(480), 281–295, doi:10.1002/qj.49711448002, 1988.

Thanwerdas, J., Saunois, M., Berchet, A., Pison, I., Vaughn, B. H., Michel, S. E., and Bousquet, P.: Variational inverse
modeling within the Community Inversion Framework v1.1 to assimilate δ13C(CH4) and CH4: a case study with
model LMDz-SACS, Geosci. Model Dev., 15, 4831–4851, https://doi.org/10.5194/gmd-15-4831-2022, 2022a.

Thanwerdas, J., Saunois, M., Pison, I., Hauglustaine, D., Berchet, A., Baier, B., Sweeney, C., and Bousquet, P.: How do
Cl concentrations matter for the simulation of CH4 and δ13C(CH4) and estimation of the CH4 budget through
atmospheric inversions?, Atmos. Chem. Phys., 22, 15489–15508, https://doi.org/10.5194/acp-22-15489-2022, 2022b.

Thanwerdas, J., Saunois, M., Berchet, A., Pison, I., and Bousquet, P.: Investigation of the renewed methane growth post-
2007 with high-resolution 3-D variational inverse modeling and isotopic constraints, Atmos. Chem. Phys., 24, 2129–
2167, https://doi.org/10.5194/acp-24-2129-2024, 2024.

Thompson, R. L., Montzka, S. A., Vollmer, M. K., Arduini, J., Crotwell, M., Krummel, P. B., Lunder, C., Mühle, J.,
O'Doherty, S., Prinn, R. G., Reimann, S., Vimont, I., Wang, H., Weiss, R. F., and Young, D.: Estimation of the
atmospheric hydroxyl radical oxidative capacity using multiple hydrofluorocarbons (HFCs), Atmos. Chem. Phys., 24,
1415–1427, https://doi.org/10.5194/acp-24-1415-2024, 2024.

Thoning, K. W., Tans, P. P. and Komhyr, W. D.: Atmospheric carbon dioxide at Mauna Loa Observatory. 2. Analysis of
the NOAA GMCC data, 1974,1985, J. Geophys. Res., 94(D6), 8549–8565, 1989.

Thorneloe, S. A., Barlaz, M. A., Peer, R., Huff, L. C., Davis, L. and Mangino, J.: Waste management, in Atmospheric
Methane: Its Role in the Global Environment, edited by M. Khalil, pp. 234–262, Springer-Verlag, New York., 2000.

Thornton, B. F., Prytherch, J., Andersson, K., Brooks, I. M., Salisbury, D., Tjernström, M. and Crill, P. M.: Shipborne
eddy covariance observations of methane fluxes constrain Arctic sea emissions, Sci. Adv., 6(5), eaay7934,



doi:10.1126/sciadv.aay7934, 2020.

Thornton B.F., Etiope G., Schwietzke S., Milkov A.V., Klusman R.W., Judd A., Oehler D.Z.: Conflicting estimates of natural geologic methane emissions. Elem. Sci. Anth., 9, 1, doi:https:// doi.org/10.1525/elementa.2021.00031, 2021

Thornton, J. A., Kercher, J. P., Riedel, T. P., Wagner, N. L., Cozic, J., Holloway, J. S., Dubé, W. P., Wolfe, G. M., Quinn, P. K., Middlebrook, A. M., Alexander, B. and Brown, S. S.: A large atomic chlorine source inferred from mid-continental reactive nitrogen chemistry, Nature, 464(7286), 271–274, doi:10.1038/nature08905, 2010.

Thorpe, A. K. , Kort, E. A. , Cusworth, D. H. , Ayasse, A. K. , Bue, B. D. ,Yadav, V. ,Thompson, D. R. , Frankenberg, C. , Herner, J. , Falk, M. ,Green, R. O. ,Miller, C. E. , and Duren, R. M.: Methane emissions decline from reduced oil, natural gas, and refinery production during COVID-19, Environmental Research Communications, 5, 021006, 2023

Tian, H., Xu, X., Liu, M., Ren, W., Zhang, C., Chen, G. and Lu, C.: Spatial and temporal patterns of $CH_4$ and $N_2O$ fluxes in terrestrial ecosystems of North America during 1979–2008: application of a global biogeochemistry model, Biogeosciences, 7(9), 2673–2694, doi:10.5194/bg-7-2673-2010, 2010.

Tian, H., Xu, X., Lu, C., Liu, M., Ren, W., Chen, G., Melillo, J. and Liu, J.: Net exchanges of $CO_2$, $CH_4$, and $N_2O$ between China's terrestrial ecosystems and the atmosphere and their contributions to global climate warming, J. Geophys. Res. Biogeosciences, 116, G2, doi:10.1029/2010jg001393, 2011.

Tian, H., Chen, G., Lu, C., Xu, X., Ren, W., Zhang, B., Banger, K., Tao, B., Pan, S., Liu, M., Zhang, C., Bruhwiler, L. and Wofsy, S.: Global methane and nitrous oxide emissions from terrestrial ecosystems due to multiple environmental changes, Ecosyst. Health Sustain., 1(1), 1–20, doi:doi:10.1890/ehs14-0015.1, 2015.

Tian, H., Lu, C., Ciais, P., Michalak, A. M., Canadell, J. G., Saikawa, E., Huntzinger, D. N., Gurney, K. R., Sitch, S., Zhang, B., Yang, J., Bousquet, P., Bruhwiler, L., Chen, G., Dlugokencky, E., Friedlingstein, P., Melillo, J., Pan, S., Poulter, B., Prinn, R., Saunois, M., Schwalm, C. R. and Wofsy, S. C.: The terrestrial biosphere as a net source of greenhouse gases to the atmosphere, Nature, 531(7593), 225–228, doi:10.1038/nature16946, 2016.

Tian, H., Xu, R., Canadell, J. G., Thompson, R. L., Winiwarter, W., Suntharalingam, P., Davidson, E. A., Ciais, P., Jackson, R. B., Janssens-Maenhout, G., Prather, M. J., Regnier, P., Pan, N., Pan, S., Peters, G. P., Shi, H., Tubiello, F. N., Zaehle, S., Zhou, F., Arneth, A., Battaglia, G., Berthet, S., Bopp, L., Bouwman, A. F., Buitenhuis, E. T., Chang, J., Chipperfield, M. P., Dangal, S. R. S., Dlugokencky, E., Elkins, J. W., Eyre, B. D., Fu, B., Hall, B., Ito, A., Joos, F., Krummel, P. B., Landolfi, A., Laruelle, G. G., Lauerwald, R., Li, W., Lienert, S., Maavara, T., MacLeod, M., Millet, D. B., Olin, S., Patra, P. K., Prinn, R. G., Raymond, P. A., Ruiz, D. J., van der Werf, G. R., Vuichard, N., Wang, J., Weiss, R. F., Wells, K. C., Wilson, C., Yang, J., and Yao, Y.: A comprehensive quantification of global nitrous oxide sources and sinks, Nature, 586, 248–256, https://doi.org/10.1038/s41586-020-2780-0, 2020.

Tian, H., Yao, Y., Li, Y., Shi, H., Pan, S., Najjar, R. G., et al. (2023). Increased terrestrial carbon export and CO2 evasion from global inland waters since the preindustrial era. *Global Biogeochemical Cycles*, 37, e2023GB007776. https://doi.org/10.1029/2023GB007776, 2023



Tibrewal, K., Ciais, P., Saunois, M. Martinez, A., Lin, X., Thanwerdas, J., Deng, Z. , Chevallier, F. , Giron, C., Albergel,
C., Tanaka, K., Patra, P., Tsuruta, A., Zheng, B., Belikov, D., Niwa, Y. , Janardanan, R. , Maksyutov, S., Segers, A.,
Tzompa-Sosa, Z. A., Bousquet, P., and Sciare, J.: Assessment of methane emissions from oil, gas and coal sectors
across inventories and atmospheric inversions, Commun Earth Environ 5, 26, https://doi.org/10.1038/s43247-023-
01190-w, 2024

Tiwari, Y. K. and Kumar, K. R.: GHG observation programs in India, Asian GAWgreenhouse Gases 3 Korea Meteorol.
Adm. Chungnam South Korea, 2012.

Tsuruta, A., Aalto, T., Backman, L., Hakkarainen, J., Laan-Luijkx, I. T. van der, Krol, M. C., Spahni, R., Houweling, S.,
Laine, M., Dlugokencky, E., Gomez-Pelaez, A. J., Schoot, M. van der, Langenfelds, R., Ellul, R., Arduini, J., Apadula,
F., Gerbig, C., Feist, D. G., Kivi, R., Yoshida, Y. and Peters, W.: Global methane emission estimates for 2000–2012
from CarbonTracker Europe-CH₄ v1.0, Geosci. Model Dev., 10(3), 1261–1289, doi:10.5194/gmd-10-1261-2017,
2017.

Tsuruta, A.; Kivimäki, E.; Lindqvist, H.; Karppinen, T.; Backman, L.; Hakkarainen, J.; Schneising, O.; Buchwitz, M.;
Lan, X.; Kivi, R.; et al. CH4 Fluxes Derived from Assimilation of TROPOMI XCH4 in CarbonTracker Europe-CH4:
Evaluation of Seasonality and Spatial Distribution in the Northern High Latitudes. Remote Sens. 2023, 15, 1620.
https://doi.org/10.3390/rs15061620, 2023

Tubiello, F. N.: Greenhouse Gas Emissions Due to Agriculture, in Elsevier Encyclopedia of Food Systems., 2019.
Tubiello, F. N., Salvatore, M., Rossi, S., Ferrara, A., Fitton, N. and Smith, P.: The FAOSTAT database of greenhouse gas
emissions from agriculture, Environ. Res. Lett., 8(1), 015009, doi:10.1088/1748-9326/8/1/015009, 2013.

Tubiello, F. N., Karl, K., Flammini, A., Gütschow, J., Obli-Laryea, G., Conchedda, G., Pan, X., Qi, S. Y., Halldórudóttir
Heiðarsdóttir, H., Wanner, N., Quadrelli, R., Rocha Souza, L., Benoit, P., Hayek, M., Sandalow, D., Mencos Contreras,
E., Rosenzweig, C., Rosero Moncayo, J., Conforti, P., and Torero, M.: Pre- and post-production processes increasingly
dominate greenhouse gas emissions from agri-food systems, Earth Syst. Sci. Data, 14, 1795–1809,
https://doi.org/10.5194/essd-14-1795-2022, 2022.

Turetsky, M. R., Kotowska, A., Bubier, J., Dise, N. B., Crill, P., Hornibrook, E. R. C., Minkkinen, K., Moore, T. R.,
Myers-Smith, I. H., Nykänen, H., Olefeldt, D., Rinne, J., Saarnio, S., Shurpali, N., Tuittila, E.-S., Waddington, J. M.,
White, J. R., Wickland, K. P. and Wilmking, M.: A synthesis of methane emissions from 71 northern, temperate, and
subtropical wetlands, Glob. Change Biol., 20(7), 2183–2197, doi:10.1111/gcb.12580, 2014.

Turetsky, M. R., Abbott, B. W., Jones, M. C., Anthony, K. W., Olefeldt, D., Schuur, E. A. G., et al.: Carbon release
through abrupt permafrost thaw. *Nature Geoscience*, 13(2), 138–143. https://doi.org/10.1038/s41561-019-0526-0,
2020

Turner, A. J., Fung, I., Naik, V., Horowitz, L. W. and Cohen, R. C.: Modulation of hydroxyl variability by ENSO in the
absence of external forcing, Proc. Natl. Acad. Sci., 115(36), 8931–8936, doi:10.1073/pnas.1807532115, 2018.



Turner, A. J., Frankenberg, C. and Kort, E. A.: Interpreting contemporary trends in atmospheric methane, Proc. Natl. Acad. Sci., 116(8), 2805, doi:10.1073/pnas.1814297116, 2019.

UNEP, United Nations Environment Programme and Climate and Clean Air Coalition. Global Methane Assessment: Benefits and Costs of Mitigating Methane Emissions. Nairobi: United Nations Environment Programme., 2021

UNEP, United Nations Environment Programme/Climate and Clean Air Coalition. Global Methane Assessment: 2030 Baseline Report. Nairobi, 2022

USEPA: Greenhouse Gas Emissions Estimation Methodologies for Biogenic Emissions from Selected Source Categories: Solid Waste Disposal Wastewater Treatment Ethanol Fermentation, Measurement Policy Group, US EPA. [online] Available from: https://www3.epa.gov/ttnchie1/efpac/ghg/GHG_Biogenic_Report_draft_Dec1410.pdf (Accessed 11 March 2020a), 2010a.

USEPA: Office of Atmospheric Programs (6207J), Methane and Nitrous Oxide Emissions From Natural Sources, U.S. Environmental Protection Agency, EPA 430-R-10-001. Available online at http://nepis.epa.gov/, Washington, DC 20460., 2010b.

USEPA: Draft: Global Anthropogenic Non-CO$_2$ Greenhouse Gas Emissions: 1990-2030. EPA 430-R-03-002, United States Environmental Protection Agency, Washington D.C., 2011.

USEPA: Global Anthropogenic Non-CO2 Greenhouse Gas Emissions 1990-2030, EPA 430-R-12-006, US Environmental Protection Agency, Washington DC., 2012.

USEPA: Draft Inventory of U.S. Greenhouse gas Emissions and Sinks: 1990-2014. EPA 430-R-16-002. February 2016. U.S. Environmental protection Agency, Washington, DC, USA., 2016.

USEPA: Global Non-CO2 Greenhouse Gas Emission Projections & Mitigation Potential: 2015-2050, EPA-430-R-19-010, U.S. Environmental protection Agency, Washington, DC, USA., 2019

Valentine, D. W., Holland, E. A. and Schimel, D. S.: Ecosystem and physiological controls over methane production in northern wetlands, J. Geophys. Res., 99(D1), 1563–1571, 1994.

Vardag, S. N., Hammer, S., O'Doherty, S., Spain, T. G., Wastine, B., Jordan, A. and Levin, I.: Comparisons of continuous atmospheric CH$_4$, CO$_2$ and N$_2$O measurements – results from a travelling instrument campaign at Mace Head, Atmospheric Chem. Phys., 14(16), 8403–8418, doi:10.5194/acp-14-8403-2014, 2014.

VODCA2GPP – a new, global, long-term (1988–2020) gross primary production dataset from microwave remote sensing, Earth Syst. Sci. Data, 14, 1063–1085, https://doi.org/10.5194/essd-14-1063-2022, 2022.

Voulgarakis, A., Naik, V., Lamarque, J. F., Shindell, D. T., Young, P. J., Prather, M. J., Wild, O., Field, R. D., Bergmann, D., Cameron-Smith, P., Cionni, I., Collins, W. J., Dals√∏ren, S. B., Doherty, R. M., Eyring, V., Faluvegi, G., Folberth, G. A., Horowitz, L. W., Josse, B., MacKenzie, I. A., Nagashima, T., Plummer, D. A., Righi, M., Rumbold, S. T., Stevenson, D. S., Strode, S. A., Sudo, K., Szopa, S. and Zeng, G.: Analysis of present day and future OH and methane



lifetime in the ACCMIP simulations, Atmospheric Chem. Phys., 13(5), 2563–2587, doi:10.5194/acp-13-2563-2013, 2013.

Voulgarakis, A., Marlier, M. E., Faluvegi, G., Shindell, D. T., Tsigaridis, K. and Mangeon, S.: Interannual variability of tropospheric trace gases and aerosols: The role of biomass burning emissions, J. Geophys. Res. Atmospheres, 120(14), 7157–7173, doi:10.1002/2014jd022926, 2015.

Wallmann, K., Pinero, E., Burwicz, E., Haeckel, M., Hensen, C., Dale, A. and Ruepke, L.: The Global Inventory of Methane Hydrate in Marine Sediments: A Theoretical Approach, Energies, 5(7), 2449, 2012.

Walter Anthony, K.M., Anthony, P., Grosse, G. and Chanton, J.: Geologic methane seeps along boundaries of Arctic permafrost thaw and melting glaciers. Nature Geoscience, 5(6), pp.419-426., DOI: 10.1038/ngeo1480, 2012

Wang, F., Maksyutov, S., Tsuruta, A., Janardanan, R., Ito, A., Sasakawa, M., Machida, T., Morino, I., Yoshida, Y., Kaiser, J. W., Janssens-Maenhout, G., Dlugokencky, E. J., Mammarella, I., Lavric, J. V. and Matsunaga, T.: Methane Emission Estimates by the Global High-Resolution Inverse Model Using National Inventories, Remote Sens., 11(21), 2489, doi:10.3390/rs11212489, 2019a.

Wang, G., X. Xia, S. Liu, L. Zhang, S. Zhang, J. Wang, N. Xi, and Q. Zhang, Intense methane ebullition from urban inland waters and its significant contribution to greenhouse gas emissions. Water Research, 189: p. 116654, 2021a

Wang, X., Jacob, D. J., Eastham, S. D., Sulprizio, M. P., Zhu, L., Chen, Q., Alexander, B., Sherwen, T., Evans, M. J., Lee, B. H., Haskins, J. D., Lopez-Hilfiker, F. D., Thornton, J. A., Huey, G. L. and Liao, H.: The role of chlorine in global tropospheric chemistry, Atmospheric Chem. Phys., 19(6), 3981–4003, doi:10.5194/acp-19-3981-2019, 2019b.

Wang, X., Jacob, D. J., Downs, W., Zhai, S., Zhu, L., Shah, V., Holmes, C. D., Sherwen, T., Alexander, B., Evans, M. J., Eastham, S. D., Neuman, J. A., Veres, P. R., Koenig, T. K., Volkamer, R., Huey, L. G., Bannan, T. J., Percival, C. J., Lee, B. H., and Thornton, J. A.: Global tropospheric halogen (Cl, Br, I) chemistry and its impact on oxidants, Atmos. Chem. Phys., 21, 13973–13996, https://doi.org/10.5194/acp-21-13973-2021, 2021b.

Wang, Z., Deutscher, N. M., Warneke, T., Notholt, J., Dils, B., Griffith, D. W. T., Schmidt, M., Ramonet, M. and Gerbig, C.: Retrieval of tropospheric column-averaged $CH_4$ mole fraction by solar absorption FTIR-spectrometry using $N_2O$ as a proxy, Atmospheric Meas. Tech., 7(10), 3295–3305, doi:10.5194/amt-7-3295-2014, 2014.

Wang, Z.-P., Gu, Q., Deng, F.-D., Huang, J.-H., Megonigal, J. P., Yu, Q., Lü, X.-T., Li, L.-H., Chang, S., Zhang, Y.-H., Feng, J.-C. and Han, X.-G.: Methane emissions from the trunks of living trees on upland soils, New Phytol., 211(2), 429–439, doi:10.1111/nph.13909, 2016.

Wania, R., I. Ross and I. C. Prentice: Implementation and evaluation of a new methane model within a dynamic global vegetation model: LPJ-WHyMe v1.3, Geosci. Model Dev. Discuss., 3, 1–59, 2010.

Wania, R., Melton, J. R., Hodson, E. L., Poulter, B., Ringeval, B., Spahni, R., Bohn, T., Avis, C. A., Chen, G., Eliseev, A. V., Hopcroft, P. O., Riley, W. J., Subin, Z. M., Tian, H., van Bodegom, P. M., Kleinen, T., Yu, Z. C., Singarayer, J. S., Zurcher, S., Lettenmaier, D. P., Beerling, D. J., Denisov, S. N., Prigent, C., Papa, F. and Kaplan, J. O.: Present state



of global wetland extent and wetland methane modelling: Methodology of a model inter-comparison project
(WETCHIMP), Geosci. Model Dev., 6(3), 617–641, 2013.
Wassmann, R., Lantin, R. S., Neue, H. U., Buendia, L. V., Corton, T. M. and Lu, Y.: Characterization of methane emissions
in Asia III: Mitigation options and future research needs, Nutr. Cycl. Agroecosystems, 58, 23–36, 2000.
Weber, T., Wiseman, N. A. and Kock, A.: Global ocean methane emissions dominated by shallow coastal waters, Nat.
Commun., 10(1), 1–10, doi:10.1038/s41467-019-12541-7, 2019.
Wells, N. S., J. J. Chen, D. T. Maher, P. Huang, D. V. Erler, M. Hipsey, and B. D. Eyre: Changing sediment and surface
water processes increase CH4 emissions from human-impacted estuaries. Geochim. Cosmochim. Acta **280**: 130–147.
doi:10.1016/j.gca.2020.04.020, 2020
van der Werf, G. R., Randerson, J. T., Giglio, L., Collatz, G. J., Mu, M., Kasibhatla, P. S., Morton, D. C., DeFries, R. S.,
Jin, Y. and van Leeuwen, T. T.: Global fire emissions and the contribution of deforestation, savanna, forest,
agricultural, and peat fires (1997-2009), Atmospheric Chem. Phys., 10(23), 11,707-11,735, 2010.
van der Werf, G. R., Randerson, J. T., Giglio, L., Leeuwen, T. T. van, Chen, Y., Rogers, B. M., Mu, M., Marle, M. J. E.
van, Morton, D. C., Collatz, G. J., Yokelson, R. J. and Kasibhatla, P. S.: Global fire emissions estimates during 1997–
2016, Earth Syst. Sci. Data, 9(2), 697–720, doi:10.5194/essd-9-697-2017, 2017.
Whalen, S. C.: Biogeochemistry of Methane Exchange between Natural Wetlands and the Atmosphere, Environ. Eng.
Sci., 22(1), 73–94, doi:10.1089/ees.2005.22.73, 2005.
Wiedinmyer, C., Kimura, Y., McDonald-Buller, E. C., Emmons, L. K., Buchholz, R. R., Tang, W., Seto, K., Joseph, M.
B., Barsanti, K. C., Carlton, A. G., and Yokelson, R.: The Fire Inventory from NCAR version 2.5: an updated global
fire emissions model for climate and chemistry applications, EGUsphere [preprint], https://doi.org/10.5194/egusphere-
2023-124, 2023.

Wik, M., Thornton, B. F., Bastviken, D., Uhlbäck, J. and Crill, P. M.: Biased sampling of methane release from northern
lakes: A problem for extrapolation, Geophys. Res. Lett., 43(3), 1256–1262, doi:10.1002/2015gl066501, 2016a.
Wik, M., Varner, R. K., Anthony, K. W., MacIntyre, S. and Bastviken, D.: Climate-sensitive northern lakes and ponds are
critical components of methane release, Nat. Geosci., 9(2), 99–105, doi:10.1038/ngeo2578, 2016b.
Wild, B., Teubner, I., Moesinger, L., Zotta, R.-M., Forkel, M., van der Schalie, R., Sitch, S., and Dorigo, W.:
VODCA2GPP – a new, global, long-term (1988–2020) gross primary production dataset from microwave remote
sensing, Earth Syst. Sci. Data, 14, 1063–1085, https://doi.org/10.5194/essd-14-1063-2022, 2022.
Winderlich, J., Chen, H., Gerbig, C., Seifert, T., Kolle, O., Lavrič, J. V., Kaiser, C., Höfer, A. and Heimann, M.:
Continuous low-maintenance $CO_2$/$CH_4$/$H_2O$ measurements at the Zotino Tall Tower Observatory (ZOTTO) in Central
Siberia, Atmospheric Meas. Tech., 3(4), 1113–1128, doi:10.5194/amt-3-1113-2010, 2010.
Wilson, C., Chipperfield, M. P., Gloor, M., Parker, R. J., Boesch, H., McNorton, J., Gatti, L. V., Miller, J. B., Basso, L.
S., and Monks, S. A.: Large and increasing methane emissions from eastern Amazonia derived from satellite data,





2010–2018, Atmos. Chem. Phys., 21, 10643–10669, https://doi.org/10.5194/acp-21-10643-2021, 2021.

Wood, T.G. and Sands, W.A. The role of termites in ecosystems. In: Brian, M.V. (Ed.), Production Ecology of Ants and
Termites. Cambridge University Press, Cambridge, UK, 245–292, 1978.

Woodward G, Perkins D.M., and Brown L. E.: Climate change and freshwater ecosystems: impacts across multiple levels
of organization, Philos Trans R Soc Lond B Biol Sci. ,365(1549), 2093-106, doi: 10.1098/rstb.2010.0055, 2010

Woodward, G., Gessner, M. O., Giller, P. S., Gulis, V., Hladyz, S., Lecerf, A., Malmqvist, B., McKie, B. G., Tiegs, S. D.,
Cariss, H., Dobson, M., Elosegi, A., Ferreira, V., Graça, M. A. S., Fleituch, T., Lacoursière, J. O., Nistorescu, M.,
Pozo, J., Risnoveanu, G., Schindler, M., Vadineanu, A., Vought, L. B.-M. and Chauvet, E.: Continental-Scale Effects
of    Nutrient    Pollution    on    Stream    Ecosystem    Functioning,    Science,    336(6087),    1438–1440,
doi:10.1126/science.1219534, 2012.

Woolway RI, Jones ID, Maberly SC, French JR, Livingstone DM, Monteith DT, et al.: Diel Surface Temperature Range
Scales with Lake Size, PLoS ONE 11(3): e0152466, doi:10.1371/journal.pone.0152466, 2016

Worden, J. R., Bloom, A. A., Pandey, S., Jiang, Z., Worden, H. M., Walker, T. W., Houweling, S. and Röckmann, T.:
Reduced biomass burning emissions reconcile conflicting estimates of the post-2006 atmospheric methane budget,
Nat. Commun., 8(1), 2227, doi:10.1038/s41467-017-02246-0, 2017.

Wu, Z., Li, J., Sun, Y. *et al.* : Imbalance of global nutrient cycles exacerbated by the greater retention of phosphorus over
nitrogen in lakes. *Nat. Geosci.* 15, 464–468, https://doi.org/10.1038/s41561-022-00958-7, 2022

Wuebbles, D. J. and Hayhoe, K.: Atmospheric methane and global change, Earth-Sci. Rev., 57(3–4), 177–210, 2002.
Wunch, D., Toon, G. C., Blavier, J.-F. L., Washenfelder, R. A., Notholt, J., Connor, B. J., Griffith, D. W. T., Sherlock, V.
and Wennberg, P. O.: The Total Carbon Column Observing Network, Philos. Trans. R. Soc. A, 369(1943),
doi:10.1098/rsta.2010.0240, 2011.

Wunch, D., Toon, G. C., Hedelius, J. K., Vizenor, N., Roehl, C. M., Saad, K. M., Blavier, J.-F. L., Blake, D. R. and
Wennberg, P. O.: Quantifying the loss of processed natural gas within California's South Coast Air Basin using long-
term measurements of ethane and methane, Atmospheric Chem. Phys., 16(22), 14091–14105, doi:10.5194/acp-16-
14091-2016, 2016.

Wunch, D., Jones, D. B. A., Toon, G. C., Deutscher, N. M., Hase, F., Notholt, J., Sussmann, R., Warneke, T., Kuenen, J.,
Denier van der Gon, H., Fisher, J. A. and Maasakkers, J. D.: Emissions of methane in Europe inferred by total column
measurements, Atmos Chem Phys, 19(6), 3963–3980, doi:10.5194/acp-19-3963-2019, 2019.

Xiao, K., F. Beulig, H. Røy, B. B. Jørgensen, and N. Risgaard-Petersen: Methylotrophic methanogenesis fuels cryptic
methane cycling in marine surface sediment. Limnol. Oceanogr. **63**: 1519–1527. doi:10.1002/lno.10788, 2018

Xu, X. F., Tian, H. Q., Zhang, C., Liu, M. L., Ren, W., Chen, G. S., Lu, C. Q. and Bruhwiler, L.: Attribution of spatial and
temporal variations in terrestrial methane flux over North America, Biogeosciences, 7(11), 3637–3655,
doi:10.5194/bg-7-3637-2010, 2010.



Xu, X., P Sharma, S Shu, TZ Lin, P Ciais, F Tubiello, P Smith, N Campbell and AK Jain (2021), Global Greenhouse Gas
Emissions from Plant- and Animal-Based Food, Nature Food, https://doi.org/10.1038/s43016-021-00358-x, 2021

Yacovitch, T. I. ,C. Daube, S. C. Herndon, and J. B. McManus: Isotopes on a Boat: Real-Time Spectroscopic Measurement
of Methane Isotopologues from Offshore Oil and Gas Emissions, OSA Optical Sensors and Sensing Congress 2021
(AIS, FTS, HISE, SENSORS, ES), S. Buckley, F. Vanier, S. Shi, K. Walker, I. Coddington, S. Paine, K. Lok Chan,
3855        W. Moses, S. Qian, P. Pellegrino, F. Vollmer, G. , J. Jágerská, R. Menzies, L. Emmenegger, and J. Westberg, eds.,
OSA Technical Digest (Optica Publishing Group, 2021), paper EW5D.3, 2021

Yan, X., Akiyama, H., Yagi, K. and Akimoto, H.: Global estimations of the inventory and mitigation potential of methane
emissions from rice cultivation conducted using the 2006 Intergovernmental Panel on Climate Change Guidelines,
Glob. Biogeochem. Cycles, 23(2), doi:10.1029/2008gb003299, 2009.

Yang, P., D. Y. F. Lai, H. Yang, and others:  Large increase in CH4 emission following conversion of coastal marsh to
aquaculture ponds caused by changing gas transport pathways. Water Res. **222**: 118882.
doi:10.1016/j.watres.2022.118882, 2022

Yin, Y., Chevallier, F., Ciais, P., Broquet, G., Fortems-Cheiney, A., Pison, I. and Saunois, M.: Decadal trends in global
CO emissions as seen by MOPITT, Atmospheric Chem. Phys., 15(23), 13433–13451, doi:10.5194/acp-15-13433-
2015, 2015.

Yoshida, Y., Kikuchi, N., Morino, I., Uchino, O., Oshchepkov, S., Bril, A., Saeki, T., Schutgens, N., Toon, G. C., Wunch,
D., Roehl, C. M., Wennberg, P. O., Griffith, D. W. T., Deutscher, N. M., Warneke, T., Notholt, J., Robinson, J.,
Sherlock, V., Connor, B., Rettinger, M., Sussmann, R., Ahonen, P., Heikkinen, P., Kyrö, E., Mendonca, J., Strong, K.,
Hase, F., Dohe, S. and Yokota, T.: Improvement of the retrieval algorithm for GOSAT SWIR XCO2 and XCH4 and
their validation using TCCON data, Atmospheric Meas. Tech., 6(6), 1533–1547, doi:10.5194/amt-6-1533-2013, 2013.

Yuan, J., J. Xiang, D. Liu, and others:  Rapid growth in greenhouse gas emissions from the adoption of industrial-scale
aquaculture. Nat. Clim. Chang. **9**: 318–322. doi:10.1038/s41558-019-0425-9, 2019

Yver Kwok, C. E., Müller, D., Caldow, C., Lebègue, B., Mønster, J. G., Rella, C. W., Scheutz, C., Schmidt, M., Ramonet,
3874        M., Warneke, T., Broquet, G. and Ciais, P.: Methane emission estimates using chamber and tracer release experiments
for a municipal waste water treatment plant, Atmospheric Meas. Tech., 8(7), 2853–2867, doi:10.5194/amt-8-2853-
2015, 2015.

Zavala-Araiza, D., Lyon, D. R., Alvarez, R. A., Davis, K. J., Harriss, R., Herndon, S. C., Karion, A., Kort, E. A., Lamb,
B. K., Lan, X., Marchese, A. J., Pacala, S. W., Robinson, A. L., Shepson, P. B., Sweeney, C., Talbot, R., Townsend-
Small, A., Yacovitch, T. I., Zimmerle, D. J. and Hamburg, S. P.: Reconciling divergent estimates of oil and gas methane
emissions, Proc. Natl. Acad. Sci. USA, 112, 15597–15602, doi:10.1073/pnas.1522126112, 2015.

Zhang: Magnitude, spatio-temporal variability and environmental controls of methane emissions from global rice fields:
Implications for water management and climate mitigation, Glob. Change Biol., 2016.



Zhang, B. and Chen, G. Q.: China's $CH_4$ and $CO_2$ Emissions: Bottom-Up Estimation and Comparative Analysis, Ecol.
Indic., 47, 112–122, doi:10.1016/j.ecolind.2014.01.022, 2014.

Zhang, L., X. Xia, S. Liu, S. Zhang, S. Li, J. Wang, G. Wang, H. Gao, Z. Zhang, Q. Wang, W. Wen, R. Liu, Z. Yang, E.H.
Stanley, and P.A. Raymond: Significant methane ebullition from alpine permafrost rivers on the East Qinghai–Tibet
Plateau. Nature Geoscience, 13(5): p. 349-354, 2020

Zhang, L., H. Tian, H. Shi, S. Pan, J. Chang, S. R. S. Dangal, X. Qin, S. Wang, F. N. Tubiello, J. G. Canadell, R. B.
Jackson: A 130-year global inventory of methane emissions from livestock: Trends, patterns, and drivers, Global
Change Biology, 28 (17), 5142-5158. https://doi.org/10.1111/gcb.16280, 2022

Zhang, Y., Xiao, X., Wu, X., Zhou, S., Zhang, G., Qin, Y., and Dong, J.: A global moderate resolution dataset of gross
primary production of vegetation for 2000–2016, Sci. Data, 4, 1–13, https://doi.org/10.1038/sdata.2017.165, 2017.

Zhang, Y., Jacob, D. J., Maasakkers, J. D., Sulprizio, M. P., Sheng, J.-X., Gautam, R., and Worden, J.: Monitoring global
tropospheric OH concentrations using satellite observations of atmospheric methane, Atmos. Chem. Phys., 18, 15959–
15973, https://doi.org/10.5194/acp-18-15959-2018, 2018.

Zhang, Z., Zimmermann, N. E., Kaplan, J. O. and Poulter, B.: Modeling spatiotemporal dynamics of global wetlands:
comprehensive evaluation of a new sub-grid TOPMODEL parameterization and uncertainties, Biogeosciences, 13(5),
1387–1408, doi:10.5194/bg-13-1387-2016, 2016.

Zhang, Z., Fluet-Chouinard, E., Jensen, K., McDonald, K., Hugelius, G., Gumbricht, T., et al. Development of the global
dataset of Wetland Area and Dynamics for Methane Modeling (WAD2M). Earth System Science Data, 13(5), 2001–
2023. https://doi.org/10.5194/essd-13-2001-2021, 2021.

Zhang, Z., Poulter, B., Feldman, A.F., Ying, Q, Ciais, P., Peng, S. and Li, X.: Recent intensification of wetland methane
feedback, Nat. Clim. Chang, 13, 430–433, https://doi.org/10.1038/s41558-023-01629-0, 2023

Zhang, Z., Poulter, B., Melton, J., Riley, W., Allen, G., Beerling D., Bousquet P., Canadell, J., Fluet-Chouinard E., Ciais,
P., Gedney, N., Hopcroft, P., Ito, A., Jackson, R., Jain, A., Jensen, K., Joos, F., Kleinen, T., Knox, S., Li., T., Li, X.,
Liu, X., McDonald, K., McNicol, G., Miller, P., Müller, J., Patra, P., Prigent, C., Peng, C., Peng, S., Qin, Z., Riggs,
R., Saunois, M., Sun Q., Tian, H., Xu, X., Yao Y., Yi, X., Zhang, W., Zhu, Q., Zhu, Q., and Zhuang, Q.: Ensemble
estimates of global wetland methane emissions over 2000-2020. Global Change Biology, in review.

Zhao, J., M. Zhang, W. Xiao, L. Jia, X. Zhang, J. Wang, Z. Zhang, Y. Xie, Y. Pu, S. Liu, Z. Feng, and X. Lee: Large
methane emission from freshwater aquaculture ponds revealed by long-term eddy covariance observation. Agricultural
and Forest Meteorology, 308-309: p. 108600, 2021

Zhao, Y., Saunois, M., Bousquet, P., Lin, X., Berchet, A., Hegglin, M. I., Canadell, J. G., Jackson, R. B., Hauglustaine,
D. A., Szopa, S., Stavert, A. R., Abraham, N. L., Archibald, A. T., Bekki, S., Deushi, M., Jöckel, P., Josse, B., Kinnison,
D., Kirner, O., Marécal, V., O'Connor, F. M., Plummer, D. A., Revell, L. E., Rozanov, E., Stenke, A., Strode, S.,
Tilmes, S., Dlugokencky, E. J. and Zheng, B.: Inter-model comparison of global hydroxyl radical (OH) distributions



and their impact on atmospheric methane over the 2000–2016 period, Atmospheric Chem. Phys., 19(21), 13701–13723, doi:10.5194/acp-19-13701-2019, 2019.

Zhao, Y., Saunois, M., Bousquet, P., Lin, X., Berchet, A., Hegglin, M. I., Canadell, J. G., Jackson, R. B., Dlugokencky, E. J., Langenfelds, R. L., Ramonet, M., Worthy, D. and Zheng, B.: Influences of hydroxyl radicals (OH) on top-down estimates of the global and regional methane budgets, Atmospheric Chem. Phys. Discuss., 1–45, doi:10.5194/acp-2019-1208, 2020.

Zhao, Y., Saunois, M., Bousquet, P., Lin, X., Hegglin, M. I., Canadell, J. G., Jackson, R. B., and Zheng, B.: Reconciling the bottom-up and top-down estimates of the methane chemical sink using multiple observations, Atmos. Chem. Phys., 23, 789–807, https://doi.org/10.5194/acp-23-789-2023, 2023.

Zheng, B., Chevallier, F., Ciais, P., Yin, Y. and Wang, Y.: On the Role of the Flaming to Smoldering Transition in the Seasonal Cycle of African Fire Emissions, Geophys. Res. Lett., 45(21), 11,998-12,007, doi:10.1029/2018GL079092, 2018a.

Zheng, B., Chevallier, F., Ciais, P., Yin, Y., Deeter, M. N., Worden, H. M., Wang, Y., Zhang, Q. and He, K.: Rapid decline in carbon monoxide emissions and export from East Asia between years 2005 and 2016, Environ. Res. Lett., 13(4), 044007, doi:10.1088/1748-9326/aab2b3, 2018b.

Zheng, B., Chevallier, F., Yin, Y., Ciais, P., Fortems-Cheiney, A., Deeter, M. N., Parker, R. J., Wang, Y., Worden, H. M., and Zhao, Y.: Global atmospheric carbon monoxide budget 2000–2017 inferred from multi-species atmospheric inversions, Earth Syst. Sci. Data, 11, 1411–1436, doi: 10.5194/essd-11-1411-2019, 2019.

Zheng, B., Ciais, P., Chevallier, F., Yang, H., Canadell, J. G., Chen, Y., van der Velde, I. R., Aben, I., Chuvieco, E., Davis, S. J., Deeter, M., Hong, C., Kong, Y., Li, H., Li, H., Lin, X., He, K., and Zhang, Q.: Record-high CO2 emissions from boreal fires in 2021, Science, 379, 912-917,doi: 10.1126/science.ade0805, 2023.

Zhu, Q., Liu, J., Peng, C., Chen, H., Fang, X., Jiang, H., Yang, G., Zhu, D., Wang, W. and Zhou, X.: Modelling methane emissions from natural wetlands by development and application of the TRIPLEX-GHG model, Geosci. Model Dev., 7(3), 981–999, doi:10.5194/gmd-7-981-2014, 2014.

Zhu, Q., Peng, C., Chen, H., Fang, X., Liu, J., Jiang, H., Yang, Y. and Yang, G.: Estimating global natural wetland methane emissions using process modelling: spatio-temporal patterns and contributions to atmospheric methane fluctuations, Glob. Ecol. Biogeogr., 24, 959–972, 2015.

Zhu, Y., K.J. Purdy, Ö. Eyice, L. Shen, S.F. Harpenslager, G. Yvon-Durocher, A.J. Dumbrell, and M. Trimmer: Disproportionate increase in freshwater methane emissions induced by experimental warming. Nature Climate Change, 10(7): p. 685-690., 2020

Zhuang, Q., Melillo, J. M., Kicklighter, D. W., Prinn, R. G., McGuire, A. D., Steudler, P. A., Felzer, B. S. and Hu, S.: Methane fluxes between terrestrial ecosystems and the atmosphere at northern high latitudes during the past century: A retrospective analysis with a process-based biogeochemistry model, Glob. Biogeochem Cycles, 18(3), GB3010,

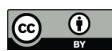



doi:10.1029/2004gb002239, 2004.

Zhuang, Q., Chen, M., Xu, K., Tang, J., Saikawa, E., Lu, Y., Melillo, J. M., Prinn, R. G. and McGuire, A. D.: Response

of global soil consumption of atmospheric methane to changes in atmospheric climate and nitrogen deposition, Glob.

Biogeochem. Cycles, 27(3), 650–663, doi:10.1002/gbc.20057, 2013.

Zhuang, Q., M. Guo, J.M. Melack, X. Lan, Z. Tan, Y. Oh, and L.R. Leung: Current and Future Global Lake Methane

Emissions: A Process-Based Modeling Analysis. Journal of Geophysical Research: Biogeosciences, 128(3): p.

e2022JG007137, 2023



**Table 1: Bottom-up (BU) models and inventories for anthropogenic and biomass burning used in this study. *Due to its limited sectoral breakdown this dataset was not used in Table 3.**

| B-U models and inventories | Contribution | Time period (resolution) | Gridded | References |
|---|---|---|---|---|
| CEDS (country based) | Fossil fuels, Agriculture and waste, Biofuel | 1970-2019 (yearly) | no | Hoesly et al. (2018) |
| CEDS (gridded)* | Fossil fuels, Agriculture and waste, Biofuel | 1970-2020 (monthly) | 0.5x0.5° | Hoesly et al. (2018) O'Rourke et al (2021) |
| EDGARv6 | Fossil fuels, Agriculture and waste, Biofuel | 1990-2018^ (yearly, monthly for some sectors) | 0.1x0.1° | Oreggioni et al. (2021), Crippa et al. (2021) |
| EDGARv7 | Fossil fuels, Agriculture and waste, Biofuel | 1990-2021 (yearly) | 0.1x0.1° | Crippa et al. (2023) |
| IIASA GAINS v4.0 | Fossil fuels, Agriculture and waste, Biofuel | 1990-2020 (yearly) | 0.5x0.5° | Höglund-Isaksson et al., (2020) |
| USEPA | Fossil fuels, Agriculture and waste, Biofuel, Biomass Burning | 1990-2030 (10-yr interval, interpolated to yearly) | no | USEPA (2019) |
| FAO-CH4 | Agriculture, Biomass Burning | 1961-2020 1990-2020 (Yearly) | no | Federici et al. (2015) ; Tubiello et al. (2013); Tubiello (2019) |
| FINNv2.5 | Biomass burning | 2002-2020 (daily) | 1km resolution | Wiedinmyer et al. (2023) |
| GFASv1.3 | Biomass burning | 2003-2020 (daily) | 0.1x0.1° | Kaiser et al. (2012) |
| GFEDv4.1s | Biomass burning | 1997-2020 (monthly) | 0.25x0.25° | Giglio et al. (2013); van der Werf et al (2017) |
| QFEDv2.5 | Biomass burning | 2000-2020 (daily) | 0.1x0.1° | Darmenov and da Silva (2015) |





**Table 2: Biogeochemical models that computed wetland emissions used in this study. Model runs were performed with two climate inputs, CRU and GSWP3-W5E5. Models were run with prognostic (using their own calculation of wetland areas) and/or diagnostic (using WAD2M (Zhang et al., 2021b)) wetland surface areas (see Sect 3.2.1).**

| Model | Institution | Prognostic | | Diagnostic | | References |
|---|---|---|---|---|---|---|
| | | CRU | GSWP3-W5E5 | CRU | GSWP3-W5E5 | |
| CH4MOD$_{wetland}$ | Institute of Atmospheric Physics, CAS | n | n | **y** | **y** | Li et al. (2010) |
| CLASSIC | Environment and Climate Change Canada | y | y* | y | y* | Arora et al. (2018); Melton and Arora (2016) |
| DLEM | Boston College | y | y | y | y | Tian et al. (2015, 2023) |
| ELM-ECA | Lawrence Berkeley National Laboratory | y | y | y | y | Riley et al. (2011) |
| ISAM | University of Illinois, Urbana-Champaign | y | y | y | y | Shu et al. (2020) Xu et al. (2021) |
| JSBACH | MPI | y | y | y | y | Kleinen et al. (2020, 2021, 2023) |
| JULES | UKMO | y | y | y | y | Gedney et al. (2019) |
| LPJ-GUESS | Lund University | n | n | y | y | McGuire et al. (2012) |
| LPJ-MPI | MPI | y | y | y | y | Kleinen et al. (2012) |
| LPJ-WSL | NASA GSFC | y | y | y | y | Zhang et al. (2016) |
| LPX-Bern | University of Bern | y | y | y | y | Spahni et al. (2011), Stocker et al. (2014) |
| ORCHIDEE | LSCE | y | y | y | y | Ringeval et al. (2011) |





| SDGVM | University of Birmingham/ University of Sheffield | y | y | y | y | Beerling & Woodward (2001), Hopcroft et al. (2011, 2020) |
|---|---|---|---|---|---|---|
| TEM-MDM | Purdue University | n | n | y | y | Zhuang et al. (2004) |
| TRIPLEX-GHG | UQAM | n | n | y | y | Zhu et al. (2014, 2015) |
| VISIT | NIES | y | y | y | y | Ito and Inatomi (2012) |

**\*CLASSIC uses GSWP3-W5E version 2 that covers the time period till 2016. All other models use GSWP-W5E5 version 3.**





**Table 3:** Global methane emissions by source type in Tg CH$_4$ yr$^{-1}$ from Saunois et al. (2020) (left column pair) and from this work using bottom-up and top-down approaches. Because top-down models cannot fully separate individual processes, only five categories of emissions are provided (see text). Uncertainties are reported as [min-max] range of reported studies. The mean, minimum and maximum values are calculated while discarding outliers, for each category of source and sink. As a result, discrepancies may occur when comparing the sum of categories and their corresponding total due to differences in outlier detections. Differences of 1 Tg CH$_4$ yr$^{-1}$ in the totals can also occur due to rounding errors. Compared to Saunois et al. (2020), emissions are split between "direct anthropogenic" emissions and "natural and indirect anthropogenic" sources. We also propose an estimate of the double-counting between bottom-up wetland and inland freshwater ecosystems emissions.

| | Saunois et al. (2020) | | This work | | | | | |
|---|---|---|---|---|---|---|---|---|
| **Period of time** | **2000-2009** | | **2000-2009** | | **2010-2019** | | **2020** | |
| **Approaches** | bottom-up | top-down | bottom-up | top-down | bottom-up | top-down | bottom-up | top-down |
| **NATURAL & indirect anthropogenic SOURCES** | | | | | | | | |
| **Combined wetlands and inland freshwaters** | **306** [229-391] | **180** [153-196] | **242** [156-355] | **158** [145-172] | **248** [159-369] | **165** [145-214] | **251** [171-364] | **175** [151-229] |
| **Wetlands** | **147** [102-179] | **180** [153-196] | **153** [116-189] (***) | **158** [145-172] | **159** [119-203] (***) | **165** [145-214] | **161** [131-198] (***) | **175** [151-229] |
| **Inland freshwaters** [a] | **159** [117-212] | | **112** [49-202] | | **112** [49-202] | | **112** [49-202] | |
| **Double counting** [b] | NA | | **-23** [-9 - -36] | | **-23** [-9 - -36] | | **-23** [-9 - -36] | |
| **Other natural sources** | **63** [26-94] | **35** [21-47] | **63** [24-93] | **44** [40-46] | **63** [24-93] | **43** [40-46] | **63** [24-93] | **44** [40-47] |
| **Land sources** | **50** [17-72] | | **51** [18-73] | | | | | |
| Geological (onshore) | 38 [13-53] | | 38 [13-53] | | | | | |
| Wild animals | 2 [1-3] | | 2 [1-3] | | | | | |
| Termites | 9 [3-15] | | 10 [4-16] | | | | | |
| Wildfires | (**) | | (**) | | | | | |
| Permafrost soils (direct) | 1 [0-1] | | 1 [0-1] | | | | | |
| Vegetation | (*) | | (*) | | | | | |
| **Coastal and Oceanic sources**[c] | **13** [9-22] | | **12** [6-20] | | | | | |
| Biogenic | 6 [4-10] | | 5 [3-10] | | | | | |
| Geological (offshore) | 7 [5-12] | | 7 [5-12] | | | | | |
| **TOTAL NATURAL & INDIRECT SOURCES** | **369** [245-485] | **215** [176-243] | **305** [180-448] | **204** [189-223] | **311** [183-462] | **206** [188-225] | **314** [195-457] | **216** [193-241] |
| **DIRECT ANTHROPOGENIC SOURCES** | | | | | | | | |
| **Agriculture and waste** | **192** [178-206] | **202** [198-219] | **194** [181-208] | **210** [197-223] | **211** [195-231] | **228** [213-242] | **211** [204-216] | **245** [232-259] |
| **Agriculture** | **132** [NA] | | **134** [125-142] | | **143** [132-155] | | **147** [143-149] | |
| Enteric ferm. & manure | 104 [93-109] | | 104 [100-110] | | 112 [107-118] | | 117 [114-124] | |
| Rice cultivation | 28 [23-34] | | 30 [24-34] | | 32 [25-37] | | 32 [29-37] | |
| **Landfills and waste** | **60** [55-63] | | **61** [52-71] | | **69** [56-80] | | **71** [60-84] | |
| **Fossil fuels** | **110** [94-129] | **101** [71-151] | **105** [97-123] | **105** [88-115] | **120** [117-125] | **115** [100-124] | **128** [120-133] | **122** [101-133] |
| Coal mining | 32 [24-42] | | (****) | | (****) | | (****) | |



|  | Saunois et al. (2020) | | This work | | | | | |
|---|---|---|---|---|---|---|---|---|
| **Period of time** | **2000-2009** | | **2000-2009** | | **2010-2019** | | **2020** | |
| Oil & Gas<br>Industry<br>Transport | 73 [60-85]<br>2 [0-6]<br>4 [1-11] |  | 30 [26-32]<br>65 [63-71]<br>4 [1-8]<br>3 [1-8] |  | 40 [37-44]<br>67 [57-74]<br>5 [1-9]<br>2 [1-3] |  | 41 [38-43]<br>74 [67-80]<br>5 [1-8]<br>2 [1-3] |  |
| **Biomass & biof. burn.** | **31** [26-46] | **29** [23-35] | **30** [22-44] | **26** [22-29] | **28** [21-39] | **27** [26-27] | **27** [20-41] | **26** [22-27] |
| Biomass burning<br>Biofuel burning | 19 [15-32]<br>12 [9-14] |  | 19 [14-29]<br>11 [8-14] |  | 17 [12-24]<br>11 [8-14] |  | 17 [13-27]<br>10 [7-14] |  |
| **TOTAL DIRECT ANTHROPOGENIC SOURCES** | **334** [d] **[321-358]** | **332** **[312-347]** | **333** [d] **[305-365]** | **341** **[319-355]** | **358** [d] **[329-387]** | **369** **[350-391]** | **372** [d] **[345-409]** | **392** **[368-409]** |
| <td colspan="9" align="center">**SINKS**</td> | | | | | | | | |
| **Total chemical loss** | **595** [489-749] | **505** [459-516] | **585** [481-716] | **504**[e] [496-511] | **602** [496-747] | **521**[e] [485-532] | 602 [496-747] | **538**[e] [503-554] |
| Tropospheric OH | 553 [476-677] |  | 546 [446-663] |  | 563 [462-663] |  | 563 [462-663] |  |
| Stratospheric loss<br>Tropospheric Cl | 31 [12-37]<br>11 [1-35] |  | 34 [10-51]<br>6 [1-13] |  | 35 [10-51]<br>6 [1-13] |  | 35 [10-51]<br>6 [1-13] |  |
| **Soil uptake** | **30** [11-49] | **34** [27-41] | **30** [11-49] | **34** [34-34] | **31** [11-49] | **35** [35-35] | **31** [11-49] | **36** [35-36] |
| **TOTAL SINKS** | **625** [500-798] | **540** [486-556] | **615** [492-765] | 538 [530-545][e] | **633** [507-796] | **554 [520-567]**[e] | 633 [507-796] | **575 [566-589]**[e] |
| <td colspan="9" align="center">**SOURCES – SINKS IMBALANCE**</td> | | | | | | | | |
| **TOTAL SOURCES** | **703** [566-842] | **547** [524-560] | **638** [485-813] | **543** [526-558] | **669** [512-849] | **575** [553-586] | **685** [540-865] | **608** [581-627] |
| **TOTAL SINKS** | **625** [500-798] | **540** [486-556] | **615** [492-765] | **538** [530-545][e] | **633** [507-796] | **554** [550-567][e] | **633** [507-796] | **575 [566-589]**[e] |
| **IMBALANCE** | **78** | **3** [-10-38] | **23** | **5** [-4-13][e] | 36 | **21** [19-33][e] | 52 | **32** [15-38][e] |
| **ATMOSPHERIC GROWTH** [f] |  | **5.8** [4.9-6.6][f] |  | **6.1** [5.2-6.9][f] |  | **20.9** [20.1-21.7][f] |  | **41.8** [40.7-42.9][f] |

(*) uncertain but likely small for upland forest and aerobic emissions, potentially large for forested wetland, but likely included elsewhere

(**) We stop reporting this value to avoid potential double counting with satellite-based products of biomass burning (see Sect. 3.1.5)

(***) Here the numbers are from prognostic runs. To ensure a fair comparison with previous budgets (Saunois et al., 2020), the numbers are 163[117-195] for 2000-2009 from diagnostic runs with CRU/CRU-JRA-55 climate inputs (see Sect. 3.2.1).

(****) Up to 8 Tg of additional emissions could account for ultra emitters (Lauvaux et al., 2022), as in Tibrewal et al. (2024), that are fully or partly missed in regular anthropogenic inventories

a: Freshwater includes lakes, ponds, reservoirs, streams and rivers, part of it is due to anthropogenic disturbances estimated in Sect.3.2.2

b: The double counting estimate is discussed in Sect. 3.2.2

c: includes flux from hydrates considered at 0 for this study, includes estuaries

d: Total anthropogenic emissions are based on estimates of full anthropogenic inventory and not on the sum of "Agriculture and Waste", "Fossil fuels" and "Biofuel and biomass burning" categories (see Sect. 3.1.2)

e: Some inversions did not provide the chemical sink. These values are derived from a subset of the inversion ensemble.

f: Atmospheric growth rates are given in the same unit Tg $CH_4$ yr$^{-1}$, based on the conversion factor of 2.75 Tg $CH_4$ ppb$^{-1}$ given by Prather et al. (2012) and the atmospheric growth rates provided in the text in ppb yr$^{-1}$.



**Table 4: Top-down studies used here with their contribution to the decadal and yearly estimates noted. For decadal means, top down studies must provide at least 8 years of data over the decade to contribute to the estimate. Details on each inverse system and inversions are provided in Table S8 to S11 in the Supplementary Material.**

| Model | Institution | Observation used | Time period | Number of inversions | 2000-2009 | 2010-2019 | 2020 | References |
|---|---|---|---|---|---|---|---|---|
| Carbon Tracker-Europe CH$_4$ | FMI | Surface stations | 2000-2020 | 4 | y | y | y | Tsuruta et al. (2017) |
| LMDz-CIF | LSCE/CEA | Surface stations | 2000-2020 | 4 | y | y | y | Thanwerdas et al. (2022a) |
| LMDz-PYVAR | LSCE/CEA/THU | GOSAT Leicester v9.0 | 2010-2020 | 4 | n | y | y | Zheng et al. (2018a, 2018b, 2019) |
| MIROC4-ACTM | JAMSTEC | Surface stations | 2000-2020 | 5 | y | y | y | Patra et al. (2018); Chandra et al. (2021) |
| NISMON-CH$_4$ | NIES/MRI | Surface stations | 2000-2020 | 2 | y | y | y | Niwa et al. (2022) |
| NIES-TM-FLEXPART (NTFVAR) | NIES | Surface stations | 2000-2020 | 2 | y | y | y | Maksyutov et al. (2020); Wang et al. (2019a) |
| NIES-TM-FLEXPART (NTFVAR) | NIES | GOSAT NIES L2 v02.95 | 2010-2020 | 1 | n | y | y | Maksyutov et al. (2020); Wang et al. (2019a) |
| TM5-CAMS | TNO/VU | Surface stations | 2000-2020 | 1 | y | y | y | Segers et al. (2022) |
| TM5-CAMS | TNO/VU | GOSAT ESA/CCI v2.3.8 (combined with surface observations) | 2010-2020 | 1 | n | y | y | Segers et al. (2022) |
| Total number of runs | | | | 24 | 18 | 24 | 24 | |





**Table 5: Global and latitudinal total methane emissions in Tg CH$_4$ yr$^{-1}$, as decadal means (2000-2009 and 2010-2019) and for the year 2020 from this work using bottom-up and top-down approaches. Global and latitudinal emissions for 2000-2009 are also compared with Saunois et al. (2016, 2020) for top-down and bottom-up approaches when available. Uncertainties are reported as [min-max] range. The mean, minimum and maximum values are calculated while discarding outliers, for each category of source and sink. As a result, discrepancies may occur when comparing the sum of categories and their corresponding total due to differences in outlier detections. Differences of 1 Tg CH$_4$ yr$^{-1}$ in the totals can also occur due to rounding errors. For the latitudinal breakdown, bottom-up anthropogenic estimates are based only on the gridded products (see Table 1). As a result, the total from the latitudinal breakdown (line called "This work (gridded BU products only") is slightly different from the values provided in Table 3 and recalled in the line "This work (all BU products)".**

| Period | 2000-2009 | | 2010-2019 | | 2020 | |
|---|---|---|---|---|---|---|
| Approach | Bottom-up | Top-down | Bottom-up | Top-down | Bottom-up | Top-down |
| **Global** | | | | | | |
| This work (all BU products) | **638 [485-813]** | **543 [526-558]** | **669 [512-849]** | **575 [553-586]** | **685 [540-865]** | **608 [581-627]** |
| This work (gridded BU products only) | **642 [501-809]** | | **676 [526-845]** | | **691 [565-862]** | |
| *S2020* | *703 [566-842]* | *547 [524-560]* | - | - | - | - |
| *S2016* | *719[583-861]* | *552[535-566]* | - | - | - | - |
| **90°S-30°N** | | | | | | |
| This work | **367 [254-487]** | **337** [311-361] | **388 [275-503]** | **364** [337-390] | **395 [292-521]** | **386** [353-425] |
| *S2020* | *408 [322-532]* | *346 [320-379]* | - | - | - | - |
| *S2016* | - | 356 [334-381] | - | - | - | - |
| **30°N-60°N** | | | | | | |
| This work | **234 [169-335]** | **182** [162-197] | **250 [184-345]** | **187** [160-204] | **256 [186-356]** | **197** [170-215] |
| *S2020* | *252 [202-342]* | *178 [159-199]* | - | - | - | - |
| *S2016* | - | *176[159-195]* | - | - | - | - |
| **60°N-90°N** | | | | | | |
| This work | **42 [22-79]** | **26** [22-33] | **38[17-73]** | **24** [18-29] | **39 [17-74]** | **25** [20-32] |
| *S2020* | *42 [28-70]* | *23 [17- 32]* | - | - | - | - |
| *S2016* | - | *20 [15-25]* | - | - | - | - |



**Table 6: Latitudinal methane emissions in Tg CH$_4$ yr$^{-1}$ for the last decade 2010-2019, based on top-down and bottom-up approaches. Uncertainties are reported as [min-max] range of reported studies. The mean, minimum, and maximum values are calculated while discarding outliers, for each category of source and sink. As a result, discrepancies may occur when comparing the sum of categories and their corresponding total due to differences in outlier detections. Differences of 1 Tg CH$_4$ yr$^{-1}$ in the totals can also occur due to rounding errors. For bottom-up approaches, natural and indirect anthropogenic sources are estimated based on available gridded data sets (see text Sect 5.2). As some emissions are missing gridded products (wild animals, permafrost, and hydrates), discrepancies may occur in terms of totals proposed in Table 3. Bottom-up direct anthropogenic estimates are based only on the gridded products (see Table 1).**

| Latitudinal band | 90°S- 30°N | | 30°N-60°N | | 60°-90°N | |
|---|---|---|---|---|---|---|
| **Approach** | Bottom-up | Top-Down | Bottom-up | Top-Down | Bottom-up | Top-Down |
| **Natural and indirect anthropogenic Sources** | **178** [95-276] | **148** [133-164] | **100** [43-188] | **42** [36-50] | **28** [9-53] | **14** [10-21] |
| Combined wetland and Inland freshwaters | 151 [85-234] | 128 [112-155] | 73 [32-147] | 27 [20-42] | 24 [9-53] | 9 [7-17] |
| Other natural | 27 [11-42] | 22 [20-29] | 27 [10-41] | 19 [16-22] | 4 [2-6] | 3 [1-5] |
| **Anthropogenic direct sources** | **210** [180-227] | **215** [191-238] | **151** [142-157] | **144** [121-162] | **10** [6-14] | **10** [6-16] |
| Agriculture & Waste | 140 [121-150] | 150 [135-168] | 81 [77-84] | 77 [56-88] | 1 [1-2] | 2 [2-2] |
| Fossil Fuels | 52 [44-65] | 46 [36-62] | 65 [61-71] | 61 [50-69] | 7 [4-10] | 7 [3-13] |
| Biomass & biofuel burning | 22 [18-30] | 19 [16-21] | 7 [4-10] | 6 [2-7] | 1 [0-1] | 1 [1-2] |
| **Sum of sources** | **388** [275-503] | **364** [337-390] | **250** [184-345] | **187** [160-204] | **38** [7-73] | **24** [18- 29] |



Table 7: Regional methane emissions (regions ranked by continent) in Tg CH₄ yr⁻¹ for the last decade 2010-2019, based on top-down and bottom-up approaches. Uncertainties are reported as [min-max] range of reported studies. Differences of 1 Tg CH₄ yr⁻¹ in the totals can occur due to rounding errors. For bottom-up approaches, natural and indirect anthropogenic sources are estimated based on available gridded data sets (see text Sect 5.2). As some emissions are missing gridded products (wild animals, permafrost, and hydrates), discrepancies may occur in terms of totals proposed in Table 3. Bottom-up direct anthropogenic estimates are based on all products (gridded and per country).

| Region | Total emissions | | Natural and indirect anthropogenic emissions | | Direct anthropogenic emissions | |
|---|---|---|---|---|---|---|
| | Bottom-up | Top-down | Bottom-up | Top-down | Bottom-up | Top-down |
| USA | 49 [27-77] | 38 [32-46] | 24 [7-43] | 12 [7-22] | 26 [19-34] | 25 [16-31] |
| Canada | 38 [14-71] | 20 [17-24] | 32 [11-63] | 14 [11-22] | 6 [3-8] | 7[5-9] |
| Central America | 18 [10-28] | 17 [14-19] | 8 [3-17] | 5 [2-6] | 10 [8-12] | 12 [11-13] |
| Northern South America | 19 [9-35] | 16 [13-20] | 10 [3-17] | 9 [7-11] | 9 [6-17] | 7 [6-8] |
| Brazil | 51 [26-79] | 47 [41-58] | 32 [11-57] | 26 [22-36] | 19 [16-22] | 21 [17-26] |
| Southwest South America | 34 [16-51] | 38 [30-48] | 21 [6-35] | 24 [16-34] | 13 [10-16] | 14 [12-17] |
| Europe | 42 [29-57] | 31 [24-36] | 17 [6-30] | 7 [5-9] | 25 [22-27] | 24 [20-31] |
| Northern Africa | 24 [18-33] | 25 [23-29] | 7 [2-13] | 6 [6-8] | 18 [16-20] | 19 [17-21] |
| Equatorial Africa | 47 [28-83] | 47 [39-59] | 23 [10-49] | 24 [20-30] | 24 [19-34] | 23 [19-29] |
| Southern Africa | 21 [5-43] | 19 [16-24] | 11 [2-29] | 8 [7-10] | 10 [3-14] | 11 [10-12] |
| Russia | 48 [24-83] | 36 [27-45] | 25 [9-47] | 14 [11-18] | 23 [15-36] | 21 [14-29] |
| Central Asia | 15 [6-29] | 10 [8-13] | 8 [2-19] | 1 [0-2] | 8 [4-10] | 9 [7-11] |
| Middle East | 35 [21-47] | 31 [24-39] | 9 [3-15] | 4 [1-6] | 26 [18-31] | 28 [20-34] |
| China | 71 [55-99] | 57 [37-72] | 15 [4-33] | 4 [3-7] | 57 [51-66] | 53 [34-66] |
| Korean-Japan | 6 [4-12] | 5 [4-6] | 3 [1-7] | 1 [1-1] | 4 [3-5] | 4 [3-5] |
| South Asia | 58 [49-72] | 52 [43-60] | 13 [5-25] | 6 [5-6] | 45 [44-47] | 45[37-49] |
| Southeast Asia | 64 [42-93] | 63 [52-71] | 32 [19-54] | 27 [20-34] | 32 [23-39] | 35 [31-46] |
| Australasia | 16 [9-26] | 13 [10-17] | 10 [4-19] | 6 [4-7] | 7 [6-7] | 7 [6-7] |

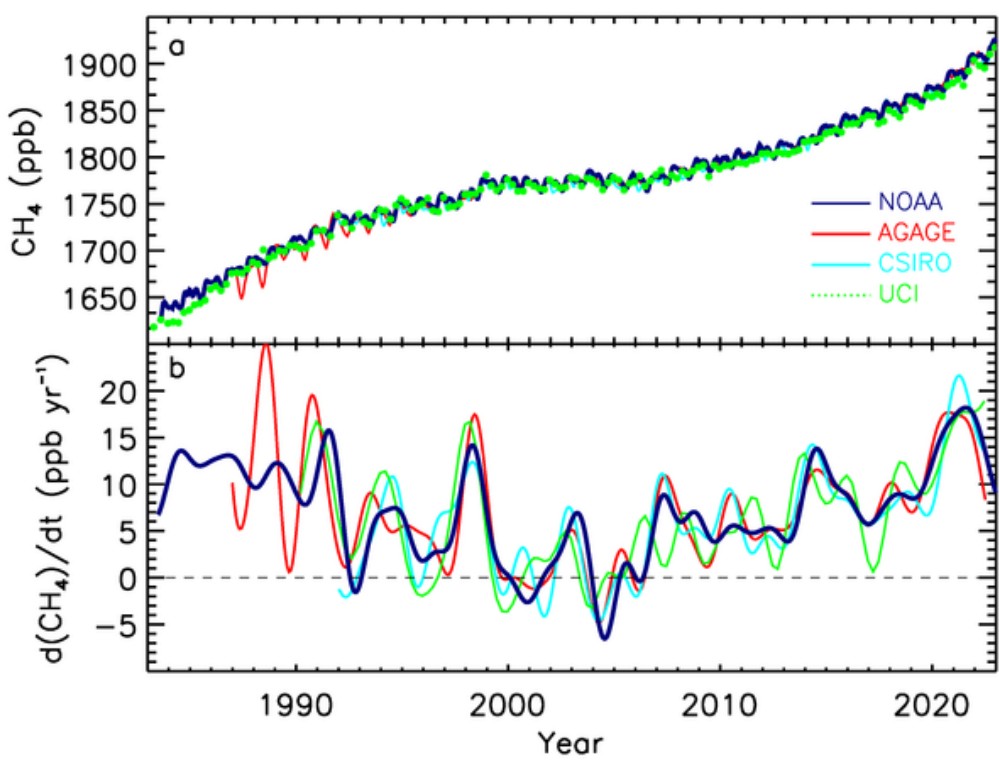

**Figure 1: Globally averaged atmospheric CH$_4$ concentrations (ppb) (a) and annual growth rates G$_{ATM}$ (ppb yr$^{-1}$) (b) between 1983 and 2022, from four measurement programs, National Oceanic and Atmospheric Administration (NOAA), Advanced Global Atmospheric Gases Experiment (AGAGE), Commonwealth Scientific and Industrial Research Organisation (CSIRO), and University of California, Irvine (UCI). Detailed descriptions of methods are given in the supplementary material of Kirschke et al. (2013).**

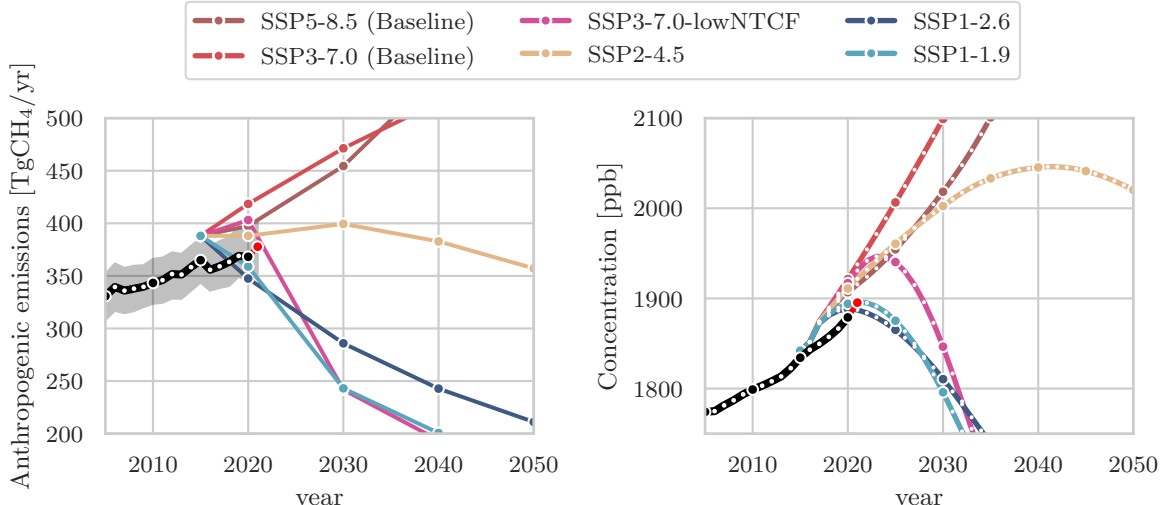

**Figure 2:** Left: Global anthropogenic methane emissions (including biomass burning) over 2005-2050 from historical inventories (black line and grey shaded area) and future projections (colored lines) (in Tg CH₄ yr⁻¹) from selected scenarios harmonized with historical emissions (CEDS) for CMIP6 activities (Gidden et al., 2019). Historical mean emissions correspond to the average of anthropogenic inventories listed in Table 1 added to the GFEDv4.1s (van der Werf et al., 2017) biomass burning historical emissions. Right: Global atmospheric methane concentrations for NOAA surface site observations (black) and projections based on SSPs (Riahi et al., 2017) with concentrations estimated using MAGICC (Meinshausen et al., 2017, 2020). Red dots show the last year available (2022 for observations).



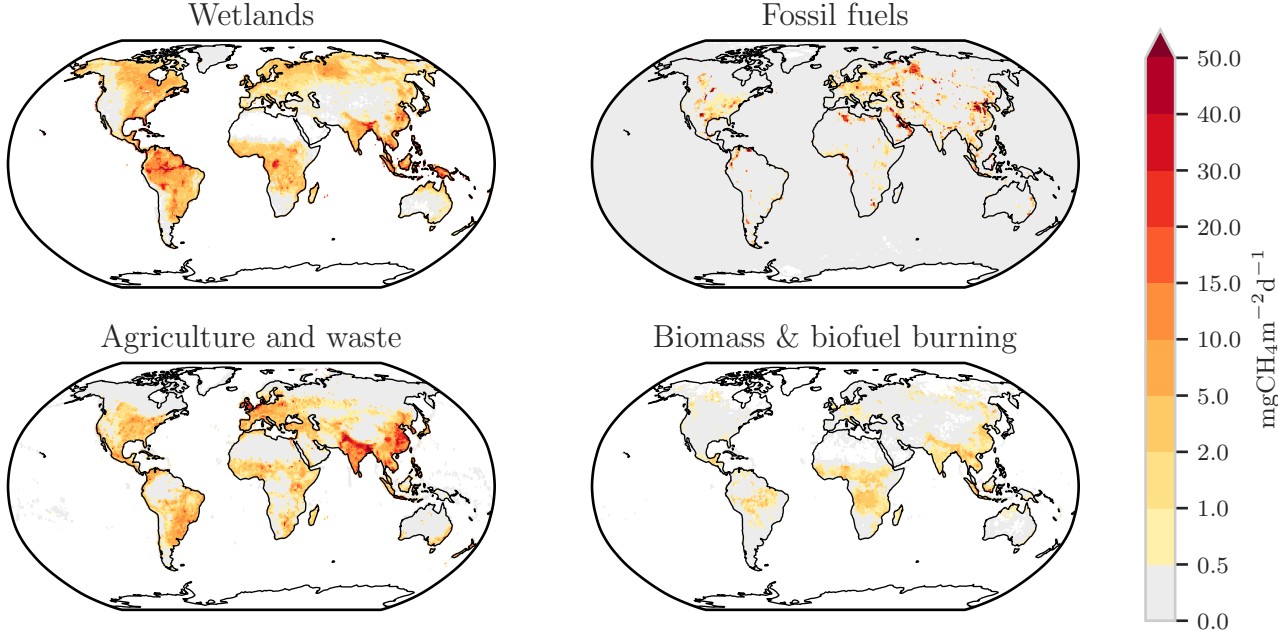

**Figure 3:** **Methane emissions from four source categories: natural wetlands (excluding lakes, ponds, and rivers), biomass and biofuel burning, agriculture and waste, and fossil fuels for the 2010-2019 decade in mg CH$_4$ m$^{-2}$ day$^{-1}$. The wetland emission map represents the mean daily emission average over the 16 biogeochemical models listed in Table 2 and over the 2010-2019 decade. Fossil fuel and Agriculture and Waste emission maps are derived from the mean estimates of gridded CEDS, EGDARv6, EDGARv7 and GAINS models. The biomass and biofuel burning map results from the mean of the biomass burning inventories listed in Table 1 added to the mean of the biofuel estimate from CEDS (O'Rourke et al., 2021), EDGARv6 (Crippa et al., 2021), EDGARv7 (Crippa et al., 2023) and GAINS (Höglund-Isaksson et al., (2020)) models.**



**Figure 4: Estimation of wetland and inland freshwater emissions over the 2010-2019 decade in Tg CH₄ yr⁻¹. The fluxes related to voluntary (such as through reservoirs or farm ponds) or involuntary (land use or eutrophication-related) perturbations of the methane cycle are shown here in orange. However, they are accounted for into the "natural and indirect anthropogenic" sources in the Table 3 budget and depicted as natural sources in Fig. 7.**





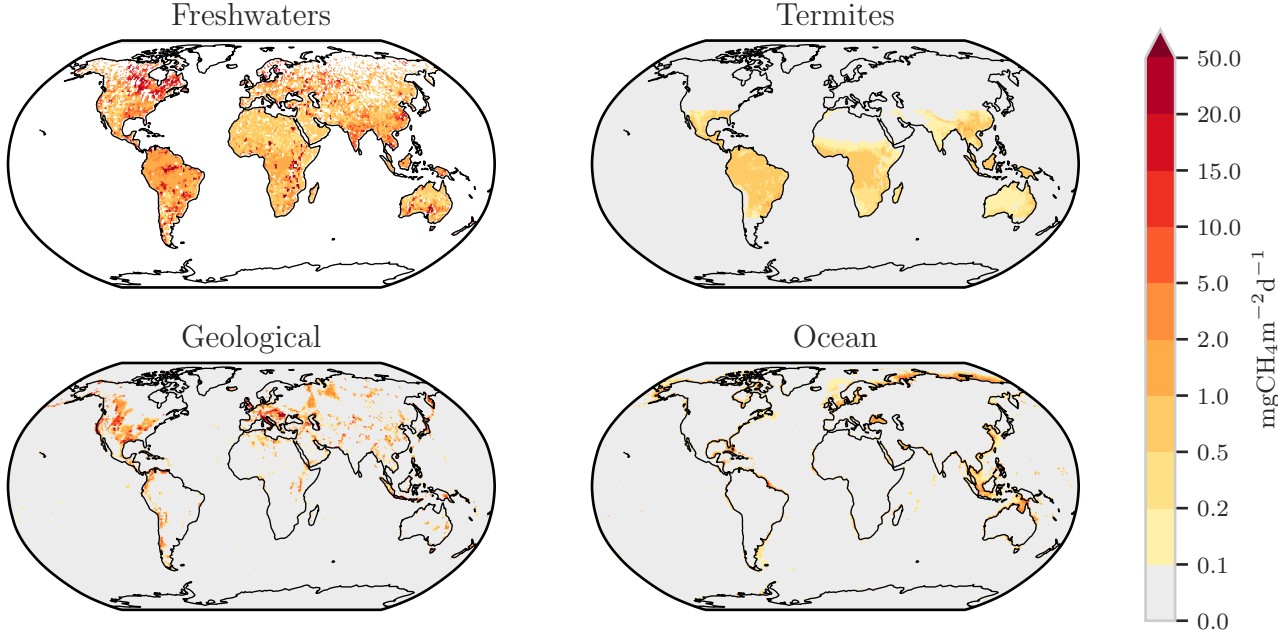

**Figure 5: Methane emissions (mg CH₄ m⁻² day⁻¹) from four natural and indirect anthropogenic sources: inland freshwaters (includes lakes, ponds (Johnson et al., 2022,), reservoirs (Johnson et al., 2021) and stream and rivers (Rocher-Ros et al., 2023) with a global total scaled to 89 Tg yr⁻¹), geological (Etiope et al., 2019), termites (this study) and oceans (Weber et al., 2019).**





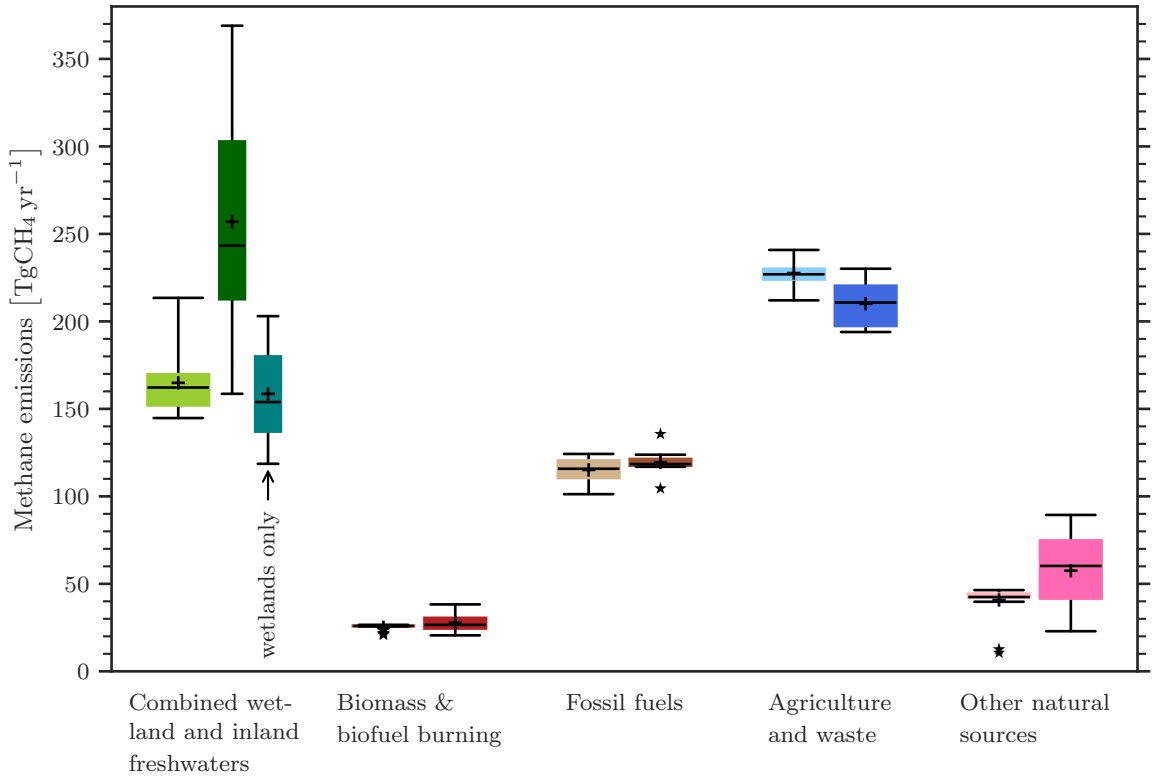

**Figure 6: Methane global emissions from five broad categories (see Sect. 2.3) for the 2010-2019 decade for top-down inversion models (left light coloured boxplots) in Tg CH4 yr⁻¹ and for bottom-up models and inventories (right dark coloured boxplots). For combined wetland and inland freshwaters three estimates are given: left = top-down estimates, middle = bottom-up estimates, right = bottom-up estimates for wetlands only. Median value, first and third quartiles are presented in the boxes. The whiskers represent the minimum and maximum values when suspected outliers are removed (see Sect. 2.2). Suspected outliers are marked with stars. Bottom-up quartiles are not available for bottom-up estimates, except for wetland emissions. Mean values are represented with "+" symbols, these are the values reported in Table 3.**



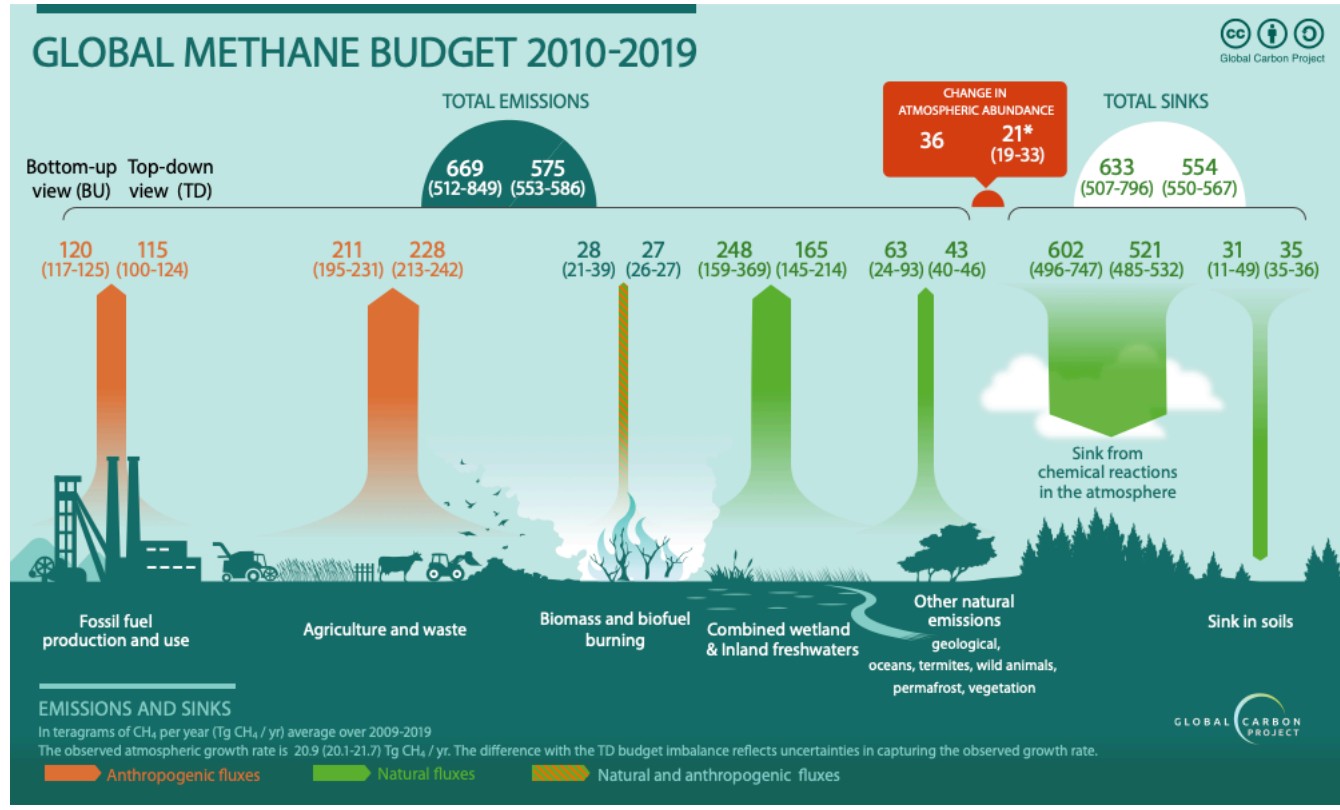

**Figure 7: Global Methane Budget for the 2010-2019 decade. Both bottom-up (left) and top-down (right) estimates are provided for each emission and sink category in Tg CH₄ yr⁻¹, as well as for total emissions and total sinks. Biomass and biofuel burning emissions are depicted here as both natural and anthropogenic emissions while they are fully included in anthropogenic emissions in the budget tables and text (Sect. 3.1.5). Combined wetland and inland freshwaters are depicted as fully natural while part has been attributed an indirect anthropogenic component (Sect. 3.2.2 and Figure 4).**

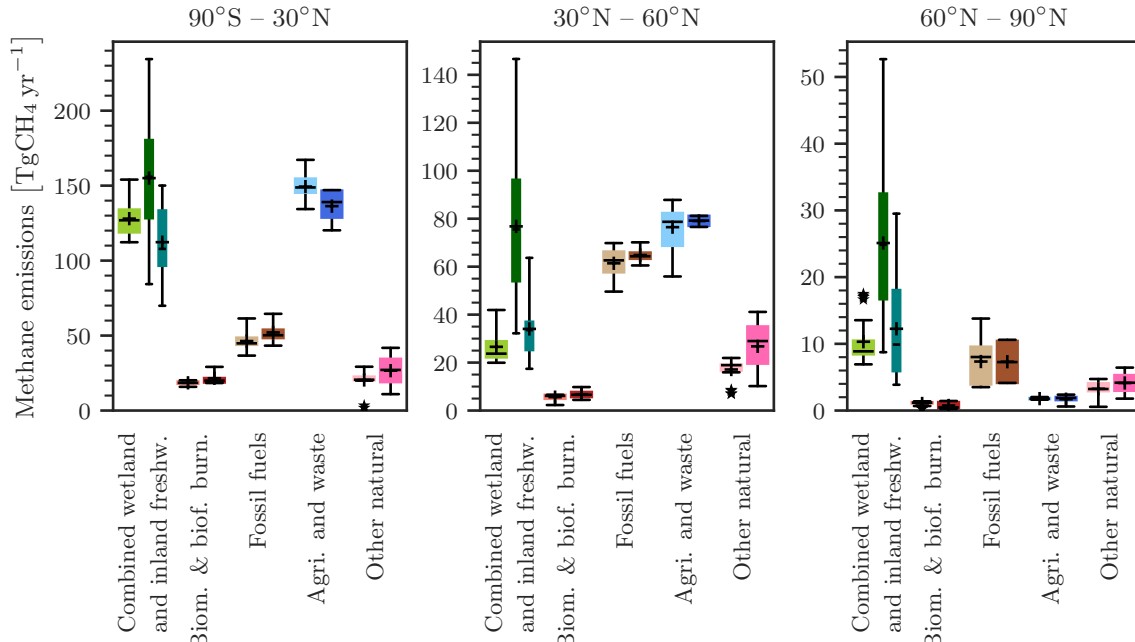

**Figure 8: Methane latitudinal emissions from five broad categories (see Sect. 2.3) for the 2010-2019 decade for top-down inversion models (left light coloured boxplots) in Tg CH₄ yr⁻¹ and for bottom-up models and inventories (right dark coloured boxplots). For combined wetland and inland freshwaters three estimates are given: left = top-down estimates, middle = bottom-up estimates, right = bottom-up estimates for wetlands only. Median value, first and third quartiles are presented in the boxes. The whiskers represent the minimum and maximum values when suspected outliers are removed (see Sect. 2.2). Suspected outliers are marked with stars. Bottom-up quartiles are not available for bottom-up estimates, except wetland emissions. Mean values are represented with "+" symbols, these are the values reported in Table 6.**




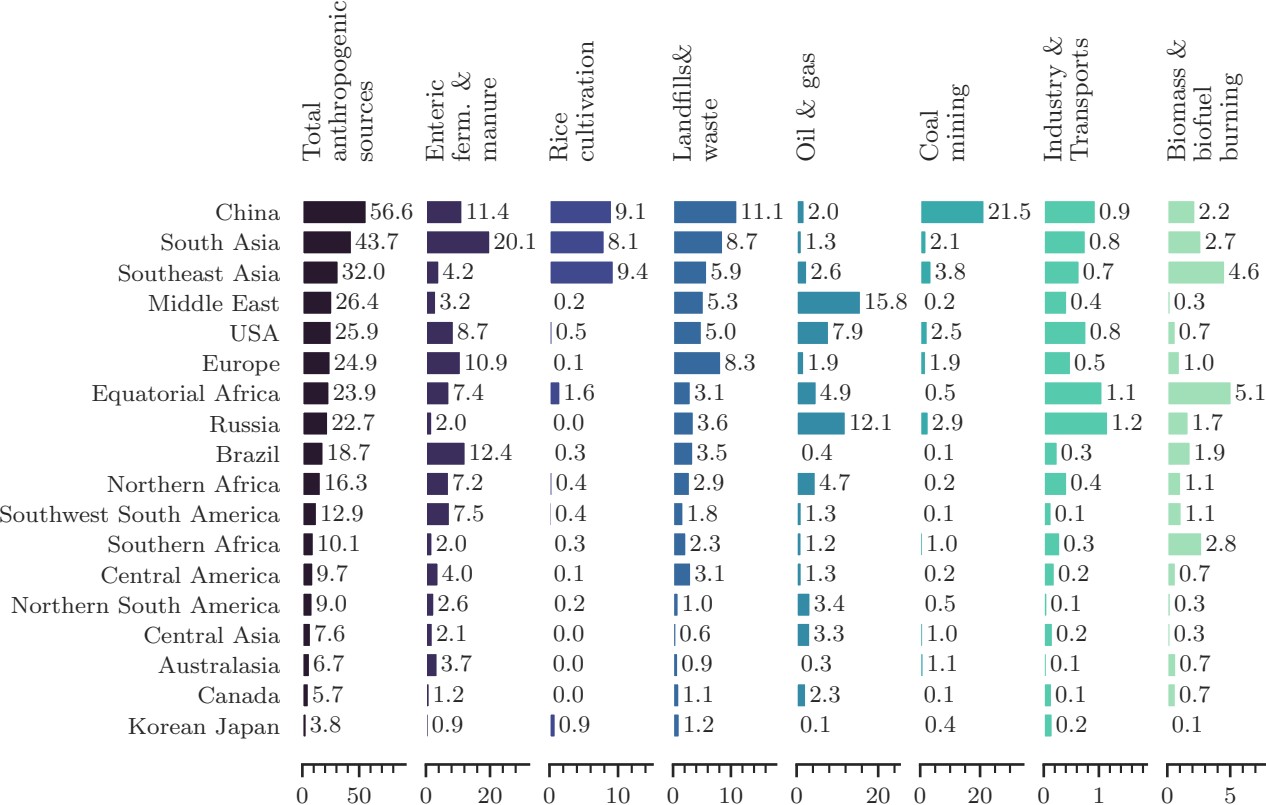

**Figure 9: Regional anthropogenic emissions for the 2010-2019 decade from bottom-up estimates in Tg CH$_4$ yr$^{-1}$. Regions are ranked by their total anthropogenic emissions. Note that each category has its own emission scale.**

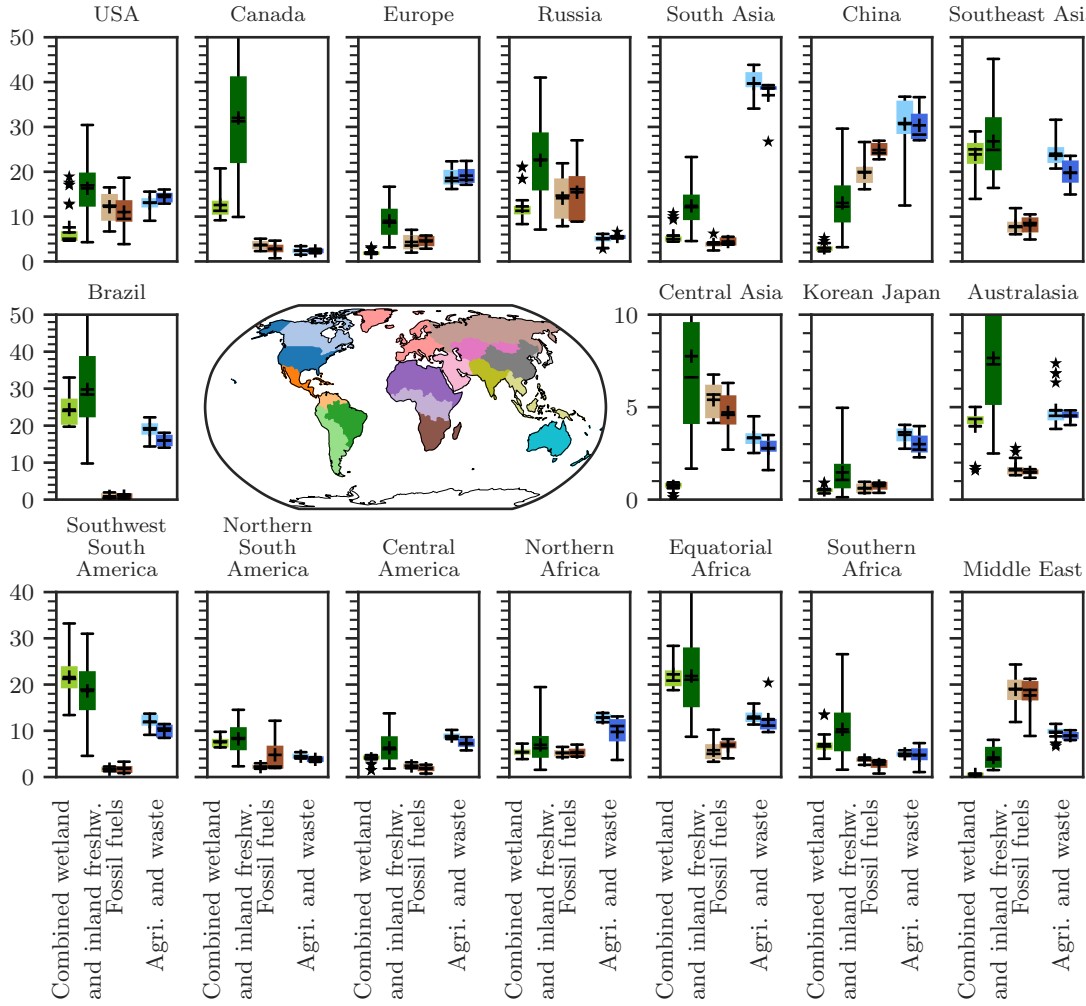

**Figure 10: Regional emissions for three broad main emissions categories for the 2010-2019 decade: Combined wetland and inland freshwaters, fossil fuel and agriculture & waste from top-down estimates (left box-plots- and bottom-up estimates (right boxplots). The inner map shows the region's distribution (see also Supplementary material, Table S1 and Fig. S3). More categories are presented in the Supplementary Material in Figure S6.**



**Table A1.** Comparison of terminologies used in this study and previous reports for methane sources.

| GCP terminology (This study) | | IPCC AR6 (Canadell et al., 2021) | National GHG inventories (used by UNFCCC according to IPCC (2006) and IPCC (2019)) | IPCC (2006, 2019) Source sector numbering |
|---|---|---|---|---|
| *Anthropogenic Sources* | | | | |
| Fossil fuels | Coal Mining | Coal Mining | Fugitive emissions from Fuels / Solid fuels | 1B1 |
| | Oil and gas | Oil and gas | Fugitive emissions from Fuels / Oil and natural gas | 1B2 |
| | Transport | Transport | Transport | 1A3 |
| | Industry | Industry | Mineral, chemical, metal industry and others | 2A, 2B, 2C, 2D, 2E |
| | | | Energy/fuel Combustion activities | 1A except 1A3 + 1B3 |
| Agriculture | Enteric fermentation and manure management | Enteric fermentation and manure management | Livestock | 3A |
| | Rice cultivation | Rice cultivation | Rice cultivation | 3C7 |
| Waste | Landfills and waste | Landfills and waste | Waste | 4 |
| Biofuel and biomass burning | Biofuel burning | Biofuel burning | Biofuel burning | 1A4b |
| | Biomass burning | Biomass burning | Biomass burning | 3C1 |
| *Natural and indirect sources* | | | | |
| Wetlands | Wetlands | Wetlands | – – | – – |
| Inland freshwaters | Reservoirs | included in Inland freshwaters | Land (incl Reservoirs) | in 3B |
| | Lakes, ponds, and rivers | incl in Inland freshwaters | only canal, ditches and ponds for human uses | in 3B |
| Other natural sources | Oceans | Oceans | – – | – – |
| | Termites | Termites | – – | – – |





| | Geological sources | Geological sources | – – | – – |
|---|---|---|---|---|




**Table A2.** Summary of methodological changes since the previous budget (Saunois et al., 2020). No significant changes have been applied to the vegetation (Sect. 3.2.8), wild animal (Sect. 3.2.5) and terrestrial permafrost and hydrates (Sect 3.2.7) estimates, though litterature has been expanded and/or updated.

| | Saunois et al. (2020) | This study |
|---|---|---|
| Regions definition (Table S1, Fig S3) | 18 continental regions + ocean | same regions except the last region including only Australia and New-Zealand and called Australasia |
| Anthropogenic global inventories (See Table 1, Sect 3.1.1) | CEDS, EDGARv4.3.2, USEPA (2012), FAO and GAINS ECLIPSE v6 | CEDS, EDGARv6 and v7, USEPA (2019), FAO, IIASA GAINS v4 Add estimate of ultra emitters from Lauvaux et al. (2022) |
| Biomass burning data sets | FINNv1.5, GFASv1.3, GFEDv4.1s, QFEDv2.5 | FINNv2.5, GFASv1.3, GFEDv4.1s, QFEDv2.5 |
| Estimate of wetland emissions (See Tables 2 and S3 and Section 3.2.1) | 13 land surface models involved, runs with either prescribed areas or based on Hydrological scheme, single meteorological forcing | 16 land surface models involved, runs with either prescribed areas or based on Hydrological scheme, two sets of meteorological forcings |
| Estimate of reservoirs emissions (Sect.3.2.2) | based on Deemer et al. (2016) | based on Johnson et al. (2021), Rosentreter et al. (2021) and Harrison et al. (2021) |
| Estimate of lakes and ponds emissions (Sect.3.2.2) | based on Bastviken et al. (2011), Wik et al. (2016b) and Tan and Zhuang (2015) | lakes > .1km2 : based on Rosentreter et al. (2021), Zhuang et al. (2023) and Johnson et al. (2022) lakes and ponds < 0.1 km2 : based on Rosentreter et al. (2021), and Johnson et al. (2022) |
| Estimates of stream and river emissions (Sect.3.2.2) | From Stanley et al. (2016) | based on Rosentreter et al. (2021) and Rocher-Ros et al. (2023) |
| Estimates of the anthropogenic perturbation component of inland freshwater emissions (Sect.3.2.2) | - - | based on several individual studies on the effect of eutrophication on emissions from lakes, and ponds (See text in Sect. 3.2.2) |
| Estimate of the double counting in the aquatic systems (Sect.3.2.2) | _ _ | due to the accounting of small lakes and ponds (<0.1km2) in the vegetated wetlands areas used in land surface models and to lateral transport from vegetated wetland to rivers. |



| Geological sources (Sect 3.2.3) - onshore and offshore | based on Etiope and Schwiezke et al. (2019) | same as in Saunois et al. (2020) |
|---|---|---|
| Termite emissions (Sect. 3.2.4) | GPP : Zhang et al. (2017)<br>termite biomass: Jung et al. (2011)<br>EF : Kirshke et al. (2013) and Fraser et al., 1986) | GPP: Wild et al. (2022)<br>termite biomass: based on different studies depending on regions (see text)<br>EF: Sugimoto et al. (1998)<br>Applied a correction factor for mound from Nauer et al. (2018) |
| Oceanic sources (Sect 3.2.6) | modern biogenic: based on Wuebbles and Hayhoe (2002) , Laruelle et al. (2013) and Rosentreter et al. (2018); geological: based on Etiope (2019) | modern biogenic: based on Rosentreter et al. (2021;2023) and Laruelle et al. (2023)<br>geological: based on Etiope (2019) |
| Tropospheric OH oxidation (Sect 3.3.2) and stratospheric loss (Sect 3.3.3) (See Supplementary Table S4) | based on results from 11 models contributing to the Chemistry Climate Model Initiative (Morgenstern et al., 2017) | based on results from 11 models contributing to the Chemistry Climate Model Initiative 2022 (Plummer et al., 2021) and the CMIP6 simulations (Collins et al., 2017) |
| Tropospheric reaction with Cl | based on Hossaini et al. (2016), Wang et al. (2019b) and Gromov (2018) | based on Hossaini et al (2016), Sherwenn et al. (2016), Wang et al (2019b, 2021b) and Gromov (2018) |
| Soil uptake (See Table S6) | based on Tian et al. (2016) | based on VISIT, JSBACH en MeMo surface models. |
| Estimates through top-down approaches (See table S7 and S8 to S11) | 9 inverse systems contributing, prior fluxes based on EDGARv4.2 or v4.3.2 for most inversions. Most inversion used constant OH. | 7 inverse systems contributing, runs with constant and varying OH, prior fluxes based on either EDGARv6 or GAINS |



**Table A3.** Funding supporting the production of the various components of the global methane budget in addition to the authors' supporting institutions (see also acknowledgements).

| Funder and grant number (where relevant) | Authors/Simulations/Observations |
| --- | --- |
| Director, Office of Science, Office of Biological and Environmental Research of the US Department of Energy under Contract No. DE-AC02-05CH11231 to Lawrence Berkeley National Laboratory as part of the RUBISCO Scientific Focus Area. | WJR, QZ, E3SM/ELM simulations |
| Funded by NASA's Interdisciplinary Research in Earth Science (IDS) Program and the NASA Terrestrial Ecology and Tropospheric Composition Programs | MSJ; lake and reservoir bottom-up methane emission data sets |
| Funded by Agence National de la Recherche through the project Advanced Methane Budget through Multi-constraints and Multi-data streams Modelling (AMB-M$^3$) - (ANR-21-CE01-0030) | AM, MS |
| The Environment Research and Technology Development Fund (JPMEERF21S20800) of the Environmental Restoration and Conservation Agency provided by Ministry of the Environment of Japan | YN, NISMON-CH$_4$ |
| Funded by the German Federal Ministry of Education and Research (BMBF) via the "PalMod" project, grant No. 01LP1921A | TK; CH$_4$ emission modelling with JSBACH and LPJ-MPI |
| Funded by the Swedish Research Council VR (2020-05338) and Swedish National Space Agency (209/19) | WZ; LPJ-GUESS simulations |
| Funded by BELSPO (project FedTwin ReCAP), EU Horizon 2020 project ESM2025 (nr. 101003536) and FRNS PDR project CH4-lake (T.0191.23) | PR; inland water, coastal and oceanic CH4 emission synthesis |
| EU H2020 (725546 ERC METLAKE and 101015825 TRIAGE) , Swedish Research Councils VR (2022-03841) and Formas (2018-01794) | DB; inland waters - data and bottom up estimation. |
| Supported by the Newton Fund through the Met Office Climate Science for Service Partnership Brazil (CSSP Brazil) | NG; JULES simulations |
| Funded by United Nations Environment Programme, Stanford University DTIE21-EN3143 | RBJ; inversions and general budget support |
| the Joint Fund for Regional Innovation and Development of the National Natural Science Foundation (Grant No. U22A20570); the Natural Sciences and Engineering Research Council of Canada (NSERC, #371706) | Changhui Peng/TRIPLEX-GHG |
| **Computing Resources** | |
| LSCE computing resources | Marielle Saunois, Philippe Bousquet, Joël Thanwerdas and Adrien Martinez |
| NASA High-End Computing (HEC) Program through the NASA Advanced Supercomputing (NAS) Division at NASA Ames Research Center | Matthew S. Johnson (MSJ) |
| Deutsches Klimarechenzentrum (DKRZ), Hamburg, Germany | Thomas Kleinen (TK) |



| | |
|---|---|
| ALICE High Performance Computing Facility at the University of Leicester | GOSAT retrievals |
| FUJITSU PRIMERGY CX2550M5 at MRI and NEC SX-Aurora TSUBASA at NIES | Yosuke Niwa (YN) |
| **Support for atmospheric observations** | |
| Australian Antarctic Division | CSIRO flask network |
| Australian Institute of Marine Science | CSIRO flask network |
| Bureau of Meteorology (Australia) | Kennaook/Cape Grim AGAGE, CSIRO flask network |
| Commonwealth Scientific and Industrial Research Organisation (CSIRO, Australia) | Kennaook/Cape Grim AGAGE, CSIRO flask network |
| Department of Climate Change, Energy, the Environment and Water (DCCEEW, Australia) | Kennaook/Cape Grim AGAGE |
| Meteorological Service of Canada | CSIRO flask network |
| NASA: grants NAG5-12669, NNX07AE89G, NNX11AF17G, NNX16AC98G and 80NSSC21K1369 to MIT with subawards to the University of Bristol (for Barbados and Mace Head) and CSIRO (for Kennaook/Cape Grim); grants NAG5-4023, NNX07AE87G, NNX07AF09G, NNX11AF15G, NNX11AF16G, NNX16AC96G, NNX16AC97G, 80NSSC21K1210 and 80NSSC21K1201 to SIO. | AGAGE calibrations and measurements at SIO, La Jolla and AGAGE station operations at Trinidad Head, Mace Head, Barbados, American Samoa, and Kennaook/Cape Grim |
| National Oceanic and Atmospheric Administration (NOAA, USA) contract RA133R15CN0008 to the University of Bristol | Barbados |
| NOAA USA | CSIRO flask network |
| Refrigerant Reclaim Australia | Kennaook/Cape Grim AGAGE |
| UK Department for Business, Energy & Industrial Strategy (BEIS) contract TRN1537/06/2018 and TRN 5488/11/2021 to the University of Bristol | Mace Head |
| National Oceanic and Atmospheric Administration (NOAA, USA) | Cape Matatula |
| Japanese Ministry of Environment | GOSAT data, Robert Parker |
| Japanese Aerospace Exploration Agency, National Institute for Environmental Studies | GOSAT data, Robert Parker |
| NERC UK: grants NE/W004895/1, NE/R016518/1 and NE/X019071/1 | GOSAT data, Robert Parker |
| The Swedish Research Council VR (2022-04839), European Space Agency projects AMPAC-Net and CCI+ permafrost, European Union's Horizon 2020 Research and Innovation Programme to the Nunataryuk project (no. 773421) | Permafrost region, Gustaf Hugelius |