# Peer review of "Global Methane Budget 2000-2020"

_Earth System Science Data, 2024_

## Author Comment (AC1)

**The Global Methane Budget: 2000-2020**

Saunois et al., ESSD, 2024

Detailed Response to Anonymous Referee#2
We acknowledge the referee for the time spent on reading and commenting on the paper. We thank Anonymous Referee #2 for the useful corrections and suggestions on the paper, which have helped clarify and improve the manuscript. Below are the responses (in black) to the comments (in italics, blue). Changes in the text follow each response in bold font.

*Global Methane Budget needs focus and discernment'*
*This manuscript represents a lot of work by a large number of co-authors. The work is massive and I admit to being unable to seriously review the entire document. It contains a lot of valuable material plus a number of errors or misguided recommendations. Given the number of authors, it would be nice if more of them spent time carefully reading/editing. I begin to wonder if the massive tomes generated so frequently by the Global Carbon Project are a benefit to the community. I suspect it is too late to raise this, but I do not think this work is ready for publication.*

*A number of the sections seem to be repeats of the same old ideas and references in Saunois 2020. Why is this not an update of what is new?*

We acknowledge the difficulty of reviewing such manuscript due to his length and amount of information and details. Indeed, this manuscript is a living document, updated every 3-4 years to produce a new budget but keeping the foundation of the main text. Such a regular budget is highly expected by the community. The first paper (Kirschke et al., 2013) has been cited 2264 times, the second (Saunois et al., 2016) has been cited 1157 times and the last update (Saunois et al., 2020) has been more cited than the 2016 paper: 1871 times.

Though some parts have been less updated than others – in terms of reference, this budget brings novel insights and latest estimates of methane sources and sinks, and uncertainties that still remain for global emissions and at the individual sector scale. These uncertainties are often overlooked in many individual studies providing a single number for methane emissions. The "repeats" from the previous budget are intentional when the information has not changed much. We believe this helps the reader not to have to read the older budgets to keep up with the background and material in the current budget. However, we agree that some old references remained in the submitted version of the text. Thanks to the constructive reviews received (mainly from reviewer #1) we improved this weakness by providing a more up-to-date review of the current knowledge for most of the concerned paragraph. Other parts may not have been updated enough but were kept for the sake of completeness of the budget.

*Section 7 appears to be a personal wish list of the authors' personal research goals, rather than a critical review. Many of the references are to a single paper, often involving one or more of the authors.*

Section 7 is not meant to be a critical review of our synthesis. Also, it has been constructed by many of the authors and reflects at least partly, priorities from the methane community. We included papers and studies known by the authors. We improved this part by adding literature and studies that were not cited in the current version.

*Abstract:*

*L115: "maintaining CH4 as…" is incorrect. Even if CH4 emissions remained constant, it would maintain CH4 as the 2nd GHG for a long while!*

Indeed. Original sentence : *"Emissions and atmospheric concentrations of CH4 continue to increase, maintaining CH4 as the second most important human-influenced greenhouse gas in terms of climate forcing after carbon dioxide (CO2)"*. This has been rephrased to: "**CH4 is the second most important human-influenced greenhouse gas in terms of climate forcing after carbon dioxide (CO2) and both emissions and atmospheric concentrations of CH4 continue to increase since 2007**"

*L117: "importance of CH4 EMISSIONS compared to THOSE OF CO2…" The atmospheric abundance is what it is, and is observed! Here I think you want to emphasize the emissions, not the abundance.*

Indeed. Thank you for your comment. This has been corrected.

*L119: "in reducing uncertainties in the factors.." is very contorted, try: "in quantifying the factors responsible for the observed atmospheric.." we do not really need 'well' here either.*

This has been modified as suggested.

*L130: this 2024 edition (help remind the reader of which edition they are reading)*

This has been modified as suggested.

*L133: either the double counting is "potential" in which case it MAY exist, or it is not 'potential' but an estimate of the REMAINING double accounting.*

This has been rephrased to " **the potential double counting that may exist**" .

*Introduction*

*L172ff: No, this is simply incorrect. CH4 is not a stronger absorber in the current atmosphere, there is more CO2 and hence CO2 is the stronger absorber (witness the ERF of each). Anyhow, what you are talking about is GWP>1 meaning stronger impact on climate over the next 100 years per kg emitted. Anyway, this paper does not have the background in radiative transfer and forcing and you should not be talking about 'absorbers', stick to climate forcing and GWPs.*

We acknowledge that this was incorrectly written. This part has been rewritten based on climate forcing and GWP only. The revised version now includes: "An **equal mass of CH₄**

**emissions has a stronger impact on climate than carbon dioxide (CO$_2$), which is reflected by its global warming potential (GWP) relative to CO$_2$ on a given time horizon**"

*L173ff: It is not 'climate feedbacks' that matter here, but 'climate and chemical composition changes' (not feedbacks!). Chemical feedback are included in the GWP.*

The GWP indeed includes indirect chemical effects. We have removed "without considering climate feedbacks" and rephrased to: "**For a 100-yr time horizon the GWP of CH$_4$ emitted by fossil sources is 29.8 (GWP of CH$_4$ emitted by microbial sources is 27), whereas the values reach 82.5 over a 20-year horizon for CH$_4$ emitted by fossil sources and 79.7 for CH$_4$ emitted by microbial sources (Forster et al., 2021)**"

*L174ff: Since you are insisting on quoting all the GWPs for methane (20 & 100 yr, fossil and non-fossil) let us get it right. OK, you are quoting GWPs from last IPCC, but it is about time we correct the maths that were done in Forster et al 2021. I know this is not the job of this CH4 budget review to fix IPCC problems, but we should avoid further propagating errors.*
*(1) The difference in fossil vs non-fossil GWP is 2.75 based on the molecular wt of CH4 and CO2. IPCC rounds to 2.8 – OK.*
*(2) This assumes that the fossil 1 kg CH4 releases immediately 2.75 kg CO2 (stoichiometric) and that the CO2 is there for the full GWP time scale.*
*(3) In reality the CO2 is released as the CH4 decays and thus the difference from non-fossil to fossil must be smaller than 2.75.*
*(4) We agree that a tropospheric CH4 pulse decays as exp(-t/11.8) since a perturbation lifetime is 11.8 yr per IPCC Table 7.15. For 20 years, the average CH4 is about 9.63 yr or about 48% of the 20-yr period. Thus, the duration of the CO2 is 0.518\*20 or only 10.37 years (i.e., add +1.4 to the GWP-20). Thus the fossil CH4 GWP-20 should be corrected to 81.1 not 82.5. This is a minor correction, but since you are quoting decimal places, we should all correct the value.*
*(5) For 100 year GWP, this correction is smaller, 29.4 not 29.8.*

As stated by the reviewer this is not the place to correct IPCC values. Small incorrections may have been done in Forster et al. (2021), and the uncertainties on the values are larger than the given precision. Though we acknowledge the reviewer to quote these errors, we keep the original Forster et al. (2021) values in the text.

*L175: "CH4-non fossil" is very poor English it is more in the French style with the noun first. I love the French language, but here try "non-fossil CH4" and "fossil CH4"*

This is not fair as a remark. This spelling is the one used in Table 7.15 of Forster et al. (2021), and it has just been reused as is. Maybe the person who did the Table was French, and none of the hundreds of co-authors and reviewers caught this? Anyway this has been rewritten in a more detailed and English way as follows: "**For a 100-yr time horizon the GWP of CH$_4$ emitted by fossil sources is 29.8 (GWP of CH$_4$ emitted by microbial sources is 27), whereas the values reach 82.5 over a 20-year horizon for CH$_4$ emitted by fossil sources and 79.7 for CH$_4$ emitted by microbial sources (Forster et al., 2021).** "

*L1467FF This section on the stratosphere is remarkably out of date and misguided. The idea that strat loss could be 10 Tg/y is ridiculous (I know that models may calculate this, but why*

*report absurd values). This gives a strat lifetime of 500 years!! Then it would be well mixed to the mesosphere. Scaling to N2O and other known species it should be ~120 yr (100-200?) = ~40 Tg, and it is uncertain to what level, maybe 30-50, but not 10-70. The large percentages for O(1D) can Cl loss must refer to regions of the stratosphere and not to total stratospheric loss? If so, drop them because they are misleading and it does not matter what drives CH4 loss at 48 km.*

Actually, we did not find recent updates on the estimation of stratospheric loss of methane on the top of the reference or simulations (e.g. CMIP6 & the latest chemistry climate estimates) used in the paper. Providing literature on the matter would be valuable to improve the manuscript. Though the first lines are very introductive, **recent references have been added (Zhang et al., 2023) and Morgenstern et al. (2018).** We acknowledge that the lower range of loss (10 Tg/yr) is not very likely, noting that average estimates are more at 30-40 Tg/yr.  We propose to exclude the single model with this low estimate from the ensemble. The text has been modified as follows: "**In this study, six chemistry climate models that contributed to CMIP6 modelling activities (Table S5) provided estimates of CH$_4$ chemical loss, including reactions with OH, O($^1$D), and Cl; CH$_4$ photolysis is also included but occurs only above the stratosphere. Considering a 200 hPa tropopause height, these six CMIP6 simulations suggest an estimate of 34 [10-51] Tg CH$_4$ yr$^{-1}$ for the CH$_4$ stratospheric sink for the 2000-2009 decade (Table S5), similar to the value derived from the previous CCMI activity reported in Saunois et al. (2020) (31 [12-41] Tg CH$_4$ yr$^{-1}$). The lowest estimate provided by a model (10 Tg CH$_4$ yr$^{-1}$ ) is quite unrealistic and would yield a methane stratospheric lifetime of several hundreds of years. As a result this outlier is excluded and we prefer to report a mean of 39 Tg CH$_4$ yr$^{-1}$ associated with a min-max range of [27-51] for 2000-2009.**
**For 2010-2019, we report here a climatological range of 28-43 Tg CH$_4$ yr$^{-1}$ associated with a mean value of 37 Tg CH$_4$ yr$^{-1}$ based on five models that contributed to CMIP6 runs (historic followed by SSP3-7.0 projections starting in 2015; Table S5).**"

Also, it does matter what drives methane loss in the stratosphere and if models get it right or wrong in terms of sinks and then in terms of concentrations. This is especially true when it comes to using total columns measured from satellites within atmospheric inversions framework to infer surface fluxes. A wrong stratospheric gradient induced overestimating the total column in the chemistry transport model not because of the fluxes at the surface (which will be wrongly corrected).

*L1617ff: This text looks like the previous 2020 edition that made the mistake of assuming that the fluctuations in CH4 abundance caused by emissions bumps would follow the budget lifetime rather than the perturbation lifetime (12 yr). Thus, the excellence of your fit to 9 yr should give you pause and admit that the variability is a mix of time scales for varying emissions and OH. The IPCC clearly discusses the different lifetimes and prominently displays the perturbation lifetime (11.8 yr) in the Table 7.15 from which you took the GWPs.*

Indeed the sentence regarding the fit made using NOAA observation has been kept as in the previous version, and corresponds to a calculation made in Dlugockenky et al. (2003) to partly discuss the plateau in the early 2000s. We decided to remove that part of the text to stick to

describing observations as done for the period after the plateau and provide the same discussion for the different periods of atmospheric methane changes.

Nonetheless, we would like to highlight that this sentence was correct. The perturbation lifetime is theoretical and does not account for the impact of fast chemistry compared to the lifetime of methane (involving NOx for example, see Nicely et al. (2018))

*L1690: "In addition, most of the top-down models use the same OH distribution from the TRANSCOM experiment (Patra et al., 2011),.." If this is so, then the top-down model mean of 575 Tg/y is really fixed by the assumed OH values. It is hardly a chemistry model, since the chemistry is fixed. This makes all the top-down numbers useless as they are fixed by Patra's choice of OH in 2011. Can you explain?*

The top-down community faces difficulties in handling OH amount and variations, two key parameters for the methane budget, that are not well-constrained. Most of the systems still use prescribed OH concentrations, some tend to try to optimize that quantity along with the fluxes (though the reviewer tends to criticize the use of satellite data for this in a following comment). Using the same OH quantity definitely constrained the overall budget and range across the simulations, though different transport and simulated methane distribution lead to slightly different loss even with the same prescribed OH. As a result the loss is not fixed at 575 Tg $CH_4$/yr. This is acknowledged in the second part of this sentence: *"…which introduces less variability to the global budget than is likely justified, and so contributes to the rather low range (10%) compared to bottom-up estimates (see below)."* The values are not useless as we recognized the limitations. Part of the simulations were done including OH changes over time – low IAV <2% and no significant trend based on Patra et al. (2021) and in agreement with Thompson et al. (2024). Differences between the two ensembles (constant versus varying OH) are much lower than the range across the inverse systems. Also two inversions used different OH fields and their results fall into the same range. If these values are useless then all the top-down inverse studies published so far could be discarded as none is really doing a better job.

Though to acknowledge such limitation we recall Zhao et al.(2020) main message in the result section 5.1.1 after stating the range from the top-down studies: "**We recall here that Zhao et al. (2020) found an uncertainty of about 17% in global methane emissions (518 to 611 Tg $CH_4$ $yr^{-1}$ for the early 2000s) due to changes in OH burden and distribution (OH ranging from 10.3 to 12.6 $10^5$ molec. $cm^{-3}$)**"

*L2052: I think we need to improve the chemistry-transport models and chemistry-climate models using full chemistry rather than try to replace interactive chemistry with a ML approximation so you can use tracer-transport models. (There is a confusion here that CTMs are tracer models with fixed OH, they are not.) It is more important to find out why the CCMs and CTMs (with interactive chemistry) may be wrong. That should be a higher recommendation than the Zhao approach that the lead author here is recommending.*

Improving the chemistry in chemistry transport model (CTM) and chemistry climate models (CCM) has been a needed path for years. Atmospheric inversions do not use interactive chemistry as it could be too costly and we would need the adjoint of the chemistry solver. This is why we use pre-calculated OH fields from CTM or CCM. However, over the last three rounds (10 years) of this exercise of methane budget, we have not noticed any improvement in the

matter: the ranges of OH quantity, trends and IAV are still of the same order of magnitude, and CTM and CCM are not converging. Biases of those models against observations are still existing. Correcting the existing OH fields based on observations could be efficient awaiting a benchmarking of OH among CTM and CCM. Zhao et al. (2024) proposed a recipe to evaluate OH based on interactive chemistry in CCM/CTMs and a box model that could be useful to improve the convergence of OH fields among CCMs if more groups were adopting it. However, we acknowledge that the recommendation spectrum was too narrow. Thus, the details provided on Zhao et al. (2023) study have been removed and other references have been added. The text is as follows:

"**These results emphasise the need to first assess, and then improve, atmospheric transport and chemistry models, especially vertically, and to integrate robust representation of OH fields in atmospheric models. Recently, numerous efforts based on satellite data have been made to constrained OH distribution, variability and trends ( e.g, Anderson et al. 2023 ;2024 ; Pimlott et al. 2022 ; Zhao et al., 2023; Zhu et al., 2022).**"
For the steps forward:
"**-      Developping benchmarking of CTM and CCM regarding simulated OH distribution and variability (as in Zhao et al. (2019) for example) to increase efforts to assess biases and improve atmospheric chemical schemes in CTM and CCM.**
**-      Developping methods to better constrain OH, numerous have been proposed: satellite $CH_4$ observations (Zhang et al., 2018; Anderson et al., 2023; 2024) could afford this but strategy is needed (see Duncan et al. preprint 2024 and references therein);... **"

*L2063: If you recommend studying the reactivity of air parcels, then you really should reference the major effort on that done by the NASA ATom mission (e.g., 2023 ESSD, doi: 10.5194/essd-15-3299-2023).*

This reference has been added to the text.

*L2067: You can recommend satellite-constrained OH and that is a very attractive area for some, but it lacks the vertical resolution needed for CH4 studies and it seems to be overhyped here.*

Indeed, this limitation is very real. We acknowledge this, we completed the text as follows:
"**-      -      Developing methods to better constrain OH. Numerous have been proposed: satellite $CH_4$ observations (Zhang et al., 2018; Anderson et al., 2023; 2024) could afford this but strategy is needed (see Duncan et al. preprint 2024 and references therein); using halogenated compounds beyond methyl chloroform (MCF), such as done in box models (Thompson et al., 2024) to derive a 3D dynamical OH. Such methods should be able to reach very low uncertainty for OH burden and trends (<2%) in order to really better constrain the $CH_4$ budget. Duncan et al. (preprint) discuss the existing satellite-based methods and propose a strategy to constrain OH from space-based approaches.**"

*L2153: Again, this whole list of recommended directions seems like the co-authors personal research agenda and hardly a critical review of where advances might come form. The idea of porting transport models to GPUs to reach sub-one-degree will not improve the overall transport characteristics of these models.  Many CTM/CCMs are already running one-degree*

*simulations. Much of the error is in the large-scale flows. Further, who is going to generate and store the petabytes needed to run the tracer models? Already we have full up CCMs running refined mesh at 3 km scales embedded in ESMs. If you want hi-res, that is where you must go. We already have CTM/CCMs with hybrid coordinates, what is so special here (L2155). The choice of odd grids will come from the dynamics of the underlying climate/forecast model rather than being designed by the tracer-transport operator.*

This paragraph and directions concern the atmospheric inversion approaches. Most of them, at the time of the submission of this paper, still run at around 2-3 degrees resolutions for the sake of computer resources and time to produce an inversion over 20 years. Aiming at finer resolution allows finer resolution for the flux retrievals, better synoptic resolution of the variations of concentrations at surface stations (especially coastal and mountain sites), and also a better vertical transport (regarding convection in particular and troposphere-stratosphere exchange). Indeed, the transport model errors consider here also the error made against the observational constraint used for the optimizations. Also, running atmospheric inversions at finer resolution on CPU as usual will take a few months which is unrealistic, porting the atmospheric inversions system to GPU will overcome (has overcome) this issue. Indeed, some atmospheric inversion systems rely on dynamical cores that now could run on odd grids, but inversion systems need to adapt to this and this is far from straightforward. Odd grids could allow better representation of the high latitudes regions, which are, considering the Arctic, important regions regarding greenhouse gas emissions and climate feedback. Finally, although we agree that some recommendations may be unadapted (e.g. hybrid coordinates (**This has been removed**)), the recommendations made in the paper come from years of analysis of the global methane budget from 80 scientists worldwide and several workshops organized by various groups among which the global carbon project. We think they mean more than personal thoughts of the authors.

The sentence has been modified as follows : "**In the long run, developments within the dynamical core of the atmospheric transport models through the implementation of hexagonal-icosaedric grid with finer resolution (Dubos et al., 2015; Niwa et al., 2017, 2022; Lloret et al., 2023), and improvements in the simulated boundary layer dynamics or troposphere-stratosphere exchanges are promising to reduce atmospheric transport errors.** "

Finally, the question of the computing resources and storage for these simulations is a common question for the whole modeling community whatever the subject, and not specific to atmospheric inversions. Though it is worth noting that many institutions are already considering the environmental impact of their research.

**References:**

Anderson, D. C., B. N. Duncan, J. M. Nicely, et al. J. Liu, S. A. Strode, and M. B. Follette-Cook. 2023. Technical note: Constraining the hydroxyl (OH) radical in the tropics with satellite observations of its drivers – first steps toward assessing the feasibility of a global observation strategy Atmospheric Chemistry and Physics 23 (11): 6319-6338, doi:10.5194/acp-23-6319-2023

Anderson, D. C., Duncan, B. N., Liu, J., Nicely, J. M., Strode, S. A., Follette-Cook, M. B., Souri, A.H., Ziemke, J.R., Gonzalez-Abad, G., and Ayazpour, Z. (2024). Trends and interannual variability of the hydroxyl radical in the remote tropics during boreal autumn inferred from satellite proxy data. Geophysical Research Letters, 51, e2024GL108531, doi:10.1029/2024GL108531

Dlugokencky, E.J., S. Houweling, L. Bruhwiler, K.A. Masarie, P.M. Lang, J.B. Miller, and P.P. Tans, Atmospheric methane levels off: Temporary pause or new steady-state?, Geophys. Res. Lett., 30 (19), doi:10.1029/2003GL018126, 2003.

Duncan, B., Anderson, D., Fiore, A., Joiner, J., Krotkov, N., Li, C., Millet, D., Nicely, J., Oman, L., St. Clair, J., Shutter, J., Souri, A., Strode, S., Weir, B., Wolfe, G., Worden, H., and Zhu, Q.: Opinion: Beyond Global Means: Novel Space-Based Approaches to Indirectly Constrain the Concentrations, Trends, and Variations of Tropospheric Hydroxyl Radical (OH), EGUsphere [preprint], https://doi.org/10.5194/egusphere-2024-2331, 2024.

Michel SE, Lan X, Miller J, Tans P, Clark JR, Schaefer H, Sperlich P, Brailsford G, Morimoto S, Moossen H, Li J. Rapid shift in methane carbon isotopes suggests microbial emissions drove record high atmospheric methane growth in 2020-2022. Proc Natl Acad Sci U S A. 2024 Oct 29;121(44):e2411212121. doi: 10.1073/pnas.2411212121. Epub 2024 Oct 21. PMID: 39432794; PMCID: PMC11536133.

Pimlott, M.A., Pope, R.J., Kerridge, B.J., Latter, B.G., Knappett, D.S., Heard, D.E., Ventress, L.J., Siddans, R., Feng, W., Chipperfield, M.P., 2022. Investigating the global OH radical distribution using steady-state approximations and satellite data. Atmos. Chem. Phys. 22, 10467–10488, doi:10.5194/acp-22-10467-2022

Prather, M. J., Guo, H., and Zhu, X.: Deconstruction of tropospheric chemical reactivity using aircraft measurements: the Atmospheric Tomography Mission (ATom) data, Earth Syst. Sci. Data, 15, 3299–3349, https://doi.org/10.5194/essd-15-3299-2023, 2023.

Zhao, Y., Saunois, M., Bousquet, P., Lin, X., Berchet, A., Hegglin, M. I., Canadell, J. G., Jackson, R. B., Dlugokencky, E. J., Langenfelds, R. L., Ramonet, M., Worthy, D., and Zheng, B.: Influences of hydroxyl radicals (OH) on top-down estimates of the global and regional methane budgets, Atmos. Chem. Phys., 20, 9525–9546, https://doi.org/10.5194/acp-20-9525-2020, 2020.

Zhu, Q., Laughner, J.L., Cohen, R.C., 2022b. Combining Machine Learning and Satellite Observations to Predict Spatial and Temporal Variation of near Surface OH in North American Cities. Environ. Sci. Technol., 56, 11, doi:10.1021/acs.est.1c05636

---

## Author Comment (AC2)

**The Global Methane Budget: 2000-2020**

Saunois et al., ESSD, 2024

**Detailed Response to Anonymous Referee #1**

We acknowledge the referee for the time spent on reading and commenting on the paper. We thank Anonymous Referee#1 for the useful corrections and suggestions provided on the paper, which have helped clarify and improve the manuscript. Below are the responses (in black) to the comments (in italics, blue). Changes in the text follow each response in bold font.

*This is a most important paper that makes a major contribution not just in terms of scientific discovery, but more generally to the wider effort of protecting the well-being of our planet. The work is extremely well documented and carefully presented.*

*I strongly recommend publication.*

*That said, I do have a number of comments that the authors may wish to take into account for minor revisions prior to final publication.*

We thank warmly the anonymous reviewer #1 for the compliment and the consideration of the importance of this work for the community.

**General comments**

*Quantifying the rapidly changing global methane budget is of the highest importance in tracking climate forcing, and of major societal value, particularly in the Global Methane Pledge, now signed by 155 nations, and in the UN Framework Convention on Climate Change, signed by all nations.*

*This paper is the most recent update in a series of authoritative and very influential assessments of the global methane budget, that have been produced by the Global Carbon Project over the past decade. Throughout the series of papers, the work of this team has been detailed, very well-documented and of high academic quality.*

*The work in the paper is comprehensive and generally appears accurate, with thoughtful evaluations of uncertainties and gaps in knowledge. Once published, the budget assessment will be immediately valuable both to the very active scientific methane community, and in the wider climate debate.*

We thank the reviewer again for this valuable comment. Indeed, the last year of the budget, 2020, is the reference year of the Global Methane Pledge (GMP) objective. The values provided by the Global Methane Budget could be further used to assess the success of the GMP.

*However, I do have a number of specific questions that the authors may wish to consider.*

*In particular, it would be useful to add at least a comment on the need for isotopic balance. There is enough $^{13}C$ isotopic information available to provide an independent test of the budget, and thereby to help to constrain uncertainties.*

Indeed we acknowledge that the budget provided here does not include any isotopic constraints, while the isotopic signal is quite clear in the atmosphere. In the past we tested the consistency of the budgets (BU and TD) with the isotopic signal (Saunois et al (2017) for the 2000-2012 period- and we also did in Zhang et al (2022), an exhaustive assessment for the 2000-2017 period. A follow-up study is planned to address this issue based on the GMB estimates. Though it would have been better to integrate this in the main budget, we could not for the sake of time. However, isotopic studies are not as straightforward and face challenges. Indeed, the uncertainties and variabilities (regional mostly but also temporal) of the isotopic signatures remain a difficulty when it comes to attributing emissions to sectors, (especially anthropogenic versus natural emissions from microbial sources). We were probably a bit too optimistic in the 2020 paper in the projection that more inversions would include isotopes capabilities. To date only 2–3 groups have published global analysis with 3D atmospheric inversions using isotopic constraints over several years (Thanwerdas et al., 2024; Basu et al., 2022). Box models may be used to try to constrain the budget with the isotopic signal but that probably would not give the same results as a full 3D inversions assimilating both $CH_4$ and delta $^{13}C$-$CH_4$ data. We still have a goal to obtain more runs and hope to be able to report an ensemble of inversion results in the next release of GCP-CH4.

A sentence at the end of the 3rd paragraph of the introduction has been added: "**Another difficulty of the $CH_4$ budget lies in the necessity to also match the isotopic balance and in particular reflect the decreasing methane isotopic signal $^{13}C$ (Nisbet et al., 2016; 2019). The previous budgets were tested against the isotopic observations (Saunois et al., 2017) and follow an exhaustive assessment (Zhang et al., 2022). To date only a couple of atmospheric inverse systems are able to assimilate both CH4 mixing ratios and stable isotopic signal to retrieve fluxes at the global scale (Thanwerdas et al., 2024; Basu et al.,2022 ), but these systems still need improvements in terms of configuration set-up and computing time resources, in addition to characterization of source signatures and chemical kinetic effect (Chandra et al., 2024). We hope to be able to report isotopic constrained budgets in the coming years , or at least test the budget against the isotopic balance.**"

*In the concluding sections it might also be interesting to compare the findings to the official UNFCCC declarations and other important assessments (e.g. IEA), and perhaps to say a little more about the remarkable changes that have taken place in the 2020s.*

1. Regarding the official UNFCCC declarations, and other assessments (such as IEA for fossil emissions), not all countries regularly report their methane emissions to UNFCCC (yet). Few studies have investigated and compared the UNFCCC reporting against bottom-up (including IEA) and top-down approaches (for example Deng et al. (2022, 2024), Tibrewal et al. (2024)). Such comparison requires thorough investigation and are stand-alone studies that are done following the publication of the Global Methane Budget. We have added the following sentences near the end of the conclusion: "**Further investigation is needed in follow-up studies to (1) compare these results to the official UNFCCC declarations and to important assessments such as those of IEA) as done previously for example in Deng et al. (2022; 2024) or more specifically for fossil fuel emissions in Tibrewal et al. (2024) and (2) further discuss the trend and interannual variability of $CH_4$ sources and sinks at sectoral and regional scales**

as in Jackson et al. (2020, 2024), Stavert et al. (2021) or RECCAP-2 related publications (e.g., Petrescu et al., 2021; 2023; Lauerwald et al., 2023b), and discuss the compatibility of the budget against the atmospheric isotopic signal such as in Saunois et al. (2017). The next budgets will be critical to assess whether the Global Methane Pledge is successful and assess methane mitigation efforts.**"

2. Regarding the recent changes in the early 2020s, the budget does not cover the period. However, we already had a Section (Section 6) addressing some of the recent changes, providing insights of studies that have investigated this.

*As a matter of presentation, it would be a lot easier on the tired-eyed reader to break up some of these enormously long paragraphs! Also, as is inevitable in such a comprehensive paper, some sections seem a bity elderly and could do with updating and a few more recent references.*

Indeed some paragraphs were not updated as much as others. Using the useful comments provided by the three reviewers, we have addressed this issue, hoping that will satisfy the reviewer.

*Conclusion*

*This is a major paper of great importance to society as a whole. It is a credit to the authorship team and should be published after minor revision. The biggest weakness is the absence of any effort to use the isotopic constraints. Perhaps that should wait until the next update in a few years' time, but it should be considered or at least mentioned here, if only in a brief section.*

The isotopic section has been removed since the budget published in 2020 as we do not (yet) use these data directly to constrain the budget. Nevertheless, and as mentioned earlier, we added a few relevant sentences to the introduction and in the last results section (Section 6)

In the introduction, we have added a few sentences at the end of the third paragraph (see above).

in section 6, we have added the following text: "**The very high records of $CH_4$ growth rate over 2020-2022 have also been accompanied by a sharp decline in the stable isotopic signal, $\delta^{13}C_{CH4}$, driven by increased emissions from microbial sources such as those found in wetlands and freshwater, agriculture, and waste systems (Michel et al., 2024).**"

*Specific Comments*

*Abstract: Line 155.The abstract will be widely cited, and many will look at it but not read the paper in detail (sadly). Thus it would help if the abstract could include more source-specific numbers detailing inputs like fossil fuels, agriculture, biomass burning and also sink estimates. If that makes the abstract too long, then maybe move lines 130-135 (which are very important, but not necessarily needed in the abstract) to the introduction?*

Following this advice, we have removed former L148-153 (about the latitudinal distribution). We have added the global decadal estimate by category after the estimates of the tropospheric OH sink and total sink: "**The tropospheric loss of methane, as the main contributor to methane lifetime, has been estimated at 563 [510-663] Tg $CH_4$ yr$^{-1}$ based on chemistry climate models. These values are slightly larger than for 2000-2009 due to the impact of the rise in atmospheric methane, and remaining large uncertainty (~25%). The total sink of $CH_4$ is estimated at 633 [507-796] Tg $CH_4$ yr$^{-1}$ by the bottom-up approaches and**

at 554 [550-567] Tg CH$_4$ yr$^{-1}$ by top-down approaches. Though, most of the top-down models use the same OH distribution, which introduces less uncertainty to the global budget than is likely justified.

For 2010-2019, agriculture and waste contributed an estimated 228 [213-242] Tg CH$_4$ yr$^{-1}$ in the top-down budget and 211 [195-231] Tg CH$_4$ yr$^{-1}$ in the bottom-up budget. Fossil fuel emissions contributed 115 [100-124] Tg CH$_4$ yr$^{-1}$ in the top-down budget and 120 [117-125] Tg CH$_4$ yr$^{-1}$ in the bottom-up budget. Biomass and biofuel burning contributed 27 [26-27] Tg CH$_4$ yr$^{-1}$ in the top-down budget and 28 [21-39] Tg CH$_4$ yr$^{-1}$ in the bottom-up budget."

*L190 – GMP now has 155 signatures.*

It is true that this number changes over time. This has been updated – 158 as of October 2024

*L208 – maybe somewhere mention Anderson, D. C., et al. (2024). Trends and interannual variability of the hydroxyl radical in the remote tropics during boreal autumn inferred from satellite proxy data. Geophysical Research Letters, 51,*

This citation has been included in Section 7, complementing the reference to their technical note published in 2023. "**Recently, numerous efforts based on satellite data have been made to constrain OH distribution, variability and trends (e.g, Anderson, 2023; 2024 ; Pimlott et al. 2022 ; Zhao et al., 2023; Zhu et al., 2022).** "

*L251 – perhaps mention Yu, X, et al. (2023) A high-resolution satellite-based map of global methane emissions reveals missing wetland, fossil fuel, and monsoon sources. Atmospheric Chemistry and Physics 23): 3325-3346.*

This citation will not be at the right place here. Here we discuss the accuracy of the satellite measurements. The paper suggested by the referee uses TROPOMI to correct prior estimates of fluxes. This correction highly depends on the prior used and also on the constraint observations.

Though the paper has been cited as an example of the use of TROPOMI in Section 7.

*L306 – EDGAR v6? – See also L424.v7, and now v8. I guess in a big synthesis exercise like this it is hard to keep up with changes to inventories. Maybe mention new work by one of the co-authors. Crippa, M., et al. (2023) Insights on the spatial distribution of global, national and sub-national GHG emissions in EDGARv8. 0. Earth System Science Data Discussions 2023: 1-28.*

Indeed, since the start of the study, the EDGAR team has released updates of their inventories. We started to use EDGARv6 as prior for the inversions. We then included EDGARv7 ending in 2021 for the BU budget - which was enough as the study spanned 2000-2020. EDGARv8 and now EDGAR_2024_GHG were released later on and provided data up to 2022 and 2023 respectively. This is used in Section 6 where we discuss the post 2020 period. We included the following statement in section 2.2 after mentioning the use of EDGARv6 as prior for the inversions: "**Though EDGARv7 (EDGAR, 2022 ;Crippa et al., 2023) spanning until 2021 was then released, and was used for the bottom-up budget. EDGARv8 (EDGAR, 2023; Crippa et al., 2023) spanning until 2022 and released in 2024, was used in Section 6 to discuss the post 2020 methane budget.** "

*L327 – good to have N2O and CH4 regions comparable.*

This is a comment.

*L348 – maybe could add a few general words on how to assess heavily human-dominated floodplain deltas like the Nile and Ganges and the Yangtse and Mekong (Tonlé Sap) floodplains.*

This assessment is indeed quite challenging, and separating natural from anthropogenic in those regions is challenging. We have added the following comment to acknowledge this issue: "**Separating natural from anthropogenic sources could be quite challenging, especially over regions where sources overlap, as over heavily human-dominated floodplain deltas for example**"

*L371 – soil uptake – major uncertainty.*

We consider this only as a comment; the uncertainty on the soil sink will be further discussed in its specific subsection (Sect. 3.3.4).

*L424 – see earlier comment on EDGAR v 6,7,8. (L306).*

This has been addressed above.

*L439 – abatement coefficient - how much uncertainty does this introduce?*

Unfortunately, emission inventories come rarely with uncertainty estimates; and in particular do not estimate the uncertainties introduced by each input variable, in particular activity data that have often no alternative dataset. However, comparing different bottom-up inventories and looking at the range of their estimates could inform on the impact brought by the methodology used to build each inventory. This abatement coefficient is very sector and region specific and looking into this would require deep analysis of the inventories at national scale for example, which is out of the scope of this study.

*L507 – Methane mitigation challenge – maybe mention Nisbet et al. 2020 Rev. Geophys.*

Indeed Nisbet et al. 2020 as well as Shindell et al. 2024 are good references here. They have been included.

*L510 – 'well within the range of scenarios' –– the paper seems to suggest the recent climb is OK and expected, so we can be complacent. But the past few years have been much more like 5-8.5. Scary! See also fig. 6 in Nisbet et al Phil Trans Roy Soc A 379 (2021): 20200457.*

Sure this is not the message we wanted to draw. This has been rephrased to: "**observations of global CH$_4$ concentrations fall well within the range of scenarios in absolute values but their trend over the past few years is closest to those of scenario SSP5-8.5** "

*L530 – FF 34% - maybe a comment on how this is changing?*

This contribution is a bit increasing over time - considering the mean estimate of the bottom-up approches. This study shows that FF contributed a bit less than 32% (31,5%) on average in 2000-2009, a bit less than 34% (33,5%) and a bit more than 34% (34.4%) for 2020. We have modified the text as follows:  "**The sector accounts on average for 34% (range 31-42%) of total global anthropogenic emissions in 2010-2019.  This contribution has slightly increased from 32% on average in 2000-2009.**"

*L563 – underground fires – real guesswork.*

This is true. This value seems to be the same for years in EDGAR inventories across any version of EDGAR, despite the increase in coal exploitation in China for example. This estimate is largely unknown but we lack information to better discuss this. Though we have added a comment on this value: "**An additional assumed very small source corresponds to fossil fuel fires, which are mostly underground coal fires. This source is estimated at around 0.15 Tg yr$^{-1}$ in EDGARv7, though this value remains the same across EDGAR versions and all years despite the changes in coal production, which could influence this estimate. However, to date, insufficient data is available to better estimate this largely unknown source.**"

*L592-594 Low fracking losses - My gut feeling is to concur with this, but it might be a bit higher. The reference list is pretty old here - maybe cite more recent studies both by the EDF supported and other teams. Examples include Zhang, Y, et al. Science advances 6 (2020): eaaz5120. Or Li,  et al.  Science of The Total Environment 912 (2024): 169645. Or Williams, et al. EGUsphere 2024 (2024): 1-31. Shen, Lu, et al.  Nature Communications 14 (2023): 4948.*

We are thankful to the reviewer for the list of recent studies to include here.

The former lines 592-594 addressed the emission factors of the Oil and gas facilities. Zhang et al., 2020 suggest that emissions represent 3.7% of the gross gas extracted from the Permian. Li et al. (2024) do not estimate an emission factor but reported more on the underestimation of emissions Alberta in the inventories in Alberta. Williams et al. (2024) discuss the loss rate depending on the production gross. Finally, Shen et al. (2023) compared the estimate of UNFCCC and other BU inventories to TROPOMI based atmospheric inversions over 24 countries and found large underestimation, though they do not specify emission factors. The suggested studies looking at the underestimation of the emissions have been used later in this section.

Looking at the literature and considering the key point we wanted to highlight the sentence (former lines 592-594) has been rephrased as follows: "**The latest studies tend to infer emission factors from the oil gas production chain of about 1% to 2% (e.g., Schneising et al., 2020; Varon et al., 2023; Zhang et al., 2020), but loss rate could be has high as more than 10% in low producing well sites (Omara et al., 2022, Williams et al., 2024 ).**"

*L611 – 613 – again, elderly references in a fast moving field…..but Lavaux ref is good.*
This has been modified to: "**Most recent studies (e.g., Zhang et al., 2020 ; Shen et al., 2023; Li et al.; 2024, Tibrewal et al., 2023; Sherwin et al., 2024) still suggest that the methane emissions from oil and gas industry are underestimated by inventories, industries, and agencies, including the USEPA and UNFCCC reporting. "**

*L624 – extra 8 Tg….perhaps – but ultra-emitters tend to be short lived.*

This is true. It would be interesting to analyze the trend of the emissions from those super-emitters, but we do not know of any study that looked into this.

*L640-642 90/10 ratio of eructation vs flatulence – give a reference for this? Or Johnson?*

We found a similar value in Hill et al. (2016), and included this reference.

*L669 – enteric 114-124 Tg – my instinct is that the uncertainty is wider than this?*

Indeed, this range is the min-max from the estimates and does not include the uncertainty of the individual estimate, which would be larger than this. We have added the following sentence : **"It is worth recalling here that the ranges provided in this study correspond to the minimum-maximum of the existing estimates and do not include the uncertainty of the individual estimate; these uncertainties could be larger than the range proposed here."**

*L737 and L769 – maybe cite some direct measurements? For example Barker, P., et al. (2023) Airborne measurements of fire emission factors for African biomass burning sampled during the MOYA campaign. ACP 20s 15443-15459.*

We have modified the text as follows: "**Small fires associated with agricultural activity, such as field burning and agricultural waste burning, are often not        detected by moderate resolution remote sensing methods and are instead estimated based on cultivated area or through in-situ measurements such as dedicated airborne campaigns (e.g., Barker et al., 2023).**"

*L834 - 842 – 853 There's a problem in depending on land surface models. Note the bad failure of land surface models in tropical African and Bolivian wetlands – see especially Fig 8 and discussion in Shaw J.T. et al. (2022) Large methane emission fluxes observed from tropical wetlands in Zambia. Global Biogeochemical Cycles 36: e2021GB007261. Also see France et al. France, J. L., et al. (2022) Very large fluxes of methane measured above Bolivian seasonal wetlands. PNAS 119 e2206345119. Also note models in Zhang, Z. et al. Nat. Clim. Change https://doi.org/10.1038/s41558-023-01629-0 (2023) and comment by Nisbet 2023 Nature Climate Change 13 421-422.*

Indeed LSMs still suffer from many uncertainties. To acknowledge this, we have included the following sentences, citing the suggested references: "**However, large uncertainty remains in both spatial and temporal emission distributions, especially over tropical wetlands where data are lacking to evaluate the models but are nevertheless a key region for climate feedbacks (Nisbet et al., 2023; Zhang et al., 2023). Direct measurement campaigns and remote sensing are providing key insights for where to improve the land surface models (e.g., France et al., 2022; Shaw  et al., 2022).**"

*L957 – note that Shaw et al (section 3.1.1) found low emissions over the wide shallow lake Bangweulu, even though the lake is shallow, warm and there is a lot of organic input – instead, the high emissions were dominated by the wetlands beside the lake. Although I have often observed warm afternoon ebullition events (sometimes dramatic) in lakes and open water bodies,  as a general but intuitive impression (i.e. personal observation not backed up by quantitative study) my 'feel' is that where water is more than about 3 to 4m deep and where plant stems (e.g. reeds,*

*papyrus) do not project to entrain methane up from mud to air in the sap, then lake emissions are fairly low. Any bubbles are taken up in the aerobic conditions in oxic upper water.*

The estimates produced here are from studies that have exhaustively collected data from the literature, or produced data using a process-based model trained on such literature data,      more than a general intuitive impression (these process-based models include an explicit representation of gas transport and bubble dissolution). The general consensus in the oceanographic literature is you need a depth of >100 meters for the methane in a bubble to equilibrate with surrounding waters. The total and ebullitive CH4 fluxes reported here for global lakes are thus based on the best empirical evidence and scientific knowledge currently available.  Of course, significant uncertainties remain (due to limited observations and incomplete understanding) and we have conveyed this by expanding on lines 995-996 of the original manuscript (see next comment).

*L975 – see comment on lakes – these numbers are pretty high and based on few observations - may be higher than reality.*

We agree that the number of observations remains limited. Yet it is important to note that if our assessment is based on a few studies/models, these efforts try to use all available observations. We nevertheless recognize that the observation ensemble might not fully capture the wide diversity of processes and environmental drivers impacting lake and reservoir CH4 emissions and have thus have expanded on uncertainties and directions for future research (line 995-96 of original ms.): " **Finally, for all inland water systems a greater scrutiny of the limiting  factors (including the impact of ice-cover, stratification of the water column and seasonality)      CH4 production, consumption and transport pathways is needed. In addition, a better understanding of the climatic, environmental and geomorphological controls on key CH4 processes (e.g., sedimentary diffusive and ebullitive production, bubble dissolution, CH4 oxidation) on the large-scale remains critically needed. For instance, the consistently lower global emissions determined by the process-based model of Zhuang et al. (2023) compared to observations, suggest that current datasets are too limited to fully capture the spatio-temporal variability in CH4 dynamics and their key control factors, possibly leading to biased-high estimates.** "

*L1020 – run-off of agricultural Nitrogen fertilizer and manure to nearly wetlands?*

The sentence has been modified: "**It is also clear that the cultural eutrophication of natural lakes driven by run-off of agricultural nitrogen fertilizer and manure is augmenting CH4 emissions (DelSontro et al., 2018; Li et al., 2021), with shallow lakes particularly likely to experience eutrophication (Qin et al., 2020).**"

*L1068. Interesting discussion of this vexed problem. Good.*

Thank you.

*L1084 – may still be too high? Very much guesswork!*

We agree that the total natural wetland (vegetated+inland water) methane emissions flux is still likely too high. The bottom-up method maintains fidelity to each of the emitting sectors, and the issue of double counting (leading to the high numbers) is gradually being reduced with each assessment. The lessons learned are carried forward each time, with this budget providing better distinction for wetlands and inland waters, and including a lateral flux correction term. The wetlands team will soon be looking at lessons learned from this budget to provide improved

estimates for the next CH$_4$ budget. The inland water team will need to continue expanding their observational database. Furthermore, these observations should help inform a wider range of (newly-developed) models resolving the spatial and temporal variability in inland water fluxes on the global scale.

*L1116 – The Petrenko group's evidence is compelling – theirs are very high quality measurements. Maybe also cite Dyonisius, Michael N., et al. (2020) Science 367(2020): 907-910. I'm not convinced by the Thornton et al (2021) arguments.*

There are still contradictory estimates of geological emissions of methane. Dyonisius et al. (2020) also suggests low geological (14C- free CH4 emissions) over the time period spanned by ice-core record (between 0 and about 15 Tg/year). This paper has been added in the discussion: "(**Dyonisius et al. (2020) also suggest a low range of geological emissions over the last deglaciation period and for the late Holocene (0-10 Tg CH$_4$yr$^{-1}$)**)".

We believe we tried to remain impartial. No improvement or reconciliation has been made since Saunois et al. (2020). Thornton et al. (2021) is still the latest paper comparing and discussing the issue. Different individual and independent studies suggest significant geological emissions at regional scale. We have further acknowledged this in the following sentence: "**This discrepancy highlights another main unresolved uncertainty in the methane budget and calls for further investigations to reconcile the different estimates and reduce the uncertainty on geological emissions.** "

*L1127 – typo – S**c**hwietzke.*

This has been corrected

*L1133 – Termites – interesting how little work has been done since Pat Zimmerman's work in the 1980s. The 10 Tg estimate looks like a placeholder! Maybe cite Chiri, E., et al (2020). Termite mounds contain soil-derived methanotroph communities kinetically adapted to elevated methane concentrations. The ISME journal, 14, 2715-2731. And Chiri, E, et al. Termite gas emissions select for hydrogenotrophic microbial communities in termite mounds. PNAS118 (2021): e2102625118.*

Indeed, the scaling up of termite emissions still lacks a sufficient number of studies to cover extensively all the uncertainty areas underlined in the introductory section on termites in this paper. Most advances concern aspects that were investigated with laboratory analyses, like the increase of data on CH4 emissions factors in vitro and also most recent interesting studies of biochemistry/genetic like the ones proposed by the reviewer by Chiri et al., which provided new insight on the complexity of the mound environment and termite- microorganisms connections. This surely is a frontier aspect that is very challenging. However a fundamental big gap of knowledge on the basics of termites ecology remains, related to the variability of termites groups, nests typologies, biodiversity distribution of the different populations (and hence trophic modes and nest types), among different ecosystems and across continents, as well as our ability to a more precisely estimate the total termite biomass of an ecosystem. This task would require a challenging      field monitoring approach at large scale, with significant international collaboration over continents among groups of entomologists, ecologists, experts of flux emissions and more.

We have included the citations in the following text : " **The species of each family and subfamily of the two major groups of lower and higher termites, listed by Sugimoto et al. (1998) were associated with EF values based on emissions from in-vitro experiments as reported by Sanderson (1996) and Eggleton et al. (1999), to which a correction factor (cf$_{MOUND}$) of 0.5 (Nauer et al., 2018; Chiri et al., 2020; 2021) was applied in order to take into account the mound effect on the CH$_4$ produced by termites, once inside the nest.** "

*L1205 – I strongly suspect this 2 Tg number is a serious underestimate. Even in cattle monocultures there are dense populations of small nocturnal antelope (in Africa) and small deer (in EU and N America), in bush and scrub forest, etc as well as larger tree clumps and forests. For Africa see for example Hempson, G.P., Archibald, S. and Bond, W.J., 2017. The consequences of replacing wildlife with livestock in Africa. Scientific reports, 7, p.17196.*

This value could seem underestimated, though not many global studies exist. We have modified the text as follows, "**However, the estimate of 1-3 Tg CH$_4$ yr$^{-1}$ seems underestimated when considering that Hempson et al. (2017) found actual CH$_4$ emissions from African wild life alone to be around 9 Tg CH$_4$ yr$^{-1}$ but without discussing the uncertainty of this value. As a result, high uncertainty remains and recalls the need for further investigation of this natural source of CH$_4$.**
**Based on these findings and waiting for further global estimates, the range adopted in this updated CH$_4$ budget is 2 [1-3] Tg CH$_4$ yr$^{-1}$ (Table 3).**"

*L1216 – gas plumes – see Fig 1 in Westbrook, Graham K., et al. (2009) Escape of methane gas from the seabed along the West Spitsbergen continental margin. Geophysical Research Letters 36. Plumes disappear quickly and even very large plumes don't manage to rise through more than about 250m of water.*

Some studies show evidence of gas bubble plumes searching the surface, even at high depth, depending on the intensity of the events. We have added the reference (and other) and modified the sentence : " **Gas bubble plumes generally reach the atmosphere in relatively shallow waters (<400 m) of continental shelves depending on the intensity of the events (e.g., Westbrook et al. 2009); however, massive deep-water seepage events could contribute significant amount of CH4 to the atmosphere even from depths > 1000m (e.g., e.g., Schmale et al., 2005; Greinert et al., 2006, Solomon et al., 2009).** "

*L1275 – the in-situ oceanic source from phosphonates is real. It may be small, but it's essentially unknown. This 5 Tg number looks a bit like the 'placeholder' in Cicerone and Oremland 1988.*

The 5 Tg is mostly based on recent studies: Rosentreter et al. (2023) and Weber et al. (2019). For such sources, the uncertainty remains large with at least +/- 50%, as shown with the 3-10 Tg yr$^{-1}$ range. We have added a reference to Resplandy et al. (2024) that discusses the possible contribution of sedimentary methane vs. direct aerobic in situ production in the surface waters (including the methylphosphonate pathway). Further studies are needed to better estimate the small coastal and oceanic emissions and their source contributions at the global scale as well as reduce uncertainties. The sentence has been revised as follows: "**Overall, these marine biogenic emissions are sustained by a mixture of sedimentary production and**

**in-situ production in the sea-surface layers (including the methylphosphonate pathway) (e.g., Karl et al., 2008; and Repeta et al., 2016; Resplandy et al., 2024).** "

*L1314 – see also L 1116 – seems too high given the Petrenko team's results.*

Indeed, this is related to the above estimate. However, no other studies provided detailed estimates of geological emissions by pathways/locations. To acknowledge this, we have included the following sentence: "**While we use the estimate by Etiope and Schwietzke (2019)   , we acknowledge that high uncertainty remains and others studies suggest lower ranges of emissions based on radiocarbon ($^{14}$C-CH$_4$) data in ice cores (e.g., Hmiel et al., 2020). The suggested estimate may overestimate this source and be part of the top-down bottom-up discrepancy as discussed in Section 5.1.2.**"

*L1374 – does this also include features like the Yamal blowout collapse structures? Bogoyavlensky, Vasily, et al. (2021) Permanent gas emission from the Seyakha Crater of gas blowout, Yamal Peninsula, Russian Arctic. Energies 14: 5345.*

This section reports permafrost direct emissions, which are difficult to estimate as stated in the text and in Hugelius et al. (2023). Individual formation exists as for the Yamal one but emission estimates are rare. Actually, we did not find quantification of the emissions of the Yamal structure in Bogoyavlensly et al (2021). Further investigation would be needed to provide a reporting of the existing thermokarst structures due to permafrost thaw associated with their emission. However, such inventory will probably be quite difficult to create.

*L1381 – Nisbet et al 2009 demonstrated transport of methane via sap – i.e. plant methane comes from underlying **an**aerobic soil methanogens, not in situ aerobic processes in the leaf. In other words the plants were acting as 'straws' (see Line1385). UV-caused emissions are tiny (Bloom et al.). Note the important work of Gauci et al and Pangala et al. (see Line 1388): are their emissions encapsulated in the other flux categories (Line 1401).*

To acknowledge this we have added Nisbet et al. 2009 in the sentence in former line 1385: "**Second, and of clearer significance, plant stems act as "straws", drawing up and releasing microbially produced CH$_4$ from anoxic soils (Cicerone and Shetter, 1981; Rice et al., 2010; Nisbet, et al. 2009).** "

In order to update this section, we have added the following sentence to mention that trees could also act as methane sink: "**Recently, field-work suggested that trees may also act as a CH$_4$ sink (Machacova et al., 2021 ; Gorgolewski et al., 2023 ; Gauci et al., 2024)**".

*L1433 some of this OH section is a bit elderly. Maybe add some new references?  E.g. (see also above L208) Anderson, D. C., et al. (2024); and  Stevenson, David S., et al. (2020) Trends in global tropospheric hydroxyl radical and methane lifetime since 1850 from AerChemMIP. Atmospheric Chemistry and Physics 20 12905-12920. Or Naus, S., et al (2021). A three-dimensional-model inversion of methyl chloroform to constrain the atmospheric oxidative capacity. Atmospheric Chemistry and Physics, 21, 4809-4824.*

It is true that some old citations have been kept. We have significantly updated this section and the discussion to highlight the most recent studies on the OH issue. Other small parts of

the text have been marginally changed. In the following, we include the most important changes made to the text: "**OH concentrations and their changes can be sensitive to climate variability (e.g., Nicely et al., 2018; Anderson et al., 2020), biomass burning (e.g., Anderson et al., 2024), and anthropogenic emissions of precursors (Peng et al., 2022; Stevenson et al., 2020). OH distributions calculated by chemistry climate models show large regional differences and various vertical profiles (Zhao et al., 2019). OH changes present also regional differences over the long term (Stevenson et al., 2020). Despite large regional changes, the global mean OH concentration was suggested to have changed only slightly from 1850 to 1980, but followed by strong (9 %) increases up to the present day (Stevenson et al., 2020). This increase simulated by models over 2000-2015 are however not in agreement with observation-based approaches (Thompson et al., 2024; Patra et al., 2020; Nicely et al., 2018; Rigby et al., 2017; Turner et al., 2017) where OH decreases or remain constant over the period. CCMI and CMIP6 models show OH interannual variability ranging from 0.9% to 1.8% over 2000-2010 (Table S4), in agreement with the values of IAV derived from some observationally constrained studies (e.g., Thompson et al., 2024; Montzka et al., 2011) but lower than value deduced from methyl chloroform measurements (Patra et al., 2021; Naus et al., 2021).** "

*L1459-1461 and 1465-1466 – maybe these numbers should be in the abstract?*

As suggested we have added the value of the tropospheric loss in the abstract with the following sentences: "**The tropospheric loss of methane, as the main contributor to methane lifetime, has been estimated at 563 [510-663] TgCH4 yr-1 based on chemistry climate models. These values are slightly larger than for 2000-2009 due to the impact of the rise in atmospheric methane, and still present large uncertainty (~25%).** "

*L1504 – interesting this number is low – the Cl sink has a strong isotopic leverage, so if the low number is correct that leverage is small.*

Indeed, recent studies tend to lower tropospheric concentrations of chlorine, leading to a smaller impact on the total methane sink than previously thought. However, the isotopic leverage of chlorine in the troposphere is still meaningful for methane isotopic modeling as discussed in Strode et al., 2020 and Thanwerdas et al., 2022b (See previous line 1500-1502).

*L1507 soil uptake. This is a major uncertainty in the budget because there are so few measurements from tropical seasonal rainfall woodlands and savannas. It's a huge unknown, linked to the termite unknown. (see Chiri papers L1133). I suspect the already wide uncertainty range on L1535 and L1543 is low!*

The authors fully agree with this comment. Studies are rare: modelling ones may miss some processes and not enough observations are available to validate and calibrate them. We also acknowledge that this section has been marginally modified due to a lack of novel global estimates. We have added the following two sentences near the end of the section: "**Indeed, Murgia-Flores et al. (2021) estimated that the global soil-uptake doubled between 1900 and 2015 and could further increase due to enhanced diffusion of CH₄ into soil as a result of increases in atmospheric CH₄ mole fraction. Further investigation of the soil uptake is required to better constrain this process at the global scale while it is highly dependent on**

**local scale microbial activity and environmental conditions (e.g., D'Imperio et al., 2023; Fest et al., 2017).** "

*L1546 – lifetime definition – burden/sink - maybe make clear that this is not the same as the 12 yr perturbation lifetime cited in some IPCC chapters. Media people get very confused between the two sorts of lifetime. Maybe cite Prather here (as well as on L1619)*

The text at former line 1619 has been removed, first to avoid confusion from another reviewer and second because it did not fit the discussion we intend to have. Regarding the comment for L1546 - start of the section on methane lifetime- We have included a clear sentence on the difference between lifetime and perturbation time: "**This value is different from what is called perturbation lifetime. Perturbation lifetime is used to determine how a one-time pulse emission may decay as a function of time as needed for the calculation of Global Warming Potentials (GWPs), and as a result is related to a theoretical concept. For $CH_4$, the corresponding perturbation lifetime that should be used in the GWP calculation is 11.8 ± 1.8 years (Forster et al., 2021). In this section, we discuss the global atmospheric lifetime (also called 'burden lifetime' or 'turnover lifetime') that characterizes the time required to turn over the global atmospheric burden and is defined by the burden divided by the removal flux.** "

*L1683 608 Tg – see also L 1836. maybe this number should be in the abstract also? It's ~10% higher than 2010's 554 Tg. See also Nisbet et al 2023 for discussion of post 2020 changes.*

We have added the following sentence to the abstract: "**The 2020 emission rate is the highest of the period and reaches 608 [581-627] Tg $CH_4$ yr$^{-1}$, which is 12% higher than the average emissions in the 2000s.**"
The article from Nisbet et al. (2023) was already cited in Section 6 discussing post 2020 changes.

*L1824 – I suspect this 3 Tg number is a major underestimate!*

Indeed, we recall this here as follows: "**Wild animal and permafrost maps do not exist and are missing from the calculation, leading to at least 3 Tg $CH_4$ yr$^{-1}$ of discrepancy. However, as aforementioned (Sections 3.2.5 and 3.2.7) this 3 Tg $CH_4$ yr$^{-1}$ estimate is probably underestimated in the bottom-up budget.**"

*L1883 regional budgets – maybe mention how these numbers contrast with recent studies like Yu, Xueying, et al. (2023) A high-resolution satellite-based map of global methane emissions reveals missing wetland, fossil fuel, and monsoon sources. Atmospheric Chemistry and Physics 23: 3325-3346.*

Comparing different studies requires covering similar periods and regional scale. Yu et al. (2023) for example provide emission estimates over 2018-2019, the year with the highest emissions from our decadal period 2010-2019, and provide national estimates while we present values for a mix of large countries (Russia, the USA, Brazil, China..) and regions (South-East Asia, ..). For example, for China we found 57 [37-72] Tg $CH_4$ yr$^{-1}$ for 2010-2019, while they found 60 Tg $CH_4$ yr$^{-1}$ for 2018-2019. For the USA, we found 38 [32-46] Tg $CH_4$ yr$^{-1}$ for 2010-2019, while they found 44 Tg $CH_4$ yr$^{-1}$ for 2018-2019. The conclusion would be that we found similar estimates, considering the range of uncertainty that is larger in the Global Methane

Budget as we use an ensemble of top-down systems and not a single one; even if the study includes sensitivity tests.

We prefer providing the ensemble estimates and let further studies investigate the differences, if any, between studies.

*L1983 ? Mention the current d13C(CH4) isotopic plunge – and concerns (e.g. Nisbet et al 2023).*

The following sentence has been added: "**The very high values of CH₄ growth rate over 2020-2022 have also been accompanied by a sharp decline in the stable isotopic signal, $\delta^{13}C_{CH4}$, which suggests that this recent increase of growth rate is at least partly explained by increased emissions from microbial sources such as wetlands, inland waters, agriculture and waste (Nisbet et al., 2023; Michel et al., 2024).** "

*L2076 – FINALLY!!! A mention of isotopes!!! That's the big missing factor in the whole study!!!! The isotopes must balance – if they don't the budget needs to be re-examined. Obviously it is too late to add a full isotopic analysis to this paper, but the need for isotopic constraint surely should be mentioned. I would suggest that a short section on isotopes should be added here at this stage in the paper, including maybe a brief one-paragraph attempt at balance, and a promise to do something about it next time.*

We fully agree that the budget must agree with the isotopic signal. This statement is also true for any top-down studies constrained by CH₄ only (even if based on TROPOMI as in Xueying et al. (2023) mentioned earlier by the reviewer). We have included the following sentence in the introduction: "**Another difficulty of the CH₄ budget lies in the necessity to also match the isotopic signal and in particular reflect the decreasing methane isotopic signal $^{13}C$ (Nisbet et al., 2016; 2019). The previous budgets were tested against the isotopic observations (Saunois et al., 2017) and followed an exhaustive assessment (Zhang et al., 2022). To date only a couple of atmospheric inverse systems are able to assimilate both CH4 mixing ratios and stable isotopic signal to retrieve fluxes at the global scale (Thanwerdas et al., 2024; Basu et al., 2022), but these systems still need improvements in terms of configuration set-up and computing time resources, in addition to characterization of source signatures and chemical kinetic effect (Chandra et al., 2024).. We hope to be able to report isotopic constrained budgets in the coming years.**"

We are working on examining the 2024 version budget against the isotopic signal through 3D transport forward modelling. The results are too preliminary to be included here but will be part of a coming paper, hopefully next year. This paper will also aim at establishing a protocol to evaluate the top-down (and bottom-up budget) against the atmospheric signal of CH4 and $\delta^{13}C_{CH4}$.

To recall that most top-down studies are only based on CH4 constraints only we have included the following sentence in Section 6 - formerly L1986: "**However, it is worth noting that almost all published top-down studies aforementioned include constraints only on CH₄, and do not discuss the consistency with the atmospheric isotopic signal.** "

In the conclusion section we had the following sentences (see previous comment): "**Further investigation is needed in follow-up studies to (1) compare these results to the official UNFCCC declarations and to important assessment (as those of IEA) as done for example in Deng et al. (2022; 2024) or more specifically for fossil fuel emissions in Tibrewal et al. (2024)**

**and (2) further discuss the trend and interannual variability of CH₄ sources and sinks at sectoral and regional scales as in Jackson et al. (2020, 2024), Stavert et al. (2021) or RECCAP-2 related publications (e.g., Petrescu et al., 2021; 2023; Lauerwald et al., 2023b), and discuss the compatibility of the budget against the atmospheric isotopic signal such as in Saunois et al. (2017). The next budgets will be critical to assess whether the Global Methane Pledge is successful and assess methane mitigation efforts.**"

*L2190 – maybe mention Nisbet et al 2023 here and/or in Line 2194?*

The citation has been added at the two mentioned lines (former L2190 et L2194).

*TABLES – these are extremely valuable and will be very heavily used. The Table 'callouts' could maybe be better integrated in the text and perhaps need to have more mentions: they'll be very prominent on publication.*

As suggested, we have included more call-out to the Tables along the text.

*FIGURES – these are excellent and again the 'callouts' maybe need some more prominence in the text. For Fig 2 see comparison with Nisbet et al 2021 Phil Trans. In Fig 3 wetlands is heavily biassed by the Congo peatlands – but somewhat misses/downplays the upper Congo wetlands and also downplays big emitters like the Pantanal and Mamore R. and the Sudd, despite lots of recent evidence for big emissions. I accept these reagions are very difficutl to study by remote sensing, as cloud cover is so pervasive in wet season. I've flown over many and seen only cloud! Also I'm very sceptical of the 'lowish' biomass burn in India where crop waste burning is horrendous (or is crop waste burning classed under 'waste'?) Also India has a huge coal mining industry and giant landfills. Fig 5 'Geological' seems improbable that central Asia is less prominent than Romania (e.g. see MEMO2 / ROMEO measurements).*

As suggested, we have included more call-out to the figures along the text. Further investigation would be needed to discuss the regional distribution of the sources and sinks, and their changes over time. We feel this is out of the scope of this study.

**References:**
Anderson, D. C., B. N. Duncan, J. M. Nicely, et al. J. Liu, S. A. Strode, and M. B. Follette-Cook. 2023. Technical note: Constraining the hydroxyl (OH) radical in the tropics with satellite observations of its drivers – first steps toward assessing the feasibility of a global observation strategy Atmospheric Chemistry and Physics 23 (11): 6319-6338, doi:10.5194/acp-23-6319-2023

Anderson, D. C., Duncan, B. N., Liu, J., Nicely, J. M., Strode, S. A., Follette-Cook, M. B., Souri, A.H., Ziemke, J.R., Gonzalez-Abad, G., and Ayazpour, Z. : Trends and interannual variability of the hydroxyl radical in the remote tropics during boreal autumn inferred from satellite proxy data. Geophysical Research Letters, 51, e2024GL108531, doi:10.1029/2024GL108531 , 2024

Basu, S., Lan, X., Dlugokencky, E., Michel, S., Schwietzke, S., Miller, J. B., Bruhwiler, L., Oh, Y., Tans, P. P., Apadula, F., Gatti, L. V., Jordan, A., Necki, J., Sasakawa, M., Morimoto, S., Di Iorio, T., Lee, H., Arduini, J., and Manca, G.: Estimating emissions of methane consistent with atmospheric measurements of methane and δ13C of methane, Atmos. Chem. Phys., 22, 15351–15377, https://doi.org/10.5194/acp-22-15351-2022, 2022.

Chandra, N., P. K. Patra, R. Fujita, L. Höglund-Isaksson, T. Umezawa, D. Goto, S. Morimoto, B. H. Vaughn, T. Röckmann: Methane emissions decreased in fossil fuel exploitation and sustainably increased in microbial source sectors during 1990–2020, Comm. Earth Environ., 5, https://doi.org/10.1038/s43247-024-01286-x, 2024

D'Imperio, L., Li, BB., Tiedje, J.M. et al. Spatial controls of methane uptake in upland soils across climatic and geological regions in Greenland. Commun Earth Environ 4, 461 (2023). https://doi.org/10.1038/s43247-023-01143-3

Dyonisius, M. N., Petrenko, V. V., Smith, A. M., Hua, Q., Yang, B., Schmitt, J., Beck, J., Seth, B., Bock, M., Hmiel, B., Vimont, I., Menking, J. A., Shackleton, S. A., Baggenstos, D., Bauska, T. K., Rhodes, R. H., Sperlich, P., Beaudette, R., Harth, C., Kalk, M., Brook, E.J., Fisher, H., Severinghaus, J.P., and Weiss, R. F.: Old carbon reservoirs were not important in the deglacial methane budget, Science, 367(6480), 907–910. https://doi.org/10.1126/science.aax0504, 2020

Fest, B. J., Hinko-Najera, N., Wardlaw, T., Griffith, D. W. T., Livesley, S. J., and Arndt, S. K.: Soil methane oxidation in both dry and wet temperate eucalypt forests shows a near-identical relationship with soil air-filled porosity, Biogeosciences, 14, 467–479, https://doi.org/10.5194/bg-14-467-2017, 2017.

France J.L., M.F. Lunt, M. Andrade, I. Moreno, A.L. Ganesan, T.Lachlan-Cope, R.E. Fisher, D. Lowry, R.J. Parker, E.G. Nisbet, A.E.Jones, Very large fluxes of methane measured above Bolivian seasonal wetlands, Proc. Natl. Acad. Sci. U.S.A. 119 (32) e2206345119, https://doi.org/10.1073/pnas.2206345119 (2022).
Gauci, V., Pangala, S.R., Shenkin, A. et al. Global atmospheric methane uptake by upland tree woody surfaces. Nature 631, 796–800 (2024). https://doi.org/10.1038/s41586-024-07592-w

Gorgolewski, A. S., Caspersen, J. P., Vantellingen, J. & Thomas, S. C. Tree foliage is a methane sink in upland temperate forests. Ecosystems 26, 174–186 (2023).

Hempson, G.P., Archibald, S. & Bond, W.J. The consequences of replacing wildlife with livestock in Africa. Sci Rep 7, 17196 (2017). https://doi.org/10.1038/s41598-017-17348-4
Machacova, K. et al. Trees as net sinks for methane ($CH_4$) and nitrous oxide ($N_2O$) in the lowland tropical rain forest on volcanic Réunion Island. New Phytol. 229, 1983–1994 (2021).

Li, H. Z., Scott P. Seymour, Katlyn MacKay, James S. Wang, Jack Warren, Luis Guanter, Daniel Zavala-Araiza, Mackenzie L. Smith, Donglai Xie, Direct measurements of methane emissions from key facilities in Alberta's oil and gas supply chain, Science of The Total Environment, Volume 912, 2024, 169645, ISSN 0048-9697, https://doi.org/10.1016/j.scitotenv.2023.169645.

Michel SE, Lan X, Miller J, Tans P, Clark JR, Schaefer H, Sperlich P, Brailsford G, Morimoto S, Moossen H, Li J.: Rapid shift in methane carbon isotopes suggests microbial emissions drove record high atmospheric methane growth in 2020-2022, Proc Natl Acad Sci ,121(44):e2411212121, doi: 10.1073/pnas.2411212121, 2024

Nisbet, E. G., Dlugokencky, E. J., Manning, M. R., Lowry, D., Fisher, R. E., France, J. L., Michel, S. E., Miller, J. B., White, J. W. C., Vaughn, B., Bousquet, P., Pyle, J. A., Warwick, N. J., Cain, M., Brownlow, R., Zazzeri, G., Lanoisellé, M., Manning, A. C., Gloor, E., Worthy, D. E. J., Brunke, E.-G., Labuschagne, C., Wolff, E. W., and Ganesan, A. L.: Rising atmospheric methane: 2007–2014 growth and isotopic shift, Global Biogeochem. Cy., 30, 1356–1370, https://doi.org/10.1002/2016gb005406, 2016.

Nisbet, E. G., Manning, M. R., Dlugokencky, E. J., Fisher, R. E., Lowry, D., Michel, S. E., Myhre, C. L., Platt, S. M., Allen, G., Bousquet, P., Brownlow, R., Cain, M., France, J. L., Hermansen, O., Hossaini, R., Jones, A. E., Levin, I., Manning, A. C., Myhre, G., Pyle, J. A., Vaughn, B. H., Warwick, N. J., and White, J. W. C.: Very Strong Atmospheric Methane Growth in the 4 Years 2014–2017: Implications for the Paris Agreement, Global Biogeochem. Cy., 33, 318–342, https://doi.org/10.1029/2018GB006009, 2019.

Nisbet, E.G. Climate feedback on methane from wetlands. Nat. Clim. Chang. 13, 421–422 (2023). https://doi.org/10.1038/s41558-023-01634-3

Omara M, Zavala-Araiza D, Lyon DR, Hmiel B, Roberts KA, Hamburg SP. Methane emissions from US low production oil and natural gas well sites. Nat Commun. 2022 Apr 19;13(1):2085. doi: 10.1038/s41467-022-29709-3. PMID: 35440563; PMCID: PMC9019036.

Pimlott, M.A., Pope, R.J., Kerridge, B.J., Latter, B.G., Knappett, D.S., Heard, D.E., Ventress, L.J., Siddans, R., Feng, W., Chipperfield, M.P., 2022. Investigating the global OH radical distribution using steady-state approximations and satellite data. Atmos. Chem. Phys. 22, 10467–10488, doi:10.5194/acp-22-10467-2022

Resplandy, L., Hogikyan, A., Müller, J.D., Najjar, R.G., Bange, H., W., Bianchi, D., Weber, T., Cai, W.-J., Doney, S. C., Fennel, K., Gehlen, M., Hauck, J., Lacroix, F., Landschützer, P., Le Quéré, C., Roobaert, A., Schwinger, J., Berthet, S., Bopp, L., Chau, T.T.T., Dai, M., Gruber, N., Ilyina, T., Kock, A., Manizza, M., Lachkar, Z., Laruelle, G. G., Liao, E., Lima, I. D., Nissen, C., Rödenbeck, C., Séférian, R., Toyama, K., Tsujino, H., and Regnier, P.: A synthesis of global coastal ocean greenhouse gas fluxes, Global Biogeochemical Cycles, 38, e2023GB007803. https://doi.org/10.1029/2023GB007803, 2024

Saunois, M., Bousquet, P., Poulter, B., Peregon, A., Ciais, P., Canadell, J. G., Dlugokencky, E. J., Etiope, G., Bastviken, D., Houweling, S., Janssens-Maenhout, G., Tubiello, F. N., Castaldi, S., Jackson, R. B., Alexe, M., Arora, V. K., Beerling, D. J., Bergamaschi, P., Blake, D. R., Brailsford, G., Bruhwiler, L., Crevoisier, C., Crill, P., Covey, K., Frankenberg, C., Gedney, N., Höglund-Isaksson, L., Ishizawa, M., Ito, A., Joos, F., Kim, H.-S., Kleinen, T., Krummel, P., Lamarque, J.-F., Langenfelds, R., Locatelli, R., Machida, T., Maksyutov, S., Melton, J. R., Morino, I., Naik, V., O'Doherty, S., Parmentier, F.-J. W., Patra, P. K., Peng, C., Peng, S., Peters, G. P., Pison, I., Prinn, R., Ramonet, M., Riley, W. J., Saito, M., Santini, M., Schroeder, R., Simpson, I. J., Spahni, R., Takizawa, A., Thornton, B. F., Tian, H., Tohjima, Y., Viovy, N., Voulgarakis, A., Weiss, R., Wilton, D. J., Wiltshire, A., Worthy, D., Wunch, D., Xu, X., Yoshida, Y., Zhang, B., Zhang, Z., and Zhu, Q.: Variability and quasi-decadal changes in the methane budget over the period 2000–2012, Atmos. Chem. Phys., 17, 11135–11161, https://doi.org/10.5194/acp-17-11135-2017, 2017.

Shaw J.T. et al. (2022) Large methane emission fluxes observed from tropical wetlands in Zambia. Global Biogeochemical Cycles 36: e2021GB007261

Sherwin, E.D., Rutherford, J.S., Zhang, Z. et al. US oil and gas system emissions from nearly one million aerial site measurements. Nature 627, 328–334 (2024). https://doi.org/10.1038/s41586-024-07117-5

Thanwerdas, J., Saunois, M., Berchet, A., Pison, I., and Bousquet, P.: Investigation of the renewed methane growth post-2007 with high-resolution 3-D variational inverse modeling and isotopic constraints, Atmos. Chem. Phys., 24, 2129–2167, https://doi.org/10.5194/acp-24-2129-2024, 2024.

Westbrook, G. K., et al. (2009), Escape of methane gas from the seabed along the West Spitsbergen continental margin, Geophys. Res. Lett., 36, L15608, doi:10.1029/2009GL039191.

Zhang Y. et al. Quantifying methane emissions from the largest oil-producing basin in the United States from space.Sci. Adv.6,eaaz5120(2020).DOI:10.1126/sciadv.aaz5120

Zhang, Z., Benjamin Poulter, Sara Knox, Ann Stavert, Gavin McNicol, Etienne Fluet-Chouinard, Aryeh Feinberg, Yuanhong Zhao, Philippe Bousquet, Josep G Canadell, Anita Ganesan, Gustaf Hugelius, George Hurtt, Robert B Jackson, Prabir K Patra, Marielle Saunois, Lena Höglund-Isaksson, Chunlin Huang, Abhishek Chatterjee, Xin Li, Anthropogenic emission is the main contributor to the rise of atmospheric methane during 1993–2017, National Science Review, Volume 9, Issue 5, nwab200, https://doi.org/10.1093/nsr/nwab200, 2022

Zhang, Z., Poulter, B., Feldman, A.F. et al. Recent intensification of wetland methane feedback. Nat. Clim. Chang. 13, 430–433 (2023). https://doi.org/10.1038/s41558-023-01629-0

Zhu, Q., Laughner, J.L., Cohen, R.C., 2022b. Combining Machine Learning and Satellite Observations to Predict Spatial and Temporal Variation of near Surface OH in North American Cities. Environ. Sci. Technol., 56, 11, doi:10.1021/acs.est.1c05636

---

## Author Comment (AC3)

**The Global Methane Budget: 2000-2020**

Saunois et al., ESSD, 2024

**Detailed Response to Anonymous Referee #3**

We acknowledge the referee for the time spent on reading and commenting on the paper. We thank him for his useful corrections and suggestions on the paper, which have helped to clarify and improve the manuscript. Below are the responses (in black) to his comments (in italics, blue). Changes in the text follow each response in bold font.

*This paper provides a comprehensive and transparent set of estimates of the global methane budget, updating previous versions of this living review and dataset. The update incorporates improved wetland and freshwater estimates compared to previous estimates, with reduced double-counting of tin bottom-up budgets. Partly as a result of this, the top-down and bottom-up budget estimates now overlap in terms of their uncertainty, when this was not previously the case. The budget dataset is publicly accessible.*

*As a potential peripheral user of this dataset (through methane lifetime being affected by atmospheric processes that I study), I had plenty to learn about the methane budget, but feel I provided a thorough check of the whole document and dataset from my perspective. I would be happy to use this dataset and consider the paper a useful reference document. The paper is long, of course, but the authors have done well to keep it readable. I would encourage the introduction of a contents section. I provide a few comments below for the authors to consider and check.*

We are really grateful to the reviewer for this encouraging comment and the positive attitude in front of such a long paper

*Minor comments*
*Contents – I think a paper this long warrants a Contents section.*

We added this comment in an earlier version. However, unfortunately, a Contents section is not allowed in ESSD. We included the Contents section of the manuscript in the first Section of the Supplementary Material. We advertise this feature at the end of the introduction adding the following sentence: " **For easier reading, the list of Contents of this manuscript is presented in the first section of the Supplementary Material**."

*Fig2 and L495-7 – The figure shows 2005 onwards not 2000. Should the figure have been altered at some point?*

This is true that Figure 7 starts in 2005. We have modified the figure and it starts now in 2000, consistent with the start of the budget as follows:

[Figure]

*L506 - "appear likely to follow the higher-emission trajectories over the next decade in terms of trend, and the peak year has not yet been reached." - this seems a bit too much of an assumption to me. since it is adding depth to a comment on the abstract, I think worth being precise. In fig2 (right) SSP1 and SSP3-7-low pathways show a change from peak growth to rapid reduction of methane over approx 10 years. So it doesn't matter what the trend is now, if it was maximum growth rate we could still have peaked, returned to approx current levels, and be declining in 10 years from now. That is quite different from "likely to follow the higher-emission trajectories". Now, perhaps you think that the preceding progress is not in place to follow those paths or the SSP scenarios are unrealistic, but I'd encourage you to explicitly say that if it's the case.*

There was probably a typo, as we meant "past" and not "next". The text has been modified as follows: "**but current emissions appear likely to follow the higher-emission trajectories over the past decade in terms of trend, and the peak year has not yet been reached. High or medium emission reduction rates as suggested by scenarios SSP1 and SSP2 have not yet happened.** "

*L1028 - "thus we estimate that ⅓" - is this purely your expert guess? If so, say so. Can you provide any other reasoning for this fraction? Could it actually be anything from 0 to 100%, or is there any reasoning that can be applied to think it's not likely to be entirely arbitrary?*

This value of ⅓ has been justified above line 1018, by citing four recent studies on the eutrophication of lakes. ⅓ is the lower bond value of the different estimates :" Several recent studies have estimated that anywhere between 30 and 50% of lakes are eutrophic (Cael et al., 2022; Qin et al., 2020; Sayers et al., 2015; Wu et al., 2022). These studies estimate numerical percentages (one by depth class: Qin et al., 2020), but none have estimated the percent of lake surface area that is eutrophic nor have any determined the extent of anthropogenic vs. natural eutrophication. Still, numerous studies have noted widespread increases in eutrophication indicators across lakes due to nutrient loading and warming (Griffiths et al., 2022; Ho et al., 2019; Taranu et al., 2015), thus we estimate that ⅓, or 11 Tg CH4 yr-1 of CH4 emissions from lakes >0.1 km2 could be anthropogenic."

As a result, we believe that the ⅓ has been justified in the text. Further studies would certainly help in reducing this uncertainty.

*Sec4.1.1 and Fig1b – Some of the clearest peaks in the growth rate are located around significant ENSO events 1997/98, 2015/16, 2020-22. Is it worth a short explanation? Example literature: https://acp.copernicus.org/articles/19/8669/2019/ And as that paper mentions, lightning can influence this variability as well as wildfire (see Murray ref within for such analysis).*

Indeed ENSO events have an impact in terms of emissions (biomass burning, or wetlands) and also sinks through OH, however this may not be systematic and depends on the intensity of the ENSO event. However the reviewer is right that we do not comment on the inter annual variability and the link to climate variability. We have added the following sentence and references: "**Both climate variability and anthropogenic emission changes are responsible for variations in atmospheric CH$_4$ growth rates. Indeed, climate variation such as El Nino Southern Oscillation induces changes in emissions such as biomass burning or wetland emission but also impact OH oxidation (e.g., Rowlinson et al., 2019 ; Zhao et al., 2020 ; Peng et al., 2022).**"

*Fig 7 – A third colour (e.g. yellow) could be used for "indirect anthropogenic fluxes" and then used to stripe the "combined wetland and inland freshwater" flux as you've done for the biomass and biofueld burning flux. Then a note in the caption equivalent to the other caption sentences.*

Indeed while reviewing Jackson et al. 2024 we modified the infographic including hatches, but we kept the orange color. Fig 4 has been redesigned so that indirect anthropogenic emissions are presented in pink, and called "indirect anthropogenic emissions" (see below). Figure 7 has also been modified to keep color consistency between the two. A darker green and pink arrow now depict natural and indirect anthropogenic fluxes - only for wetland and freshwater emissions.

[Figure]

*L1830 - "tropical" - you do not have a tropical category. You have a tropics + Southern Hemisphere category. I suggest you reword to be more precise at least in first reference. If you then want a sentence to say that you believe this category is dominated by tropical emissions and that you refer to it as "tropical" thereafter, then so be it.*

Indeed, this was a shortcut. This has been rephrased to : "**The latitudinal breakdown of emissions inferred from atmospheric inversions reveals a dominance of emissions in the latitudinal band 90S-**

**30N of 364Tg…As emissions in the Tropics (30S-30N) dominate this latitudinal contribution, we may refer to 90S-3N as the "Tropics" in the following**."

*L1830-1837 – Your regional categories are over different sized areas. I think it would be worth noting the area of each here. Your regional percentages may not be proportional to area necessarily, but they are roughly following it.*

This is really pertinent! We have calculated the land surfaces (emissions are mostly on land) in $km^2$:

| 90°S - 30°N ($km^2$) | 30°N - 60°N($km^2$) | 60°N - 90°N ($km^2$) | Total ($km^2$) |
|---|---|---|---|
| 7.094483e+07 | 4.635364e+07 | 1.731775e+07 | 1.346162e+08 |
| 52,7% | 34,4% | 12,8% | 100% |

In the text, we stated:" The latitudinal breakdown of emissions inferred from atmospheric inversions reveals a dominance of tropical emissions of 364 [337-390] Tg CH4 yr-1 , representing 64% of the global total (Table 5 and 6). 32% of the emissions are from the mid latitudes (187  [160-204] Tg CH4 yr-1 ) and 4% from high latitudes (above 60°N)." We have added the following sentence:
"**While the amounts of emissions depend on the surface area of the regions, the relative contribution of the emissions is much larger (12 points of percent) than the relative importance of the surface areas for the 90°S-30°N region, on the contrary the boreal regions (60°N-90°N) emissions contribute significantly less than the relative importance of their surface areas (9 points of percent).**"

*L2042 – Hopefully the research under the new NERC highlight topic on tropical oxidation will provide some good analysis to feed into the next GMB https://www.ukri.org/opportunity/addressing-environmental-challenges-nerc-highlight-topics-2024/*

Indeed, the Tropics play a major role regarding atmosphere oxidation capacity. More data and modeling effort on that topic will be of great help for the global methane budget as OH remains one of the main issues and source of uncertainties for top-down studies.

***Technical comments***
*L205 – typo "s estimated" and a bit confusing how "estimated" also used later in sentence.*
The typo has been corrected and the sentence cut in two and rephrased. "**The uncertainty in the chemical loss of $CH_4$ by OH, the predominant sink of atmospheric $CH_4$, has been estimated using Prather et al. (2012) and Rigby et al. (2017). The former study estimated this uncertainty at ~10% from the uncertainty in the reaction rate between $CH_4$ and OH, and the latter study was based on methyl-chloroform measurements.**"

*L265 – typo? "Saunosi"*
The typo has been corrected

*Fig4 - "XX to XX" for farm ponds – I don't think this has any general meaning. I suggest just removing, or using question marks, but in the least explaining this term in the caption.*
Indeed, we noticed this later on. The figure has been corrected.

*L1082 – A bit random how "BU" suddenly starts being used now, given that "bottom-up" is used earlier in the paragraph. I see the abbreviation is used elsewhere. At least define it at first use.*

We decided not to use the abbreviation in the text. I have searched for and replaced any remaining BU (or TD) abbreviation.

*L1353 - "Increased seepage of geogenic CH4 gas seeps along permafrost boundaries and lake beds may also be considered a direct flux" – please check the phrasing on this text, I'm not sure it makes sense. I wonder if "gas" should be "as".*

Indeed. This has been rephrased to "**Increased release of CH4 from deep geogenic sources that occurs as seepage along permafrost boundaries and lake beds may also be considered a direct flux**"

*L1971 - "inter annual", "inter-annual" or "interannual" are inconsistently used throughout.*
Thank you for spotting this, the text being long with multiple contributors, it happens that there are such inconsistencies. This has been corrected and we consistently now use "interannual".

*L2189 - "are sustained increase" typo... "is a" or "increases"*
This has been corrected

*L2211 – Is the following a normal requirement for a dataset connected to an ESSD publication? I'd normally assume that published data can be used freely for research (without requiring further permission), assuming correct acknowledgement/citation. - "The free availability of the data does not constitute permission for publication of the data."*
This is true. The sentence has been removed.

*Supp text 1 – subscript missed on "xa" and "Pa"*
This has been corrected

*Supp materials – Should "Plumer" be "Plummer" throughout references?*
This has been corrected

*fig_maps_wetlands_anthropogenic.nc - strange how "fos" metadata is poor compared to the other flux variables in this netcdf. I had to go to the supplement to find a reference to "fos".*

Indeed, we forgot to add Attributes to the variable "fos" in the submitted files. This has been corrected and the variable has been renamed "flux_ch4_fossils_fuels" for clarity. We really thank the reviewer for spotting this error. The modified files will be uploaded.

**References**

Peng, S., Lin, X., Thompson, R.L. et al. Wetland emission and atmospheric sink changes explain methane growth in 2020. Nature 612, 477–482 (2022). https://doi.org/10.1038/s41586-022-05447-w

Rowlinson, M. J., Rap, A., Arnold, S. R., Pope, R. J., Chipperfield, M. P., McNorton, J., Forster, P., Gordon, H., Pringle, K. J., Feng, W., Kerridge, B. J., Latter, B. L., and Siddans, R.: Impact of El Niño–Southern Oscillation on the interannual variability of methane and tropospheric ozone, Atmos. Chem. Phys., 19, 8669–8686, https://doi.org/10.5194/acp-19-8669-2019, 2019.

Zhao, Y., Saunois, M., Bousquet, P., Lin, X., Berchet, A., Hegglin, M. I., Canadell, J. G., Jackson, R. B., Deushi, M., Jöckel, P., Kinnison, D., Kirner, O., Strode, S., Tilmes, S., Dlugokencky, E. J., and Zheng, B.: On the role of trend

and variability in the hydroxyl radical (OH) in the global methane budget, Atmos. Chem. Phys., 20, 13011–13022, https://doi.org/10.5194/acp-20-13011-2020, 2020.

---

## Author Comment (AC4)

**The Global Methane Budget: 2000-2020**

Saunois et al., ESSD, 2024

Detailed Response to Dr Plummer.
We acknowledge Mr Plummer for his comments on the supplementary document and spotting errors in the Table.
Below are the responses (in black) to his comments (in italics, blue). Changes in the text follow each response in bold font.

*A very minor comment on what I believe is an error in Table S5. There is a heading for 'CMIP6 (2000-2009) - Hist' and a list of models given below, and then the heading 'CCMI (2000-2009)' but the models listed below the CCMI heading are the same models listed under CMIP6. The numbers are not the same, but should the models listed under CCMI be more in line with what is given in Table S4?*

Actually, we did not process CCMI-2022 outputs to calculate CH4 chemical loss, as we did for OH (Table S4). In Table S5, values are only from CMIP6 simulations. Chemical loss were not mandatories outputs for CCMI-2022 and we did not get enough data to proceed to the calculation.

Though, the caption of Table S5 was incorrect and has been corrected as follows:

" **Methane chemical loss in the troposphere and stratosphere in Tg CH4 yr-1 on average for the decade 2000-2009 and 2010-2019 as calculated from the model outputs contributing to CMIP6 (Collins et al., 2021) modeling activities. Values are provided for both the historical run (Hist- SSP3-7.0 starting in 2015), and the AMIP experiments. The AMIP experiments use prescribed SSTs and Sea Ice rather than a coupled ocean.** »

---

## Author Response (AR2)

**The Global Methane Budget: 2000-2020**

Saunois et al., ESSD, 2024

**Detailed Response to Anonymous Referee #3 – Review #2**

We warmly thank the referee for the time spent on reading and commenting on the paper for a second time. We thank him for his useful corrections and suggestions on the paper, which have helped to clarify and improve the manuscript. Below are the point by point responses (in black) to his comments (in italics, blue). Changes in the text follow each response in bold font.

Thank you to the authors for the changes they have made. I would be happy to see this published. A couple of minor points are below that will need consideration before publication.

On the changed text: "but current emissions appear likely to follow the higher-emission trajectories over the past decade in terms of trend, and the peak year has not yet been reached."

Whilst I'm happy with the sentiment of the change, I'm not sure the new sentence makes sense. I propose something like:

"but current emissions appear likely to follow the higher-emission trajectories, given that over the past decade the trend has followed such trajectories, and because the peak emission year has not yet been reached."

This attempts to correct for, what seem to me, clashing uses of tense within the sentence.

We have corrected the sentence has suggested by the reviewer: "After 2015, the SSPs span a range of possible outcomes, but current emissions appear likely to follow the higher-emission trajectories, given that over the past decade their trend has followed such trajectories, and because the peak emission year has not yet been reached. »

On the changed text: "While the amounts of emissions depend on the surface area of the regions, the relative

contribution of the emissions is much larger (12 points of percent) than the relative importance of the surface areas for the 90°S-30°N region, on the contrary the boreal regions (60°N-90°N) emissions contribute significantly less than the relative importance of their surface areas (9 points of percent)."

I am happy with the sentiment but the meaning of the new text is a bit unclear so I'm going to share how I read it, in case it helps you clarify it.

Firstly, if you are using land surface area as your response suggests, you should say "land surface area" not "surface area", as there is no reason to assume land.

Secondly, It's not obvious to me what's wrong with my interpretation below, but if there's isn't anything wrong, I don't understand how you can have a negative value for surface area percentage for the boreal region.

My interpretation: 90S-30N produces 64% of global emissions. Emissions are 12 percentage points higher than area percentage, so the percent of global land this region represents is 52%? (that's fine by me). 60N-90N produces 4% of global emissions. Emissions are 9 percentage points fewer than area percentage, so the percent of global land this region represents is -5%? (What?!) Please aim to clarify your text so that any error in my interpretation is less likely to be made.

We have clarified this part of the text as follows: "The amounts of emissions depend on the land surface area of the region, however the 90°S-30°N latitudinal band represents 53% of global land surfaces and the boreal region 60°N-90°N around 13%. Hence, the relative contribution of the emissions from the 90°S-30°N region is much larger (11 points of percent more) than the percentage of its land surface areas, on the contrary the boreal region (60°N-90°N) emissions contribute significantly less than the surface area percentage of this region (9 points of percent less)."